# Robust Neural Contextual Bandit against Adversarial Corruptions

**Yunzhe Qi, Yikun Ban, Arindam Banerjee, Jingrui He**
University of Illinois at Urbana-Champaign
Champaign, IL 61820
{yunzheq2,yikunb2,arindamb,jingrui}@illinois.edu

## Abstract

Contextual bandit algorithms aim to identify the optimal arm with the highest reward among a set of candidates, based on the accessible contextual information. Among these algorithms, neural contextual bandit methods have shown generally superior performances against linear and kernel ones, due to the representation power of neural networks. However, similar to other neural network applications, neural bandit algorithms can be vulnerable to adversarial attacks or corruptions on the received labels (i.e., arm rewards), which can lead to unexpected performance degradation without proper treatments. As a result, it is necessary to improve the robustness of neural bandit models against potential reward corruptions. In this work, we propose a novel neural contextual bandit algorithm named R-NeuralUCB, which utilizes a novel context-aware Gradient Descent (GD) training strategy to improve the robustness against adversarial reward corruptions. Under over-parameterized neural network settings, we provide regret analysis for R-NeuralUCB to quantify reward corruption impacts, without the commonly adopted arm separateness assumption in existing neural bandit works. We also conduct experiments against baselines on real data sets under different scenarios, in order to demonstrate the effectiveness of our proposed R-NeuralUCB.

## 1 Introduction

Contextual bandits refer to one specific type of multi-armed bandit (MAB) problems, where the learner can access the arm context information during the decision-making process. Contextual bandit algorithms have been commonly applied in various real-world applications, including online content recommendation [59, 79, 8], and medical experiments [30, 73, 7]. While these algorithms have been proved effective for numerous online learning tasks, they can be susceptible to the malicious feedback from the environment, such as malicious user feedback in recommender systems [64], and corrupted labels under active learning settings [61]. This can potentially impair the model performance and interfere with the internal decision-making logic. One renowned research direction formulates this problem as *contextual bandits with adversarial corruptions* [57, 42, 16], where received *arm rewards* can be potentially "corrupted" by the unknown adversary. In this case, bandit algorithms need to be robust against such adversarial corruptions, otherwise they can lead to sub-optimal results. Existing works on contextual bandits with corruptions are mainly based on linear [17, 29, 57] and kernelized bandits [15, 16], where the unknown reward mapping function is assumed to be linear, or lies in a specified Reproducing Kernel Hilbert Space (RKHS). However, one key challenge is that these assumptions can evidently fail under real-world application scenarios [86], when we have little prior knowledge regarding this mapping function, or it becomes increasingly complex.

In the face of this challenge, neural bandit algorithms [86, 84, 11, 12] have been proposed to relax the assumptions on reward functions. By leveraging the representation power of neural networks, neural contextual bandit algorithms are able to deal with complex reward functions irrespective of whether

they are linear or non-linear, along with suitable exploration strategies for tackling the exploitation-exploration dilemma [5, 59]. While neural bandit algorithms have been proved effective [9, 68, 60], they can be sensitive to adversarial corruptions as well. From perspectives of trustworthiness, it is well known that neural models can be susceptible to "label attacks" [71, 62, 65], which is akin to reward corruptions in bandit settings. Failure to comply with robustness requirements can impair the feasibility under real-world application scenarios like recommender systems [82, 27, 87, 78], and therefore it is necessary for neural bandit methods to be robust against potential adversarial corruptions. Furthermore, existing neural bandit works generally require *arm separateness assumptions* (e.g., assuming a positive-definite Neural Tangent Kernel [NTK] Gram matrix [86, 84, 26], or positive arm Euclidean distances [11, 67]), which will require no duplicate arm contexts are observed (or chosen) by the learner. This can lead to additional vulnerabilities when arm contexts are intentionally chosen by the adversary (e.g., assigning duplicate arms in the candidate pool across different rounds), making the arm separateness assumption fail in such adversarial environments.

Motivated by aforementioned challenges, in this paper, we propose a novel neural contextual bandit algorithm named Robust Neural-UCB (R-NeuralUCB), which can model the discrepancy among candidate arms and adopt arm-specific context-aware Gradient Descent (GD) to enhance model robustness against reward corruptions. Instead of applying ordinary GD to update the network parameters, R-NeuralUCB utilizes a fine-grained GD strategy by modeling the importance level of training samples, to reduce the impact of potential adversarial reward corruptions. Meanwhile, to improve the model performance from both the theoretical and empirical perspectives, R-NeuralUCB simultaneously perceives the uncertainty levels of candidate arms, and adaptively customizes network parameters for each of these candidates. To deal with the exploitation-exploration dilemma, R-NeuralUCB is equipped with an informative exploration mechanism based on Upper Confidence Bound (UCB) to achieve principled exploration. In addition, we present regret analysis without the commonly adopted *arm separateness assumption*, which reinforces R-NeuralUCB's theoretical robustness under adversarial scenarios. Our contributions can be summarized as follows:

- **Problem Settings and Proposed algorithm:** We study a novel neural bandit problem, where the received arm reward can be potentially corrupted by the unknown adversary. To deal with this problem, we propose a novel neural bandit algorithm called R-NeuralUCB, which leverages a refined context-aware Gradient Descent training strategy to improve the model robustness against potential arm reward corruptions. While we consider all the observed arms are governed by the same unknown reward mapping function as in (1) similar to existing neural bandit works, our R-NeuralUCB interestingly maintains separate model parameters specific to the different candidate arms, for a fine-grained way of improving the robustness. This casts lights on our contributions of novel algorithmic designs, compared to related existing methods without an arm-specific modeling (e.g., [42] and our base algorithm NeuralUCB-WGD in Appendix E).

- **Theoretical Analysis:** With over-parameterized neural networks, we present the regret analysis for R-NeuralUCB. Given finite horizon $T$, effective dimension of NTK Gram matrix $\widetilde{d}$, and corruption level $C$, R-NeuralUCB enjoys a data-dependent regret bound of $\widetilde{\mathcal{O}}(\widetilde{d}\sqrt{T} + C\widetilde{d})$. In addition, to ensure R-NeuralUCB is capable of handling contexts specified by the adversary (e.g., duplicate arms across different rounds), our analysis removes the *arm separateness assumption* dependency, a widely adopted assumption for neural bandit literature, which can be of independent interest.

- **Experiments:** We conduct experiments on publicly available real-world data sets with various specifications. Under different types of reward corruptions, our R-NeuralUCB can achieve better performance, and is less vulnerable to reward corruptions than baselines.

## 2 Related Works

**Contextual Bandits with Adversarial Corruptions.** To begin with, there have been numerous studies [76, 17, 29, 57, 63, 31, 22, 54] working on tackling adversarial reward corruptions under linear contextual bandit settings [59, 24]. A related topic is bandits with mis-specifications [36, 55, 33, 53, 83, 75], where the deviation of reward estimation comes from problem modeling instead of the adversary. On the other hand, kernelized bandits [15, 40, 16] extend the adversarial corruption problem to non-linear cases by assuming the reward mapping is a functional in the specified RKHS [72], while comparable ideas are also applicable for robust Bayesian Optimization [52]. Adversarial corruptions are also studied for other formulations, such as Lipschitz bandits [49, 89] and MAB without contexts [18, 80]. However, compared with neural bandit methods, these works generally require assumptions on the reward function prior, which may not be satisfied in real-world scenarios.

**Neural Contextual Bandits.** Neural contextual bandits algorithms are proposed to leverage the representation power of neural networks, and relax the assumptions on the reward mapping functions that can be linear or non-linear. Neural-UCB [86] applies a fully-connected (FC) neural network for reward estimation and utilizes corresponding network gradients for principled exploration. Comparable ideas have been leveraged by other neural bandit works [84, 50, 25, 10, 37, 46, 9, 60, 11], and adopted under various application scenarios such as active learning [74, 13, 6], and bandit-based graph learning [66, 67, 51] with graph neural networks [77, 34, 35]. Alternatively, [81] utilizes the neural network to embed original arm contexts for regression. [26] utilizes inverse reward gap for exploration, and [48] achieves exploration with the reward perturbation. However, as these methods are not designed to defend against reward corruptions and widely require arm separateness assumptions, they can fail to meet the robustness requirements in an adversarial environment.

## 3 Problem Definition

Let $T$ be the finite horizon. In round $t \in [T]$, the learner receives $K$ candidate arms $\mathcal{X}_t$, $|\mathcal{X}_t| = K$, and each arm $\boldsymbol{x}_{i,t} \in \mathcal{X}_t$ with arm index $i \in [K]$ is described by a $d$-dimensional vector $\boldsymbol{x}_{i,t} \in \mathbb{R}^d$. The learner will then choose one arm $\boldsymbol{x}_t \in \mathcal{X}_t$ and receive its reward $r_t$. The index of $\boldsymbol{x}_t$ is denoted by $i_t \in [K]$, s.t. $\boldsymbol{x}_t = \boldsymbol{x}_{i_t,t}$. Here, similar to existing works (e.g., [42, 16, 15, 86, 84]), we define *corruption-free* arm reward $\widetilde{r}_{i,t}$ for each candidate arm $\boldsymbol{x}_{i,t} \in \mathcal{X}_t$, as well as *corrupted* arm reward $r_t$ for chosen arm $\boldsymbol{x}_t \in \mathcal{X}_t$, as

$$\widetilde{r}_{i,t} = h(\boldsymbol{x}_{i,t}) + \epsilon_{i,t}, \qquad r_t = \widetilde{r}_t + c_t = h(\boldsymbol{x}_t) + \epsilon_t + c_t, \qquad (1)$$

where $h : \mathbb{R}^d \mapsto \mathbb{R}$ is an unknown reward mapping function that can be either linear or non-linear. $\epsilon_{i,t} \in \mathbb{R}$ stands for zero-mean $\nu$-sub-Gaussian random noise which is standard for stochastic contextual bandit works (e.g., [24, 72, 86]), and $c_t \in \mathbb{R}$ is the unknown adversarial corruption imposed by the adversary. While kernelized bandit works (e.g., [16, 72]) assume $h(\cdot)$ belongs to the RKHS induced by specified kernels, we alternatively consider $h(\cdot)$ as an arbitrary unknown function, and utilize the neural model to learn this mapping with flexibility.

Taking expectation w.r.t. zero-mean noise $\epsilon$, for the chosen arm $\boldsymbol{x}_t \in \mathcal{X}_t$, we denote its expected perturbed reward $\mathbb{E}[r_t] = h(\boldsymbol{x}_t) + c_t$; meanwhile, for each candidate arm $\boldsymbol{x}_{i,t} \in \mathcal{X}_t$, its expected corruption-free reward $\mathbb{E}[\widetilde{r}_{i,t}] = h(\boldsymbol{x}_{i,t})$. Here, we consider $\mathbb{E}[r]$ and $\mathbb{E}[\widetilde{r}]$ both fall into value range $[0, 1]$, analogous to existing works (e.g., [86, 84, 11, 50]). This is intuitive as numerous real-world applications work with bounded rewards (e.g., online recommendation tasks with normalized rating [67] or binary feedback [24]); and the adversary also needs its attack to be stealthy, by ensuring perturbed rewards fall into the normal value range. With previously chosen arms $\{\boldsymbol{x}_\tau\}_{\tau \in [t]}$ up to round $t$, we denote received context-reward tuples with *perturbed rewards* as $\mathcal{P}_t := \{\boldsymbol{x}_\tau, r_\tau\}_{\tau \in [t]} = \{\boldsymbol{x}_{i_\tau,\tau}, r_{i_\tau,\tau}\}_{\tau \in [t]}$, and the corresponding context-reward tuples with *corruption-free rewards* as $\widetilde{\mathcal{P}}_t := \{\boldsymbol{x}_\tau, \widetilde{r}_\tau\}_{\tau \in [t]} = \{\boldsymbol{x}_{i_\tau,\tau}, \widetilde{r}_{i_\tau,\tau}\}_{\tau \in [t]}$, where each corruption-free but imaginary unobserved reward is $\widetilde{r}_\tau = h(\boldsymbol{x}_\tau) + \epsilon_\tau, \tau \in [t]$ based on (1).

**Learning Objective.** Our objective is to minimize cumulative pseudo-regret for $T$ rounds:

$$R(T) = \sum\nolimits_{t=1}^{T} \mathbb{E}[\widetilde{r}_t^* - \widetilde{r}_t], \qquad (2)$$

where $\mathbb{E}[\widetilde{r}_t] = h(\boldsymbol{x}_t)$ is the expected corruption-free reward of the chosen arm $\boldsymbol{x}_t \in \mathcal{X}_t$, and $\mathbb{E}[\widetilde{r}_t^*] = \max_{\boldsymbol{x}_{i,t} \in \mathcal{X}_t} [h(\boldsymbol{x}_{i,t})]$ stands for that of the optimal arm $\boldsymbol{x}_t^* \in \mathcal{X}_t$.

**Corruption Level.** If the adversary determines the reward corruption $c_{i,t}$ for each candidate arm $\boldsymbol{x}_{i,t} \in \mathcal{X}_t$ beforehand, without observing the learner's choice $\boldsymbol{x}_t$, some works (e.g., [38]) formulate the corruption level measurement as $C' = \sum_{t \in [T]} [\max_{i \in [K]} |c_{i,t}|]$. In this work, similar to [42, 16], we alternatively consider reward corruptions are determined w.r.t. particular chosen arms $\{\boldsymbol{x}_t\}_{t \in [T]}$, and formulate the corruption level as $C = \sum_{t \in [T]} |c_t|$. This leads to $C \leq C'$.

## 4 Proposed Algorithm: Robust Neural-UCB (R-NeuralUCB)

Recall that in (1), arm rewards under neural bandit settings are governed by the unknown reward mapping function $h(\cdot)$, where $h(\cdot)$ can be an arbitrary function. For our proposed R-NeuralUCB, we adopt a neural network $f(\cdot)$ to approximate $h(\cdot)$ for reward estimation.

**Network Structure.** We use $f(\cdot; \boldsymbol{\theta})$ to denote an FC network with depth $L \geq 2$ and width $m \in \mathbb{N}^+$:

$$f(\boldsymbol{x}; \boldsymbol{\theta}) := \sqrt{m} \boldsymbol{\theta}_L \sigma(\boldsymbol{\theta}_{L-1} \sigma(\boldsymbol{\theta}_{L-2} \ldots \sigma(\boldsymbol{\theta}_1 \boldsymbol{x}))) \qquad (3)$$

Table 1: Comparison of $T$-round regret bounds with adversarial corruption level $C$.

| Algorithm | Reward Function | Corruption $C$ | Regret Bound* |
|---|---|---|---|
| Robust OFUL [76] | Linear | Known | $\widetilde{\mathcal{O}}(d\sqrt{T} + \sqrt{T\sum_{t=1}^{T}c_t^2})$ |
| CW-OFUL [42] | Linear | Known | $\widetilde{\mathcal{O}}(d\sqrt{T} + Cd)$ |
| COBE + OFUL [76] | Linear | Unknown | $\widetilde{\mathcal{O}}(d\sqrt{T} + \sqrt{T\sum_{t=1}^{T}c_t^2})$ |
| CW-OFUL ($\bar{C} = \sqrt{T}$) [42] | Linear | Unknown | $\widetilde{\mathcal{O}}(d\sqrt{T})$, if $C \leq \sqrt{T}$. Otherwise $\mathcal{O}(T)$ |
| Fast-slow GP-UCB [15] | Kernelized | Known | $\widetilde{\mathcal{O}}(\widetilde{d}\sqrt{T} + C\widetilde{d}\sqrt{T})$ |
| RGB-PE [16] | Kernelized | Known | $\widetilde{\mathcal{O}}(\widetilde{d}\sqrt{T} + C\widetilde{d}^{3/2})$ |
| Fast-slow GP-UCB [15] | Kernelized | Unknown | $\widetilde{\mathcal{O}}(\widetilde{d}\sqrt{T} + C\widetilde{d}\sqrt{T})$ |
| NeuralUCB-WGD (Thm. E.1) | Arbitrary | Known | $\widetilde{\mathcal{O}}(\widetilde{d}\sqrt{T} + C\widetilde{d}^{3/2})$ |
| R-NeuralUCB (Thm. 5.6) | Arbitrary | Unknown | $\widetilde{\mathcal{O}}(\widetilde{d}\sqrt{T} + C\beta^{-1}\widetilde{d})$ |

* $d$: context dimension; $\widetilde{d}$: NTK matrix effective dimension or kernel information gain; $\beta$: data-dependent gradient deviation term.

where $\sigma(\cdot)$ is element-wise ReLU activation, and we have trainable weight matrices $\boldsymbol{\theta}_1 \in \mathbb{R}^{m \times d}$, $\boldsymbol{\theta}_l \in \mathbb{R}^{m \times m}, 2 \leq l \leq L-1, \boldsymbol{\theta}_L \in \mathbb{R}^{1 \times m}$. For the ease of notation, we denote vectorized parameters

$$\boldsymbol{\theta} := [\text{vec}(\boldsymbol{\theta}_1)^\intercal, \text{vec}(\boldsymbol{\theta}_2)^\intercal, \ldots, \boldsymbol{\theta}_L]^\intercal \in \mathbb{R}^p,$$

with the dimensionality of $p$, and randomly initialized parameters are denoted by $\boldsymbol{\theta}_0$. Then, we let $g(\boldsymbol{x}; \boldsymbol{\theta}) = \text{vec}(\nabla_{\boldsymbol{\theta}} f(\boldsymbol{x}; \boldsymbol{\theta})) \in \mathbb{R}^p$ be vectorized network gradients w.r.t. input $\boldsymbol{x}$ and parameters $\boldsymbol{\theta}$.

We motivate our proposed R-NeuralUCB by first mentioning a base algorithm named Neural-UCB with Weighted GD (NeuralUCB-WGD), which is elaborated in Appendix E. To begin with, NeuralUCB-WGD measures the uncertainty level of training samples (i.e., previously received arm-reward pairs) through their UCB values, as the UCB essentially measures *arm uncertainty levels* in terms of reward estimation [24, 72, 86]. Then, different from conventional neural bandit methods that treat all training samples equally [86, 84], inspired by [42], NeuralUCB-WGD utilizes a weighted GD process to train neural model $f(\cdot)$ for estimating arm rewards, where training samples with high uncertainty levels will be downplayed. The main idea is that although we do not know which training samples are corrupted, we instead aim to reduce potentially severe impacts caused by adversarial corruptions, by paying relatively more attention on the training samples (arm-reward pairs) with low uncertainty levels, for a stable GD training process. We also present corresponding regret analysis for NeuralUCB-WGD in Appendix E.2, as well as experiments in Section 6.

However, notice that the neural model will also perceive varying uncertainty levels for different candidate arms in terms of reward estimation. In this case, simply applying the same exploitation-exploration strategy across all candidate arms can overlook this discrepancy, leading to insufficient granularity w.r.t. reward estimation. For instance, regarding candidate arms with low uncertainty levels, it can be more beneficial to adequately leverage existing training samples for estimating their rewards, instead of sharing an identical exploitation-exploration strategy with other high-uncertainty candidate arms. Meanwhile, analogous to existing works (e.g., [42, 16, 15]), NeuralUCB-WGD supposes a known corruption level $C$ for regret analysis (Theorem E.1), which can be difficult to satisfy if we have limited knowledge regarding the unknown adversary. With the above motivations, we propose R-NeuralUCB as a refined solution to further enhance neural model robustness against potential reward corruptions. For readers' reference, we also compare our proposed R-NeuralUCB and NeuralUCB-WGD with some regret results from existing works in Table 1.

### 4.1 R-NeuralUCB: Robust Neural-UCB

Our R-NeuralUCB formulates a novel context-aware GD process, by taking the uncertainty information of both candidate arms and training samples into account, for neural network training and decision making. Here, R-NeuralUCB customizes individual sets of network parameters $\boldsymbol{\theta}_{i,t-1}$ for each candidate arm $\boldsymbol{x}_{i,t} \in \mathcal{X}_t, i \in [K]$, before the actual arm recommendation. Afterwards, these arm-specific networks are applied for arm reward estimation, along with an informative arm-specific UCB-based exploration mechanism. The pseudo-code is presented in Algorithm 1.

**Arm Weight Formulation.** Inspired by NTK-based exploration mechanisms [86, 51, 66, 11, 84], we measure arm uncertainty levels with the weighted gradient norm of arms. With a regularization parameter $\lambda > 0$, we first define a weight-free gradient covariance matrix $\overline{\boldsymbol{\Sigma}}_{t-1} = \lambda \mathbf{I} + \sum_{\tau \in [t-1]} g(\boldsymbol{x}_\tau; \boldsymbol{\theta}_{\tau-1}) g(\boldsymbol{x}_\tau; \boldsymbol{\theta}_{\tau-1})^\intercal / m$. Here, $\boldsymbol{\theta}_{\tau-1}$ is the shorthand of $\boldsymbol{\theta}_{i_\tau, \tau-1}$, representing the network parameters of the previously chosen arm $\boldsymbol{x}_\tau = \boldsymbol{x}_{i_\tau, \tau}, \tau \in [t-1]$. Then, for each

---

**Algorithm 1** Robust Neural-UCB (R-NeuralUCB)

---

1: **Input:** Time horizon $T$. GD iterations $J$. Learning rate $\eta$. Exploration coefficient $\nu$. Scaling parameter $\alpha$. Norm parameter $S$. Regularization parameter $\lambda$.
2: **Initialization:** Parameters $\boldsymbol{\theta}_0$. Weight-free covariance matrix $\bar{\boldsymbol{\Sigma}}_0 = \lambda \mathbf{I}$. Records $\mathcal{P}_0 = \emptyset$.
3: **for** each round $t \in [T]$ **do**
4:     Observe a collection of $K$ candidate arms $\mathcal{X}_t = \{\boldsymbol{x}_{i,t}\}_{i \in [K]}$.
5:     **for** each candidate arm $\boldsymbol{x}_{i,t} \in \mathcal{X}_t$ **do**
6:         **if** $t$ equals to 1 **then**
7:             $\boldsymbol{\theta}_{i,t-1} \leftarrow \boldsymbol{\theta}_0$.
8:         **else**
9:             With arm weights $\{w_{i,t}^{(\tau)}\}_{\tau \in [t-1]}$ in (4), train parameters $\boldsymbol{\theta}_{i,t-1}$ with GD and the arm-specific loss function in (5) based on received records $\mathcal{P}_{t-1}$.
10:         **end if**
11:         For candidate arm $\boldsymbol{x}_{i,t}$, calculate its benefit score $U(\boldsymbol{x}_{i,t})$ based on (6).
12:     **end for**
13:     Choose arm $\boldsymbol{x}_t = \arg\max_{\boldsymbol{x}_{i,t} \in \mathcal{X}_t} [U(\boldsymbol{x}_{i,t})]$ with the highest benefit score.
14:     Receive arm reward $r_t$, and update the records, such that $\mathcal{P}_t \leftarrow \mathcal{P}_{t-1} \cup \{(\boldsymbol{x}_t, r_t)\}$.
15:     Update the shorthand $\boldsymbol{\theta}_{t-1} \leftarrow \boldsymbol{\theta}_{i_t,t-1}$, and matrix $\bar{\boldsymbol{\Sigma}}_t \leftarrow \bar{\boldsymbol{\Sigma}}_{t-1} + g(\boldsymbol{x}_t; \boldsymbol{\theta}_{t-1}) g(\boldsymbol{x}_t; \boldsymbol{\theta}_{t-1})^\mathsf{T}/m$.
16: **end for**

---

candidate arm $\boldsymbol{x}_{i,t} \in \mathcal{X}_t$, we formulate its weight w.r.t. previously chosen arm $\boldsymbol{x}_\tau, \tau \in [t-1]$ as

$$
w_{i,t}^{(\tau)} = \min \left\{ 1, \ \frac{\alpha \cdot \min_{\boldsymbol{x} \in \mathcal{X}_t} \|g(\boldsymbol{x}; \boldsymbol{\theta}_{t-1})/\sqrt{m}\|_{\bar{\boldsymbol{\Sigma}}_{t-1}^{-1}}^2}{g_\tau \cdot \|g(\boldsymbol{x}_{i,t}; \boldsymbol{\theta}_{t-1})/\sqrt{m}\|_{(\bar{\boldsymbol{\Sigma}}_{t-1}^{(\kappa)})^{-1}}} \right\}, \tag{4}
$$

with $\kappa^2$-scaled covariance matrix being $\bar{\boldsymbol{\Sigma}}_{t-1}^{(\kappa)} := \lambda \mathbf{I} + \kappa^2 \cdot \sum_{\tau \in [t-1]} g(\boldsymbol{x}_\tau; \boldsymbol{\theta}_{\tau-1}) g(\boldsymbol{x}_\tau; \boldsymbol{\theta}_{\tau-1})^\mathsf{T}/m$, for a constant $\kappa \in (0,1)$. Alternatively, we also denote $w_{i,t}^{(\tau)} = \min\{1, \alpha \cdot \mathsf{frac}_\tau(\boldsymbol{x}_{i,t}; \mathcal{X}_t, \bar{\boldsymbol{\Sigma}}_{t-1})\}$, with $\mathsf{frac}_\tau(\cdot)$ being a shorthand that integrally represents the fraction term in (4). A tunable parameter $\alpha > 0$ and the squared round-wise minimum weighted norm $\min_{\boldsymbol{x} \in \mathcal{X}_t} \|g(\boldsymbol{x}; \boldsymbol{\theta}_{t-1})/\sqrt{m}\|_{\bar{\boldsymbol{\Sigma}}_{t-1}^{-1}}^2$ in the numerator are applied for scaling purposes. We also include complementary discussions for arm weight scaling in Appendix B.5. Meanwhile, we have $g_\tau = \|g(\boldsymbol{x}_\tau; \boldsymbol{\theta}_{\tau-1})/\sqrt{m}\|_{(\bar{\boldsymbol{\Sigma}}_{\tau-1}^{(\kappa)})^{-1}}, \tau \in [t-1]$ quantifying uncertainty levels of previously chosen arms (training samples) motivated by UCB-based exploration strategies (e.g., [86, 42]). Since previous $\{g_\tau\}_{\tau \in [t-1]}$ values can be reused, we only need to compute and store $g_t$ for current round $t$. As a result, if candidate arm $\boldsymbol{x}_{i,t}$ is of high uncertainty (i.e., large $\|g(\boldsymbol{x}_{i,t}; \boldsymbol{\theta}_{t-1})/\sqrt{m}\|_{\bar{\boldsymbol{\Sigma}}_{t-1}^{-1}}$ value), its arm weights $\{w_{i,t}^{(\tau)}\}_{\tau \in [t-1]}$ will become small.

**Model Training with Context-aware GD.** According to line 9 in Algorithm 1, we perform model training *before* the actual arm recommendation in each round $t \in \{2, \dots, T\}$. For each candidate $\boldsymbol{x}_{i,t} \in \mathcal{X}_t$, we train its arm-specific parameters $\boldsymbol{\theta}_{i,t-1}$ with $J$ iterations of GD and received records $\mathcal{P}_{t-1} = \{(\boldsymbol{x}_\tau, r_\tau)\}_{\tau \in [t-1]}$. Starting from initialization $\boldsymbol{\theta}_{i,t-1}^{(0)} = \boldsymbol{\theta}_0$, we have $j$-th GD iteration $(j \in [J])$ being $\boldsymbol{\theta}_{i,t-1}^{(j)} = \boldsymbol{\theta}_{i,t-1}^{(j-1)} - \eta \nabla_{\boldsymbol{\theta}} \mathcal{L}_{i,t}(\mathcal{P}_{t-1}; \boldsymbol{\theta}_{i,t-1}^{(j-1)})$, where $\eta > 0$ refers to the learning rate. We formulate a loss function $\mathcal{L}_{i,t}(\cdot; \cdot), i \in [K]$ specified to candidate arm $\boldsymbol{x}_{i,t} \in \mathcal{X}_t$ as

$$
\mathcal{L}_{i,t}(\mathcal{P}_{t-1}; \boldsymbol{\theta}) = \sum_{(\boldsymbol{x}_\tau, r_\tau) \in \mathcal{P}_{t-1}} \frac{w_{i,t}^{(\tau)}}{2} \cdot \left| f(\boldsymbol{x}_\tau; \boldsymbol{\theta}) - r_\tau \right|^2 + \frac{m\lambda}{2} \cdot \|\boldsymbol{\theta} - \boldsymbol{\theta}_0\|_2^2, \tag{5}
$$

where the $L_2$ loss is scaled by arm weights $w_{i,t}^{(\tau)}, \tau \in [t-1]$ from (4). Intuitively, if arm weights $w_{i,t}^{(\tau)}$ are large (i.e., low uncertainty level), we proceed to train a neural model that adequately fits the collected training data (i.e., previously received records) $\mathcal{P}_{t-1}$, instead of staying around the random initialization $\boldsymbol{\theta}_0$ given the $L_2$ regularization. On the other hand, if arm weights $w_{i,t}^{(\tau)}$ are small, it means that the uncertainty level in terms of reward estimation is high. In this case, we prefer being relatively conservative to prevent potentially large impacts caused by adversarial corruptions. As a result, R-NeuralUCB will focus more on the training samples in $\mathcal{P}_{t-1}$ with low uncertainty levels, and stay relatively close to the random initialization $\boldsymbol{\theta}_0$ due to the regularization term in (5).

In practice, instead of starting from $\boldsymbol{\theta}_0$ in each round $t \in \{2, \ldots, T\}$, we can alternatively initiate the GD process from the existing trained parameters to reduce computational cost, inspired by the concept of warm-start GD [13]. Here, we can start from $\boldsymbol{\theta}_{t-2}$, the parameters of the previously chosen arm $\boldsymbol{x}_{t-1}$, and fine-tune arm-specific parameters $\boldsymbol{\theta}_{i,t-1}$ for each candidate arm $\boldsymbol{x}_{i,t} \in \mathcal{X}_t$, based on its loss function $\mathcal{L}_{i,t}(\cdot; \cdot)$ and a small batch of samples from $\mathcal{P}_{t-1}$. Further details are elaborated in Appendix B.6, and this approach is also applied for the experiments in Section 6.

**Arm Selection.** For candidate arm $\boldsymbol{x}_{i,t} \in \mathcal{X}_t$ and arm weights $\{w_{i,t}^{(\tau)}\}_{\tau \in [t-1]}$, we formulate its arm-specific gradient covariance matrix $\boldsymbol{\Sigma}_{i,t-1} = \lambda \mathbf{I} + \sum_{\tau \in [t-1]} w_{i,t}^{(\tau)} \cdot g(\boldsymbol{x}_\tau; \boldsymbol{\theta}_{\tau-1}) g(\boldsymbol{x}_\tau; \boldsymbol{\theta}_{\tau-1})^{\mathsf{T}} / m$. Here, if the variance proxy value $\nu$ in (1) is unknown, similar to existing works (e.g., [86, 84]), we deem $\nu \geq 0$ as a tunable parameter to control the exploration intensity. With our UCB-type exploration motivated by Appendix Lemma C.11, we formulate the benefit score for arm $\boldsymbol{x}_{i,t} \in \mathcal{X}_t$ as

$$U(\boldsymbol{x}_{i,t}) = f(\boldsymbol{x}_{i,t}; \boldsymbol{\theta}_{i,t-1}) + \gamma_{i,t-1} \cdot \sqrt{g(\boldsymbol{x}_{i,t}; \boldsymbol{\theta}_{i,t-1})^{\mathsf{T}} \boldsymbol{\Sigma}_{i,t-1}^{-1} g(\boldsymbol{x}_{i,t}; \boldsymbol{\theta}_{i,t-1}) / m}, \qquad (6)$$

where the confidence coefficient $\gamma_{i,t-1} = \zeta \cdot \left( \nu \sqrt{\log \frac{\det(\boldsymbol{\Sigma}_{i,t-1})}{\det(\lambda \mathbf{I})} - 2 \log(\delta)} + \sqrt{\lambda} S \right)$, along with a constant $\zeta > 0$ from Lemma C.11. Afterwards, we choose $\boldsymbol{x}_t = \arg\max_{\boldsymbol{x}_{i,t} \in \mathcal{X}_t} \left[ U(\boldsymbol{x}_{i,t}) \right]$ (line 13, Algorithm 1), based on calculated arm benefit scores in (6). After receiving reward $r_t$, the collected records will be updated by $\mathcal{P}_t \leftarrow \mathcal{P}_{t-1} \cup \{(\boldsymbol{x}_t, r_t)\}$ (line 14, Algorithm 1). We also update the shorthand for model parameters of the chosen arm as $\boldsymbol{\theta}_{t-1} \leftarrow \boldsymbol{\theta}_{i_t, t-1}$, and the weight-free covariance matrix $\bar{\boldsymbol{\Sigma}}_t \leftarrow \bar{\boldsymbol{\Sigma}}_{t-1} + g(\boldsymbol{x}_t; \boldsymbol{\theta}_{t-1}) g(\boldsymbol{x}_t; \boldsymbol{\theta}_{t-1})^{\mathsf{T}} / m$ for next round $t+1$ (line 15, Algorithm 1).

In summary, the primary goal of R-NeuralUCB is to customize individual learning objectives (i.e., loss functions) for different candidate arms by leveraging arm uncertainty information before pulling an arm. For candidate arms with high uncertainty levels, the neural model may lack confidence in estimating rewards based on current records, due to potential reward corruptions, which can lead to significant estimation errors. In this situation, by using the regularization term $\frac{m\lambda}{2} \|\boldsymbol{\theta} - \boldsymbol{\theta}_0\|_2^2$, we prefer to adopt a relatively conservative approach, training a model close to random initialization to mitigate the potentially large impacts of adversarial corruptions. This approach is inspired by existing work on enhancing model robustness through regularization techniques (e.g., [69, 23]). On the other hand, for candidate arms with low uncertainty, we aim to train neural models that fully utilize the received records for reward estimation. Since the model is confident in its estimation, the received samples can provide adequate reference. With larger arm weights, the loss function can focus more on the training samples, instead of staying closely around $\boldsymbol{\theta}_0$.

## 5 Theoretical Analysis

To the best of our knowledge, we provide the first theoretical results under the neural bandit settings with adversarial reward corruptions, and our proof flow is distinct from those of linear and kernelized bandit works. In particular, as our ReLU activation in (3) is not Lipschitz smooth [3, 21], it leads to additional challenges for our theoretical analysis, since a small perturbation on rewards can lead to drastic changes of network gradients. As a result, even with a small corruption level $C$, the corrupted model parameters trained by GD can significantly deviate from the imaginary network parameters trained with corresponding corruption-free rewards. Therefore, it is non-trivial to quantify corruption impacts from theoretical perspectives, which simultaneously makes our proof flow differ significantly from that of the vanilla Neural-UCB [86]. We include additional discussions on analysis distinctions and our contributions in Appendix B.3. To begin with, we first introduce some preliminaries.

**Parameter Initialization.** Analogous to existing works [86, 21, 3, 9, 84], for an $L$-layer network of width $m$ in (3), we let its intermediate-layer matrices $\boldsymbol{\theta}_l = \left( \begin{smallmatrix} \boldsymbol{\Lambda} & \mathbf{0} \\ \mathbf{0} & \boldsymbol{\Lambda} \end{smallmatrix} \right)$, $l \in [L-1]$, where each element of matrix $\boldsymbol{\Lambda}$ is drawn from Gaussian distribution $\mathcal{N}(0, 4/m)$. Similarly, let $\boldsymbol{\theta}_L = (\boldsymbol{w}^{\mathsf{T}}, -\boldsymbol{w}^{\mathsf{T}})$, where each element of vector $\boldsymbol{w}$ is drawn from $\mathcal{N}(0, 2/m)$.

**Arm Context Normalization.** To ensure arm contexts are of unit length (i.e., $\|\boldsymbol{x}_{i,t}\| = 1, \forall i \in [K], t \in [T]$) as in existing neural bandit works [86, 84, 11, 67, 50], we can apply the following transformation inspired by existing works [3, 86, 84] without loss of generality: with unprocessed context $\widetilde{\boldsymbol{x}}_{i,t}$, we formulate the corresponding normalized arm context $\boldsymbol{x}_{i,t} = [\frac{\widetilde{\boldsymbol{x}}_{i,t}}{2 \cdot \|\widetilde{\boldsymbol{x}}_{i,t}\|_2}, \frac{1}{2}, \frac{\widetilde{\boldsymbol{x}}_{i,t}}{2 \cdot \|\widetilde{\boldsymbol{x}}_{i,t}\|_2}, \frac{1}{2}]$. It can be verified that we have three properties: (i) $\|\boldsymbol{x}_{i,t}\|_2 = 1$; (ii) no two normalized arm contexts will be in opposite directions; and (iii) $f(\boldsymbol{x}_{i,t}; \boldsymbol{\theta}_0) = 0$ with the randomly initialized $\boldsymbol{\theta}_0$.

**Definitions of NTK Matrices.** First, we denote imaginary corruption-free models as $f(\cdot; \widetilde{\boldsymbol{\theta}}_{t-1}), t \in [T]$ which are trained on corruption-free records $\widetilde{\mathcal{P}}_t = \{\boldsymbol{x}_\tau, \widetilde{r}_\tau\}_{\tau \in [t]}, t \in [T]$, for the sake of theoretical analysis, and the learner does not need to own the imaginary model in practice. Let $\{\boldsymbol{x}_t\}_{t \in [T]} = \{\boldsymbol{x}_{i_t,t}\}_{t \in [T]}$ be arms chosen by the corrupted model $f(\cdot; \boldsymbol{\theta})$, and $\{\widetilde{\boldsymbol{x}}_t\}_{t \in [T]} = \{\boldsymbol{x}_{\widetilde{i}_t,t}\}_{t \in [T]}$ be those chosen by the corruption-free model $f(\cdot; \widetilde{\boldsymbol{\theta}})$ respectively. Then, define a union set $\breve{\mathcal{A}}_T := (\{\boldsymbol{x}_t\}_{t=1}^T \cup \{\boldsymbol{x}_t^*\}_{t=1}^T \cup \{\widetilde{\boldsymbol{x}}_t\}_{t=1}^T)$, based on: (i) the chosen arms $\{\boldsymbol{x}_t\}_{t=1}^T$, (ii) the optimal arms $\{\boldsymbol{x}_t^*\}_{t=1}^T$ according to (2), and (iii) arms $\{\widetilde{\boldsymbol{x}}_t\}_{t=1}^T$ chosen by the imaginary corruption-free models. Here, $\breve{\mathcal{A}}_T$ naturally contains *unique arms* from these three arm collections, with cardinality $|\breve{\mathcal{A}}_T| \le 3T$. Meanwhile, we simply merge these three arm collections to form $\mathcal{A}_T$ (with cardinality $|\mathcal{A}_T| = 3T$), which allows duplicate arms. Afterwards, we have the following *two formulations* of the NTK Gram matrix: (i) The NTK Gram matrix $\mathbf{H}$ with *possibly duplicate* arms based on the collection $\mathcal{A}_T$; (ii) the NTK matrix for *non-duplicate* arms $\breve{\mathbf{H}}$ built upon the set $\breve{\mathcal{A}}_T$.

**Definition 5.1** (NTK Gram Matrix with *Possibly Duplicate* Arms). Let $\mathcal{N}$ be the Gaussian distribution. With layer index $l \in [L]$ and subscripts $i, j \in \{1, \ldots, |\mathcal{A}_T|\}$ for enumerating across arms, comparable to [47, 86], define the following recursive process

$$
\begin{aligned}
\mathbf{H}_{i,j}^0 = \boldsymbol{\Psi}_{i,j}^0 = \langle \boldsymbol{x}_i, \boldsymbol{x}_j \rangle, \qquad\qquad
& \mathbf{N}_{i,j}^l = \begin{pmatrix} \boldsymbol{\Psi}_{i,i}^l & \boldsymbol{\Psi}_{i,j}^l \\ \boldsymbol{\Psi}_{j,i}^l & \boldsymbol{\Psi}_{j,j}^l \end{pmatrix}, \\
\boldsymbol{\Psi}_{i,j}^l = 2\mathbb{E}_{a,b \sim \mathcal{N}(\mathbf{0}, \mathbf{N}_{i,j}^{l-1})}[\sigma(a)\sigma(b)], \quad
& \mathbf{H}_{i,j}^l = 2\mathbf{H}_{i,j}^{l-1}\mathbb{E}_{a,b \sim \mathcal{N}(\mathbf{0}, \mathbf{N}_{i,j}^{l-1})}[\sigma'(a)\sigma'(b)] + \boldsymbol{\Psi}_{i,j}^l.
\end{aligned}
\tag{7}
$$

With $\mathcal{A}_T$ containing *possibly duplicate arms*, we denote the NTK Gram matrix $\mathbf{H} = (\mathbf{H}^L + \boldsymbol{\Psi}^L)/2 \in \mathbb{R}^{3T \times 3T}$, and expected reward vector $\boldsymbol{h} = [h(\boldsymbol{x})]_{\boldsymbol{x} \in \mathcal{A}_T} \in \mathbb{R}^{3T}$. Existing works with the *arm separateness assumption* (e.g., [86, 84, 9, 25, 81]) generally assume $\mathbf{H} \succ \mathbf{0}$, while we do not.

**Definition 5.2** (NTK Gram Matrix with *Non-duplicate* Arms). Follow the recursive process in (7). With set $\breve{\mathcal{A}}_T$ containing *non-duplicate arms*, we denote the corresponding NTK matrix $\breve{\mathbf{H}} = (\breve{\mathbf{H}}^L + \breve{\boldsymbol{\Psi}}^L)/2 \in \mathbb{R}^{|\breve{\mathcal{A}}_T| \times |\breve{\mathcal{A}}_T|}$, and expected reward vector $\breve{\boldsymbol{h}} = [h(\boldsymbol{x})]_{\boldsymbol{x} \in \breve{\mathcal{A}}_T} \in \mathbb{R}^{|\breve{\mathcal{A}}_T|}$, with $|\breve{\mathcal{A}}_T| \le 3T$.

**Remark 5.3** (No Arm Separateness Assumption). Existing neural bandit works generally impose separateness assumptions regarding the arm contexts: NTK-based approaches (e.g., [86, 84, 9, 51, 50]) commonly assume $\mathbf{H} \succ \mathbf{0}$ which requires no two arms are parallel among $\{\boldsymbol{x}_{i,t}\}_{i \in [K], t \in [T]}$; meanwhile, some other works (e.g., [11, 67]) assume the Euclidean separateness: $\|\boldsymbol{x}_{i,t} - \boldsymbol{x}_{i',t'}\|_2 > 0$ if $(i, t) \ne (i', t'), \forall i, i' \in [K], t, t' \in [T]$. To avoid the arm separateness assumption, since $\breve{\mathcal{A}}_T$ contains all the unique arms from $\mathcal{A}_T$, we alternatively build the confidence ellipsoid upon the NTK matrix $\breve{\mathbf{H}}$, and the ellipsoid will also hold for all the arms in $\mathcal{A}_T$ for regret analysis (Lemma C.1). This also leads to our tighter definition of NTK norm term $S$ (Theorem 5.6, Remark 5.8).

**Fact 5.4.** Let $\breve{\lambda}_0$ be the minimum eigenvalue of matrix $\breve{\mathbf{H}}$, and $\lambda_0$ be that of NTK matrix $\mathbf{H}$. We have (i) $\breve{\lambda}_0 = \lambda_{\min}(\breve{\mathbf{H}}) > 0$; and, (ii) $\breve{\lambda}_0 \ge \lambda_0 \ge 0$.

For (i) in Fact 5.4, since $\breve{\mathcal{A}}_T$ contains no parallel arms, matrix $\breve{\mathbf{H}}$ will be full-rank, leading to $\breve{\lambda}_0 > 0$. For (ii), if $\breve{\mathcal{A}}_T \ne \mathcal{A}_T$, then $\mathcal{A}_T$ contains duplicate arms and matrix $\mathbf{H}$ will be singular, s.t. $\breve{\lambda}_0 > \lambda_0 = 0$. Otherwise, if $\breve{\mathcal{A}}_T = \mathcal{A}_T$, it will naturally lead to $\breve{\mathbf{H}} = \mathbf{H}$ and $\breve{\lambda}_0 = \lambda_0$. Next, similar to existing neural bandit works (e.g., [86, 84]), we define the NTK Gram matrix effective dimension $\widetilde{d}$, which essentially measures the vanishing speed of NTK Gram matrix eigenvalues.

**Definition 5.5** (Effective Dimension of NTK Matrix [86, 84]). Given the NTK matrix $\mathbf{H}$ with possibly duplicate arms (Def. 5.1), its effective dimension is defined as $\widetilde{d} = \frac{\log \det(\mathbf{I} + \mathbf{H}/\lambda)}{\log(1 + TK/\lambda)}$.

### 5.1 Regret Analysis for R-NeuralUCB

We follow the pseudo-regret $R(T) = \sum_{t=1}^T \mathbb{E}[\widetilde{r}_t^* - \widetilde{r}_t]$ in (2), which is defined based on the expected corruption-free reward of chosen arms and optimal arms across $T$ rounds.

**Instance-dependent Gradient Deviation Term $\beta$.** Recall that for a candidate arm $\boldsymbol{x}_{i,t} \in \mathcal{X}_t$ in round $t \in [T]$, its arm weight w.r.t. previously chosen arm $\boldsymbol{x}_\tau, \tau \in [t-1]$ in (4) can be represented by $w_{i,t}^{(\tau)} = \min\{1, \alpha \cdot \mathsf{frac}_\tau(\boldsymbol{x}_{i,t}; \mathcal{X}_t, \bar{\boldsymbol{\Sigma}}_{t-1})\}$, with the scaling parameter $\alpha > 0$. Here, we define a minimum fraction value as $\beta = \min_{t \in [T], \tau \in [t-1]} \left[\min\{\mathsf{frac}_\tau(\boldsymbol{x}_t; \mathcal{X}_t, \bar{\boldsymbol{\Sigma}}_{t-1}), \mathsf{frac}_\tau(\widetilde{\boldsymbol{x}}_t; \mathcal{X}_t, \bar{\boldsymbol{\Sigma}}_{t-1})\}\right]$, which

is formulated to quantify the gradient deviation among arms. Here, the learner is not required to know $\beta$, and we can adjust the scaling parameter $\alpha$ in each round $t \in [T]$ to constrain the round-wise minimum weight value $\min\{w_{i,t}^{(\tau)}\}_{i \in [K], \tau \in [t-1]}$ (Subsection B.5), which leads to Theorem 5.6.

**Theorem 5.6.** *With finite horizon $T \in \mathbb{N}^+$, denote $S \geq \sqrt{2\check{h}^\top \check{H}^{-1} \check{h}}, \beta > 0$. Suppose $\lambda \geq S^{-2}, \eta \leq \mathcal{O}((TmL+m\lambda)^{-1}), J \geq \widetilde{\mathcal{O}}(TL/\lambda)$. Let $f(\cdot)$ be an L-layer FC network with width $m$, and adjust the scaling parameter $\alpha$, s.t. $\min\{w_{i,t}^{(\tau)}\}_{i \in [K], \tau \in [t-1]} = \kappa^2, \forall t \in [T]$, for a tunable constant $\kappa \in (0,1)$ from (4). With $\delta \in (0,1)$, let network width $m \geq \Omega(poly(T, L, \kappa^{-1}, \check{\lambda}_0^{-1}, \lambda^{-1}, S^{-1}) \log(\delta^{-1}))$. With probability at least $1 - \delta$, R-NeuralUCB achieves the regret bound of*

$$R(T) \leq \mathcal{O}\left(\nu\sqrt{\widetilde{d}\log(\frac{\lambda + TK}{\lambda}) - 2\log(\delta)} + S\sqrt{\lambda}\right)\widetilde{\mathcal{O}}\left(\sqrt{T\widetilde{d}/\kappa^2}\right) + \mathcal{O}\left(C\widetilde{d}\beta^{-1}\kappa^2\log(\frac{\lambda + TK}{\lambda})\right).$$

The proof of Theorem 5.6 is in Appendix C. The first term on the RHS refers to the corruption-independent regret upper bound, which comparably matches the bound $\widetilde{\mathcal{O}}(\widetilde{d}\sqrt{T} + S\sqrt{\widetilde{d}T})$ in existing corruption-free neural bandit works [86, 84]. Here, our corruption-dependent term is free of the NTK norm $S$, which measures the complexity of reward mapping $h(\cdot)$ (Appendix B.4). This is different from existing works (e.g., [15]) that include a parameter norm (similar to our NTK norm $S$) in their corruption-dependent terms, as the estimation error of confidence ellipsoids. In addition, inspired by [84], we can derive a $T$-independent upper bound for the $\beta^{-1}$ term, when the arm contexts are nearly spreading within some low-dimensional subspace of the NTK-induced RKHS (Appendix C.9), as it will lead to small effective dimension $\widetilde{d}$ and small eigenvalues of NTK matrix $\mathbf{H}$ [84]. Meanwhile, compared with the regret bound of our base algorithm NeuralUCB-WGD (Theorem E.1), Theorem 5.6 removes the assumption of known corruption $C$; and, reduces the order of effective dimension $\widetilde{d}$ as well as the dependency of NTK norm $S$ for corruption-dependent terms.

**Remark 5.7** (Unknown corruption level $C$). For Theorem 5.6, we do not assume $C$ is known to the learner in advance, as practitioners can have little prior knowledge regarding the unknown adversary. This makes our regret analysis more challenging, compared with the existing works (e.g., [16]) where $C$ is assumed known for setting hyper-parameters to achieve tight regret bounds.

**Remark 5.8** (Tighter definition for NTK norm $S$). For existing works (e.g., [86, 84]), the NTK Gram matrix is generally defined with all the $TK$ observed candidate arms, i.e., $\{\boldsymbol{x}_{i,t}\}_{i \in [K], t \in [T]}$, while our NTK matrices (Def. 5.1 and 5.2) only rely on arm collection $\mathcal{A}_T$ and set $\check{\mathcal{A}}_T$, with cardinality $|\check{\mathcal{A}}_T| \leq |\mathcal{A}_T| = 3T$. This results in our parameter norm $S$ that can be tighter compared to existing works (e.g., [84, 86]), because when constructing the confidence ellipsoid around the initialization $\boldsymbol{\Theta}_0$ in Lemma C.1, our ellipsoid is intuitively tighter, as it only needs to ensure Eq. C.1 holds for arms in $\check{\mathcal{A}}_T$ (with cardinality $|\check{\mathcal{A}}_T| \leq 3T$), rather than for all $TK$ candidate arms.

**Remark 5.9** (Reducing the order of $\widetilde{d}$ and removing the dependency of $S$ for corruption-dependent terms). For corruption-dependent terms involving $C$, we have $\widetilde{\mathcal{O}}(C\beta^{-1}\widetilde{d})$. Using NTK to align the information gain definition [16] with our effective dimension $\widetilde{d}$, our result improves latest kernelized bandit works from $\widetilde{\mathcal{O}}(\widetilde{d}^{3/2})$ to $\widetilde{\mathcal{O}}(\widetilde{d})$ for corruption-dependent terms, given the NTK-induced RKHS and an indefinite arm space (Corollary 7 in [16]). Meanwhile, our corruption-dependent term is free of the NTK norm $S$, while for some existing works with UCB-type exploration (e.g., [15]), they involve comparable parameter norms in their corruption-dependent regret terms, in order to quantify corruption impacts w.r.t. the reward mapping function complexity.

## 6 Experiments

We evaluate R-NeuralUCB and the base algorithm NeuralUCB-WGD (Appendix E) with experiments on three real data sets, under different adversarial corruption scenarios. Following definition in (2), we record the cumulative regret in terms of corruption-free rewards $R(T) = \sum_{t \in [T]} \left[\widetilde{r}_t^* - \widetilde{r}_t\right]$. Note that the learner will still only have access to the potentially corrupted rewards $r_t, t \in [T]$. Our baselines consist of linear algorithms: Lin-UCB [24], CW-OFUL [42]; and conventional neural algorithms: Neural-UCB [86], Neural-TS [84]. Complementary experiment details are in Appendix A.

**MovieLens and Amazon Data Sets.** From *"MovieLens 20M rating data set"* [41], we choose 5,000 movies and 10,000 users with most reviews to form the user-movie matrix, and the entries are user ratings. Then, we consider the arm (user-item pair) features as the concatenation of corresponding

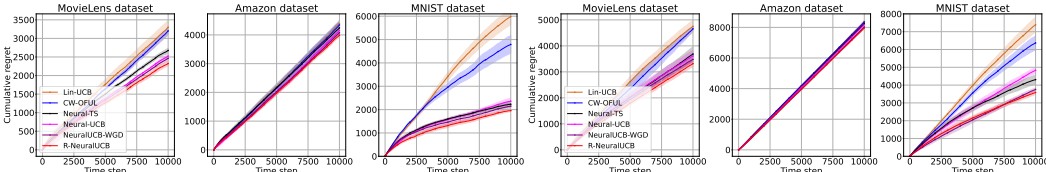

Figure 1: Regret results on real data sets. (Left three figures: For MovieLens and Amazon, corrupt the chosen arm reward with 20% probability. For MNIST, consider $C = 2000$ and randomly sample 2000 rounds for attack); (Right three figures: For MovieLens and Amazon: we corrupt reward with 50% probability; For MNIST: $C = 4000$ and randomly sample 4000 corrupted rounds).

user features and item features, which are obtained by singular value decomposition (SVD) and extracting item genome-scores respectively, with $K = 10$ and $d = 41$. The corruption-free arm rewards $\widetilde{r}_{i,t}$ are user ratings normalized into range $[0, 1]$. Here, we consider the "exaggerated reward corruption". If one pulled arm $\boldsymbol{x}_t$ is attacked and its corruption-free reward $\widetilde{r}_t \geq 0.5$, we exaggerate its reward to $r_t = 1$. Otherwise, if one pulled arm $\boldsymbol{x}_t$ is attacked and $\widetilde{r}_t < 0.5$, we set its reward $r_t = 0$. *Amazon Recommendation data set* [43] consists of user reviews and corresponding ratings. With each piece of review (user-item pair) as an arm, we vectorize the review as the arm features using the "Sentire" package [85, 58], with $K = 10$ and $d = 41$. Similarly, the corruption-free arm rewards $\widetilde{r}_{i,t}$ are normalized user ratings with the value range $[0, 1]$. Different from MovieLens data set, we here consider the "reverse exaggerated corruption": if the pulled arm $\boldsymbol{x}_t$ is attacked and its corruption-free reward $\widetilde{r}_t \geq 0.5$, we downplay its reward to $r_t = 0$; or if the pulled arm $\boldsymbol{x}_t$ is attacked and $\widetilde{r}_t < 0.5$, we alternatively set the corrupted reward $r_t = 1$.

**MNIST Data Set.** To perform online classification with bandit feedback experiment, we adopt the MNIST data set [56] which consists of 10 image classes. Similar to previous works (e.g., [86, 84]), given a sample $\boldsymbol{x} \in \mathbb{R}^{d'}$ in each round, we transform it into $K = 10$ arms, denoted by $\boldsymbol{x}_1 = (\boldsymbol{x}, \boldsymbol{0}, \ldots, \boldsymbol{0}), \boldsymbol{x}_2 = (\boldsymbol{0}, \boldsymbol{x}, \ldots, \boldsymbol{0}), \ldots, \boldsymbol{x}_{10} = (\boldsymbol{0}, \boldsymbol{0}, \ldots, \boldsymbol{x}) \in \mathbb{R}^{10 \times d'}$, s.t. $d = 10 \times d'$. The arm index that the learner chooses will be its predicted class, and the reward is $1$ if the sample $\boldsymbol{x}$ belongs to this class; otherwise, the reward will be $0$. Here, we consider the symmetric "label-flipping" attack [39]. For example, when a sample from digit class 2 is attacked, its corrupted label will be switched to digit class $9 - 2 = 7$, and the corrupted arm rewards will also change accordingly.

**Experiment Results.** The experiment results are shown in Fig. 1, and we also include a parameter study in Appendix A.2. Due to the representation power of neural networks, neural algorithms generally perform better than linear ones. In particular, for the MNIST data set, since the reward mapping can be relatively more complex, neural algorithms manage to achieve more significant improvements over the linear algorithms. Here, compared with conventional neural methods, our proposed NeuralUCB-WGD and R-NeuralUCB are more robust against adversarial reward corruptions. In particular, we see that R-NeuralUCB outperforms NeuralUCB-WGD on these three data sets, which helps support our claim that it is beneficial to involve the uncertainty information in terms of both training samples and candidate arms. When we increase the corruption intensity (three figures on the right), the overall results tend to be consistent with previous findings. Notice that the performance gap among algorithms on the Amazon data set tends to be smaller, as this setting becomes significantly more difficult (i.e., with up to $\sim 8000$ regret) when we increase the corruption probability to $50\%$. Meanwhile, for MNIST, when we increase $C$ to 4000, the performance gap between our proposed algorithms and the conventional neural methods tends to increase, as the task becomes increasingly more complex. We also see that R-NeuralUCB still outperforms NeuralUCB-WGD given the increased corruption intensity, showing the benefit of involving the candidate arm information and customizing arm-specific model parameters.

## 7  Conclusion and Future Direction

In this paper, we propose a novel neural bandit algorithm named R-NeuralUCB to address potential adversarial corruption issues on arm rewards. To enhance model robustness against reward corruptions, R-NeuralUCB applies a refined, context-aware Gradient Descent procedure that incorporates arm uncertainty information. To demonstrate its effectiveness, we present a regret analysis of R-NeuralUCB to quantify the impacts of adversarial corruption. Furthermore, to ensure that R-NeuralUCB can handle arm contexts deliberately chosen by an adversary (e.g., duplicate arms across different rounds), our analysis avoids the commonly adopted arm separateness assumption in neural bandit literature, which can be of independent interest. Empirical evaluations on real datasets with varied specifications show the effectiveness of our proposed solution over baseline methods. A challenging future direction is to derive the theoretical lower bound for neural bandits with corruption, and we provide complementary discussions in Appendix B.7.

## Acknowledgments and Disclosure of Funding

This work is supported by National Science Foundation under Award No. IIS-2117902, and Agriculture and Food Research Initiative (AFRI) grant no. 2020-67021-32799/project accession no.1024178 from the USDA National Institute of Food and Agriculture. The work is also supported in part by the National Science Foundation through awards IIS 21-31335, OAC 21-30835, DBI 20-21898, as well as a C3.ai research award. The views and conclusions are those of the authors and should not be interpreted as representing the official policies of the funding agencies or the government.

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

# A Experiment Settings and Additional Experiments

## A.1 Experiment settings

For all UCB-based baselines, we choose the exploration parameter through grid search over the range $\{0.01, 0.1, 1\}$. We set $L = 2$ for all deep learning models, including our proposed NeuralUCB-WGD and R-NeuralUCB, and set the network width to $m = 200$. The learning rate for all neural algorithms is chosen by grid search from the range $\{0.0001, 0.001, 0.01\}$. For all methods, we select the regularization parameter $\lambda$ from the range $\{0.0001, 0.001, 0.01\}$. The scaling parameter $\alpha$ for NeuralUCB-WGD and R-NeuralUCB is chosen from $\{0.2, 0.5, 1\}$. All experiments are conducted on a server with an Intel Xeon CPU and NVIDIA V100 GPUs. Additionally, we provide further details on our baseline methods, which include two linear algorithms and two conventional neural algorithms:

- Lin-UCB [24, 59] uses linear regression as the reward estimation model and employs a UCB-based strategy for exploration.

- CW-OFUL [42] applies weighted linear ridge regression in instead of the standard one from Lin-UCB, with weights assigned to selected samples in proportion to reward estimation confidence.

- Neural-UCB [86] employs a single neural network to estimate arm rewards and calculates the UCB based on network gradients for exploration.

- Neural-TS [84] utilizes a fully connected network for arm reward estimation, along with the Thompson Sampling strategy [2] for exploration.

**Additional Data Processing Details for Recommendation Data Sets.** Here, we provide additional details about our data processing procedure. For the first data set, *MovieLens 20M rating data set* (`https://grouplens.org/datasets/movielens/20m/`), we initially select 5,000 movies and 10,000 users with the **most reviews** to form a user-movie matrix, where the entries represent user ratings. The user features $\boldsymbol{x}_u \in \mathbb{R}^{d'}$ are derived via singular value decomposition (SVD) with a dimensionality of $d' = 20$. Using the genome scores provided for each movie, we select the 20 tags with the highest variance and used their corresponding scores as movie features $\boldsymbol{v}_i \in \mathbb{R}^{d'}$. At each time step $t$, given a user $u_t$, we encode user information into the arm contexts following the Generalized Matrix Factorization (GMF) approach [44, 88] by concatenating the features $\boldsymbol{x}_{i,t} = [\boldsymbol{x}_{u_t}; \boldsymbol{v}_i] \in \mathbb{R}^{2d'}$, where $c \in \mathcal{C}_t$ and $i \in [K]$, with $K = 10$. Finally, we concatenate a constant 0.01 to each $\boldsymbol{x}_{i,t}$ and normalize the entire vector to obtain $\boldsymbol{x}_{i,t} \in \mathbb{R}^d$, where $d = 41$. The corruption-free arm rewards $\widetilde{r}_{i,t}$ are user ratings normalized to the range $[0, 1]$. We consider the scenario of "exaggerated reward corruption": If a pulled arm $\boldsymbol{x}_t$ is attacked and its corruption-free reward $\widetilde{r}_t \geq 0.5$, we exaggerate its reward to $r_t = 1$. Conversely, if a pulled arm $\boldsymbol{x}_t$ is attacked and its corruption-free reward $\widetilde{r}_t < 0.5$, we downplay its reward to $r_t = 0$.

For the Amazon Recommendation data set (`https://jmcauley.ucsd.edu/data/amazon/index_2014.html`), each user-item pair is associated with a review and the corresponding user rating. We transform the review text into vector representations to derive the arm contexts, following the text processing procedure in the "Sentires" package [85, 58]. We then set $d = 41$ and apply $L_2$ normalization, with an arm pool size of $K = 10$. Similarly, the corruption-free arm rewards $\widetilde{r}_{i,t}$ are normalized user ratings in the range $[0, 1]$. Unlike the MovieLens data set, we apply a "reverse exaggerated corruption" approach here: If the pulled arm $\boldsymbol{x}_t$ is attacked and its corruption-free reward $\widetilde{r}_t \geq 0.5$, we downplay its reward to $r_t = 0$. Conversely, if the pulled arm $\boldsymbol{x}_t$ is attacked and its corruption-free reward $\widetilde{r}_t < 0.5$, we set the corrupted reward to $r_t = 1$.

## A.2 Additional experiments: parameter study

We also include additional experiments with different regularization parameter values $\lambda$ and exploration parameter values $\nu$. On the MNIST data set (corruption level $C = 2000$), we conduct experiments for NeuralUCB-WGD and R-NeuralUCB. We present the parameter study results in Tables 2 and 3. For both of our proposed algorithms, setting $\nu \in (0.1, 0.5]$ generally yields the best performance. However, with an overly small exploration coefficient (e.g., $\nu = 0.05$), optimal empirical performance may not be achievable. Meanwhile, setting $\lambda$ to smaller values, such as 0.001 or 0.0001, tends to result in the best performance. Increasingly large regularization parameter values

| Algorithm \ $\lambda$ value | $\lambda = 0.1$ | $\lambda = 0.01$ | $\lambda = 0.001$ | $\lambda = 0.0001$ |
|---|---|---|---|---|
| NeuralUCB-WGD | 3244 $\pm$95 | 2782 $\pm$90 | 2156 $\pm$105 | 2103 $\pm$119 |
| R-NeuralUCB | 2933 $\pm$101 | 2501 $\pm$93 | 1989 $\pm$67 | 2127 $\pm$92 |

Table 2: Regret results for different exploration regularization parameter values $\lambda$ (with std.)

| Algorithm \ $\nu$ value | $\nu = 1$ | $\nu = 0.5$ | $\nu = 0.1$ | $\nu = 0.05$ |
|---|---|---|---|---|
| NeuralUCB-WGD | 2510 $\pm$102 | 2154 $\pm$121 | 2103 $\pm$119 | 2278 $\pm$85 |
| R-NeuralUCB | 2793 $\pm$104 | 2163 $\pm$88 | 1989 $\pm$67 | 2197 $\pm$65 |

Table 3: Regret results for different exploration parameter values $\nu$ (with std.)

can cause the trained model parameters $\theta$ to remain close to their random initialization $\boldsymbol{\theta}_0$, which can overly constrain the neural network's capacity to fit the underlying reward mapping function. Thus, practitioners can adjust the $\lambda$ value based on specific application needs, as is common in other neural bandit studies (e.g., [86]). In practice, starting with small values like $10^{-4}$ and performing a grid search to identify the optimal $\lambda$ is a reasonable approach for R-NeuralUCB and NeuralUCB-WGD.

# B Complementary Discussions on the Content of the Main Body

In this section, we provide additional discussion to complement the main body content.

## B.1 Boarder impacts

Since our objective is to deal with the potential adversarial attacks in machine learning applications, this work can contribute to the goal of achieving trustworthy machine learning for general practitioners. Therefore, we do not perceive significant negative societal impacts that can be generated by this work.

## B.2 Limitations

One limitation of this work is the absence of a theoretical lower bound for neural bandits with adversarial corruptions. We would like to mention that this problem is significantly challenging and non-trivial. Given that we deal with an arbitrary reward function $h(\cdot)$, which is considerably different from linear [42] and kernelized bandits [16], deriving the lower bound itself can lead to substantial contributions, potentially leading to a separate line of research works (e.g., [70] under kernelized bandit settings). Therefore, we consider the derivation of such a lower bound for neural bandits with adversarial corruptions as an interesting and challenging future direction of this work. Additional discussions on the lower bound can be found in Subsec. B.7.

## B.3 Theoretical contributions and comparisons with vanilla Neural-UCB

Recall that we propose deriving the regret bound using NTK-based regression techniques. Unlike linear bandit approaches (e.g., [42]) and kernel bandit methods (e.g., [16]), our regression is conducted on the network gradients $g(\cdot; \boldsymbol{\theta}) := \text{vec}(\nabla_{\boldsymbol{\theta}} f(\cdot; \boldsymbol{\theta}))$, which serve as the mapping for gradient-based NTK. In this framework, even for the same arm $\boldsymbol{x}$, the NTK-embedded arm contexts can differ due to the corrupted parameters $g(\boldsymbol{x}; \boldsymbol{\theta})$ and the corruption-free parameters $g(\boldsymbol{x}; \widetilde{\boldsymbol{\theta}})$. To address this challenge, we define two sets of regression parameters corresponding to the corrupted model and the corruption-free model respectively. Then, using the corruption-free model $f(\cdot; \widetilde{\boldsymbol{\theta}})$, we derive the confidence ellipsoid around its parameters $\widetilde{\boldsymbol{\theta}}$. This serves as a proxy to quantify the parameter shift of the trained corrupted model parameters $\boldsymbol{\theta}$, enabling us to establish the regret upper bound.

For the theoretical analysis of R-NeuralUCB presented in Theorem 5.6, we considerably modify the regret analysis workflow due to the following reasons: (i) To achieve improved performance, R-NeuralUCB differs from conventional neural bandit approaches by tuning separate sets of network parameters for each candidate arm after perceiving arm context information; (ii) To achieve a tighter regret bound and eliminate the assumption of a known corruption level $C$, unlike in Theorem E.1,

we cannot quantify the impact of $C$ using the confidence ellipsoid. To address this, let $x_t^*$ and $x_t$ represent the optimal arm and the chosen arm by the corrupted model, respectively. We decompose the single-round pseudo-regret $R_t = \min\{h(x_t^*) - h(x_t), 1\}$ into three components: (i) The prediction error of the corruption-free model; (ii) The reward estimation discrepancy between the corruption-free model and the corrupted model on the same arm; (iii) The arm selection discrepancy of the corrupted model induced by adversarial corruptions. Next, we apply carefully designed arm weights in (4) to guide the gradient descent process and mitigate the impact of adversarial corruptions. As a result, R-NeuralUCB achieves non-trivial theoretical improvements: (i) removing the assumption of a known corruption level $C$ in regret analysis; (ii) eliminating the dependency on the NTK norm term $S$ for corruption-dependent terms in the regret bound; (iii) reducing the order of corruption-dependent terms in the regret bound to the effective dimension $\widetilde{d}$, from $\mathcal{O}(\widetilde{d}^{3/2})$ to $\mathcal{O}(\widetilde{d})$, compared with our base algorithm NeuralUCB-WGD and existing kernelized bandit algorithms (e.g., [16]).

In addition, as mentioned in Remark 5.3, existing neural bandit approaches typically impose separateness assumptions on observed arm contexts, whereas we do not. For example, Neural-UCB [86] assumes $\mathbf{H} \succ \mathbf{0}$, which requires that no two arms are parallel among $\{\boldsymbol{x}_{i,t}\}_{i\in[K],t\in[T]}$. In contrast, by formulating our NTK Gram matrices (Definitions 5.1 and 5.2), we complete our proof without the separateness assumption, reinforcing the theoretical robustness of our approach against possible arm contexts selected by an adversary (e.g., duplicate arm contexts across time steps). As in Remark 5.8, existing methods, including Neural-UCB [86, 84], generally define their NTK Gram matrices over all $TK$ observed arms, i.e., $\{\boldsymbol{x}_{i,t}\}_{i\in[K],t\in[T]}$. However, our NTK matrices (Definitions 5.1 and 5.2) are based on $\mathcal{A}_T$ and $\breve{\mathcal{A}}_T$, where $|\breve{\mathcal{A}}_T| \leq |\mathcal{A}_T| = 3T$. This formulation can result in a tighter NTK norm $S$ compared with existing methods.

**Upper bound for vanilla Neural-UCB.** Meanwhile, to provide insights into the regret bound of Neural-UCB, one possible approach is to follow a similar analysis to the regret bound of NeuralUCB-WGD. The key idea here is to quantify the impact of adversarial corruptions on the confidence ellipsoid around the trained parameters. Referring to the derivations in Lemma F.1, and denoting the corruption-free confidence radius in round $t$ as $\tilde{\gamma}_{t-1}$, we obtain the corrupted confidence ellipsoid for Neural-UCB as $\mathcal{C}_{t-1} = \{\theta : \|\theta - \theta_{t-1}\|_{\Gamma_{t-1}} \leq \gamma_{t-1}/\sqrt{m}\}$, where $\gamma_{t-1} = \tilde{\gamma}_{t-1} + \mathcal{O}(CL\lambda^{-1/2})$. This result is derived by setting $w_\tau = 1$ for $\tau \in [t-1]$ and applying the fact that $\sum_{\tau\in[t]} c_\tau \leq C$, along with Lemma G.2 and the initialization of the gradient covariance matrix $\Gamma$. Following the proof flow of Lemma 5.3 in [86], we obtain a regret upper bound of $\tilde{\mathcal{O}}(\tilde{d}\sqrt{T} + \sqrt{S\tilde{d}T} + CL\sqrt{\tilde{d}T/\lambda})$, which introduces an additional $\tilde{\mathcal{O}}(\sqrt{T})$ to the corruption-dependent term.

**Over-parameterization.** For most neural bandit works with experiments (e.g., [84, 86, 25, 9, 11]), a gap exists between experiments and theoretical analysis. On one hand, as the number of layers $L$ and hidden dimension $m$ increase, neural networks become progressively harder to train, more time-consuming in inference, and more resource-intensive. To make neural bandits feasible for practical applications, these works generally use a neural network of ordinary size for experiments. It has been shown that even with ordinary-sized neural networks, neural bandit algorithms achieve notable performance gains over linear and kernel-based methods [86, 84, 9, 11, 67]. On the other hand, from a theoretical standpoint, neural networks need to be over-parameterized, with $m \geq \mathcal{O}(\text{poly}(T))$, to approximate any arbitrary reward mapping function $h(\cdot)$. Additionally, with over-parameterization, the difference between NTK-based regression models and neural networks becomes sufficiently small for regret analysis, which is essential in neural bandit research. Therefore, we use a two-layer fully connected network for experiments while performing theoretical analysis under over-parameterized settings, as in most existing neural bandit works (e.g., [86, 84, 9, 11, 67]).

## B.4 The definition and order of NTK norm parameter $S$

Recall that we have the NTK norm $S$ defined as the upper bound of the weighted norm $S \geq \|\breve{\mathbf{h}}\|_{\breve{\mathbf{H}}^{-1}}$, where $\mathbf{h}$ refers to the vector of expected rewards and $\breve{\mathbf{H}}$ refers to the NTK Gram matrix (Definition 5.2). We can follow existing neural bandit works [86, 50, 51, 84, 48] by considering that the reward mapping function $h(\cdot)$ in (1) belongs to the Reproducing Kernel Hilbert Space (RKHS) $\mathcal{H}$ induced by NTK. In this case, we can upper bound $S$ with the RKHS norm, such that $\|h\|_{\mathcal{H}} \geq S$, and the RKHS norm $\|h\|_{\mathcal{H}}$ will not grow along with the finite horizon $T$ (Remark 4.8 in [86]).

Meanwhile, we also would like to mention that this is a common formulation, and nearly all the neural bandit works (e.g., [86, 50, 51, 84, 9, 48]) will include a comparable NTK norm term in the regret bound. This is because the regret analysis of neural bandits is generally depending on the NTK regression approach. In this case, when constructing the confidence ellipsoid within the NTK-induced RKHS, we will need to involve the RKHS norm as the cost. Analogously, for the kernelized contextual bandits works (e.g., [72, 28, 16]), they inevitably involve the RKHS norm into the regret bound. For linear bandit works (e.g., [42]), they will also need to include an assumed upper bound of the true parameter $\boldsymbol{\theta}^*$ norm in the Euclidean space, such that $\|\boldsymbol{\theta}^*\|_2$ is bounded by a constant. Meanwhile, since the NTK norm term $S$ stays invariant across candidate arms $x_{i,t} \in \mathcal{X}_t$, we can treat $S$ as a constant in practice (e.g., setting $S = 1$), and control the exploration intensity by tuning the exploration parameter $\nu$.

## B.5 Details regarding the scaling of arm weights $w$

Recall that when defining the sample weights $w_{i,t}^{(\tau)}$ in (4), we use the minimum gradient norm in the numerator to scale weights across the current candidate arms $\mathcal{X}_t$, while introducing the scaling parameter $\alpha > 0$ to provide additional control from the practitioner's perspective. Under the stochastic contextual bandit settings, the learner receives the candidate arm pool $\mathcal{X}_t$ in each round $t$ from the environment, having little control over the minimum gradient norm, as $\mathcal{X}_t$ is only revealed at round $t$. To address this, we introduce a tunable parameter $\alpha$ to control the minimum value of $w_{i,t}^{(\tau)}$, aiming for a more stable learning process. Additionally, when deriving the regret bound for R-NeuralUCB (Theorem 5.6), we scale the $\alpha$ values to ensure that the minimum weight value is $\kappa^2$. Without this scaling, such as by setting $\alpha = 1$, an extra corruption-independent term $\mathcal{O}(\sqrt{\beta^{-1} T \widetilde{d} \log(1 + TK/\lambda)})$ would be added to the current regret bound, making the overall bound less tight. Therefore, the scaling parameter $\alpha$ is essential for R-NeuralUCB.

Furthermore, the denominator of (4) consists of the product of two gradient norms: (i) the norm of the previously chosen arm $g_\tau$, and (ii) the norm of the candidate arm $\|g(\boldsymbol{x}_{i,t};\boldsymbol{\theta})/\sqrt{m}\|_{\boldsymbol{\Sigma}^{-1}}$. Here, we use the squared norm in the numerator to balance with the norm product in the denominator. This design is also critical for deriving the regret bound in Theorem 5.6. Without using the squared norm, our current derivation would yield a corruption-dependent term of $\widetilde{\mathcal{O}}(\widetilde{d}\sqrt{T}\beta^{-1}C)$, rather than the current $\widetilde{\mathcal{O}}(\widetilde{d}\beta^{-1}C)$.

To be specific, for scaling the arm weight based on $\kappa$, we first recall that in round $t \in [T]$, we have arm weights $w_{i,t}^{(\tau)}, \tau \in [t-1], i \in [K]$. We can also denote $w_{i,t}^{(\tau)} = \min\left\{1, \alpha \cdot \mathsf{frac}_\tau(x_{i,t}; \mathcal{X}_t, \bar{\Sigma}_{t-1})\right\}$. Here, instead of deeming $\alpha$ as a fixed value across horizon $T$, we can consider $\alpha$ to be varying across different rounds, denoted by $\alpha_t, t \in [T]$. With a shorthand for minimum fraction value $\mathsf{frac}_t^{\min} = \min_{i \in [K], \tau \in [t-1]} \left[\mathsf{frac}_\tau(x_{i,t}; \mathcal{X}_t, \bar{\Sigma}_{t-1})\right]$, we can set each $\alpha_t = \kappa^2/\mathsf{frac}_t^{\min}, \kappa \in (0,1)$. As a result, we can consequently have $\min\{w_{i,t}^{(\tau)}\}_{i \in [K], \tau \in [t-1]} = \kappa^2, \forall t \in [T]$.

## B.6 Warm-start training for candidate arms

Recall that in each round $t \in \{2, \ldots, T\}$, we need to train different sets of arm-specific parameters $\boldsymbol{\theta}_{i,t-1}, i \in [K]$ according to Algorithm 1, for each of the candidate arms $\boldsymbol{x}_{i,t} \in \mathcal{X}_t, i \in [K]$. As we have mentioned in the main body, we can adopt the idea of warm-start GD [13] in practice. Here, instead of training each set of parameters $\boldsymbol{\theta}_{i,t-1}$ from the randomly initialized $\boldsymbol{\theta}_0$, we tune arm-specific parameters for each $\boldsymbol{x}_{i,t} \in \mathcal{X}_t$ with a small number of samples from current received records $\mathcal{P}_{t-1}$. With the formulated arm weights $w_{i,t}^{(\tau)}$ in (4), we first recall the arm-specific loss function associated with arm $\boldsymbol{x}_{i,t}$ as

$$\mathcal{L}_{i,t}(\mathcal{P}_{t-1}; \boldsymbol{\theta}) = \sum_{(\boldsymbol{x}_\tau, r_\tau) \in \mathcal{P}_{t-1}} \frac{w_{i,t}^{(\tau)}}{2} \cdot \left|f(\boldsymbol{x}_\tau; \boldsymbol{\theta}) - r_\tau\right|^2 + \frac{m\lambda}{2} \cdot \|\boldsymbol{\theta} - \boldsymbol{\theta}_0\|_2^2.$$

The idea is that instead of starting from $\boldsymbol{\theta}_0$, we can initiate the GD process from the existing network parameters $\boldsymbol{\theta}_{t-2}$ from the previous round $t-1$, where $\boldsymbol{\theta}_{t-2} = \boldsymbol{\theta}_{i_{t-1}, t-2}$ represents the parameters associated with the chosen arm $\boldsymbol{x}_{t-1} = \boldsymbol{x}_{i_{t-1}, t-1}$ in round $t-1$. The pseudo-code for this arm-specific warm-start GD process is provided in Algorithm 2.

---
**Algorithm 2** Warm-start training for R-NeuralUCB
---
1: **Input:** Candidate arm $\boldsymbol{x}_{i,t} \in \mathcal{X}_t$. Training steps $\bar{J}$. Learning rates $\eta$. Batch size $B$. Regularization parameter $\lambda$. Network parameters $\boldsymbol{\theta}_{t-2}$ from round $t-1$. Received records $\mathcal{P}_{t-1}$.
2: **Output:** Trained arm-specific network parameters $\boldsymbol{\theta}_{i,t-1}$ for arm $\boldsymbol{x}_{i,t} \in \mathcal{X}_t$.
3: Sample a batch of training samples from $\mathcal{P}_{t-1}$, denoted by $\widehat{\mathcal{P}}_{t-1} \subseteq \mathcal{P}_{t-1}$, where $|\widehat{\mathcal{P}}_{t-1}| = B$. Following (4), calculate arm weights $w_{i,t}^{(\tau)}$ for samples in $\widehat{\mathcal{P}}_{t-1}$.
4: $\boldsymbol{\theta}_{i,t-1}^{(0)} \leftarrow \boldsymbol{\theta}_{t-2}$.
5: **for** each training step $j \in \bar{J}$ **do**
6: $\quad \boldsymbol{\theta}_{i,t-1}^{(j)} = \boldsymbol{\theta}_{i,t-1}^{(j-1)} - \eta \nabla_{\boldsymbol{\theta}} \mathcal{L}_{i,t}(\widehat{\mathcal{P}}_{t-1}; \boldsymbol{\theta}_{i,t-1}^{(j-1)})$
7: **end for**
8: $\boldsymbol{\theta}_{i,t-1} \leftarrow \boldsymbol{\theta}_{i,t-1}^{(\bar{J})}$.
9: Return arm-specific network parameters $\boldsymbol{\theta}_{i,t-1}$.
---

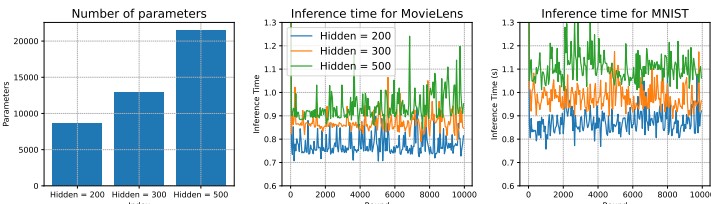

Figure 2: Number of parameters with hidden dimensions $m$. Inference time with warm-start.

As a result, in our experiments, to balance computational costs and model performance, we implement the following strategies: (1) Inspired by meta-learning approaches [32], we apply warm-start gradient descent (GD) by adapting previously trained network parameters $\theta_{t-2}$ for each candidate arm, using a small number of training samples rather than starting from $\theta_0$ with a large sample size; (2) Based on our formulation of the warm-start GD process, we sample a fixed number of mini-batch training samples (i.e., received arm-reward pairs) for each candidate arm to compute arm weights and perform GD. Using a fixed number of training samples helps keep round-wise inference time relatively stable, avoiding a drastic increase with $T$. Figure 2 illustrates the parameter count and inference time across different hidden dimensions. As shown, inference time remains relatively stable due to the fixed number of adaptation samples used for warm-start GD.

### B.7 Additional discussions on the lower bound

Under linear bandit settings, there is a model-agnostic lower bound of corruption-dependent term $\Omega(Cd)$ with probability at least $1/2$ [17], which will also hold for neural bandit works as our $h(\cdot)$ can be an arbitrary function. Meanwhile, the lower bounds for kernelized bandits tend to vary depending on kernel characteristics, e.g., $\Omega(C(\log(T))^{d/2})$ for the SE kernel and $\Omega(C^{\frac{v}{d+v}} T^{\frac{v}{d+v}})$ for the $v$-Matérn kernel [70, 16]. In this case, the order of term $C$ and whether the lower bound depends on non-logarithmic $T$, will both depend on the kernel properties. Therefore, given close connections between NTK-based regression and over-parameterized networks, we hypothesize that such a lower bound for neural bandits with corruption can depend on NTK properties. However, it will require significant efforts and a well-established existing knowledge base (e.g., number of functions $M$ needed for the functional separateness condition [70, 19] regarding specified kernels) to obtain such a lower bound for non-linear cases, especially considering few restrictions are imposed for reward mapping $h(\cdot)$ for neural bandits. Since there are no existing works from neural bandits or NTK perspectives, it can lead to a different line of research work by proving these results. Therefore, we consider providing such a corruption-dependent regret lower bound as a challenging future direction.

Meanwhile, when $C = 0$, we obtain a corruption-free regret of $\tilde{\mathcal{O}}(\tilde{d}\sqrt{T} + S\sqrt{\tilde{d}T})$. By setting $C = \Omega(R_T/d)$, the regret bound becomes $\tilde{\mathcal{O}}\left((\tilde{d}^2\sqrt{T} + S\tilde{d}^{3/2}\sqrt{T}) \cdot C\beta^{-1}d^{-1}\right)$. We note that, following the proof flow of Theorem 4.12 in [42] and the learning problem defined in Assumption

2.1 of [42], our effective dimension term $\tilde{d}^2$ may depend on the horizon $T$ and can grow with $T$ [26]. Consequently, although the regret bound contains only $\sqrt{T}$ terms, the overall order of the regret bound could reach or exceed $\mathcal{O}(T)$ due to the effective dimension $\tilde{d}^2$, as discussed in [26]. This behavior differs from that of linear bandits, where regret bounds generally depend on the horizon $T$ and other $T$-independent terms, such as context dimension $d$ and a fixed $T$-independent linear parameter norm $\|\theta^*\|_2$. Therefore, our Theorem 5.6 will not contradict Theorem 4.12 in [42].

# C Regret Analysis for R-NeuralUCB

To begin with, recall that we aim to minimize the pseudo-regret for $T$ rounds, denoted by

$$R(T) = \sum_{t=1}^{T} R_t = \sum_{t=1}^{T} \left[ h(\boldsymbol{x}_t^*) - h(\boldsymbol{x}_t) \right]$$

where the second equality is due to the definition of reward mapping $h$ in (1). We denote $f(\cdot)$ as the bandit model we currently possess, which is trained with corrupted records $\mathcal{P}_{t-1}$ up to round $t$. Similarly, we can also suppose a corresponding imaginary corruption-free bandit model, which is trained with corruption-free records $\widetilde{\mathcal{P}}_{t-1}$. Similarly, the model parameters of our possessed $f(\cdot)$ are denoted as $\boldsymbol{\theta}$, while the parameters of the imaginary corruption-free model will be denoted as $\widetilde{\boldsymbol{\theta}}$.

**Subsections outline and proof sketch.** The content in this section is organized into the following sub-components: In Subsection C.1, we first present theoretical properties related to our definitions of the NTK Gram matrix (Definitions 5.1 and 5.2); In Subsection C.2, we decompose the single-round objective $R_t, t \in [T]$ into its sub-components: (i) the first component represents the prediction error of the corruption-free model; (ii) the second component measures the potential arm selection discrepancy of the corrupted model caused by adversarial corruptions; and (iii) the third component captures the estimation discrepancy between the corruption-free model and the corrupted model. Next, in Subsection C.3, we bound the cumulative pseudo-regret $R(T)$ (proof of Theorem 5.6). Using the auxiliary sequence introduced in Subsection C.4, we bound the components of the single-round regret in Subsections C.5 through C.8. Finally, we discuss bounding the minimum fraction term $\beta$ in Subsection C.9, particularly when the observed arm contexts lie nearly within a low-dimensional subspace of the RKHS induced by NTK.

## C.1 Theoretical Results with NTK Gram Matrices

We will first introduce some results, in order to link the NTK matrices (Def. 5.1 and Def. 5.2) with the reward mapping function $h(\cdot)$ and the gradient covariance matrix $\boldsymbol{\Sigma}$ (Algorithm 1).

**Lemma C.1.** *With probability at least $1 - \delta$, if network width $m$ satisfies the condition in Theorem 5.6, for any $\boldsymbol{x} \in \mathcal{A}_T$, there exists a set of parameters $\boldsymbol{\theta}^*$ such that*

$$h(\boldsymbol{x}) = \langle g(\boldsymbol{x}; \boldsymbol{\theta}_0), \boldsymbol{\theta}^* - \boldsymbol{\theta}_0 \rangle \tag{C.1}$$

*where parameters $\boldsymbol{\theta}^*$ satisfy $\|\boldsymbol{\theta}^* - \boldsymbol{\theta}_0\| \leq S/\sqrt{m}$, along with the NTK norm $S \geq \sqrt{2\breve{\boldsymbol{h}}^\mathsf{T} \breve{\mathbf{H}}^{-1} \breve{\boldsymbol{h}}}$.*

**Proof.** The proof of this lemma is inspired by that of Lemma 5.1 in [86]. However, we build our proof upon the non-duplicate arms $\breve{\mathcal{A}}_T$ and the corresponding NTK Gram matrix $\breve{\mathbf{H}}$ (Def. 5.2), instead of imposing the full-rank assumption on the conventional NTK matrix $\mathbf{H}$ (Def. 5.1). Here, we recall that the matrix $\breve{\mathbf{H}}$ is naturally positive definite ($\breve{\lambda}_0 = \lambda_{\min}(\breve{\mathbf{H}}) > 0$), as it is the NTK Gram matrix built upon a set of distinct arms that are not parallel (Fact 5.4).

Then, consider the gradient matrix with no-duplicate arms $\breve{\mathbf{G}} = [g(\boldsymbol{x}; \boldsymbol{\theta}_0)]_{\boldsymbol{x} \in \breve{\mathcal{A}}_T} / \sqrt{m} \in \mathbb{R}^{p \times |\breve{\mathcal{A}}_T|}$, where $p$ represents the total number of parameters in the neural network. As a result, by applying conclusion from Lemma C.3 and due to the fact that $|\breve{\mathcal{A}}_T| \leq 3T$, with the network width $m \geq \Omega(L^6 \log(TL/\delta)/\epsilon)$, $\forall \epsilon > 0$ and the probability at least $1 - \delta$, we will have

$$\|\breve{\mathbf{G}}^\mathsf{T} \breve{\mathbf{G}} - \breve{\mathbf{H}}\|_F \leq |\breve{\mathcal{A}}_T| \cdot \epsilon.$$

By setting $\epsilon = \frac{\breve{\lambda}_0}{2|\breve{\mathcal{A}}_T|}$, we will have

$$\breve{\mathbf{G}}^\mathsf{T} \breve{\mathbf{G}} \succeq \breve{\mathbf{H}} - \|\breve{\mathbf{G}}^\mathsf{T} \breve{\mathbf{G}} - \breve{\mathbf{H}}\|_F \mathbf{I} \succeq \breve{\mathbf{H}} - \breve{\lambda}_0/2\mathbf{I} \succeq \breve{\mathbf{H}}/2 \succ \mathbf{0}.$$

where the last two inequalities are due to Fact 5.4 that $\breve{\mathbf{H}} \succeq \breve{\lambda}_0 \mathbf{I} \succ \mathbf{0}$. Analogous to Lemma 5.1 in [86], we consider the singular value decomposition of $\breve{\mathbf{G}}$ being $\breve{\mathbf{G}} = \breve{\mathbf{P}} \breve{\mathbf{A}} \breve{\mathbf{Q}}^\mathsf{T}$, where we naturally have $\breve{\mathbf{A}} \succ \mathbf{0}$ since $\breve{\mathbf{H}}$ is positive definite. Then, with the expected reward vector $\breve{\boldsymbol{h}}$ (Def. 5.2), we have

$$\breve{\boldsymbol{h}} = (\breve{\mathbf{Q}} \breve{\mathbf{A}} \breve{\mathbf{P}}^\mathsf{T}) \cdot (\breve{\mathbf{P}} \breve{\mathbf{A}}^{-1} \breve{\mathbf{Q}}^\mathsf{T}) \cdot \breve{\boldsymbol{h}} = \sqrt{m} \cdot \breve{\mathbf{G}}^\mathsf{T} (\boldsymbol{\theta}^* - \boldsymbol{\theta}_0),$$

by considering there exists a set of parameters $\boldsymbol{\theta}^* = \boldsymbol{\theta}_0 + (\breve{\mathbf{P}}\breve{\mathbf{A}}^{-1}\breve{\mathbf{Q}}^\intercal) \cdot \breve{\boldsymbol{h}}/\sqrt{m}$.

Therefore, since $\breve{\boldsymbol{h}} = \sqrt{m} \cdot \breve{\mathbf{G}}^\intercal(\boldsymbol{\theta}^* - \boldsymbol{\theta}_0)$, we will have $\forall \boldsymbol{x} \in \breve{\mathcal{A}}_T$,

$$h(\boldsymbol{x}) = \langle g(\boldsymbol{x}; \boldsymbol{\theta}_0), \boldsymbol{\theta}^* - \boldsymbol{\theta}_0 \rangle.$$

Meanwhile, since $\breve{\mathbf{G}}^\intercal\breve{\mathbf{G}} \succeq \breve{\mathbf{H}}/2$, by applying Lemma G.8, we will have the distance

$$m \cdot \|\boldsymbol{\theta}^* - \boldsymbol{\theta}_0\|_2^2 = \boldsymbol{h}^\intercal\breve{\mathbf{Q}}\breve{\mathbf{A}}^{-1}\breve{\mathbf{P}}^\intercal \cdot \breve{\mathbf{P}}\breve{\mathbf{A}}^{-1}\breve{\mathbf{Q}}^\intercal\boldsymbol{h} = \boldsymbol{h}^\intercal(\breve{\mathbf{G}}^\intercal\breve{\mathbf{G}})^{-1}\boldsymbol{h} \leq 2\boldsymbol{h}^\intercal\breve{\mathbf{H}}^{-1}\boldsymbol{h}.$$

Finally, since the above results holds $\forall \boldsymbol{x} \in \breve{\mathcal{A}}_T$, due to the fact that $\breve{\mathcal{A}}_T$ contains all the *unique arms* of the collection $\mathcal{A}_T$, we will directly have the above results regarding parameters $\boldsymbol{\theta}^*$ feasible $\forall \boldsymbol{x} \in \mathcal{A}_T$. This completes the proof.

$\square$

**Lemma C.2.** *Suppose $m$ satisfies the conditions in Theorem 5.6. Suppose the gradient matrix with randomly initialized parameters is $\boldsymbol{\Sigma}^{(0)} = \lambda\mathbf{I} + \sum_{\boldsymbol{x} \in \mathcal{A}} g(\boldsymbol{x}; \boldsymbol{\theta}_0) \cdot g(\boldsymbol{x}; \boldsymbol{\theta}_0)^\intercal/m$, upon an arbitrary subset $\mathcal{A} \subseteq \mathcal{A}_T$ of arm collection $\mathcal{A}_T$. With probability at least $1 - \delta$ over the initialization, the result holds:*

$$\log\left(\frac{\det \boldsymbol{\Sigma}^{(0)}}{\det \lambda\mathbf{I}}\right) \leq \widetilde{d}\log(1 + TK/\lambda) + 1.$$

**Proof.** First, recall that we have $\mathcal{A}_T$ as the arm collection of: (i) the chosen arms $\{\boldsymbol{x}_t\}_{t=1}^T$; (ii) the optimal arms $\{\boldsymbol{x}_t^*\}_{t=1}^T$; (iii) and the imaginary ones $\{\widetilde{\boldsymbol{x}}_t\}_{t=1}^T$ chosen by the corruption-free model. This makes its cardinality $|\mathcal{A}_T| = 3T$. Thus, for the left hand side, we have

$$\log\frac{\det(\boldsymbol{\Sigma}^{(0)})}{\det(\lambda\mathbf{I})} \leq \log\det(\lambda\mathbf{I} + \sum_{\boldsymbol{x} \in \mathcal{A}_T} g(\boldsymbol{x}; \boldsymbol{\theta}_0)g(\boldsymbol{x}; \boldsymbol{\theta}_0)^\intercal/m) = \det(\lambda\mathbf{I} + \mathbf{G}_0\mathbf{G}_0^\intercal),$$

where we define gradient matrix $\mathbf{G}_0 = \left[g(\boldsymbol{x}; \boldsymbol{\theta}_0)/\sqrt{m}\right]_{\boldsymbol{x} \in \mathcal{A}_T} \in \mathbb{R}^{p \times (3T)}$ based on arm collection $\mathcal{A}_T$. Here, based on Lemma C.3, we can bound the distance between the Gradient matrix product $\mathbf{G}_0^\intercal\mathbf{G}_0$ and the NTK matrix $\mathbf{H}$ (Def. 5.1), as

$$\|\mathbf{G}_0^\intercal\mathbf{G}_0 - \mathbf{H}\| \leq 3T \cdot \frac{1}{3T \cdot \mathcal{O}(\sqrt{T}/\lambda)} = \frac{1}{\mathcal{O}(\sqrt{T}/\lambda)},$$

by setting $\epsilon = \frac{1}{3T \cdot \mathcal{O}(\sqrt{T}/\lambda)}$. The above results will hold, as long as we have the network width $m \geq \Omega((TL)^6 \log(TL/\delta)/\lambda^4)$, matching the conditions in Theorem 5.6. As a result, we can have

$$\log\det(\mathbf{I} + \mathbf{G}_0^\intercal\mathbf{G}_0/\lambda)$$
$$= \log\det(\mathbf{I} + \mathbf{H}/\lambda + (\mathbf{G}_0^\intercal\mathbf{G}_0 - \mathbf{H})/\lambda)$$
$$\leq \log\det(\mathbf{I} + \mathbf{H}/\lambda) + \langle(\mathbf{I} + \mathbf{H}/\lambda)^{-1}, (\mathbf{G}_0^\intercal\mathbf{G}_0 - \mathbf{H})/\lambda\rangle$$
$$\leq \log\det(\mathbf{I} + \mathbf{H}/\lambda) + \|(\mathbf{I} + \mathbf{H}/\lambda)^{-1}\|_F \|\mathbf{G}_0^\intercal\mathbf{G}_0 - \mathbf{H}\|_F/\lambda$$
$$\leq \log\det(\mathbf{I} + \mathbf{H}/\lambda) + \mathcal{O}(\sqrt{T}/\lambda) \cdot \|\mathbf{G}_0^\intercal\mathbf{G}_0 - \mathbf{H}\|_F$$
$$\leq \log\det(\mathbf{I} + \mathbf{H}/\lambda) + 1$$
$$= \widetilde{d}\log(1 + TK/\lambda) + 1.$$

The first inequality is because the concavity of $\log\det(\cdot)$ function; The third inequality is due to $\|(\mathbf{I} + \mathbf{H}\lambda)^{-1}\|_F \leq \|\mathbf{I}^{-1}\|_F \leq \sqrt{T}$; The fourth inequality is by applying the above distance upper bound $\|\mathbf{G}_0^\intercal\mathbf{G}_0 - \mathbf{H}\| \leq \frac{1}{\mathcal{O}(\sqrt{T}/\lambda)}$. The last inequality is because of the choice the $m$; The last equality is because of the Definition of $\widetilde{d}$. The proof is completed.

$\square$

**Lemma C.3.** *With the randomly initialized network parameters $\boldsymbol{\theta}_0 \in \mathbb{R}^p$ and a collection of arms $\mathcal{A} \subset \mathbb{R}^d$, define the gradient matrix $\mathbf{G}_\mathcal{A} = [g(\boldsymbol{x}; \boldsymbol{\theta}_0)]_{\boldsymbol{x} \in \mathcal{A}} \in \mathbb{R}^{p \times |\mathcal{A}|}$. Following the recursive procedure in Def. 5.1 (7), construct the NTK Gram matrix $\mathbf{H}_\mathcal{A}$ based on arms $\mathcal{A}$. Then, with the probability at least $1 - \delta$, we will have*

$$\|\mathbf{G}_\mathcal{A}^\intercal\mathbf{G}_\mathcal{A} - \mathbf{H}_\mathcal{A}\|_F \leq |\mathcal{A}| \cdot \epsilon,$$

*with the network width $m \geq \Omega(L^6 \log(|\mathcal{A}|L/\delta)/\epsilon^4)$.*

**Proof.** The proof of this lemma is analogous to the proof of Lemma B.1 in [86]. Based on Theorem 3.1 from [4], we have that for any two arms $\boldsymbol{x}, \boldsymbol{x}' \in \mathcal{A}$, as the network width $m \geq \Omega(L^6 \log(L/\delta)/\epsilon^4)$, we will have $|\langle g(\boldsymbol{x}; \boldsymbol{\theta}_0), g(\boldsymbol{x}'; \boldsymbol{\theta}_0) \rangle / m - \mathbf{H}_{\mathcal{A}}[\boldsymbol{x}, \boldsymbol{x}']| \leq \epsilon$, where $\mathbf{H}_{\mathcal{A}}[\boldsymbol{x}, \boldsymbol{x}']$ represents the element in NTK Gram matrix $\mathbf{H}_{\mathcal{A}}$ that corresponds to arms $\boldsymbol{x}$ and $\boldsymbol{x}'$.

Next, taking the union bound over all the arms in $\mathcal{A}$, we will have

$$\|\mathbf{G}_{\mathcal{A}}^{\mathsf{T}} \mathbf{G}_{\mathcal{A}} - \mathbf{H}_{\mathcal{A}}\|_F = \sqrt{\sum_{\boldsymbol{x} \in \mathcal{A}} \sum_{\boldsymbol{x}' \in \mathcal{A}} |\langle g(\boldsymbol{x}; \boldsymbol{\theta}_0), g(\boldsymbol{x}'; \boldsymbol{\theta}_0) \rangle / m - \mathbf{H}_{\mathcal{A}}[\boldsymbol{x}, \boldsymbol{x}']|^2} \leq |\mathcal{A}| \cdot \epsilon,$$

as long as the network width $m \geq \Omega(L^6 \log(|\mathcal{A}| L / \delta)/\epsilon^4)$.

$\square$

## C.2 Bounding single-round regret

With the above results linking the NTK to the neural model, we proceed to bound the single-round regret $R_t$ for $t \in [T]$. This single-round regret will then be aggregated to obtain the cumulative regret $R(T) = \sum_{t \in [T]} R_t$.

Based on Lemma C.1, we have the expected reward of an arm $\boldsymbol{x} \in \mathcal{X}_t$ being

$$\mathbb{E}[r|x] = h(x) = \langle g(x; \boldsymbol{\theta}_0), \boldsymbol{\theta}^* - \boldsymbol{\theta}_0 \rangle$$

where there exist parameters $\boldsymbol{\theta}^*$ such that $\|\boldsymbol{\theta}^* - \boldsymbol{\theta}_0\| \leq S/\sqrt{m}$. Meanwhile, apart from the trained parameters $\boldsymbol{\theta}_{t-1}$ based on chosen arms as well as the corresponding received rewards $\{\boldsymbol{x}_\tau, r_\tau\}_{\tau \in [t-1]}$, we also denote the imaginary corruption-free parameters $\widetilde{\boldsymbol{\theta}}_{t-1}$, which is trained with the chosen arms along with their unknown corruption-free rewards $\{\boldsymbol{x}_\tau, \widetilde{r}_\tau\}_{\tau \in [t-1]}$, for the sake of analysis.

### C.2.1 Decomposing the single-round regret

To bound the single-round regret $R_t$, we first decompose the objective into several individual terms, and then bound them individually. For reference, the parameters $\boldsymbol{\theta}_{t-1}$, covariance matrix $\boldsymbol{\Sigma}_{t-1}$, confidence ellipsoid $\mathcal{C}_{t-1}$, and weights $w_t$ pertain to the chosen arm $\boldsymbol{x}_t$, with the arm index $i \in [K]$ omitted for simplicity of notation.

**Arm selection scores.** First, recall that based on the arm pulling mechanism (line 13, Algorithm 1) and the benefit score (6) (Lemma C.11), the chosen arm $\boldsymbol{x}_t \in \mathcal{X}_t$ is selected by

$$\boldsymbol{x}_t = \arg \max_{\boldsymbol{x}_{i,t} \in \mathcal{X}_t} \left[ f(\boldsymbol{x}_{i,t}; \boldsymbol{\theta}_{i,t-1}) + \gamma_{i,t-1} \cdot \sqrt{g(\boldsymbol{x}_{i,t}; \boldsymbol{\theta}_{i,t-1})^{\mathsf{T}} \boldsymbol{\Sigma}_{i,t-1}^{-1} g(\boldsymbol{x}_{i,t}; \boldsymbol{\theta}_{i,t-1})/m} \right]$$
$$= \arg \max_{\boldsymbol{x}_{i,t} \in \mathcal{X}_t} U(\boldsymbol{x}_{i,t}),$$

where we denote the corresponding score shorthand as

$$U(\boldsymbol{x}_{i,t}) = f(\boldsymbol{x}_{i,t}; \boldsymbol{\theta}_{i,t-1}) + \gamma_{i,t-1} \cdot \sqrt{g(\boldsymbol{x}_{i,t}; \boldsymbol{\theta}_{i,t-1})^{\mathsf{T}} \boldsymbol{\Sigma}_{i,t-1}^{-1} g(\boldsymbol{x}_{i,t}; \boldsymbol{\theta}_{i,t-1})}. \quad \text{(C.2)}$$

Analogously, with $\widetilde{\boldsymbol{\theta}}_{i,t-1}$ being the parameters trained on same set of chosen arms and the corresponding corruption-free rewards, we denote

$$\widetilde{U}(\boldsymbol{x}_{i,t}) = f(\boldsymbol{x}_{i,t}; \widetilde{\boldsymbol{\theta}}_{i,t-1}) + \widetilde{\gamma}_{i,t-1} \cdot \sqrt{g(\boldsymbol{x}_{i,t}; \widetilde{\boldsymbol{\theta}}_{i,t-1})^{\mathsf{T}} \widetilde{\boldsymbol{\Sigma}}_{i,t-1}^{-1} g(\boldsymbol{x}_{i,t}; \widetilde{\boldsymbol{\theta}}_{i,t-1})}, \quad \text{(C.3)}$$

with corresponding covariance matrix $\widetilde{\boldsymbol{\Sigma}}_{i,t-1} = \lambda \mathbf{I} + \sum_{\tau \in [t-1]} w_{i,t}^{(\tau)} \cdot g(\boldsymbol{x}_\tau; \widetilde{\boldsymbol{\theta}}_{\tau-1}) g(\boldsymbol{x}_\tau; \widetilde{\boldsymbol{\theta}}_{\tau-1})^{\mathsf{T}}/m$, and the coefficient $\widetilde{\gamma}_{i,t-1}$ based on $\widetilde{\boldsymbol{\Sigma}}_{i,t-1}$ following the definition from (6). It is obvious that the arm selection depends on the trained network parameters, and thus if the corruption makes the makes the trained network parameters $\boldsymbol{\theta}_{i,t-1}$ deviate from the corruption-free ones $\widetilde{\boldsymbol{\theta}}_{i,t-1}$, it will lead to discrepancy in terms of arm selection decisions.

**Alternative forms of selection scores.** On the other hand, to maintain the consistency with the form of (C.1) in Lemma C.1, we can consider an alternative form of arm selection being

$$V(\boldsymbol{x}_{i,t}) = \langle g(\boldsymbol{x}_{i,t}; \boldsymbol{\theta}_0), \boldsymbol{\theta}_{i,t-1} - \boldsymbol{\theta}_0 \rangle + \gamma_{i,t-1} \cdot \sqrt{g(\boldsymbol{x}_{i,t}; \boldsymbol{\theta}_{i,t-1})^{\mathsf{T}} \boldsymbol{\Sigma}_{i,t-1}^{-1} g(\boldsymbol{x}_{i,t}; \boldsymbol{\theta}_{i,t-1})/m}$$
$$= \max_{\boldsymbol{\theta} \in \mathcal{C}_{i,t-1}} \langle g(\boldsymbol{x}_{i,t}; \boldsymbol{\theta}_0), \boldsymbol{\theta} - \boldsymbol{\theta}_0 \rangle \quad \text{(C.4)}$$

where we have $\mathcal{C}_{i,t-1} := \{\boldsymbol{\theta} : \|\boldsymbol{\theta} - \boldsymbol{\theta}_{i,t-1}\|_{\boldsymbol{\Sigma}_{i,t-1}} \leq \gamma_{i,t-1}/\sqrt{m}, \gamma_{i,t-1} > 0\}$ being the confidence ellipsoid of for the actual trained parameters $\boldsymbol{\theta}_{i,t-1}$, trained with possibly corrupted records. $\gamma_{i,t-1} > 0$ represents the radius of the confidence ellipsoid; and the last equality in (C.4) is due to $\max_{\boldsymbol{x}:\|\boldsymbol{x}-\boldsymbol{b}\|_{\mathbf{A}} \leq c} \langle \boldsymbol{a}, \boldsymbol{x} \rangle = \langle \boldsymbol{a}, \boldsymbol{b} \rangle + c \cdot \sqrt{\boldsymbol{a}^{\intercal}\mathbf{A}^{-1}\boldsymbol{a}}$ [86]. Analogously, for the corruption-free parameters $\widetilde{\boldsymbol{\theta}}_{i,t-1}$, we also define its confidence ellipsoid $\widetilde{\mathcal{C}}_{i,t-1} := \{\boldsymbol{\theta} : \|\boldsymbol{\theta} - \widetilde{\boldsymbol{\theta}}_{i,t-1}\|_{\widetilde{\boldsymbol{\Sigma}}_{i,t-1}} \leq \widetilde{\gamma}_{i,t-1}/\sqrt{m}, \widetilde{\gamma}_{i,t-1} > 0\}$, with $\widetilde{\gamma}_{i,t-1} > 0$ being the radius of the confidence ellipsoid, and by Lemma C.10, we will have $\boldsymbol{\theta}^* \in \widetilde{\mathcal{C}}_{i,t-1}$. As a result, we can also formulate an alternative form for arm selection as

$$
\begin{aligned}
\widetilde{V}(\boldsymbol{x}_{i,t}) &= \langle g(\boldsymbol{x}_{i,t}; \boldsymbol{\theta}_0), \widetilde{\boldsymbol{\theta}}_{i,t-1} - \boldsymbol{\theta}_0 \rangle + \widetilde{\gamma}_{i,t-1} \cdot \sqrt{g(\boldsymbol{x}_{i,t}; \widetilde{\boldsymbol{\theta}}_{i,t-1})^{\intercal} \widetilde{\boldsymbol{\Sigma}}_{i,t-1}^{-1} g(\boldsymbol{x}_{i,t}; \widetilde{\boldsymbol{\theta}}_{i,t-1})/m} \\
&= \max_{\widetilde{\boldsymbol{\theta}} \in \widetilde{\mathcal{C}}_{i,t-1}} \langle g(\boldsymbol{x}_{i,t}; \boldsymbol{\theta}_0), \widetilde{\boldsymbol{\theta}} - \boldsymbol{\theta}_0 \rangle.
\end{aligned}
\tag{C.5}
$$

Here, the radius of the confidence ellipsoid for the corruption-free parameters $\widetilde{\gamma}_{i,t-1}$, is provided in Lemma C.10, ensuring that $\boldsymbol{\theta}^* \in \widetilde{\mathcal{C}}_{i,t-1}$. On the other hand, deriving the radius for the trained model, $\gamma_{i,t-1}$ in face of potential corruptions, is considerably more challenging. With our carefully designed arm weights in (4), we manage to establish the updated confidence ellipsoid in Lemma C.12, along with the corresponding UCB for reward estimation (Lemma C.11).

Note that our UCB-based exploration score (6) is also motivated by Lemma C.11. For a specific arm $\boldsymbol{x}_{i,t} \in \mathcal{X}_t$, our UCB-type exploration score includes only terms related to $\boldsymbol{x}_{i,t}$, omitting constant and arm-invariant parts. This approach is intuitive, as arm-independent terms do not influence arm selection (line 13, Algorithm 1), and they will remain the same across candidate arms $\mathcal{X}_t$. Moreover, with a sufficiently large network width $m$ as in Theorem 5.6, a majority of these terms can be further reduced to $\mathcal{O}(1)$.

**Decomposing the single-round objective.** Up to the time step $t \in [T]$, denoting $\boldsymbol{x}_t, \boldsymbol{x}_t^* \in \mathcal{X}_t$ being the chosen arm and the optimal arm in each round respectively, we will have the corresponding regret bound for step $t$ as:

$$
\begin{aligned}
R_t &= \min\left\{ h(\boldsymbol{x}_t^*) - h(\boldsymbol{x}_t), 1 \right\} \\
&= \min\left\{ h(\boldsymbol{x}_t^*) - \widetilde{V}(\boldsymbol{x}_t) + \underbrace{\widetilde{V}(\boldsymbol{x}_t) - h(\boldsymbol{x}_t)}_{\widetilde{\mathrm{UCB}}_t(\boldsymbol{x}_t)}, 1 \right\} \\
&= \min\left\{ \widetilde{\mathrm{UCB}}_t(\boldsymbol{x}_t) + \underbrace{h(\boldsymbol{x}_t^*) - V(\boldsymbol{x}_t)}_{I_{R_1}} + \underbrace{V(\boldsymbol{x}_t) - \widetilde{V}(\boldsymbol{x}_t)}_{I_{R_2}}, 1 \right\}
\end{aligned}
\tag{C.6}
$$

where the first equality is because we have the expected rewards in bounded value range $[0, 1]$. Here, with the three terms after decomposing the single-round regret $R_t$, we can bound the first term $\widetilde{\mathrm{UCB}}_t(\boldsymbol{x}_t)$ with Lemma C.9, while bounding the two error terms $I_{R_1}$ and $I_{R_2}$ with Lemma C.6 and Lemma C.4 respectively.

### C.3 Bounding cumulative regret $R(T)$ (Proof of Theorem 5.6)

First, we would like to mention that since both the expected corrupted reward $\mathbb{E}[r]$ and the expected corruption-free reward $\mathbb{E}[\widetilde{r}]$ fall within the range $[0, 1]$, as defined in (1), it follows that $C \leq T$. In this case, as long as the network width requirement $m \geq \Omega(\mathrm{poly}(T))$ in Theorem 5.6 is adequately satisfied, this naturally implies $m \geq \Omega(\mathrm{poly}(C,T))$ with a sufficiently large network width $m$. This also indicates that knowledge of $C$ is not mandatory for setting $m$ before the online learning process begins. The above clarification will also be used to establish the final regret bound.

Then, with the derived results in terms of single-round regret $R_t, t \in [T]$, we can then proceed to bound the cumulative regret over $T$ rounds. By definition in (C.6), we have $R(T) = \sum_{t=1}^{T} \min\left\{ \widetilde{\mathrm{UCB}}_t(\boldsymbol{x}_t) + \underbrace{h(\boldsymbol{x}_t^*) - V(\boldsymbol{x}_t)}_{I_{R_1}} + \underbrace{V(\boldsymbol{x}_t) - \widetilde{V}(\boldsymbol{x}_t)}_{I_{R_2}}, 1 \right\}$. Recall that the first term $\widetilde{\mathrm{UCB}}_t(\boldsymbol{x}_t)$ can be bounded with Lemma C.9, while we bound the other two error terms $I_{R_1}$ and $I_{R_2}$

with Lemma C.6 and Lemma C.4 respectively. Next, with the conclusions from above lemmas, we will have

$$R(T) = \sum_{t=1}^{T} \min\left\{\widetilde{\mathsf{UCB}}_t(\boldsymbol{x}_t) + \underbrace{h(\boldsymbol{x}_t^*) - V(\boldsymbol{x}_t)}_{I_{R_1}} + \underbrace{V(\boldsymbol{x}_t) - \widetilde{V}(\boldsymbol{x}_t)}_{I_{R_2}}, 1\right\}$$

$$\leq \sum_{t=1}^{T} \min\left\{2\widetilde{\gamma}_{t-1} \cdot \|g(\boldsymbol{x}_t; \widetilde{\boldsymbol{\theta}}_{t-1})/\sqrt{m}\|_{\widetilde{\boldsymbol{\Sigma}}_{t-1}^{-1}} + \gamma_{t-1} \cdot \|g(\boldsymbol{x}_t; \boldsymbol{\theta}_{t-1})/\sqrt{m}\|_{\boldsymbol{\Sigma}_{t-1}^{-1}}\right.$$

$$+ \mathcal{O}(\alpha C) \cdot \left\|g(\boldsymbol{x}_t; \boldsymbol{\theta}_{t-1})/\sqrt{m}\right\|_{(\bar{\boldsymbol{\Sigma}}_{t-1})^{-1}}^2 + \mathcal{O}(\sqrt{\lambda}S) \cdot \left\|g(\widetilde{\boldsymbol{x}}_t; \boldsymbol{\theta}_{\widetilde{i}_t, t-1})/\sqrt{m}\right\|_{(\boldsymbol{\Sigma}'_{t-1})^{-1}}$$

$$+ \mathcal{O}(\alpha C) \cdot \left\|g(\boldsymbol{x}_t; \boldsymbol{\theta}_{t-1})/\sqrt{m}\right\|_{(\bar{\boldsymbol{\Sigma}}_{t-1})^{-1}}^2 + \mathcal{O}(\sqrt{\lambda}S) \cdot \left\|g(\boldsymbol{x}_t; \boldsymbol{\theta}_{t-1})/\sqrt{m}\right\|_{(\bar{\boldsymbol{\Sigma}}_{t-1})^{-1}}$$

$$+ \mathcal{O}(m^{-2/3}\log(m)L^{7/2}t^{5/3}\lambda^{-5/3}(1 + \sqrt{t/\lambda}))$$

$$+ \mathcal{O}(m^{-1/6}\sqrt{\log(m)}t^{1/6}\lambda^{-7/6}L^{2/7}) + \mathcal{O}(Sm^{-1/6}\sqrt{\log(m)}t^{1/6}\lambda^{-1/6}L^{2/7})$$

$$+ \nu \cdot \sqrt{1 + \mathcal{O}(m^{-1/6}\sqrt{\log(m)}L^4 t^{7/6}\lambda^{-7/6})} \cdot \mathcal{O}(m^{-1/12}\log^{1/4}(m)L^3 t^{5/6}\lambda^{-7/12})$$

$$+ \mathcal{O}(m^{-1/6}t^{2/3}\lambda^{-2/3}\sqrt{\log(m)}L^{7/2}) + \mathcal{O}(m^{-1/6}t^{1/6}\lambda^{-7/6}\sqrt{\log(m)}L^{7/2})$$

$$\left. + \gamma_{t-1} \cdot \mathcal{O}(m^{-1/12}t^{7/12}\lambda^{-13/12}\log^{1/4}(m)L^{t/2}) + \mathcal{O}(m^{-2/3}\log(m)L^{7/2}t^{5/3}\lambda^{-5/3}(1 + \sqrt{t/\lambda})), 1\right\}$$

where "weight-free" covariance matrices are $\bar{\boldsymbol{\Sigma}}_{t-1} = \lambda\mathbf{I} + \sum_{\tau\in[t-1]} g(\boldsymbol{x}_\tau; \boldsymbol{\theta}_{\tau-1})g(\boldsymbol{x}_\tau; \boldsymbol{\theta}_{\tau-1})^{\mathsf{T}}/m$, and $\boldsymbol{\Sigma}'_{t-1} = \lambda\mathbf{I} + \sum_{\tau\in[t-1]} g(\widetilde{\boldsymbol{x}}_\tau; \boldsymbol{\theta}_{\widetilde{i}_\tau, \tau-1})g(\widetilde{\boldsymbol{x}}_\tau; \boldsymbol{\theta}_{\widetilde{i}_\tau, \tau-1})^{\mathsf{T}}/m$. Afterwards, with sufficient network width $m$ that satisfies the conditions in Theorem 5.6, the majority of the terms on the RHS of (C.7), which contain $m$ to the negative order, can be reduced to $\mathcal{O}(1)$. Thus, we will then have

$$R(T) \leq \mathcal{O}(1) + \sum_{t=1}^{T} \min\left\{2\widetilde{\gamma}_{t-1} \cdot \|g(\boldsymbol{x}_t; \widetilde{\boldsymbol{\theta}}_{t-1})/\sqrt{m}\|_{\widetilde{\boldsymbol{\Sigma}}_{t-1}^{-1}} + \gamma_{t-1} \cdot \|g(\boldsymbol{x}_t; \boldsymbol{\theta}_{t-1})/\sqrt{m}\|_{\boldsymbol{\Sigma}_{t-1}^{-1}}\right.$$

$$+ \mathcal{O}(\alpha C) \cdot \left\|g(\boldsymbol{x}_t; \boldsymbol{\theta}_{t-1})/\sqrt{m}\right\|_{(\bar{\boldsymbol{\Sigma}}_{t-1})^{-1}}^2 + \mathcal{O}(\sqrt{\lambda}S)\left\|g(\widetilde{\boldsymbol{x}}_t; \boldsymbol{\theta}_{\widetilde{i}_t, t-1})/\sqrt{m}\right\|_{(\boldsymbol{\Sigma}'_{t-1})^{-1}}$$

$$\left. + \mathcal{O}(\alpha C) \cdot \left\|g(\boldsymbol{x}_t; \boldsymbol{\theta}_{t-1})/\sqrt{m}\right\|_{(\bar{\boldsymbol{\Sigma}}_{t-1})^{-1}}^2 + \mathcal{O}(\sqrt{\lambda}S) \cdot \left\|g(\boldsymbol{x}_t; \boldsymbol{\theta}_{t-1})/\sqrt{m}\right\|_{(\bar{\boldsymbol{\Sigma}}_{t-1})^{-1}}, 1\right\}$$

$$\leq \mathcal{O}(1) + \sum_{t=1}^{T} \min\left\{2\widetilde{\gamma}_{t-1} \cdot \|g(\boldsymbol{x}_t; \widetilde{\boldsymbol{\theta}}_{t-1})/\sqrt{m}\|_{\widetilde{\boldsymbol{\Sigma}}_{t-1}^{-1}} + \gamma_{t-1} \cdot \|g(\boldsymbol{x}_t; \boldsymbol{\theta}_{t-1})/\sqrt{m}\|_{\boldsymbol{\Sigma}_{t-1}^{-1}}, 1\right\}$$

$$+ \mathcal{O}(\alpha C) \cdot \sum_{t=1}^{T} \min\left\{\left\|g(\boldsymbol{x}_t; \boldsymbol{\theta}_{t-1})/\sqrt{m}\right\|_{(\bar{\boldsymbol{\Sigma}}_{t-1})^{-1}}^2, 1\right\}$$

$$+ \mathcal{O}(\sqrt{\lambda}S)\sqrt{T\sum_{t=1}^{T} \min\left\{\left\|g(\widetilde{\boldsymbol{x}}_t; \boldsymbol{\theta}_{\widetilde{i}_t, t-1})/\sqrt{m}\right\|_{(\boldsymbol{\Sigma}'_{t-1})^{-1}}^2, 1\right\}}$$

$$+ \mathcal{O}(\sqrt{\lambda}S) \cdot \sqrt{T\sum_{t=1}^{T} \min\left\{\left\|g(\boldsymbol{x}_t; \boldsymbol{\theta}_{t-1})/\sqrt{m}\right\|_{(\bar{\boldsymbol{\Sigma}}_{t-1})^{-1}}^2, 1\right\}}$$

$$\tag{C.7}$$

where the second inequality is by applying the triangular inequality. Then, denote $w_{\min} = \min[\{w_t^{(\tau)}, w_{\widetilde{i}_t, t}^{(\tau)}\}_{t\in[T], \tau\in[t-1]}]$. Analogously, we also have the round-wise minimum weight value being $w_t^{\min} = \min\{w_{i,t}^{(\tau)}\}_{i\in[K], \tau\in[t-1]} = \kappa^2 < 1$, $\forall t \in [T]$, which will be used to determine our scaling parameter $\alpha$. With our notation from Theorem 5.6, this intuitively leads to $\alpha \leq \frac{\kappa^2}{\beta}$. As a

result, we can therefore have

$$R(T) \leq \mathcal{O}(1) + \sum_{t=1}^{T} \min \left\{ 2\widetilde{\gamma}_{t-1} \cdot \|g(\boldsymbol{x}_t; \widetilde{\boldsymbol{\theta}}_{t-1})/\sqrt{m}\|_{\widetilde{\boldsymbol{\Sigma}}_{t-1}^{-1}} + \gamma_{t-1} \cdot \|g(\boldsymbol{x}_t; \boldsymbol{\theta}_{t-1})/\sqrt{m}\|_{\boldsymbol{\Sigma}_{t-1}^{-1}}, 1 \right\}$$

$$+ \mathcal{O}(C\beta^{-1}\kappa^2) \cdot \log \frac{\det(\bar{\boldsymbol{\Sigma}}_T)}{\det(\lambda \mathbf{I})} + \mathcal{O}(\sqrt{\lambda}S) \cdot \sqrt{T} \log \frac{\det(\bar{\boldsymbol{\Sigma}}_T)}{\det(\lambda \mathbf{I})} + \mathcal{O}(\sqrt{\lambda}S) \cdot \sqrt{T} \log \frac{\det(\boldsymbol{\Sigma}'_T)}{\det(\lambda \mathbf{I})}$$

$$\leq \mathcal{O}(1) + \sum_{t=1}^{T} \min \left\{ 2\widetilde{\gamma}_{t-1} \cdot \|g(\boldsymbol{x}_t; \widetilde{\boldsymbol{\theta}}_{t-1})/\sqrt{m}\|_{\widetilde{\boldsymbol{\Sigma}}_{t-1}^{-1}} + \gamma_{t-1} \cdot \|g(\boldsymbol{x}_t; \boldsymbol{\theta}_{t-1})/\sqrt{m}\|_{\boldsymbol{\Sigma}_{t-1}^{-1}}, 1 \right\}$$

$$+ \mathcal{O}(C\beta^{-1}\kappa^2) \cdot \widetilde{d} \log(1 + TK/\lambda) + \mathcal{O}(\sqrt{\lambda}S) \cdot \sqrt{T\widetilde{d} \log(1 + TK/\lambda)}$$

$$\leq \mathcal{O}(1) + \sum_{t=1}^{T} \min \left\{ 2\frac{\widetilde{\gamma}_{t-1}}{\sqrt{w_{\mathsf{min}}}} \cdot \|\sqrt{w_{\mathsf{min}}}g(\boldsymbol{x}_t; \widetilde{\boldsymbol{\theta}}_{t-1})/\sqrt{m}\|_{\widetilde{\boldsymbol{\Sigma}}_{t-1}^{-1}}, 1 \right\}$$

$$+ \sum_{t=1}^{T} \min \left\{ \frac{\gamma_{t-1}}{\sqrt{w_{\mathsf{min}}}} \cdot \|\sqrt{w_{\mathsf{min}}}g(\boldsymbol{x}_t; \boldsymbol{\theta}_{t-1})/\sqrt{m}\|_{\boldsymbol{\Sigma}_{t-1}^{-1}}, 1 \right\}$$

$$+ \mathcal{O}(C\beta^{-1}\kappa^2) \cdot \widetilde{d} \log(1 + TK/\lambda) + \mathcal{O}(\sqrt{\lambda}S) \cdot \sqrt{T\widetilde{d} \log(1 + TK/\lambda)} \tag{C.8}$$

where the first inequality is because the auxiliary matrices $\bar{\boldsymbol{\Sigma}}_{t-1}^{(0)}$ and $\boldsymbol{\Sigma}'_{t-1}$ do not involve arm weights, thus we can directly applying the Lemma G.7 and Lemma G.6. The second inequality is by applying Lemma C.2. Meanwhile, since $\boldsymbol{\Sigma}'_{t-1}$ is not defined w.r.t. randomly initialized $\boldsymbol{\theta}_0$, we additional apply the Lemma G.6 in terms of the matrix determinant difference to derive the results, where the extra term will also be reduced to $\mathcal{O}(1)$ with sufficiently large $m$. Afterwards, for the rest of the summation terms in (C.8), we can have

$$R(T) \leq \mathcal{O}(\frac{\widetilde{\gamma}_T}{\sqrt{w_{\mathsf{min}}}}) \sqrt{T \sum_{t=1}^{T} \min \left\{ \|\sqrt{w_{\mathsf{min}}}g(\boldsymbol{x}_t; \widetilde{\boldsymbol{\theta}}_{t-1})/\sqrt{m}\|^2_{\widetilde{\boldsymbol{\Sigma}}_{t-1}^{-1}}, 1 \right\}}$$

$$+ \mathcal{O}(\frac{\gamma_T}{\sqrt{w_{\mathsf{min}}}}) \sqrt{T \sum_{t=1}^{T} \min \left\{ \|\sqrt{w_{\mathsf{min}}}g(\boldsymbol{x}_t; \boldsymbol{\theta}_{t-1})/\sqrt{m}\|^2_{\boldsymbol{\Sigma}_{t-1}^{-1}}, 1 \right\}}$$

$$+ \mathcal{O}(C\beta^{-1}\kappa^2) \cdot \widetilde{d} \log(1 + TK/\lambda) + \mathcal{O}(\sqrt{\lambda}S) \cdot \sqrt{T\widetilde{d} \log(1 + TK/\lambda)} + \mathcal{O}(1)$$

$$\leq \frac{1}{\sqrt{w_{\mathsf{min}}}} \mathcal{O}\left( \nu \sqrt{\log \frac{\det(\widetilde{\boldsymbol{\Sigma}}_T)}{\det(\lambda \mathbf{I})} - 2\log(\delta)} + \lambda^{1/2}S + \nu \sqrt{\log \frac{\det(\boldsymbol{\Sigma}_T)}{\det(\lambda \mathbf{I})} - 2\log(\delta)} + \lambda^{1/2}S \right)$$

$$\cdot \sqrt{T\widetilde{d} \log(1 + TK/\lambda)}$$

$$+ \mathcal{O}(C\beta^{-1}\kappa^2) \cdot \widetilde{d} \log(1 + TK/\lambda) + \mathcal{O}(\sqrt{\lambda}S) \cdot \sqrt{T\widetilde{d} \log(1 + TK/\lambda)} + \mathcal{O}(1)$$

$$\leq \frac{1}{\sqrt{w_{\mathsf{min}}}} \mathcal{O}\left( \lambda^{1/2}S + \nu \sqrt{\log \frac{\det(\widetilde{\boldsymbol{\Sigma}}_T^{(0)})}{\det(\lambda \mathbf{I})} + \mathcal{O}(m^{-1/6}\sqrt{\log(m)}L^4 T^{5/3}\lambda^{-1/6}) - 2\log(\delta)} \right.$$

$$\left. + \nu \sqrt{\log \frac{\det(\boldsymbol{\Sigma}_T^{(0)})}{\det(\lambda \mathbf{I})} + \mathcal{O}(m^{-1/6}\sqrt{\log(m)}L^4 T^{5/3}\lambda^{-1/6}) - 2\log(\delta)} \right) \sqrt{T\widetilde{d} \log(1 + TK/\lambda)}$$

$$+ \mathcal{O}(C\beta^{-1}\kappa^2) \cdot \widetilde{d} \log(1 + TK/\lambda) + \mathcal{O}(\sqrt{\lambda}S) \cdot \sqrt{T\widetilde{d} \log(1 + TK/\lambda)} + \mathcal{O}(1), \tag{C.9}$$

where the second inequality is by applying Lemma G.7 and Lemma C.2, as well as the definition of coefficients $\gamma_T, \widetilde{\gamma}_T$, along with the fact that $\gamma_T \geq \gamma_t, \widetilde{\gamma}_T \geq \widetilde{\gamma}_t, t \in [T]$. The last inequality is by applying Lemma G.6. With sufficiently large network width $m$ as in Theorem 5.6, we can have $\mathcal{O}(m^{-1/6}\sqrt{\log(m)}L^4 T^{5/3}\lambda^{-1/6}) \leq \mathcal{O}(1)$. Then, due to the fact that $\bar{\boldsymbol{\Sigma}}_T^{(0)} \succeq \widetilde{\boldsymbol{\Sigma}}_T^{(0)}, \boldsymbol{\Sigma}_T^{(0)}$, applying

the Lemma C.2, it will then lead to

$$
\begin{aligned}
R(T) &\leq \frac{1}{\sqrt{w_{\mathsf{min}}}} \mathcal{O}\!\left(\nu\sqrt{\widetilde{d}\log(1+TK/\lambda)-2\log(\delta)}+\lambda^{1/2}S\right)\sqrt{T\widetilde{d}\log(1+TK/\lambda)} \\
&\quad + \mathcal{O}\!\left(C\beta^{-1}\kappa^2\right)\cdot\widetilde{d}\log(1+TK/\lambda)+\mathcal{O}(1) \\
&\leq \mathcal{O}\!\left(\nu\sqrt{\widetilde{d}\log(1+TK/\lambda)-2\log(\delta)}+\lambda^{1/2}S\right)\sqrt{T\widetilde{d}\log(1+TK/\lambda)/\kappa^2} \\
&\quad + \mathcal{O}\!\left(C\beta^{-1}\widetilde{d}\kappa^2\log(1+TK/\lambda)\right)
\end{aligned}
\tag{C.10}
$$

where the second inequality is by setting the tunable parameter $\alpha$ in each round accordingly, in order to ensure $w_{\mathsf{min}} = \kappa^2 < 1$. Here, we remind that our regret analysis does not require the learner to know the minimum fraction value $\beta$ before the online learning process, where $\beta = \min_{t\in[T],\tau\in[t-1]}\big[\min\{\mathsf{frac}_\tau(\boldsymbol{x}_t;\mathcal{X}_t,\bar{\boldsymbol{\Sigma}}_{t-1}),\mathsf{frac}_\tau(\widetilde{\boldsymbol{x}}_t;\mathcal{X}_t,\bar{\boldsymbol{\Sigma}}_{t-1})\}\big]$. As we have mentioned, in practice, the learner can scale the $\alpha$ values in each round to make sure the round-wise minimum weight value $w_t^{\mathsf{min}} = \min\{w_{i,t}^{(\tau)}\}_{i\in[K],\tau\in[t-1]} = \kappa^2 < 1,\ \forall t\in[T]$ (Subsec. B.5).

### C.4  Auxiliary sequences: Regression parameters and gradient descent parameters

To bridge neural models with NTK regression, we have two different routes to decouple the effects of adversarial corruptions from the received arm rewards. First, we define the gradient-based ridge regression parameters specific to a candidate arm $\boldsymbol{x}_{i,t}\in\mathcal{X}_t$ in round $t\in[T]$, as

$$
\begin{aligned}
\boldsymbol{\Sigma}_{i,t-1}^{(0)} &= \lambda\mathbf{I} + \sum_{\tau\in[t-1]} w_{i,t}^{(\tau)} g(\boldsymbol{x}_\tau;\boldsymbol{\theta}_0)\cdot g(\boldsymbol{x}_\tau;\boldsymbol{\theta}_0)^\mathsf{T}/m, \\
\boldsymbol{\Sigma}_{i,t-1} &= \lambda\mathbf{I} + \sum_{\tau\in[t-1]} w_{i,t}^{(\tau)} g(\boldsymbol{x}_\tau;\boldsymbol{\theta}_{\tau-1})\cdot g(\boldsymbol{x}_\tau;\boldsymbol{\theta}_{\tau-1})^\mathsf{T}/m, \\
\widetilde{\boldsymbol{\Sigma}}_{i,t-1} &= \lambda\mathbf{I} + \sum_{\tau\in[t-1]} w_{i,t}^{(\tau)} g(\boldsymbol{x}_\tau;\widetilde{\boldsymbol{\theta}}_{\tau-1})\cdot g(\boldsymbol{x}_\tau;\widetilde{\boldsymbol{\theta}}_{\tau-1})^\mathsf{T}/m, \\
\boldsymbol{b}_{i,t-1}^{(0)} &= \sum_{\tau\in[t-1]} w_{i,t}^{(\tau)} g(\boldsymbol{x}_\tau;\boldsymbol{\theta}_0)\cdot r_\tau/\sqrt{m}, \qquad\qquad \boldsymbol{b}_{i,t-1} = \sum_{\tau\in[t-1]} w_{i,t}^{(\tau)} g(\boldsymbol{x}_\tau;\boldsymbol{\theta}_{\tau-1})\cdot r_\tau/\sqrt{m}, \\
\widetilde{\boldsymbol{b}}_{i,t-1}^{(0)} &= \sum_{\tau\in[t-1]} w_t^{(\tau)} g(\boldsymbol{x}_\tau;\boldsymbol{\theta}_0)\cdot \widetilde{r}_\tau/\sqrt{m}, \qquad\qquad \widetilde{\boldsymbol{b}}_{i,t-1} = \sum_{\tau\in[t-1]} w_{i,t}^{(\tau)} g(\boldsymbol{x}_\tau;\boldsymbol{\theta}_{\tau-1})\cdot \widetilde{r}_\tau/\sqrt{m},
\end{aligned}
\tag{C.11}
$$

where $\{\boldsymbol{x}_\tau,r_\tau\},\tau\in[t]$ respectively stand for the chosen arms as well as their rewards, while $\{\boldsymbol{x}_\tau,\widetilde{r}_\tau\},\tau\in[t]$ refer to chosen arms and their imaginary corruption-free rewards. For notation simplicity, we use $w_t^{(\tau)},\tau\in[t-1]$ to denote the arm weights for $\boldsymbol{x}_t$.

Meanwhile, given a candidate arm $\boldsymbol{x}_{i,t}\in\mathcal{X}_t$ with arm weight $w_{i,t}^{(\tau)}$ defined in (4), we can try to bound the $I_2$ term by decomposing the adversarial corruptions with a series auxiliary gradient sequences $\{\boldsymbol{\theta}_0,\boldsymbol{\Theta}^{(1)},\ldots,\boldsymbol{\Theta}^{(J)}\}$ as in Lemma D.1, such that for $j$-th iteration

$$
\boldsymbol{\Theta}^{(j+1)} = \boldsymbol{\Theta}^{(j)} - \eta\cdot\left[\mathbf{J}^{(0)}\cdot\mathbf{W}\cdot\left([\mathbf{J}^{(0)}]^\mathsf{T}(\boldsymbol{\Theta}^{(j)}-\boldsymbol{\theta}_0)-\boldsymbol{y}\right)+m\lambda(\boldsymbol{\Theta}^{(j)}-\boldsymbol{\theta}_0)\right]
$$

as well as an analogous sequence for corruption-free parameters

$$
\widetilde{\boldsymbol{\Theta}}^{(j+1)} = \widetilde{\boldsymbol{\Theta}}^{(j)} - \eta\cdot\left[\mathbf{J}^{(0)}\cdot\mathbf{W}\cdot\left([\mathbf{J}^{(0)}]^\mathsf{T}(\widetilde{\boldsymbol{\Theta}}^{(j)}-\boldsymbol{\theta}_0)-\widetilde{\boldsymbol{y}}\right)+m\lambda(\widetilde{\boldsymbol{\Theta}}^{(j)}-\boldsymbol{\theta}_0)\right]
$$

where $\mathbf{J}^{(0)} := \big(g(\boldsymbol{x}_1;\boldsymbol{\theta}_0),g(\boldsymbol{x}_2;\boldsymbol{\theta}_0),\ldots,g(\boldsymbol{x}_{t-1};\boldsymbol{\theta}_0)\big)\in\mathbb{R}^{p\times(t-1)}$, and $\mathbf{W}$ refers to the diagonal matrix of arm weights $\{w_{i,t}^{(\tau)}\}_{\tau\in[t-1]}$, along with the reward vectors $\boldsymbol{y},\widetilde{\boldsymbol{y}}\in\mathbb{R}^{t-1}$ separately being the vector of received rewards and corruption-free rewards. In particular, with $[\mathbf{J}^{(0)}]_\tau$ being the $\tau$-th column of matrix $\mathbf{J}^{(0)}$, the auxiliary sequence $\boldsymbol{\Theta}^{(j)}$ can be deemed as applying Gradient Descent to

solve the following optimization problem

$$\min_{\boldsymbol{\Theta}} \mathcal{L}(\boldsymbol{\Theta}) = \sum_{\tau \in [t-1]} \frac{1}{2} \cdot w_{i,t}^{(\tau)} \cdot \left\| [\mathbf{J}^{(0)}]_\tau^\intercal (\boldsymbol{\Theta} - \boldsymbol{\theta}_0) - y_\tau \right\|_2^2 + \frac{1}{2} \cdot m\lambda \cdot \left\| \boldsymbol{\Theta} - \boldsymbol{\theta}_0 \right\|_2^2 \qquad \text{(C.12)}$$

Analogously, we can also derive the optimization problem for the sequence of corruption-free auxiliary parameters $\widetilde{\boldsymbol{\Theta}}^{(j)}$, by applying the same definition of weight matrix $\mathbf{W}$. Since the arm weights $w \leq 1$ by definition, we will also have the diagonal matrix norm $\|\mathbf{W}\|_2 \leq 1$.

**Notation simplicity.** For reference, we remind that the parameters $\boldsymbol{\theta}_{t-1}$, covariance matrix $\boldsymbol{\Sigma}_{t-1}$, confidence ellipsoid $\mathcal{C}_{t-1}$, and weights $w_t$ pertain to the chosen arm $\boldsymbol{x}_t$, with the arm index $i \in [K]$ omitted for simplicity of notation. Meanwhile, the gradinet covariance matrix $\boldsymbol{\Sigma}_{i,t-1}$, confidence ellipsoid $\mathcal{C}_{i,t-1}$, and weights $w_{i,t}$ are associated with each candidate arm $\boldsymbol{x}_{i,t}$ for $i \in [K]$.

## C.5 Bounding the error term $I_{R_2}$ in (C.6)

Recall that to derive the upper bound for the single-round regret $R_t$, we need to respectively bound the three error terms on the RHS of (C.6). Here, we first bound term $I_{R_2}$ with the following Lemma C.4.

**Lemma C.4.** *Suppose the imaginary neural network $f(\cdot; \widetilde{\boldsymbol{\theta}}_{t-1})$ in round $t \in [T]$ has been trained on corruption-free rewards $\{\boldsymbol{x}_\tau, \widetilde{r}_\tau\}_{\tau \in [t-1]}$. Meanwhile, $f(\cdot)$ is an $L$-layer FC network with width $m$. Suppose we have $m, J, \eta$ satisfying the conditions in Theorem 5.6. Then, for the chosen arm $\boldsymbol{x}_t \in \mathcal{X}_t$, with the probability at least $1 - \delta$, we will have*

$$I_{R_2} = V(\boldsymbol{x}_t) - \widetilde{V}(\boldsymbol{x}_t)$$
$$\leq \gamma_{t-1} \cdot \| g(\boldsymbol{x}_t; \boldsymbol{\theta}_{t-1}) / \sqrt{m} \|_{\boldsymbol{\Sigma}_{t-1}^{-1}} + \mathcal{O}(m^{-1/6}\sqrt{\log(m)}t^{1/6}\lambda^{-7/6}L^{2/7})$$

$$+ \mathcal{O}(\alpha C) \cdot \left\| g(\boldsymbol{x}_t; \boldsymbol{\theta}_{t-1}) / \sqrt{m} \right\|_{(\bar{\boldsymbol{\Sigma}}_{t-1})^{-1}}^2 + \mathcal{O}(\sqrt{\lambda}S) \left\| g(\boldsymbol{x}_t; \boldsymbol{\theta}_{t-1}) / \sqrt{m} \right\|_{(\bar{\boldsymbol{\Sigma}}_{t-1})^{-1}}$$
$$+ \mathcal{O}(m^{-2/3}\log(m)L^{7/2}t^{5/3}\lambda^{-5/3}(1 + \sqrt{t/\lambda})) + \mathcal{O}(Cm^{-1/6}\sqrt{\log(m)}t^{7/6}\lambda^{-1/6}L^5)$$
$$+ \mathcal{O}(Cm^{-1/6}\sqrt{\log(m)}t^{1/6}\lambda^{-7/6}L^4)$$
$$+ \mathcal{O}(m^{-1/6}\sqrt{\log(m)}t^{2/3}\lambda^{-2/3}L^{7/2}).$$

*By definition, we have $\gamma_{t-1} = \mathcal{O}\left(\nu\sqrt{\log\frac{\det(\boldsymbol{\Sigma}_{t-1})}{\det(\lambda\mathbf{I})} - 2\log(\delta)} + \lambda^{1/2}S\right)$, as well as the gradient covariance matrix $\bar{\boldsymbol{\Sigma}}_{t-1} = \lambda\mathbf{I} + \sum_{\tau \in [t-1]} g(\boldsymbol{x}_\tau; \boldsymbol{\theta}_{\tau-1})g(\boldsymbol{x}_\tau; \boldsymbol{\theta}_{\tau-1})^\intercal / m$. The minimum round-wise arm weight $w_t^{min} = \min\{w_{i,t}^{(\tau)}\}_{i \in [K], \tau \in [t-1]} = \kappa^2 < 1, \forall t \in [T]$ by scaling parameter $\alpha$.*

**Proof.** For the error term $I_{R_2}$, we have

$$I_{R_2} = V(\boldsymbol{x}_t) - \widetilde{V}(\boldsymbol{x}_t)$$
$$= \max_{\boldsymbol{\theta} \in \mathcal{C}_{t-1}} \langle g(\boldsymbol{x}_t; \boldsymbol{\theta}_0), \boldsymbol{\theta} - \boldsymbol{\theta}_0 \rangle - \max_{\widetilde{\boldsymbol{\theta}} \in \widetilde{\mathcal{C}}_{t-1}} \langle g(\boldsymbol{x}_t; \boldsymbol{\theta}_0), \widetilde{\boldsymbol{\theta}} - \boldsymbol{\theta}_0 \rangle$$
$$\leq \max_{\boldsymbol{\theta} \in \mathcal{C}_{t-1}} \langle g(\boldsymbol{x}_t; \boldsymbol{\theta}_0), \boldsymbol{\theta} - \boldsymbol{\theta}_0 \rangle - \langle g(\boldsymbol{x}_t; \boldsymbol{\theta}_0), \widetilde{\boldsymbol{\theta}}_{t-1} - \boldsymbol{\theta}_0 \rangle$$
$$\leq \max_{\boldsymbol{\theta} \in \mathcal{C}_{t-1}} \langle g(\boldsymbol{x}_t; \boldsymbol{\theta}_0), \boldsymbol{\theta} - \boldsymbol{\theta}_0 \rangle - \langle g(\boldsymbol{x}_t; \boldsymbol{\theta}_{t-1}), \widetilde{\boldsymbol{\theta}}_{t-1} - \boldsymbol{\theta}_0 \rangle + \mathcal{O}(m^{-1/6}\sqrt{\log(m)}t^{2/3}\lambda^{-2/3}L^{7/2})$$
$$= \max_{\boldsymbol{\theta} \in \mathcal{C}_{t-1}} \langle g(\boldsymbol{x}_t; \boldsymbol{\theta}_0), \boldsymbol{\theta} - \boldsymbol{\theta}_{t-1} \rangle - \langle g(\boldsymbol{x}_t; \boldsymbol{\theta}_{t-1}), \widetilde{\boldsymbol{\theta}}_{t-1} - \boldsymbol{\theta}_{t-1} \rangle + \mathcal{O}(m^{-1/6}\sqrt{\log(m)}t^{2/3}\lambda^{-2/3}L^{7/2})$$
$$= \underbrace{\max_{\boldsymbol{\theta} \in \mathcal{C}_{t-1}} \langle g(\boldsymbol{x}_t; \boldsymbol{\theta}_0), \boldsymbol{\theta} - \boldsymbol{\theta}_{t-1} \rangle}_{\text{Projection difference}} + |\underbrace{\langle g(\boldsymbol{x}_t; \boldsymbol{\theta}_{t-1}), \widetilde{\boldsymbol{\theta}}_{t-1} - \boldsymbol{\theta}_{t-1} \rangle}_{I_2}| + \mathcal{O}(m^{-1/6}\sqrt{\log(m)}t^{2/3}\lambda^{-2/3}L^{7/2})$$

$$\text{(C.13)}$$

where the first inequality is because of the definition of confidence ellipsoids $\widetilde{\mathcal{C}}_{t-1}$ and $\mathcal{C}_{t-1}$. The second inequality is by applying Lemma G.3 and Lemma G.4. Here, we have the first term on

the RHS is bounded by Corollary C.5, where this term is used to represent the gradient projection difference, between the confidence ellipsoid center parameters $\boldsymbol{\theta}_{t-1}$ and the other parameters in this confidence ellipsoid $\boldsymbol{\theta} \in \mathcal{C}_{t-1}$. Meanwhile, term $I_2$ will be bounded by Corollary C.8.

$\square$

**Corollary C.5.** *Suppose the imaginary neural network $f(\cdot; \widetilde{\boldsymbol{\theta}}_{t-1})$ in round $t \in [T]$ has been trained on corruption-free rewards $\{\boldsymbol{x}_\tau, \widetilde{r}_\tau\}_{\tau \in [t-1]}$. Meanwhile, $f(\cdot)$ is an L-layer FC network with width $m$. Suppose we have $m, J, \eta$ satisfying the conditions in Theorem 5.6. Then, for the chosen arm $\boldsymbol{x}_t \in \mathcal{X}_t$, with the probability at least $1 - \delta$, we will have*

$$\max_{\boldsymbol{\theta} \in \mathcal{C}_{t-1}} \langle g(\boldsymbol{x}_t; \boldsymbol{\theta}_0),\ \boldsymbol{\theta} - \boldsymbol{\theta}_{t-1} \rangle \le \gamma_{t-1} \cdot \|g(\boldsymbol{x}_t; \boldsymbol{\theta}_{t-1})/\sqrt{m}\|_{\boldsymbol{\Sigma}_{t-1}^{-1}} + \mathcal{O}(m^{-1/6}\sqrt{\log(m)}t^{1/6}\lambda^{-7/6}L^{2/7}),$$

*and the corresponding summation value being*

$$\sum_{t \in [T]} \min \Big\{ \max_{\boldsymbol{\theta} \in \mathcal{C}_{t-1}} \langle g(\boldsymbol{x}_t; \boldsymbol{\theta}_0),\ \boldsymbol{\theta} - \boldsymbol{\theta}_{t-1} \rangle, 1 \Big\}$$

$$\le \frac{1}{\sqrt{w_t^{min}}} \mathcal{O}\left( \nu \sqrt{\log \frac{\det(\boldsymbol{\Sigma}_T)}{\det(\lambda \mathbf{I})} - 2\log(\delta)} + \lambda^{1/2} S \right) \cdot \sqrt{2T \cdot \log \frac{\det(\boldsymbol{\Sigma}_T)}{\det(\lambda \mathbf{I})}}.$$

$$+ \mathcal{O}(Sm^{-1/6}\sqrt{\log(m)}T^{7/6}\lambda^{-1/6}L^{2/7}) + \mathcal{O}(m^{-1/6}\sqrt{\log(m)}T^{7/6}\lambda^{-7/6}L^{2/7}).$$

*By definition, we have the radius of $\mathcal{C}_{t-1}$ being $\gamma_{t-1} = \mathcal{O}\big(\nu \cdot \sqrt{\log \frac{\det(\boldsymbol{\Sigma}_{t-1})}{\det(\lambda \mathbf{I})} - 2\log(\delta)} + \lambda^{1/2} S\big)$, and the gradient covariance matrix $\boldsymbol{\Sigma}_{t-1} = \lambda \mathbf{I} + \sum_{\tau \in [t-1]} w_t^{(\tau)} \cdot g(\boldsymbol{x}_\tau; \boldsymbol{\theta}_{\tau-1})g(\boldsymbol{x}_\tau; \boldsymbol{\theta}_{\tau-1})^{\intercal}/m$.*

**Proof.** The proof of this corollary follows an analogous approach as Lemma C.9. By definition, we have the gradient inner product for the chosen arm $\boldsymbol{x}_t$ in round $t$ as

$$\max_{\boldsymbol{\theta} \in \mathcal{C}_{t-1}} \langle g(\boldsymbol{x}_t; \boldsymbol{\theta}_0),\ \boldsymbol{\theta} - \boldsymbol{\theta}_{t-1} \rangle$$

$$\le \max_{\boldsymbol{\theta} \in \mathcal{C}_{t-1}} \|\boldsymbol{\theta} - \boldsymbol{\theta}_{t-1}\|_{\boldsymbol{\Sigma}_{t-1}} \cdot \|g(\boldsymbol{x}_t; \boldsymbol{\theta}_0)\|_{\boldsymbol{\Sigma}_{t-1}^{-1}} \qquad (C.14)$$

$$\le \gamma_{t-1} \cdot \|g(\boldsymbol{x}_t; \boldsymbol{\theta}_{t-1})/\sqrt{m}\|_{\boldsymbol{\Sigma}_{t-1}^{-1}} + \mathcal{O}(m^{-1/6}\sqrt{\log(m)}t^{1/6}\lambda^{-7/6}L^{2/7})$$

where the first inequality is due to Holder's inequality. The second inequality is by applying the definition of confidence ellipsoid $\mathcal{C}_{t-1}$, as well as Lemma G.4 and Lemma G.6. Then, similarly, we will also need to bound the summation over $T$ rounds. Following an analogous procedure as in Lemma C.9, we will have

$$\sum_{t \in [T]} \min\Big\{ \max_{\boldsymbol{\theta} \in \mathcal{C}_{t-1}} \langle g(\boldsymbol{x}_t; \boldsymbol{\theta}_0),\ \boldsymbol{\theta} - \boldsymbol{\theta}_{t-1} \rangle,\ 1 \Big\}$$

$$\le \frac{1}{\sqrt{w_t^{min}}} \mathcal{O}\left( \nu \sqrt{\log \frac{\det(\boldsymbol{\Sigma}_T)}{\det(\lambda \mathbf{I})} - 2\log(\delta)} + \lambda^{1/2} S \right) \cdot \sqrt{2T \cdot \log \frac{\det(\boldsymbol{\Sigma}_T)}{\det(\lambda \mathbf{I})}}.$$

$$+ \mathcal{O}(Sm^{-1/6}\sqrt{\log(m)}T^{7/6}\lambda^{-1/6}L^{2/7}) + \mathcal{O}(m^{-1/6}\sqrt{\log(m)}T^{7/6}\lambda^{-7/6}L^{2/7}).$$

By the definition $\beta = \min_{t \in [T], \tau \in [t-1]}[\min\{\mathsf{frac}_\tau(\boldsymbol{x}_t; \mathcal{X}_t, \bar{\boldsymbol{\Sigma}}_{t-1}), \mathsf{frac}_\tau(\widetilde{\boldsymbol{x}}_t; \mathcal{X}_t, \bar{\boldsymbol{\Sigma}}_{t-1})\}]$, we have the lower bound of arm weights being $w_t^{min} \ge \alpha \cdot \beta$. Note that we also scale the $\alpha$ parameter to ensure $w_t^{min} = \kappa^2$ as indicated in the Subsec. B.5, which requires no prior knowledge for $\beta$.

$\square$

## C.6 Bounding the error term $I_{R_1}$ in (C.6)

In this subsection, we bound the error term $I_{R_1}$ in (C.6), with the following Lemma C.6. Meanwhile, we denote an extra weight-free gradient covariance matrix for the chosen arms $\{\widetilde{\boldsymbol{x}}_\tau\}_{\tau \in [t]}$ of the corruption-free model. The corresponding arm index is denoted as $\widetilde{i}_t, t \in [T]$, such that $\boldsymbol{x}_{\widetilde{i}_t,t} = \widetilde{\boldsymbol{x}}_t$.

**Lemma C.6.** *Suppose the imaginary neural network $f(\cdot; \widetilde{\boldsymbol{\theta}}_{t-1})$ in round $t \in [T]$ has been trained on corruption-free rewards $\{\boldsymbol{x}_\tau, \widetilde{r}_\tau\}_{\tau \in [t-1]}$. Meanwhile, $f(\cdot)$ is an L-layer FC network with width $m$.*

*Suppose we have $m, J, \eta$ satisfying the conditions in Theorem 5.6. Then, for the chosen arm $\boldsymbol{x}_t \in \mathcal{X}_t$, with the probability at least $1 - \delta$, we will have*

$$I_{R_1} = h(\boldsymbol{x}_t^*) - V(\boldsymbol{x}_t)$$

$$\leq \mathcal{O}(\alpha C)\left\|g(\boldsymbol{x}_t; \boldsymbol{\theta}_{t-1})/\sqrt{m}\right\|_{(\bar{\boldsymbol{\Sigma}}_{t-1})^{-1}}^2 + \mathcal{O}(\sqrt{\lambda}S) \cdot \left\|g(\widetilde{\boldsymbol{x}}_t; \boldsymbol{\theta}_{\widetilde{i}_t, t-1})/\sqrt{m}\right\|_{(\boldsymbol{\Sigma}'_{t-1})^{-1}}$$

$$+ \mathcal{O}(m^{-2/3}\log(m)L^{7/2}t^{5/3}\lambda^{-5/3}(1 + \sqrt{t/\lambda})) + \mathcal{O}(Cm^{-1/6}\sqrt{\log(m)}t^{7/6}\lambda^{-1/6}L^5)$$

$$+ \mathcal{O}(Cm^{-1/6}\sqrt{\log(m)}t^{1/6}\lambda^{-7/6}L^4)$$

$$+ \nu \cdot \sqrt{1 + \mathcal{O}(m^{-1/6}\sqrt{\log(m)}L^4t^{7/6}\lambda^{-7/6})} \cdot \mathcal{O}(m^{-1/12}\log^{1/4}(m)L^3t^{5/6}\lambda^{-7/12})$$

$$+ \mathcal{O}(m^{-1/6}t^{2/3}\lambda^{-2/3}\sqrt{\log(m)}L^{7/2}) + \mathcal{O}(m^{-1/6}t^{1/6}\lambda^{-7/6}\sqrt{\log(m)}L^{7/2})$$

$$+ \gamma_{\widetilde{i}_t, t-1} \cdot \mathcal{O}(m^{-1/12}t^{7/12}\lambda^{-13/12}\log^{1/4}(m)L^{t/2}).$$

*By definition, we have the coefficient $\gamma_{\widetilde{i}_t, t-1} = \mathcal{O}\big(\nu\sqrt{\log\frac{\det(\boldsymbol{\Sigma}_{\widetilde{i}_t, t-1})}{\det(\lambda\mathbf{I})} - 2\log(\delta)} + \lambda^{1/2}S\big)$, as well as the gradient covariance matrices $\boldsymbol{\Sigma}'_{t-1} = \lambda\mathbf{I} + \sum_{\tau \in [t-1]} g(\widetilde{\boldsymbol{x}}_\tau; \boldsymbol{\theta}_{\widetilde{i}_\tau, \tau-1})g(\widetilde{\boldsymbol{x}}_\tau; \boldsymbol{\theta}_{\widetilde{i}_\tau, \tau-1})^\intercal/m$, and $\bar{\boldsymbol{\Sigma}}_{t-1} = \lambda\mathbf{I} + \sum_{\tau \in [t-1]} g(\boldsymbol{x}_\tau; \boldsymbol{\theta}_{\tau-1})g(\boldsymbol{x}_\tau; \boldsymbol{\theta}_{\tau-1})^\intercal/m$. The minimum round-wise arm weight $w_t^{min} = \min\{w_{i,t}^{(\tau)}\}_{i \in [K], \tau \in [t-1]} = \kappa^2 < 1, \forall t \in [T]$ by scaling parameter $\alpha$.*

**Proof.** By definition, we can have $I_{R_1} = h(\boldsymbol{x}_t^*) - U(\boldsymbol{x}_t) + U(\boldsymbol{x}_t) - V(\boldsymbol{x}_t)$. Here, in terms of the distance between $U(\boldsymbol{x}_{i,t})$ and $V(\boldsymbol{x}_{i,t})$ given a candidate arm $\boldsymbol{x}_{i,t} \in \mathcal{X}_t$, we have

$$|U(\boldsymbol{x}_{i,t}) - V(\boldsymbol{x}_{i,t})| = |f(\boldsymbol{x}_{i,t}; \boldsymbol{\theta}_{i,t-1}) - \langle g(\boldsymbol{x}_{i,t}; \boldsymbol{\theta}_0), \boldsymbol{\theta}_{i,t-1} - \boldsymbol{\theta}_0\rangle|$$

$$\leq |f(\boldsymbol{x}_{i,t}; \boldsymbol{\theta}_{i,t-1}) - \langle g(\boldsymbol{x}_{i,t}; \boldsymbol{\theta}_{i,t-1}), \boldsymbol{\theta}_{i,t-1} - \boldsymbol{\theta}_0\rangle|$$

$$+ |\langle g(\boldsymbol{x}_{i,t}; \boldsymbol{\theta}_{i,t-1}), \boldsymbol{\theta}_{i,t-1} - \boldsymbol{\theta}_0\rangle - \langle g(\boldsymbol{x}_{i,t}; \boldsymbol{\theta}_0), \boldsymbol{\theta}_{i,t-1} - \boldsymbol{\theta}_0\rangle|$$

$$\leq \mathcal{O}(m^{-1/6}\sqrt{\log(m)}t^{2/3}\lambda^{-2/3}L^3) + \mathcal{O}(m^{-1/6}t^{2/3}\lambda^{-2/3}\sqrt{\log(m)}L^{7/2}).$$

where the first inequality is by triangular inequality, and the second inequality is by applying a similar approach as in (B.12) from [86] as well as Lemma G.4. Next, we proceed to bound term $I_{R_1}$. Here, for a candidate arm $\boldsymbol{x}_{i,t} \in \mathcal{X}_t$ and the associated corruption-free confidence ellipsoid $\widetilde{\mathcal{C}}_{i,t-1}$, we have $\boldsymbol{\theta}^* \in \widetilde{\mathcal{C}}_{i,t-1}$ based on Lemma C.10. Thus, with $\widetilde{\boldsymbol{x}}_t = \arg\max_{\boldsymbol{x}_{i,t} \in \mathcal{X}_t} \widetilde{V}(\boldsymbol{x}_{i,t})$ being the arm chosen by the corruption-free neural model, with the highest score $\widetilde{V}(\widetilde{\boldsymbol{x}}_t)$, there will be

$$I_{R_1} = h(\boldsymbol{x}_t^*) - U(\boldsymbol{x}_t) + U(\boldsymbol{x}_t) - V(\boldsymbol{x}_t)$$

$$= \langle g(\boldsymbol{x}_t^*; \boldsymbol{\theta}_0), \boldsymbol{\theta}^* - \boldsymbol{\theta}_0\rangle - U(\boldsymbol{x}_t) + U(\boldsymbol{x}_t) - V(\boldsymbol{x}_t)$$

$$\leq \max_{\widetilde{\boldsymbol{\theta}} \in \widetilde{\mathcal{C}}_{\widetilde{i}_t, t-1}} \langle g(\boldsymbol{x}_t^*; \boldsymbol{\theta}_0), \widetilde{\boldsymbol{\theta}} - \boldsymbol{\theta}_0\rangle - U(\boldsymbol{x}_t) + U(\boldsymbol{x}_t) - V(\boldsymbol{x}_t)$$

$$\leq \max_{\widetilde{\boldsymbol{\theta}} \in \widetilde{\mathcal{C}}_{\widetilde{i}_t, t-1}} \langle g(\widetilde{\boldsymbol{x}}_t; \boldsymbol{\theta}_0), \widetilde{\boldsymbol{\theta}} - \boldsymbol{\theta}_0\rangle - U(\boldsymbol{x}_t) + |U(\boldsymbol{x}_t) - V(\boldsymbol{x}_t)|,$$

where $\widetilde{\mathcal{C}}_{\widetilde{i}_t, t-1}$ refers to the confidence ellipsoid of corruption-free parameters $\widetilde{\boldsymbol{\theta}}_{\widetilde{i}_t, t-1}$, associated to arm $\widetilde{\boldsymbol{x}}_t \in \mathcal{X}_t$. Afterwards, it further leads to

$$I_{R_1} \leq \max_{\widetilde{\boldsymbol{\theta}} \in \widetilde{\mathcal{C}}_{\widetilde{i}_t, t-1}} \langle g(\widetilde{\boldsymbol{x}}_t; \boldsymbol{\theta}_0), \widetilde{\boldsymbol{\theta}} - \boldsymbol{\theta}_0\rangle - U(\boldsymbol{x}_t)$$

$$+ \mathcal{O}(m^{-1/6}\sqrt{\log(m)}t^{2/3}\lambda^{-2/3}L^3) + \mathcal{O}(m^{-1/6}t^{2/3}\lambda^{-2/3}\sqrt{\log(m)}L^{7/2})$$

$$\leq \max_{\widetilde{\boldsymbol{\theta}} \in \widetilde{\mathcal{C}}_{\widetilde{i}_t, t-1}} \langle g(\widetilde{\boldsymbol{x}}_t; \boldsymbol{\theta}_0), \widetilde{\boldsymbol{\theta}} - \boldsymbol{\theta}_0\rangle - V(\boldsymbol{x}_t)$$

$$+ \mathcal{O}(m^{-1/6}\sqrt{\log(m)}t^{2/3}\lambda^{-2/3}L^3) + \mathcal{O}(m^{-1/6}t^{2/3}\lambda^{-2/3}\sqrt{\log(m)}L^{7/2})$$

$$\leq \max_{\widetilde{\boldsymbol{\theta}} \in \widetilde{\mathcal{C}}_{\widetilde{i}_t, t-1}} \langle g(\widetilde{\boldsymbol{x}}_t; \boldsymbol{\theta}_0), \widetilde{\boldsymbol{\theta}} - \boldsymbol{\theta}_0\rangle - V(\widetilde{\boldsymbol{x}}_t)$$

$$+ \mathcal{O}(m^{-1/6}\sqrt{\log(m)}t^{2/3}\lambda^{-2/3}L^3) + \mathcal{O}(m^{-1/6}t^{2/3}\lambda^{-2/3}\sqrt{\log(m)}L^{7/2})$$

$$= \widetilde{V}(\widetilde{\boldsymbol{x}}_t) - V(\widetilde{\boldsymbol{x}}_t) + \mathcal{O}(m^{-1/6}\sqrt{\log(m)}t^{2/3}\lambda^{-2/3}L^3) + \mathcal{O}(m^{-1/6}t^{2/3}\lambda^{-2/3}\sqrt{\log(m)}L^{7/2})$$

To bound the output difference between $V(\boldsymbol{x}_{i,t})$ and $\widetilde{V}(\boldsymbol{x}_{i,t})$ for an arm $\boldsymbol{x}_{i,t}$, we can decompose them into separate terms. Recall the definition of $V(\cdot)$ in (C.4), and we can also define the analogous $\widetilde{V}(\cdot)$ by applying the corruption-free parameters $\widetilde{\boldsymbol{\theta}}$ and covariance matrix $\widetilde{\boldsymbol{\Sigma}}$. Therefore, for arm $\boldsymbol{x}_{i,t} \in \mathcal{X}_t$, we have

$$V(\boldsymbol{x}_{i,t}) - \widetilde{V}(\boldsymbol{x}_{i,t}) \le |V(\boldsymbol{x}_{i,t}) - V^{(0)}(\boldsymbol{x}_{i,t})| + |V^{(0)}(\boldsymbol{x}_{i,t}) - \widetilde{V}^{(0)}(\boldsymbol{x}_{i,t})| + |\widetilde{V}^{(0)}(\boldsymbol{x}_{i,t}) - \widetilde{V}(\boldsymbol{x}_{i,t})|, \tag{C.15}$$

where for the sake of analysis, we define two variants with randomly initialized parameters $\boldsymbol{\theta}_0$ being: $V^{(0)}(\boldsymbol{x}_{i,t}) = \langle g(\boldsymbol{x}_{i,t}; \boldsymbol{\theta}_0), \ \boldsymbol{\theta}_{t-1} - \boldsymbol{\theta}_0 \rangle + \gamma_{t-1} \cdot \sqrt{g(\boldsymbol{x}_{i,t}; \boldsymbol{\theta}_0)^\mathsf{T} (\boldsymbol{\Sigma}_{t-1}^{(0)})^{-1} g(\boldsymbol{x}_{i,t}; \boldsymbol{\theta}_0)/m}$, and $\widetilde{V}^{(0)}(\boldsymbol{x}_{i,t}) = \langle g(\boldsymbol{x}_{i,t}; \boldsymbol{\theta}_0), \ \widetilde{\boldsymbol{\theta}}_{t-1} - \boldsymbol{\theta}_0 \rangle + \widetilde{\gamma}_{t-1} \cdot \sqrt{g(\boldsymbol{x}_{i,t}; \boldsymbol{\theta}_0)^\mathsf{T} (\boldsymbol{\Sigma}_{t-1}^{(0)})^{-1} g(\boldsymbol{x}_{i,t}; \boldsymbol{\theta}_0)/m}$. With this result, our objective then is to derive the upper bounds for the three terms on the RHS of (C.15).

**Bounding the second term** $|V^{(0)}(\boldsymbol{x}_{i,t}) - \widetilde{V}^{(0)}(\boldsymbol{x}_{i,t})|$ **in (C.15).** For the second term, we have

$$
\begin{aligned}
&|V^{(0)}(\boldsymbol{x}_{i,t}) - \widetilde{V}^{(0)}(\boldsymbol{x}_{i,t})| \\
&\le |\langle g(\boldsymbol{x}_{i,t}; \boldsymbol{\theta}_0), \ \boldsymbol{\theta}_{i,t-1} - \widetilde{\boldsymbol{\theta}}_{i,t-1} \rangle| \\
&\quad + |\gamma_{i,t-1} \cdot \sqrt{g(\boldsymbol{x}_{i,t}; \boldsymbol{\theta}_0)^\mathsf{T} (\boldsymbol{\Sigma}_{i,t-1}^{(0)})^{-1} g(\boldsymbol{x}_{i,t}; \boldsymbol{\theta}_0)/m} - \widetilde{\gamma}_{i,t-1} \cdot \sqrt{g(\boldsymbol{x}_{i,t}; \boldsymbol{\theta}_0)^\mathsf{T} (\boldsymbol{\Sigma}_{i,t-1}^{(0)})^{-1} g(\boldsymbol{x}_{i,t}; \boldsymbol{\theta}_0)/m}| \\
&= |\langle g(\boldsymbol{x}_{i,t}; \boldsymbol{\theta}_0), \ \boldsymbol{\theta}_{i,t-1} - \widetilde{\boldsymbol{\theta}}_{i,t-1} \rangle| + |\gamma_{i,t-1} - \widetilde{\gamma}_{i,t-1}| \cdot \sqrt{g(\boldsymbol{x}_{i,t}; \boldsymbol{\theta}_0)^\mathsf{T} (\boldsymbol{\Sigma}_{i,t-1}^{(0)})^{-1} g(\boldsymbol{x}_{i,t}; \boldsymbol{\theta}_0)/m} \\
&\le |\langle g(\boldsymbol{x}_{i,t}; \boldsymbol{\theta}_0), \ \boldsymbol{\theta}_{i,t-1} - \widetilde{\boldsymbol{\theta}}_{i,t-1} \rangle| + |\gamma_{i,t-1} - \widetilde{\gamma}_{i,t-1}| \cdot \mathcal{O}(L/\sqrt{\lambda}).
\end{aligned}
$$

Due to the fact that $|\sqrt{a} - \sqrt{b}| \le \sqrt{|a-b|}$, based on the definition of $\gamma_{i,t-1}$ and $\widetilde{\gamma}_{i,t-1}$, we will have

$$
\begin{aligned}
|\gamma_{i,t-1} - \widetilde{\gamma}_{i,t-1}| &\le \nu \cdot \sqrt{1 + \mathcal{O}(m^{-1/6}\sqrt{\log(m)}L^4 t^{7/6} \lambda^{-7/6})} \\
&\quad \cdot \sqrt{\left| \log\left(\frac{\det \boldsymbol{\Sigma}_{i,t-1}}{\det \lambda \mathbf{I}}\right) - \log\left(\frac{\det \boldsymbol{\Sigma}_0}{\det \lambda \mathbf{I}}\right) + \log\left(\frac{\det \boldsymbol{\Sigma}_0}{\det \lambda \mathbf{I}}\right) - \log\left(\frac{\det \widetilde{\boldsymbol{\Sigma}}_{i,t-1}}{\det \lambda \mathbf{I}}\right) \right|} \\
&\le \nu \cdot \sqrt{1 + \mathcal{O}(m^{-1/6}\sqrt{\log(m)}L^4 t^{7/6} \lambda^{-7/6})} \cdot \mathcal{O}(m^{-1/12} \log^{1/4}(m) L^2 t^{5/6} \lambda^{-1/12}),
\end{aligned}
$$

based on Lemma G.6. Therefore, we will end up with

$$
\begin{aligned}
|V^{(0)}(\boldsymbol{x}_{i,t}) - \widetilde{V}^{(0)}(\boldsymbol{x}_{i,t})| &\le |\langle g(\boldsymbol{x}_{i,t}; \boldsymbol{\theta}_0), \ \boldsymbol{\theta}_{i,t-1} - \widetilde{\boldsymbol{\theta}}_{i,t-1} \rangle| \\
&\quad + \nu \cdot \sqrt{1 + \mathcal{O}(m^{-1/6}\sqrt{\log(m)}L^4 t^{7/6} \lambda^{-7/6})} \cdot \mathcal{O}(m^{-1/12} \log^{1/4}(m) L^3 t^{5/6} \lambda^{-7/12}).
\end{aligned}
$$

**Bounding Term** $|V(\boldsymbol{x}_{i,t}) - V^{(0)}(\boldsymbol{x}_{i,t})|$ **and term** $|\widetilde{V}^{(0)}(\boldsymbol{x}_{i,t}) - \widetilde{V}(\boldsymbol{x}_{i,t})|$ **in (C.15).** On the other hand, for the first term on the RHS, $|V(\boldsymbol{x}_{i,t}) - V^{(0)}(\boldsymbol{x}_{i,t})|$, we can bound this difference term by

$$
\begin{aligned}
&V(\boldsymbol{x}_{i,t}) - V^{(0)}(\boldsymbol{x}_{i,t}) \\
&= \frac{\gamma_{i,t-1}}{\sqrt{m}} \left( \sqrt{g(\boldsymbol{x}_{i,t}; \boldsymbol{\theta}_{i,t-1})^\mathsf{T} \boldsymbol{\Sigma}_{i,t-1}^{-1} g(\boldsymbol{x}_{i,t}; \boldsymbol{\theta}_{i,t-1})} - \sqrt{g(\boldsymbol{x}_{i,t}; \boldsymbol{\theta}_0)^\mathsf{T} (\boldsymbol{\Sigma}_{i,t-1}^{(0)})^{-1} g(\boldsymbol{x}_{i,t}; \boldsymbol{\theta}_0)} \right) \\
&= \frac{\gamma_{i,t-1}}{\sqrt{m}} \left( \left\| g(\boldsymbol{x}_{i,t}; \boldsymbol{\theta}_{i,t-1}) \right\|_{\boldsymbol{\Sigma}_{i,t-1}^{-1}} - \left\| g(\boldsymbol{x}_{i,t}; \boldsymbol{\theta}_0) \right\|_{(\boldsymbol{\Sigma}_{i,t-1}^{(0)})^{-1}} \right) \\
&\le \frac{\gamma_{i,t-1}}{\sqrt{m}} \left( \left\| g(\boldsymbol{x}_{i,t}; \boldsymbol{\theta}_{i,t-1}) \right\|_{\boldsymbol{\Sigma}_{i,t-1}^{-1}} - \left\| g(\boldsymbol{x}_{i,t}; \boldsymbol{\theta}_{i,t-1}) \right\|_{(\boldsymbol{\Sigma}_{i,t-1}^{(0)})^{-1}} + \left\| g(\boldsymbol{x}_{i,t}; \boldsymbol{\theta}_0) - g(\boldsymbol{x}_{i,t}; \boldsymbol{\theta}_{i,t-1}) \right\|_{(\boldsymbol{\Sigma}_{i,t-1}^{(0)})^{-1}} \right) \\
&\le \mathcal{O}(m^{-1/6} t^{1/6} \lambda^{-7/6} \sqrt{\log(m)} L^{7/2}) + \frac{\gamma_{i,t-1}}{\sqrt{m}} \cdot \left( \left\| g(\boldsymbol{x}_{i,t}; \boldsymbol{\theta}_{i,t-1}) \right\|_{\boldsymbol{\Sigma}_{i,t-1}^{-1}} - \left\| g(\boldsymbol{x}_{i,t}; \boldsymbol{\theta}_{i,t-1}) \right\|_{(\boldsymbol{\Sigma}_{i,t-1}^{(0)})^{-1}} \right)
\end{aligned}
$$

$$\leq \mathcal{O}(m^{-1/6}t^{1/6}\lambda^{-7/6}\sqrt{\log(m)}L^{7/2})$$
$$+ \frac{\gamma_{i,t-1}}{\sqrt{m}} \cdot \left( \sqrt{g(\boldsymbol{x}_{i,t};\boldsymbol{\theta}_{i,t-1})^{\mathsf{T}}\boldsymbol{\Sigma}_{i,t-1}^{-1}g(\boldsymbol{x}_{i,t};\boldsymbol{\theta}_{i,t-1}) - g(\boldsymbol{x}_{i,t};\boldsymbol{\theta}_{i,t-1})^{\mathsf{T}}(\boldsymbol{\Sigma}_{i,t-1}^{(0)})^{-1}g(\boldsymbol{x}_{i,t};\boldsymbol{\theta}_{i,t-1})} \right)$$
$$\leq \mathcal{O}(m^{-1/6}t^{1/6}\lambda^{-7/6}\sqrt{\log(m)}L^{7/2}) + \frac{\gamma_{i,t-1}}{\sqrt{m}} \cdot \left( \sqrt{\langle g(\boldsymbol{x}_{i,t};\boldsymbol{\theta}_{i,t-1}),\ (\boldsymbol{\Sigma}_{i,t-1}^{-1} - (\boldsymbol{\Sigma}_{i,t-1}^{(0)})^{-1})g(\boldsymbol{x}_{i,t};\boldsymbol{\theta}_{i,t-1})\rangle} \right)$$
$$= \mathcal{O}(m^{-1/6}t^{1/6}\lambda^{-7/6}\sqrt{\log(m)}L^{7/2})$$
$$+ \frac{\gamma_{i,t-1}}{\sqrt{m}} \cdot \left( \sqrt{\left\langle g(\boldsymbol{x}_{i,t};\boldsymbol{\theta}_{i,t-1}),\ \left( \boldsymbol{\Sigma}_{i,t-1}^{-1} \cdot (\boldsymbol{\Sigma}_{i,t-1}^{(0)} - \boldsymbol{\Sigma}_{i,t-1}) \cdot (\boldsymbol{\Sigma}_{i,t-1}^{(0)})^{-1} \right) \cdot g(\boldsymbol{x}_{i,t};\boldsymbol{\theta}_{i,t-1}) \right\rangle} \right)$$
$$\leq \mathcal{O}(m^{-1/6}t^{1/6}\lambda^{-7/6}\sqrt{\log(m)}L^{7/2}) + \gamma_{i,t-1} \cdot \mathcal{O}(m^{-1/12}t^{7/12}\lambda^{-13/12}\log^{1/4}(m)L^{t/2})$$

where the first inequality is because of the triangular inequality. The second inequality is by applying Lemma G.4. The third inequality is again the application of triangular inequality, and the last inequality is the application of Lemma G.6. Since the similar procedure can also be applied to bound the third term $|\widetilde{V}^{(0)}(\boldsymbol{x}_{i,t}) - \widetilde{V}(\boldsymbol{x}_{i,t})|$, after summing up the results, we will have

$$|V(\boldsymbol{x}_{i,t}) - V^{(0)}(\boldsymbol{x}_{i,t})|,\ |\widetilde{V}^{(0)}(\boldsymbol{x}_{i,t}) - \widetilde{V}(\boldsymbol{x}_{i,t})|$$
$$\leq \mathcal{O}(m^{-1/6}t^{2/3}\lambda^{-2/3}\sqrt{\log(m)}L^{7/2}) + \mathcal{O}(m^{-1/6}t^{1/6}\lambda^{-7/6}\sqrt{\log(m)}L^{7/2})$$
$$+ \gamma_{i,t-1} \cdot \mathcal{O}(m^{-1/12}t^{7/12}\lambda^{-13/12}\log^{1/4}(m)L^{t/2}).$$

**Summing up results.** Combining the results, we finally have the upper bound for $I_{R_1}$ being

$$I_{R_1} \leq |\langle g(\widetilde{\boldsymbol{x}}_t;\boldsymbol{\theta}_0),\ \boldsymbol{\theta}_{\widetilde{i}_t,t-1} - \widetilde{\boldsymbol{\theta}}_{\widetilde{i}_t,t-1}\rangle|$$
$$+ \nu \cdot \sqrt{1 + \mathcal{O}(m^{-1/6}\sqrt{\log(m)}L^4t^{7/6}\lambda^{-7/6})} \cdot \mathcal{O}(m^{-1/12}\log^{1/4}(m)L^3t^{5/6}\lambda^{-7/12})$$
$$+ \mathcal{O}(m^{-1/6}t^{2/3}\lambda^{-2/3}\sqrt{\log(m)}L^{7/2}) + \mathcal{O}(m^{-1/6}t^{1/6}\lambda^{-7/6}\sqrt{\log(m)}L^{7/2})$$
$$+ \gamma_{\widetilde{i}_t,t-1} \cdot \mathcal{O}(m^{-1/12}t^{7/12}\lambda^{-13/12}\log^{1/4}(m)L^{t/2})$$
$$\leq |\langle g(\widetilde{\boldsymbol{x}}_t;\boldsymbol{\theta}_{\widetilde{i}_t,t-1}),\ \boldsymbol{\theta}_{\widetilde{i}_t,t-1} - \widetilde{\boldsymbol{\theta}}_{\widetilde{i}_t,t-1}\rangle|$$
$$+ \nu \cdot \sqrt{1 + \mathcal{O}(m^{-1/6}\sqrt{\log(m)}L^4t^{7/6}\lambda^{-7/6})} \cdot \mathcal{O}(m^{-1/12}\log^{1/4}(m)L^3t^{5/6}\lambda^{-7/12})$$
$$+ \mathcal{O}(m^{-1/6}t^{2/3}\lambda^{-2/3}\sqrt{\log(m)}L^{7/2})$$
$$+ \mathcal{O}(m^{-1/6}t^{2/3}\lambda^{-2/3}\sqrt{\log(m)}L^{7/2}) + \mathcal{O}(m^{-1/6}t^{1/6}\lambda^{-7/6}\sqrt{\log(m)}L^{7/2})$$
$$+ \gamma_{\widetilde{i}_t,t-1} \cdot \mathcal{O}(m^{-1/12}t^{7/12}\lambda^{-13/12}\log^{1/4}(m)L^{t/2}),$$

and therefore we can formulate our objective to bound as

$$I_{R_1} \leq |\underbrace{\langle g(\widetilde{\boldsymbol{x}}_t;\boldsymbol{\theta}_{\widetilde{i}_t,t-1}),\ \boldsymbol{\theta}_{\widetilde{i}_t,t-1} - \widetilde{\boldsymbol{\theta}}_{\widetilde{i}_t,t-1}\rangle}_{I_1}|$$
$$+ \nu \cdot \sqrt{1 + \mathcal{O}(m^{-1/6}\sqrt{\log(m)}L^4t^{7/6}\lambda^{-7/6})} \cdot \mathcal{O}(m^{-1/12}\log^{1/4}(m)L^3t^{5/6}\lambda^{-7/12})$$
$$+ \mathcal{O}(m^{-1/6}t^{2/3}\lambda^{-2/3}\sqrt{\log(m)}L^{7/2}) + \mathcal{O}(m^{-1/6}t^{1/6}\lambda^{-7/6}\sqrt{\log(m)}L^{7/2})$$
$$+ \gamma_{\widetilde{i}_t,t-1} \cdot \mathcal{O}(m^{-1/12}t^{7/12}\lambda^{-13/12}\log^{1/4}(m)L^{t/2}), \tag{C.16}$$

where we recall that $\nu$ is the pre-defined exploration parameter that echoes the sub-Gaussian noise variance proxy. Finally, using the upper bound for term $I_1$ from Lemma C.7 will finish the proof.

$\square$

## C.7  Bounding the terms $I_1$, $I_2$

Here, we see that there are still two terms in $R_t, t \in [T]$ that need to be bounded, which are

$$I_1 = \langle g(\widetilde{\boldsymbol{x}}_t;\boldsymbol{\theta}_{\widetilde{i}_t,t-1}),\ \boldsymbol{\theta}_{\widetilde{i}_t,t-1} - \widetilde{\boldsymbol{\theta}}_{\widetilde{i}_t,t-1}\rangle, \qquad I_2 = \langle g(\boldsymbol{x}_t;\boldsymbol{\theta}_{t-1}),\ \widetilde{\boldsymbol{\theta}}_{t-1} - \boldsymbol{\theta}_{t-1}\rangle,$$

where we recall $\boldsymbol{x}_t \in \mathcal{X}_t$ is the actual chosen arm that is selected by the corrupted model $\boldsymbol{\theta}_{t-1}$, while we have $\widetilde{\boldsymbol{x}}_t \in \mathcal{X}_t$ being the imaginary arm chosen by the corruption-free model $\widetilde{\boldsymbol{\theta}}_{t-1}$. In this subsection, the error term $I_1$ will be bounded by Lemma C.7, while term $I_2$ will be bounded by Corollary C.8.

**Recap of auxiliary parameter definitions.** With definitions in Subsection C.4, we can have two different alternatives to decouple the adversarial corruptions from arm rewards: (i) the gradient-based regression parameters; and, (ii) the auxiliary sequence of gradient descent. For reference, recall that we denote the a series of gradient-based regression parameters, specified to candidate arm $\boldsymbol{x}_{i,t} \in \mathcal{X}_t$, as

$$\boldsymbol{\Sigma}_{i,t-1}^{(0)} = \lambda \mathbf{I} + \sum_{\tau \in [t-1]} w_{i,t}^{(\tau)} \cdot g(\boldsymbol{x}_\tau; \boldsymbol{\theta}_0) \cdot g(\boldsymbol{x}_\tau; \boldsymbol{\theta}_0)^\intercal / m,$$

$$\boldsymbol{\Sigma}_{i,t-1} = \lambda \mathbf{I} + \sum_{\tau \in [t-1]} w_{i,t}^{(\tau)} \cdot g(\boldsymbol{x}_\tau; \boldsymbol{\theta}_{\tau-1}) \cdot g(\boldsymbol{x}_\tau; \boldsymbol{\theta}_{\tau-1})^\intercal / m,$$

$$\boldsymbol{b}_{i,t-1}^{(0)} = \sum_{\tau \in [t-1]} w_{i,t}^{(\tau)} \cdot g(\boldsymbol{x}_\tau; \boldsymbol{\theta}_0) \cdot r_\tau / \sqrt{m}, \qquad \boldsymbol{b}_{i,t-1} = \sum_{\tau \in [t-1]} w_{i,t}^{(\tau)} \cdot g(\boldsymbol{x}_\tau; \boldsymbol{\theta}_{\tau-1}) \cdot r_\tau / \sqrt{m},$$

$$\widetilde{\boldsymbol{b}}_{i,t-1}^{(0)} = \sum_{\tau \in [t-1]} w_{i,t}^{(\tau)} \cdot g(\boldsymbol{x}_\tau; \boldsymbol{\theta}_0) \cdot \widetilde{r}_\tau / \sqrt{m}, \qquad \widetilde{\boldsymbol{b}}_{i,t-1} = \sum_{\tau \in [t-1]} w_{i,t}^{(\tau)} \cdot g(\boldsymbol{x}_\tau; \boldsymbol{\theta}_{\tau-1}) \cdot \widetilde{r}_\tau / \sqrt{m},$$

where $\{\boldsymbol{x}_\tau, r_\tau\}, \tau \in [t]$ respectively stands for the chosen arms as well as their received rewards, while $\{\boldsymbol{x}_\tau, \widetilde{r}_\tau\}, \tau \in [t]$ refer to the imaginary corruption-free rewards.

We also recall that with the chosen arm $\boldsymbol{x}_t \in \mathcal{X}_t$ in round $t$ with arm weights $w_t^{(\tau)}, \tau \in [t-1]$, defined in (4), we will have a series of auxiliary gradient sequences $\{\boldsymbol{\theta}_0, \boldsymbol{\Theta}^{(1)}, \dots, \boldsymbol{\Theta}^{(J)}\}$ as in Lemma D.1, such that for $j$-th, $j \in [J]$, iteration

$$\boldsymbol{\Theta}^{(j+1)} = \boldsymbol{\Theta}^{(j)} - \eta \cdot \left[ \mathbf{J}^{(0)} \cdot \mathbf{W} \big( [\mathbf{J}^{(0)}]^\intercal (\boldsymbol{\Theta}^{(j)} - \boldsymbol{\theta}_0) - \boldsymbol{y} \big) + m\lambda (\boldsymbol{\Theta}^{(j)} - \boldsymbol{\theta}_0) \right]$$

where the diagonal weight matrix is made up with the arm weights, as $\mathbf{W} = \operatorname{diag}\big( [w_t^{(\tau)}]_{\tau \in [t-1]} \big) \in \mathbb{R}^{(t-1) \times (t-1)}$. Similarly, we will also have the sequence definition for corruption-free parameters $\widetilde{\boldsymbol{\Theta}}^{(j+1)} = \widetilde{\boldsymbol{\Theta}}^{(j)} - \eta \cdot \left[ \mathbf{J}^{(0)} \cdot \mathbf{W} \big( [\mathbf{J}^{(0)}]^\intercal (\widetilde{\boldsymbol{\Theta}}^{(j)} - \boldsymbol{\theta}_0) - \widetilde{\boldsymbol{y}} \big) + m\lambda (\widetilde{\boldsymbol{\Theta}}^{(j)} - \boldsymbol{\theta}_0) \right]$, along with the Jacobian matrix $\mathbf{J}^{(0)} := \big( g(\boldsymbol{x}_1; \boldsymbol{\theta}_0), g(\boldsymbol{x}_2; \boldsymbol{\theta}_0), \dots, g(\boldsymbol{x}_{t-1}; \boldsymbol{\theta}_0) \big) \in \mathbb{R}^{p \times (t-1)}$. Here, we have $\boldsymbol{y}, \widetilde{\boldsymbol{y}} \in \mathbb{R}^{t-1}$ separately being the vector of received rewards and that of the imaginary corruption-free rewards. In particular, with $[\mathbf{J}^{(0)}]_\tau$ being the $\tau$-th column of matrix $\mathbf{J}^{(0)}$, the auxiliary sequence $\boldsymbol{\Theta}^{(j)}$ can be deemed as applying Gradient Descent to solve the following optimization problem

$$\min_{\boldsymbol{\Theta}} \mathcal{L}(\boldsymbol{\Theta}) = \sum_{\tau \in [t-1]} \frac{1}{2} \cdot w_{i,t}^{(\tau)} \left\| [\mathbf{J}^{(0)}]_\tau^\intercal (\boldsymbol{\Theta} - \boldsymbol{\theta}_0) - y_\tau \right\|_2^2 + \frac{1}{2} \cdot m\lambda \left\| \boldsymbol{\Theta} - \boldsymbol{\theta}_0 \right\|_2^2$$

Consequently, we can follow an analogous approach as in Lemma D.1 with the above optimization problem, to bound the difference between gradient-based parameters and the auxiliary sequence.

### C.7.1 Bounding the error term $I_1$

We bound the error term using Lemma C.7. Following the notation in the main body, let $\boldsymbol{\theta}_{t-1}$ denote the trained parameters associated with the chosen arm $\boldsymbol{x}_t$, and let $\boldsymbol{\theta}_{\widetilde{i}_t, t-1}$ represent the trained parameters of the arm $\widetilde{\boldsymbol{x}}_t = \boldsymbol{x}_{\widetilde{i}_t, t}, t \in [T]$, which corresponds to the chosen arm of the hypothetical corruption-free model $f(\cdot; \widetilde{\boldsymbol{\theta}})$. Additionally, we define a weight-free gradient covariance matrix for the arm collection $\{\widetilde{\boldsymbol{x}}_\tau\}_{\tau \in [t]}$ containing arms selected by the corruption-free model, which will be denoted by $\boldsymbol{\Sigma}_{t-1}' = \lambda \mathbf{I} + \sum_{\tau \in [t-1]} g(\widetilde{\boldsymbol{x}}_\tau; \boldsymbol{\theta}_{\widetilde{i}_\tau, \tau-1}) g(\widetilde{\boldsymbol{x}}_\tau; \boldsymbol{\theta}_{\widetilde{i}_\tau, \tau-1})^\intercal / m$.

**Lemma C.7.** *Suppose the imaginary corruption-free neural network $f(\cdot; \widetilde{\boldsymbol{\theta}}_{i,t-1})$ for arm $\boldsymbol{x}_{i,t} \in \mathcal{X}_t$ has been trained on corruption-free rewards $\{\boldsymbol{x}_\tau, \widetilde{r}_\tau\}_{\tau \in [t-1]}$, while the other trained network $f(\cdot; \boldsymbol{\theta}_{i,t-1})$ is trained on the received records $\{\boldsymbol{x}_\tau, r_\tau\}_{\tau \in [t-1]}$. Suppose $f(\cdot)$ is an $L$-layer FC*

*network with width $m$ that satisfies the conditions in Theorem 5.6. Then, for the arms $\boldsymbol{x}_t, \widetilde{\boldsymbol{x}}_t \in \mathcal{X}_t$, with the probability at least $1 - \delta$, we will have*

$$I_1 = \left| \langle g(\widetilde{\boldsymbol{x}}_t; \boldsymbol{\theta}_{\widetilde{i}_t, t-1}), \; \boldsymbol{\theta}_{\widetilde{i}_t, t-1} - \widetilde{\boldsymbol{\theta}}_{\widetilde{i}_t, t-1} \rangle \right|$$

$$\leq \mathcal{O}(\alpha C) \cdot \left\| g(\boldsymbol{x}_t; \boldsymbol{\theta}_{t-1})/\sqrt{m} \right\|_{(\bar{\boldsymbol{\Sigma}}_{t-1})^{-1}}^2 + \mathcal{O}(\sqrt{\lambda} S) \cdot \left\| g(\widetilde{\boldsymbol{x}}_t; \boldsymbol{\theta}_{\widetilde{i}_t, t-1})/\sqrt{m} \right\|_{(\boldsymbol{\Sigma}'_{t-1})^{-1}}$$

$$+ \mathcal{O}(m^{-2/3} \log(m) L^{7/2} t^{5/3} \lambda^{-5/3}(1 + \sqrt{t/\lambda})) + \mathcal{O}(C m^{-1/6} \sqrt{\log(m)} t^{7/6} \lambda^{-1/6} L^5)$$

$$+ \mathcal{O}(C m^{-1/6} \sqrt{\log(m)} t^{1/6} \lambda^{-7/6} L^4).$$

*where $\boldsymbol{\theta}_{t-1}$ are the parameters associated to chosen arm $\boldsymbol{x}_t$, and $\boldsymbol{\theta}_{\widetilde{i}_t, t-1}$ are those parameters of arm $\widetilde{\boldsymbol{x}}_t$. The gradient matrix is defined as $\boldsymbol{\Sigma}'_{t-1} = \lambda \mathbf{I} + \sum_{\tau \in [t-1]} g(\widetilde{\boldsymbol{x}}_\tau; \boldsymbol{\theta}_{\widetilde{i}_\tau, \tau-1}) g(\widetilde{\boldsymbol{x}}_\tau; \boldsymbol{\theta}_{\widetilde{i}_\tau, \tau-1})^\mathsf{T}/m$, and we also have $\bar{\boldsymbol{\Sigma}}_{t-1} = \lambda \mathbf{I} + \sum_{\tau \in [t-1]} g(\boldsymbol{x}_\tau; \boldsymbol{\theta}_{\tau-1}) g(\boldsymbol{x}_\tau; \boldsymbol{\theta}_{\tau-1})^\mathsf{T}/m$.*

**Proof.** Since the only difference between term $I_1$ and term $I_2$ is the gradients w.r.t. different arms, we begin with term $I_1 = \langle g(\widetilde{\boldsymbol{x}}_t; \boldsymbol{\theta}_{t-1}), \; \widetilde{\boldsymbol{\theta}}_{t-1} - \boldsymbol{\theta}_{t-1} \rangle$, and the results can be readily generalized to term $I_2$. Here, recall that we have $\|\boldsymbol{\theta}^* - \widetilde{\boldsymbol{\theta}}_t\|_{\widetilde{\boldsymbol{\Sigma}}_{t-1}} \leq \widetilde{\gamma}_t/\sqrt{m}$ based on Lemma C.10, as well as $\|\boldsymbol{\theta}_t - \boldsymbol{\theta}_0\|_2, \|\widetilde{\boldsymbol{\theta}}_t - \boldsymbol{\theta}_0\|_2 \leq \mathcal{O}(\sqrt{t/m\lambda})$ based on lemma G.3.

**Simplifying the notation.** For the following proof, for the sake of notation simplicity, we directly use $\widetilde{\boldsymbol{\Theta}}^{(J)}, \boldsymbol{\Theta}^{(J)}$ to respectively represent $\widetilde{\boldsymbol{\Theta}}_{\widetilde{i}_t, t-1}^{(J)}, \boldsymbol{\Theta}_{\widetilde{i}_t, t-1}^{(J)}$, which are the gradient descent based parameters associated with the arm $\widetilde{\boldsymbol{x}}_t$. On the other hand, for the terms that belong to arm $\widetilde{\boldsymbol{x}}_t = \boldsymbol{x}_{\widetilde{i}_t, t}$, we use $\boldsymbol{\theta}_{\widetilde{i}_t}, \Sigma_{\widetilde{i}_t}$ and $\boldsymbol{b}_{\widetilde{i}_t}$ to separately represent $\boldsymbol{\theta}_{\widetilde{i}_t, t-1}, \Sigma_{\widetilde{i}_t, t-1}$ and $\boldsymbol{b}_{\widetilde{i}_t, t-1}$ by omitted the subscript of time step $t - 1$ to simplify the notation.

Afterwards, it can further lead to

$$I_1 = \left| \langle g(\widetilde{\boldsymbol{x}}_t; \boldsymbol{\theta}_{\widetilde{i}_t}), \; \boldsymbol{\theta}_{\widetilde{i}_t} - \widetilde{\boldsymbol{\theta}}_{\widetilde{i}_t} \rangle \right|$$

$$\leq \left| \langle g(\widetilde{\boldsymbol{x}}_t; \boldsymbol{\theta}_{\widetilde{i}_t}), \; \boldsymbol{\theta}_{\widetilde{i}_t} - \boldsymbol{\Theta}^{(J)} \rangle \right| + \left| \langle g(\widetilde{\boldsymbol{x}}_t; \boldsymbol{\theta}_{\widetilde{i}_t}), \; \widetilde{\boldsymbol{\Theta}}^{(J)} - \widetilde{\boldsymbol{\theta}}_{\widetilde{i}_t} \rangle \right| + \underbrace{\left| \langle g(\widetilde{\boldsymbol{x}}_t; \boldsymbol{\theta}_{\widetilde{i}_t}), \; \widetilde{\boldsymbol{\Theta}}^{(J)} - \boldsymbol{\Theta}^{(J)} \rangle \right|}_{I_{1.1}}$$

$$\leq \mathcal{O}(m^{-2/3} \log(m) L^{7/2} t^{5/3} \lambda^{-5/3}(1 + \sqrt{t/\lambda})) + \underbrace{\left| \langle g(\widetilde{\boldsymbol{x}}_t; \boldsymbol{\theta}_{\widetilde{i}_t}), \; \boldsymbol{\Theta}^{(J)} - \widetilde{\boldsymbol{\Theta}}^{(J)} \rangle \right|}_{I_{1.1}},$$

where the first inequality is due to triangular inequality, and the last inequality is by applying Lemma C.4 in [86], with the optimization problem being (C.12) which bounds the difference between gradient-based parameters and the auxiliary sequence. Here, term $I_{1.1}$ measures the distance between the auxiliary sequence trained with corrupted records, and that trained by imaginary corruption-free records.

Next, recall that with randomly initialized network parameters $\boldsymbol{\theta}_0$, we can formulate the least square parameters as $(\boldsymbol{\Sigma}_{\widetilde{i}_t}^{(0)})^{-1} \boldsymbol{b}_{\widetilde{i}_t}^{(0)}/\sqrt{m}$, while the least square parameters trained by corruption-free rewards are analogously denoted by $(\boldsymbol{\Sigma}_{\widetilde{i}_t}^{(0)})^{-1} \widetilde{\boldsymbol{b}}_{\widetilde{i}_t}^{(0)}/\sqrt{m}$. In this case, the term $I_{1.1}$ can be alternatively transformed to

$$I_{1.1} = \left| \left\langle g(\widetilde{\boldsymbol{x}}_t; \boldsymbol{\theta}_{\widetilde{i}_t}), \; \boldsymbol{\Theta}^{(J)} - \widetilde{\boldsymbol{\Theta}}^{(J)} \right\rangle \right|$$

$$\leq \left| \left\langle g(\widetilde{\boldsymbol{x}}_t; \boldsymbol{\theta}_{\widetilde{i}_t}), \; (\boldsymbol{\Theta}^{(J)} - \boldsymbol{\theta}_0 - (\boldsymbol{\Sigma}_{\widetilde{i}_t}^{(0)})^{-1} \boldsymbol{b}_{\widetilde{i}_t}^{(0)}/\sqrt{m}) - (\widetilde{\boldsymbol{\Theta}}^{(J)} - \boldsymbol{\theta}_0 - (\boldsymbol{\Sigma}_{\widetilde{i}_t}^{(0)})^{-1} \widetilde{\boldsymbol{b}}_{\widetilde{i}_t}^{(0)}/\sqrt{m}) \right\rangle \right|$$

$$\leq \left| \left\langle g(\widetilde{\boldsymbol{x}}_t; \boldsymbol{\theta}_{\widetilde{i}_t}), \; \boldsymbol{\Theta}^{(J)} - \boldsymbol{\theta}_0 - (\boldsymbol{\Sigma}_{\widetilde{i}_t}^{(0)})^{-1} \boldsymbol{b}_{\widetilde{i}_t}^{(0)}/\sqrt{m} \right\rangle \right|$$

$$+ \left| \left\langle g(\widetilde{\boldsymbol{x}}_t; \boldsymbol{\theta}_{\widetilde{i}_t}), \; \widetilde{\boldsymbol{\Theta}}^{(J)} - \boldsymbol{\theta}_0 - (\boldsymbol{\Sigma}_{\widetilde{i}_t}^{(0)})^{-1} \widetilde{\boldsymbol{b}}_{\widetilde{i}_t}^{(0)}/\sqrt{m} \right\rangle \right|$$

which further leads to

$$
\begin{aligned}
I_{1.1} \leq & \left| \left\langle g(\widetilde{\boldsymbol{x}}_t; \boldsymbol{\theta}_{\widetilde{i}_t}), \ \boldsymbol{\Theta}^{(J)} - \boldsymbol{\theta}_0 - (\boldsymbol{\Sigma}_{\widetilde{i}_t}^{(0)})^{-1} \boldsymbol{b}_{\widetilde{i}_t}^{(0)} / \sqrt{m} \right\rangle \right| \\
& + \left| \left\langle g(\widetilde{\boldsymbol{x}}_t; \boldsymbol{\theta}_{\widetilde{i}_t}), \ \widetilde{\boldsymbol{\Theta}}^{(J)} - \boldsymbol{\theta}_0 - (\boldsymbol{\Sigma}_{\widetilde{i}_t}^{(0)})^{-1} \widetilde{\boldsymbol{b}}_{\widetilde{i}_t}^{(0)} / \sqrt{m} \right\rangle \right| \\
& + \underbrace{\left| \left\langle g(\widetilde{\boldsymbol{x}}_t; \boldsymbol{\theta}_{\widetilde{i}_t}), \ (\boldsymbol{\Sigma}_{\widetilde{i}_t}^{(0)})^{-1} ( \sum_{\tau \in [t-1]} w_{\widetilde{i}_t, t}^{(\tau)} \cdot g(\boldsymbol{x}_\tau; \boldsymbol{\theta}_0) \cdot c_\tau) / m \right\rangle \right|}_{I_{1.2}}
\end{aligned}
\tag{C.17}
$$

where the inequality is due to the definition of gradient-based regression parameters and triangular inequality. It is obvious that for the third term on the RHS, the only thing we have control on is $\boldsymbol{\theta}_{\widetilde{i}_t}$. When trying to bound the first two terms on the RHS for the arm $\widetilde{\boldsymbol{x}}_t$, we can first apply Holder's inequality with $\boldsymbol{\Sigma}'_{t-1} = \lambda \mathbf{I} + \sum_{\tau \in [t-1]} g(\widetilde{\boldsymbol{x}}_\tau; \boldsymbol{\theta}_{\widetilde{i}_\tau, \tau-1}) g(\widetilde{\boldsymbol{x}}_\tau; \boldsymbol{\theta}_{\widetilde{i}_\tau, \tau-1})^\intercal / m$. Note that for the purpose of analysis and different from previous ones defined in (C.11), the new matrix $(\boldsymbol{\Sigma}'_{t-1})$ contains gradients of the sequence of arms $\{\widetilde{\boldsymbol{x}}_\tau\}_{\tau \in [t-1]}$ chosen by the corruption-free model $f(\cdot; \widetilde{\boldsymbol{\theta}}_{\tau-1}), \tau \in [t-1]$, with the parameters $\widetilde{\boldsymbol{\theta}}_{\tau-1}, \tau \in [t-1]$.

In this case, we use the Holder's inequality to the first term on the RHS of (C.17) as

$$
\begin{aligned}
& \left\langle g(\widetilde{\boldsymbol{x}}_t; \boldsymbol{\theta}_{\widetilde{i}_t}), \ \boldsymbol{\Theta}^{(J)} - \boldsymbol{\theta}_0 - (\boldsymbol{\Sigma}_{\widetilde{i}_t}^{(0)})^{-1} \boldsymbol{b}_{\widetilde{i}_t}^{(0)} / \sqrt{m} \right\rangle \\
& \leq \left\| g(\widetilde{\boldsymbol{x}}_t; \boldsymbol{\theta}_{\widetilde{i}_t}) / \sqrt{m} \right\|_{(\boldsymbol{\Sigma}'_{t-1})^{-1}} \cdot \sqrt{m} \cdot \left\| \boldsymbol{\Theta}^{(J)} - \boldsymbol{\theta}_0 - (\boldsymbol{\Sigma}_{\widetilde{i}_t}^{(0)})^{-1} \boldsymbol{b}_{\widetilde{i}_t}^{(0)} / \sqrt{m} \right\|_{(\boldsymbol{\Sigma}'_{t-1})} \\
& \leq \left\| g(\widetilde{\boldsymbol{x}}_t; \boldsymbol{\theta}_{\widetilde{i}_t}) / \sqrt{m} \right\|_{(\boldsymbol{\Sigma}'_{t-1})^{-1}} \cdot \sqrt{m} \cdot \left\| \boldsymbol{\Theta}^{(J)} - \boldsymbol{\theta}_0 - (\boldsymbol{\Sigma}_{\widetilde{i}_t}^{(0)})^{-1} \boldsymbol{b}_{\widetilde{i}_t}^{(0)} / \sqrt{m} \right\|_2 \cdot \left\| (\boldsymbol{\Sigma}'_{t-1}) \right\|_2 \\
& \leq \left\| g(\widetilde{\boldsymbol{x}}_t; \boldsymbol{\theta}_{\widetilde{i}_t}) / \sqrt{m} \right\|_{(\boldsymbol{\Sigma}'_{t-1})^{-1}} \cdot \mathcal{O}(\lambda + tL) \cdot \left\| \sqrt{m} (\boldsymbol{\Theta}^{(J)} - \boldsymbol{\theta}_0) - (\boldsymbol{\Sigma}_{\widetilde{i}_t}^{(0)})^{-1} \boldsymbol{b}_{\widetilde{i}_t}^{(0)} \right\|_2 \\
& \leq \left\| g(\widetilde{\boldsymbol{x}}_t; \boldsymbol{\theta}_{\widetilde{i}_t}) / \sqrt{m} \right\|_{(\boldsymbol{\Sigma}'_{t-1})^{-1}} \cdot \mathcal{O}(\sqrt{\lambda + tL}) \\
& \qquad \cdot \mathcal{O}\left( (1 - \eta m \lambda)^{J/2} \sqrt{t/\lambda} + m^{-1/6} \sqrt{\log(m)} L^{7/2} t^{5/3} \lambda^{-5/3} (1 + \sqrt{t/\lambda}) \right) \\
& \leq \left\| g(\widetilde{\boldsymbol{x}}_t; \boldsymbol{\theta}_{\widetilde{i}_t}) / \sqrt{m} \right\|_{(\boldsymbol{\Sigma}'_{t-1})^{-1}} \cdot \mathcal{O}(\sqrt{\lambda} S)
\end{aligned}
$$

where the first two inequalities are by the Holder's inequality and Cauchy-Schwartz inequality. The third inequality is by Lemma G.6. The fourth inequality is by applying Lemma D.1, with the optimization problem being (C.12) bounding the difference between gradient-based parameters and the auxiliary sequence. Finally, with $w_t \leq 1$ and the conditions in Theorem 5.6, applying the conclusion from Remark 4.7 in [86] will give the last inequality. Following a similar approach can also lead to the identical upper bound for the second term on the RHS of (C.17).

**Bounding Term $I_{1.2}$ Based on Arm Weights.** Next, we proceed to bound term $I_{1.2}$. Recall that we need to derive the upper bound for the following term

$$
\begin{aligned}
I_{1.2} = & \left| \left\langle g(\widetilde{\boldsymbol{x}}_t; \boldsymbol{\theta}_{\widetilde{i}_t}), \ (\boldsymbol{\Sigma}_{\widetilde{i}_t}^{(0)})^{-1} ( \sum_{\tau \in [t-1]} w_{\widetilde{i}_t, t}^{(\tau)} \cdot g(\boldsymbol{x}_\tau; \boldsymbol{\theta}_0) \cdot c_\tau) / m \right\rangle \right| \\
& \leq \left| \sum_{\tau \in [t-1]} \left\langle g(\widetilde{\boldsymbol{x}}_t; \boldsymbol{\theta}_{\widetilde{i}_t}), \ (\boldsymbol{\Sigma}_{\widetilde{i}_t}^{(0)})^{-1} \cdot w_{\widetilde{i}_t, t}^{(\tau)} \cdot g(\boldsymbol{x}_\tau; \boldsymbol{\theta}_0) \cdot c_\tau / m \right\rangle \right| \\
& \leq \left| \sum_{\tau \in [t-1]} \left\langle g(\widetilde{\boldsymbol{x}}_t; \boldsymbol{\theta}_{\widetilde{i}_t}), \ w_{\widetilde{i}_t, t}^{(\tau)} \cdot (\boldsymbol{\Sigma}_{\widetilde{i}_t}^{(0)})^{-1} g(\boldsymbol{x}_\tau; \boldsymbol{\theta}_{\tau-1}) c_\tau / m \right\rangle \right| \\
& \quad + \left| \sum_{\tau \in [t-1]} \left\langle g(\widetilde{\boldsymbol{x}}_t; \boldsymbol{\theta}_{\widetilde{i}_t}), \ w_{\widetilde{i}_t, t}^{(\tau)} \cdot (\boldsymbol{\Sigma}_{\widetilde{i}_t}^{(0)})^{-1} (g(\boldsymbol{x}_\tau; \boldsymbol{\theta}_0) - g(\boldsymbol{x}_\tau; \boldsymbol{\theta}_{\tau-1})) c_\tau / m \right\rangle \right| \\
& \leq \left| \sum_{\tau \in [t-1]} \left\langle g(\widetilde{\boldsymbol{x}}_t; \boldsymbol{\theta}_{\widetilde{i}_t}), \ w_{\widetilde{i}_t, t}^{(\tau)} \cdot (\boldsymbol{\Sigma}_{\widetilde{i}_t}^{(0)})^{-1} g(\boldsymbol{x}_\tau; \boldsymbol{\theta}_{\tau-1}) \cdot c_\tau / m \right\rangle \right| + \mathcal{O}(C m^{-1/6} \sqrt{\log(m)} t^{1/6} \lambda^{-7/6} L^4)
\end{aligned}
$$

$$\leq \left| \sum_{\tau \in [t-1]} \left\langle g(\widetilde{\boldsymbol{x}}_t; \boldsymbol{\theta}_{\widetilde{i}_t}), \ w_{\widetilde{i}_t,t}^{(\tau)} \cdot (\boldsymbol{\Sigma}_{\widetilde{i}_t})^{-1} g(\boldsymbol{x}_\tau; \boldsymbol{\theta}_{\tau-1}) \cdot c_\tau / m \right\rangle \right| + \mathcal{O}(Cm^{-1/6}\sqrt{\log(m)}t^{1/6}\lambda^{-7/6}L^4)$$

$$+ \left| \sum_{\tau \in [t-1]} \left\langle g(\widetilde{\boldsymbol{x}}_t; \boldsymbol{\theta}_{\widetilde{i}_t}), \ w_{\widetilde{i}_t,t}^{(\tau)} \cdot ((\boldsymbol{\Sigma}_{\widetilde{i}_t}^{(0)})^{-1} - (\boldsymbol{\Sigma}_{\widetilde{i}_t})^{-1}) g(\boldsymbol{x}_\tau; \boldsymbol{\theta}_{\tau-1}) \cdot c_\tau / m \right\rangle \right|$$

$$\leq \underbrace{\left| \sum_{\tau \in [t-1]} \left\langle g(\widetilde{\boldsymbol{x}}_t; \boldsymbol{\theta}_{\widetilde{i}_t}), \ w_{\widetilde{i}_t,t}^{(\tau)} \cdot (\boldsymbol{\Sigma}_{\widetilde{i}_t})^{-1} g(\boldsymbol{x}_\tau; \boldsymbol{\theta}_{\tau-1}) \cdot c_\tau / m \right\rangle \right|}_{I_{1.3}} + \mathcal{O}(Cm^{-1/6}\sqrt{\log(m)}t^{7/6}\lambda^{-1/6}L^5)$$

$$+ \mathcal{O}(Cm^{-1/6}\sqrt{\log(m)}t^{1/6}\lambda^{-7/6}L^4)$$

where the second and fourth inequality is due to triangular inequality. The third inequality is by Lemma G.4, and the last inequality is due to Lemma G.6.

In particular, we aim to train separate neural models $f(\cdot; \boldsymbol{\theta}_{i,t})$ for each candidate arm $\boldsymbol{x}_{i,t} \in \mathcal{X}_t$, such that the term $I_{1.2}$ can be minimized. Recall that we denote $\bar{\boldsymbol{\Sigma}}_{t-1} = \lambda \mathbf{I} + \sum_{\tau \in [t-1]} g(\boldsymbol{x}_\tau; \boldsymbol{\theta}_{\tau-1}) g(\boldsymbol{x}_\tau; \boldsymbol{\theta}_{\tau-1})^\intercal / m$ as the "vanilla" gradient covariance matrix without the arm weights, in terms of the potentially corrupted network parameters $\boldsymbol{\theta}_{\tau-1}$.

In this case, with $w_{\widetilde{i}_t,t}^{(\tau)}$ referring to the weight of arm $\widetilde{\boldsymbol{x}}_t$, the above formulation of term $I_{1.3}$ can be further transformed into

$$I_{1.3} \leq \left| \sum_{\tau \in [t-1]} \frac{w_{\widetilde{i}_t,t}^{(\tau)} \cdot c_\tau}{m} \cdot \left\langle g(\widetilde{\boldsymbol{x}}_t; \boldsymbol{\theta}_{\widetilde{i}_t}), \ (\boldsymbol{\Sigma}_{\widetilde{i}_t})^{-1} \cdot g(\boldsymbol{x}_\tau; \boldsymbol{\theta}_{\tau-1}) \right\rangle \right|$$

$$\leq \left| \sum_{\tau \in [t-1]} w_{\widetilde{i}_t,t}^{(\tau)} \cdot c_\tau \cdot \left\| g(\widetilde{\boldsymbol{x}}_t; \boldsymbol{\theta}_{\widetilde{i}_t})/\sqrt{m} \right\|_{(\boldsymbol{\Sigma}_{\widetilde{i}_t})^{-1}} \cdot \left\| (\boldsymbol{\Sigma}_{\widetilde{i}_t})^{-1} \cdot g(\boldsymbol{x}_\tau; \boldsymbol{\theta}_{\tau-1})/\sqrt{m} \right\|_{\boldsymbol{\Sigma}_{\widetilde{i}_t}} \right|$$

$$\leq \left| \sum_{\tau \in [t-1]} w_{\widetilde{i}_t,t}^{(\tau)} \cdot c_\tau \cdot \left\| g(\widetilde{\boldsymbol{x}}_t; \boldsymbol{\theta}_{\widetilde{i}_t})/\sqrt{m} \right\|_{(\boldsymbol{\Sigma}_{\widetilde{i}_t})^{-1}} \cdot \left\| g(\boldsymbol{x}_\tau; \boldsymbol{\theta}_{\tau-1})/\sqrt{m} \right\|_{(\boldsymbol{\Sigma}_{\widetilde{i}_t})^{-1}} \right|$$

$$\leq \left| \sum_{\tau \in [t-1]} w_{\widetilde{i}_t,t}^{(\tau)} \cdot c_\tau \cdot \left\| g(\widetilde{\boldsymbol{x}}_t; \boldsymbol{\theta}_{\widetilde{i}_t})/\sqrt{m} \right\|_{(\bar{\boldsymbol{\Sigma}}_{t-1}^{(\kappa)})^{-1}} \cdot \left\| g(\boldsymbol{x}_\tau; \boldsymbol{\theta}_{\tau-1})/\sqrt{m} \right\|_{(\bar{\boldsymbol{\Sigma}}_{\tau-1}^{(\kappa)})^{-1}} \right|$$

$$\leq \alpha C \cdot \min_{\boldsymbol{x} \in \mathcal{X}_t} \left\| g(\boldsymbol{x}; \boldsymbol{\theta}_{t-1})/\sqrt{m} \right\|_{(\bar{\boldsymbol{\Sigma}}_{t-1})^{-1}}^2$$

$$\leq \alpha C \cdot \left\| g(\boldsymbol{x}_t; \boldsymbol{\theta}_{t-1})/\sqrt{m} \right\|_{(\bar{\boldsymbol{\Sigma}}_{t-1})^{-1}}^2$$

where the first inequality follows from applying Holder's inequality, and the second and third inequalities are obtained by using Lemma G.8, along with the fact that $\boldsymbol{\Sigma}_{\widetilde{i}_t} \succeq \bar{\boldsymbol{\Sigma}}_{t-1}^{(\kappa)} \succeq \bar{\boldsymbol{\Sigma}}_{\tau-1}^{(\kappa)}$ by definition. The last two inequalities are derived from the definition of the arm weight $w_{\widetilde{i}_t,t}^{(\tau)}$ and the corruption level $C = \sum_{t \in [T]} |c_t|$. Recall that for each arm $\boldsymbol{x}_{i,t} \in \mathcal{X}_t$, we define its weight as $w_{i,t}^{(\tau)} = \min\left\{ 1, \frac{\alpha \cdot \min_{\boldsymbol{x} \in \mathcal{X}_t} \|g(\boldsymbol{x};\boldsymbol{\theta}_{t-1})/\sqrt{m}\|_{\bar{\boldsymbol{\Sigma}}_{t-1}^{-1}}^2}{g_\tau \cdot \|g(\boldsymbol{x}_{i,t};\boldsymbol{\theta}_{t-1})/\sqrt{m}\|_{(\bar{\boldsymbol{\Sigma}}_{t-1}^{(\kappa)})^{-1}}} \right\}$, where $\alpha > 0$ is the tunable parameter. As a result, we will have the upper bound for $I_{1.2}$, being

$$I_{1.2} \leq \mathcal{O}(\alpha C) \cdot \left\| g(\boldsymbol{x}_t; \boldsymbol{\theta}_{t-1})/\sqrt{m} \right\|_{(\bar{\boldsymbol{\Sigma}}_{t-1})^{-1}}^2 + \mathcal{O}(Cm^{-1/6}\sqrt{\log(m)}t^{7/6}\lambda^{-1/6}L^5)$$

$$+ \mathcal{O}(Cm^{-1/6}\sqrt{\log(m)}t^{1/6}\lambda^{-7/6}L^4)$$

In this case, since the weights are lower bounded by $\kappa^2$, the only way to ensure this is by adjusting the tunable parameter $\alpha > 0$ accordingly. We remind that this does not require the learner to have a global view of the minimum fraction value $\beta$. In practice, the learner can adjust the $\alpha$ values in each round to ensure that the round-wise minimum weight value $w_t^{\mathsf{min}} = \min\{w_{i,t}^{(\tau)}\}_{i \in [K]} = \kappa^2 < 1$, for

all $t \in [T]$ and $\tau \in [t-1]$, by tuning the parameter $\alpha$. By summing up all the results, we obtain the single-round bound for term $I_1$, which leads to

$$I_1 \leq \mathcal{O}(\alpha C) \cdot \left\| g(\boldsymbol{x}_t; \boldsymbol{\theta}_{t-1})/\sqrt{m} \right\|^2_{(\bar{\boldsymbol{\Sigma}}_{t-1})^{-1}} + \mathcal{O}(\sqrt{\lambda}S) \cdot \left\| g(\widetilde{\boldsymbol{x}}_t; \boldsymbol{\theta}_{\tilde{i}_t, t-1})/\sqrt{m} \right\|_{(\boldsymbol{\Sigma}'_{t-1})^{-1}}$$
$$+ \mathcal{O}(m^{-2/3} \log(m) L^{7/2} t^{5/3} \lambda^{-5/3} (1 + \sqrt{t/\lambda})) + \mathcal{O}(C m^{-1/6} \sqrt{\log(m)} t^{7/6} \lambda^{-1/6} L^5)$$
$$+ \mathcal{O}(C m^{-1/6} \sqrt{\log(m)} t^{1/6} \lambda^{-7/6} L^4).$$

(C.18)

$\square$

### C.7.2 Bounding the error term $I_2$

Similarly, for the term $I_2$ related to the chosen arm $\boldsymbol{x}_t \in \mathcal{X}_t$, we can follow the below procedure to obtain a comparable bound as for term $I_1$.

**Corollary C.8.** *Suppose the imaginary corruption-free neural network $f(\cdot; \widetilde{\boldsymbol{\theta}}_{i,t-1})$ for arm $\boldsymbol{x}_{i,t} \in \mathcal{X}_t$ has been trained on corruption-free rewards $\{\boldsymbol{x}_\tau, \widetilde{r}_\tau\}_{\tau \in [t-1]}$. $f(\cdot)$ is an L-layer FC network with width $m$ that satisfy the conditions in Theorem 5.6. Then, for the chosen arm $\boldsymbol{x}_t \in \mathcal{X}_t$, with the probability at least $1 - \delta$, we will have*

$$I_2 \leq \mathcal{O}(\alpha C) \cdot \left\| g(\boldsymbol{x}_t; \boldsymbol{\theta}_{t-1})/\sqrt{m} \right\|^2_{(\bar{\boldsymbol{\Sigma}}_{t-1})^{-1}} + \mathcal{O}(\sqrt{\lambda}S) \cdot \left\| g(\boldsymbol{x}_t; \boldsymbol{\theta}_{t-1})/\sqrt{m} \right\|_{(\bar{\boldsymbol{\Sigma}}_{t-1})^{-1}}$$
$$+ \mathcal{O}(m^{-2/3} \log(m) L^{7/2} t^{5/3} \lambda^{-5/3} (1 + \sqrt{t/\lambda})) + \mathcal{O}(C m^{-1/6} \sqrt{\log(m)} t^{7/6} \lambda^{-1/6} L^5)$$
$$+ \mathcal{O}(C m^{-1/6} \sqrt{\log(m)} t^{1/6} \lambda^{-7/6} L^4).$$

*where $\boldsymbol{\theta}_{t-1}$ are the trained parameters associated to chosen arm $\boldsymbol{x}_t \in \mathcal{X}_t$ in round t, and $\boldsymbol{\theta}_{\tilde{i}_t, t-1}$ are the trained parameters of arm $\widetilde{\boldsymbol{x}}_t$, along with the corresponding weight-free gradient covariance matrix $\bar{\boldsymbol{\Sigma}}_{t-1} = \lambda \mathbf{I} + \sum_{\tau \in [t-1]} g(\boldsymbol{x}_\tau; \boldsymbol{\theta}_{\tau-1}) g(\boldsymbol{x}_\tau; \boldsymbol{\theta}_{\tau-1})^{\mathsf{T}}/m$.*

**Proof.** The proof of this corollary follows an analogous procedure as in Lemma C.7.

**Simplifying the notation.** For notation simplicity, we apply $\widetilde{\boldsymbol{\Theta}}^{(J)}$ and $\boldsymbol{\Theta}^{(J)}$ to represent the gradient descent-based parameters associated with the arm $\boldsymbol{x}_t$. For terms specific to the arm $\boldsymbol{x}_t = \boldsymbol{x}_{i_t, t}$, we simplify notation by using $\boldsymbol{\theta}_{t-1}, \Sigma_{t-1}$, and $\boldsymbol{b}_{t-1}$ to denote $\boldsymbol{\theta}_{i_t, t-1}, \Sigma_{i_t, t-1}$, and $\boldsymbol{b}_{i_t, t-1}$, respectively.

Then, with the simplified notation, it leads to

$$I_2 = \left| \langle g(\boldsymbol{x}_t; \boldsymbol{\theta}_{t-1}), \boldsymbol{\theta}_{t-1} - \widetilde{\boldsymbol{\theta}}_{t-1} \rangle \right|$$
$$\leq \left| \langle g(\boldsymbol{x}_t; \boldsymbol{\theta}_{t-1}), \boldsymbol{\theta}_{t-1} - \boldsymbol{\Theta}^{(J)} \rangle \right| + \left| \langle g(\boldsymbol{x}_t; \boldsymbol{\theta}_{t-1}), \widetilde{\boldsymbol{\Theta}}^{(J)} - \widetilde{\boldsymbol{\theta}}_{t-1} \rangle \right| + \underbrace{\left| \langle g(\boldsymbol{x}_t; \boldsymbol{\theta}_{t-1}), \widetilde{\boldsymbol{\Theta}}^{(J)} - \boldsymbol{\Theta}^{(J)} \rangle \right|}_{I_{2.1}}$$
$$\leq \mathcal{O}(m^{-2/3} \log(m) L^{7/2} t^{5/3} \lambda^{-5/3} (1 + \sqrt{t/\lambda})) + \underbrace{\left| \langle g(\boldsymbol{x}_t; \boldsymbol{\theta}_{t-1}), \boldsymbol{\Theta}^{(J)} - \widetilde{\boldsymbol{\Theta}}^{(J)} \rangle \right|}_{I_{2.1}}$$

where the first inequality is due to triangular inequality, and the last inequality is by applying Lemma C.4 in [86], with the optimization problem being (C.12) bounding the difference between gradient-based parameters and the auxiliary sequence.

Next, recall that with randomly initialized network parameters $\boldsymbol{\theta}_0$, we can formulate the least square parameters as $(\boldsymbol{\Sigma}^{(0)}_{t-1})^{-1} \boldsymbol{b}^{(0)}_{t-1}/\sqrt{m}$, while the least square parameters trained by corruption-free

rewards are $(\boldsymbol{\Sigma}_{t-1}^{(0)})^{-1}\widetilde{\boldsymbol{b}}_{t-1}^{(0)}/\sqrt{m}$. In this case, the term $I_{2.1}$ can be alternatively transformed to

$$
\begin{aligned}
I_{2.1} &= \left| \left\langle g(\boldsymbol{x}_t; \boldsymbol{\theta}_{t-1}),\ \boldsymbol{\Theta}^{(J)} - \widetilde{\boldsymbol{\Theta}}^{(J)} \right\rangle \right| \\
&\leq \left| \left\langle g(\boldsymbol{x}_t; \boldsymbol{\theta}_{t-1}),\ \boldsymbol{\Theta}^{(J)} - \boldsymbol{\theta}_0 - (\boldsymbol{\Sigma}_{t-1}^{(0)})^{-1}\boldsymbol{b}_{t-1}^{(0)}/\sqrt{m} \right\rangle \right| \\
&\quad + \left| \left\langle g(\boldsymbol{x}_t; \boldsymbol{\theta}_{t-1}),\ \widetilde{\boldsymbol{\Theta}}^{(J)} - \boldsymbol{\theta}_0 - (\boldsymbol{\Sigma}_{t-1}^{(0)})^{-1}\widetilde{\boldsymbol{b}}_{t-1}^{(0)}/\sqrt{m} \right\rangle \right| \\
&\quad + \underbrace{\left| \left\langle g(\boldsymbol{x}_t; \boldsymbol{\theta}_{t-1}),\ (\boldsymbol{\Sigma}_{t-1}^{(0)})^{-1}(\sum_{\tau \in [t-1]} w_t^{(\tau)} g(\boldsymbol{x}_\tau; \boldsymbol{\theta}_0) \cdot c_\tau)/m \right\rangle \right|}_{I_{2.2}}
\end{aligned}
\tag{C.19}
$$

where the inequality is due to the definition of gradient-based regression parameters.

When trying to bound the first two terms on the RHS, we first apply Holder's inequality with $\bar{\boldsymbol{\Sigma}}_{t-1} = \lambda \mathbf{I} + \sum_{\tau \in [t-1]} g(\widetilde{\boldsymbol{x}}_\tau; \boldsymbol{\theta}_{\tau-1}) g(\widetilde{\boldsymbol{x}}_\tau; \boldsymbol{\theta}_{\tau-1})^\mathsf{T}/m$, and it further leads to

$$
\begin{aligned}
&\left\langle g(\boldsymbol{x}_t; \boldsymbol{\theta}_{t-1}),\ \boldsymbol{\Theta}^{(J)} - \boldsymbol{\theta}_0 - (\boldsymbol{\Sigma}_{t-1}^{(0)})^{-1}\boldsymbol{b}_{t-1}^{(0)}/\sqrt{m} \right\rangle \\
&\leq \left\| g(\boldsymbol{x}_t; \boldsymbol{\theta}_{t-1})/\sqrt{m} \right\|_{(\bar{\boldsymbol{\Sigma}}_{t-1})^{-1}} \cdot \sqrt{m} \cdot \left\| \boldsymbol{\Theta}^{(J)} - \boldsymbol{\theta}_0 - (\boldsymbol{\Sigma}_{t-1}^{(0)})^{-1}\boldsymbol{b}_{t-1}^{(0)}/\sqrt{m} \right\|_{(\bar{\boldsymbol{\Sigma}}_{t-1})} \\
&\leq \left\| g(\boldsymbol{x}_t; \boldsymbol{\theta}_{t-1})/\sqrt{m} \right\|_{(\bar{\boldsymbol{\Sigma}}_{t-1})^{-1}} \cdot \sqrt{m} \cdot \left\| \boldsymbol{\Theta}^{(J)} - \boldsymbol{\theta}_0 - (\boldsymbol{\Sigma}_{t-1}^{(0)})^{-1}\boldsymbol{b}_{t-1}^{(0)}/\sqrt{m} \right\|_2 \cdot \left\| (\bar{\boldsymbol{\Sigma}}_{t-1}) \right\|_2 \\
&\leq \left\| g(\boldsymbol{x}_t; \boldsymbol{\theta}_{t-1})/\sqrt{m} \right\|_{(\bar{\boldsymbol{\Sigma}}_{t-1})^{-1}} \cdot \mathcal{O}(\lambda + tL) \cdot \left\| \sqrt{m}(\boldsymbol{\Theta}^{(J)} - \boldsymbol{\theta}_0) - (\boldsymbol{\Sigma}_{t-1}^{(0)})^{-1}\boldsymbol{b}_{t-1}^{(0)} \right\|_2 \\
&\leq \left\| g(\boldsymbol{x}_t; \boldsymbol{\theta}_{t-1})/\sqrt{m} \right\|_{(\bar{\boldsymbol{\Sigma}}_{t-1})^{-1}} \cdot \mathcal{O}(\sqrt{\lambda + tL}) \\
&\qquad \cdot \mathcal{O}\left( (1 - \eta m \lambda)^{J/2}\sqrt{t/\lambda} + m^{-1/6}\sqrt{\log(m)}L^{7/2}t^{5/3}\lambda^{-5/3}(1 + \sqrt{t/\lambda}) \right) \\
&\leq \left\| g(\boldsymbol{x}_t; \boldsymbol{\theta}_{t-1})/\sqrt{m} \right\|_{(\bar{\boldsymbol{\Sigma}}_{t-1})^{-1}} \cdot \mathcal{O}(\sqrt{\lambda}S)
\end{aligned}
$$

where the first two inequalities is by Holder's inequality and Cauchy-Schwartz inequality. The third inequality is by Lemma G.6. The fourth inequality is by applying Lemma D.1, with the optimization problem being (C.12), in terms of the difference between gradient-based parameters and the auxiliary sequence. Finally, with $w_t \leq 1$ and the conditions in Theorem 5.6, applying the conclusion from Remark 4.7 in [86] will give the last inequality. Following a similar approach can also lead to the upper bound for the second term on the RHS of (C.19).

**Bounding Term $I_{2.2}$ Based on Arm Weights.** Next, we need to bound term $I_{2.2}$. Recall that we want to have an upper bound for the following term

$$
\begin{aligned}
I_{2.2} &= \left| \left\langle g(\boldsymbol{x}_t; \boldsymbol{\theta}_{t-1}),\ (\boldsymbol{\Sigma}_{t-1}^{(0)})^{-1}(\sum_{\tau \in [t-1]} g(\boldsymbol{x}_\tau; \boldsymbol{\theta}_0) \cdot c_\tau)/m \right\rangle \right| \\
&\leq \left| \sum_{\tau \in [t-1]} \left\langle g(\boldsymbol{x}_t; \boldsymbol{\theta}_{t-1}),\ (\boldsymbol{\Sigma}_{t-1}^{(0)})^{-1}g(\boldsymbol{x}_\tau; \boldsymbol{\theta}_{\tau-1}) \cdot c_\tau/m \right\rangle \right| + \mathcal{O}(Cm^{-1/6}\sqrt{\log(m)}t^{1/6}\lambda^{-7/6}L^4) \\
&\leq \left| \sum_{\tau \in [t-1]} \left\langle g(\boldsymbol{x}_t; \boldsymbol{\theta}_{t-1}),\ (\boldsymbol{\Sigma}_{t-1})^{-1}g(\boldsymbol{x}_\tau; \boldsymbol{\theta}_{\tau-1}) \cdot c_\tau/m \right\rangle \right| + \mathcal{O}(Cm^{-1/6}\sqrt{\log(m)}t^{1/6}\lambda^{-7/6}L^4) \\
&\quad + \left| \sum_{\tau \in [t-1]} \left\langle g(\boldsymbol{x}_t; \boldsymbol{\theta}_{t-1}),\ ((\boldsymbol{\Sigma}_{t-1}^{(0)})^{-1} - (\boldsymbol{\Sigma}_{t-1})^{-1})g(\boldsymbol{x}_\tau; \boldsymbol{\theta}_{\tau-1}) \cdot c_\tau/m \right\rangle \right| \\
&\leq \underbrace{\left| \sum_{\tau \in [t-1]} \left\langle g(\boldsymbol{x}_t; \boldsymbol{\theta}_{t-1}),\ (\boldsymbol{\Sigma}_{t-1})^{-1}g(\boldsymbol{x}_\tau; \boldsymbol{\theta}_{\tau-1}) \cdot c_\tau/m \right\rangle \right|}_{I_{2.3}} + \mathcal{O}(Cm^{-1/6}\sqrt{\log(m)}t^{7/6}\lambda^{-1/6}L^5) \\
&\quad + \mathcal{O}(Cm^{-1/6}\sqrt{\log(m)}t^{1/6}\lambda^{-7/6}L^4)
\end{aligned}
$$

where the first inequality follows from applying the triangle inequality and Lemma G.4. The second inequality is also due to the triangle inequality, and the last inequality follows from Lemma G.6. In particular, we aim to train separate neural models $f(\cdot; \boldsymbol{\theta}_{i,t})$ for each candidate arm $\boldsymbol{x}_{i,t} \in \mathcal{X}_t$ to minimize the term $I_{2.2}$. Similarly, recall that $\bar{\boldsymbol{\Sigma}}_{t-1} = \lambda\mathbf{I} + \sum_{\tau \in [t-1]} g(\boldsymbol{x}_\tau; \boldsymbol{\theta}_{\tau-1})g(\boldsymbol{x}_\tau; \boldsymbol{\theta}_{\tau-1})^\mathsf{T}/m$ is the gradient covariance matrix without arm weights, and $w_t^{(\tau)}$ is the weight of arm $\boldsymbol{x}_t$, defined as

$$w_t^{(\tau)} = \min\{1, \frac{\alpha \cdot \min_{\boldsymbol{x} \in \mathcal{X}_t} \|g(\boldsymbol{x}; \boldsymbol{\theta}_{t-1})/\sqrt{m}\|_{\bar{\boldsymbol{\Sigma}}_{t-1}^{-1}}^2}{g_\tau \cdot \|g(\boldsymbol{x}_t; \boldsymbol{\theta}_{t-1})/\sqrt{m}\|_{(\bar{\boldsymbol{\Sigma}}_{t-1}^{(\kappa)})^{-1}}}\}, \text{ where } \alpha > 0 \text{ is the tunable parameter. Similar to}$$

the derivation of term $I_{1.3}$, it can further lead to

$$
\begin{aligned}
I_{2.3} &\leq \left| \sum_{\tau \in [t-1]} \frac{w_t^{(\tau)} \cdot c_\tau}{m} \cdot \left\langle g(\boldsymbol{x}_t; \boldsymbol{\theta}_{t-1}), (\boldsymbol{\Sigma}_{t-1})^{-1} \cdot g(\boldsymbol{x}_\tau; \boldsymbol{\theta}_{\tau-1}) \right\rangle \right| \\
&\leq \left| \sum_{\tau \in [t-1]} w_t^{(\tau)} \cdot c_\tau \cdot \left\| g(\boldsymbol{x}_t; \boldsymbol{\theta}_{t-1})/\sqrt{m} \right\|_{(\boldsymbol{\Sigma}_{t-1})^{-1}} \cdot \left\| (\boldsymbol{\Sigma}_{t-1})^{-1} \cdot g(\boldsymbol{x}_\tau; \boldsymbol{\theta}_{\tau-1})/\sqrt{m} \right\|_{\boldsymbol{\Sigma}_{t-1}} \right| \\
&\leq \left| \sum_{\tau \in [t-1]} w_t^{(\tau)} \cdot c_\tau \cdot \left\| g(\boldsymbol{x}_t; \boldsymbol{\theta}_{t-1})/\sqrt{m} \right\|_{(\boldsymbol{\Sigma}_{t-1})^{-1}} \cdot \left\| g(\boldsymbol{x}_\tau; \boldsymbol{\theta}_{\tau-1})/\sqrt{m} \right\|_{(\boldsymbol{\Sigma}_{t-1})^{-1}} \right| \\
&\leq \left| \sum_{\tau \in [t-1]} w_t^{(\tau)} \cdot c_\tau \cdot \left\| g(\boldsymbol{x}_t; \boldsymbol{\theta}_{t-1})/\sqrt{m} \right\|_{(\bar{\boldsymbol{\Sigma}}_{t-1}^{(\kappa)})^{-1}} \cdot \left\| g(\boldsymbol{x}_\tau; \boldsymbol{\theta}_{\tau-1})/\sqrt{m} \right\|_{(\bar{\boldsymbol{\Sigma}}_{\tau-1}^{(\kappa)})^{-1}} \right| \\
&\leq \alpha C \cdot \min_{\boldsymbol{x} \in \mathcal{X}_t} \left\| g(\boldsymbol{x}; \boldsymbol{\theta}_{t-1})/\sqrt{m} \right\|_{(\bar{\boldsymbol{\Sigma}}_{t-1})^{-1}}^2 \\
&\leq \alpha C \cdot \left\| g(\boldsymbol{x}_t; \boldsymbol{\theta}_{t-1})/\sqrt{m} \right\|_{(\bar{\boldsymbol{\Sigma}}_{t-1})^{-1}}^2
\end{aligned}
$$

where the first inequality is by applying the Holder's inequality, and the second and third inequalities are by applying Lemma G.8 with the fact that $\boldsymbol{\Sigma}_{t-1} \succeq \bar{\boldsymbol{\Sigma}}_{t-1}^{(\kappa)} \succeq \bar{\boldsymbol{\Sigma}}_{\tau-1}^{(\kappa)}$ by definition. The last two inequalities are due to the definition of arm weight $w_t^{(\tau)}$, as well as the definition of corruption level $C = \sum_{t \in [T]} |c_t|$. As a result, by combining the upper bounds for the terms $I_{2.4}$ and $I_{2.5}$, we obtain the following upper bound for $I_{2.2}$:

$$
\begin{aligned}
I_{2.2} \leq \mathcal{O}(\alpha C) \cdot \left\| g(\boldsymbol{x}_t; \boldsymbol{\theta}_{t-1})/\sqrt{m} \right\|_{(\bar{\boldsymbol{\Sigma}}_{t-1})^{-1}}^2 &+ \mathcal{O}(Cm^{-1/6}\sqrt{\log(m)}t^{7/6}\lambda^{-1/6}L^5) \\
&+ \mathcal{O}(Cm^{-1/6}\sqrt{\log(m)}t^{1/6}\lambda^{-7/6}L^4).
\end{aligned}
$$

In this case, since we need to ensure that the minimum weight is $\kappa^2$, we scale the tunable parameter $\alpha > 0$ accordingly. Recall that this does not require the learner to have the prior knowledge of the minimum fraction value $\beta$, and the learner can adjust the $\alpha$ values in each round to ensure that the round-wise minimum weight $w_t^{\min} = \min\{w_{i,t}^{(\tau)}\}_{i \in [K]} < 1$, for all $t \in [T]$ and $\tau \in [t-1]$. By summing up all the results, we have

$$
\begin{aligned}
I_2 \leq \mathcal{O}(\alpha C) \cdot \left\| g(\boldsymbol{x}_t; \boldsymbol{\theta}_{t-1})/\sqrt{m} \right\|_{(\bar{\boldsymbol{\Sigma}}_{t-1})^{-1}}^2 &+ \mathcal{O}(\sqrt{\lambda}S) \cdot \left\| g(\boldsymbol{x}_t; \boldsymbol{\theta}_{t-1})/\sqrt{m} \right\|_{(\bar{\boldsymbol{\Sigma}}_{t-1})^{-1}} \\
&+ \mathcal{O}(m^{-2/3}\log(m)L^{7/2}t^{5/3}\lambda^{-5/3}(1 + \sqrt{t/\lambda})) + \mathcal{O}(Cm^{-1/6}\sqrt{\log(m)}t^{7/6}\lambda^{-1/6}L^5) \\
&+ \mathcal{O}(Cm^{-1/6}\sqrt{\log(m)}t^{1/6}\lambda^{-7/6}L^4).
\end{aligned}
$$

$\square$

## C.8 Deriving the UCB and confidence ellipsoid for corruption-free parameters and corrupted parameters

In this subsection, we provide upper bounds for $\widetilde{\mathsf{UCB}}_t(\boldsymbol{x}_t)$ in terms of the corruption-free parameters $\widetilde{\boldsymbol{\theta}}_{t-1}$. Recall that without the arm index $i \in [K]$, the parameters $\boldsymbol{\theta}$, covariance matrix $\boldsymbol{\Sigma}_{t-1}$,

confidence ellipsoid $\mathcal{C}_{t-1}$, and weights $w_t$ pertain to the chosen arm $\boldsymbol{x}_t$. For the hypothetical corruption-free parameters $\widetilde{\boldsymbol{\theta}}$, trained with corruption-free rewards, Lemma C.9 provides the corresponding UCB result, and Lemma C.10 introduces the associated confidence ellipsoid. For the trained parameters $\boldsymbol{\theta}_{i,t-1}$, based on the received records $\mathcal{P}_{t-1}$ with potentially corrupted rewards, Lemma C.11 presents the corresponding UCB for each candidate arm $\boldsymbol{x}_{i,t} \in \mathcal{X}_t$, while Lemma C.12 provides the corresponding confidence ellipsoid. Note that Lemmas C.11 and C.12 are used solely to motivate the design of our UCB-type exploration strategy and are not applied to derive the cumulative regret analysis result.

**Lemma C.9.** *Suppose the imaginary neural network $f(\cdot; \widetilde{\boldsymbol{\theta}}_{t-1})$ in round $t \in [T]$ has been trained on corruption-free rewards $\{\boldsymbol{x}_\tau, \widetilde{r}_\tau\}_{\tau \in [t-1]}$, and $f(\cdot)$ is an $L$-layer FC network with width $m$. Suppose we have $m, J, \eta$ satisfying the conditions in Theorem 5.6. Then, for the chosen arm $\boldsymbol{x}_t \in \mathcal{X}_t$, with the probability at least $1 - \delta$, we will have*

$$\widetilde{UCB}_t(\boldsymbol{x}_t) = \widetilde{V}(\boldsymbol{x}_t) - h(\boldsymbol{x}_t) \le 2\widetilde{\gamma}_{t-1} \cdot \|g(\boldsymbol{x}_t; \widetilde{\boldsymbol{\theta}}_{t-1})/\sqrt{m}\|_{\widetilde{\boldsymbol{\Sigma}}_{t-1}^{-1}} + \mathcal{O}(Sm^{-1/6}\sqrt{\log(m)}t^{1/6}\lambda^{-1/6}L^{2/7})$$
$$+ \mathcal{O}(m^{-1/6}\sqrt{\log(m)}t^{1/6}\lambda^{-7/6}L^{2/7}),$$

*and the corresponding summation value across $T$ rounds will be*

$$\sum_{t \in [T]} \min\{\widetilde{UCB}_t(\boldsymbol{x}_t), 1\} = \sum_{t \in [T]} \min\{\widetilde{V}(\boldsymbol{x}_t) - h(\boldsymbol{x}_t), 1\}$$

$$\le \frac{1}{\sqrt{w_t^{min}}} \mathcal{O}\left(\nu\sqrt{\log\frac{\det(\widetilde{\boldsymbol{\Sigma}}_T)}{\det(\lambda\mathbf{I})} - 2\log(\delta)} + \lambda^{1/2}S\right) \cdot \sqrt{2T \cdot \log\frac{\det(\widetilde{\boldsymbol{\Sigma}}_T)}{\det(\lambda\mathbf{I})}}$$
$$+ \mathcal{O}(Sm^{-1/6}\sqrt{\log(m)}T^{7/6}\lambda^{-1/6}L^{2/7}) + \mathcal{O}(m^{-1/6}\sqrt{\log(m)}T^{7/6}\lambda^{-7/6}L^{2/7})$$

*By definitions in Lemma C.10, we have the corresponding radius term for confidence ellipsoid $\widetilde{\mathcal{C}}_{t-1}$ as $\widetilde{\gamma}_{t-1} = \mathcal{O}\left(\nu \cdot \sqrt{\log\frac{\det(\widetilde{\boldsymbol{\Sigma}}_{t-1})}{\det(\lambda\mathbf{I})} - 2\log(\delta)} + \lambda^{1/2}S\right)$, and the gradient covariance matrix as $\widetilde{\boldsymbol{\Sigma}}_{t-1} = \lambda\mathbf{I} + \sum_{\tau \in [t-1]} w_t^{(\tau)} \cdot g(\boldsymbol{x}_\tau; \widetilde{\boldsymbol{\theta}}_{\tau-1})g(\boldsymbol{x}_\tau; \widetilde{\boldsymbol{\theta}}_{\tau-1})^\intercal/m$.*

**Proof.** With the confidence ellipsoid around the corruption-free parameters $\widetilde{\mathcal{C}}_{t-1} := \{\boldsymbol{\theta} : \|\boldsymbol{\theta} - \widetilde{\boldsymbol{\theta}}_{t-1}\|_{\widetilde{\boldsymbol{\Sigma}}_{t-1}} \le \widetilde{\gamma}_{t-1}/\sqrt{m}, \widetilde{\gamma}_{t-1} > 0\}$, we have $\boldsymbol{\theta}^* \in \widetilde{\mathcal{C}}_{t-1}$ according to Lemma C.10. In this case, with the coefficient $\widetilde{\gamma}_{t-1} = \mathcal{O}\left(\nu \cdot \sqrt{\log\frac{\det(\widetilde{\boldsymbol{\Sigma}}_{t-1})}{\det(\lambda\mathbf{I})} - 2\log(\delta)} + \lambda^{1/2}S\right)$ and the gradient covariance matrix $\widetilde{\boldsymbol{\Sigma}}_{t-1} = \lambda\mathbf{I} + \sum_{\tau \in [t-1]} w_t^{(\tau)} \cdot g(\boldsymbol{x}_\tau; \widetilde{\boldsymbol{\theta}}_{\tau-1})g(\boldsymbol{x}_\tau; \widetilde{\boldsymbol{\theta}}_{\tau-1})^\intercal/m$, we have

$$\begin{aligned}
\widetilde{UCB}_t(\boldsymbol{x}_t) &= \widetilde{V}(\boldsymbol{x}_t) - h(\boldsymbol{x}_t) \\
&= \max_{\boldsymbol{\theta} \in \widetilde{\mathcal{C}}_{t-1}} \langle g(\boldsymbol{x}_t; \boldsymbol{\theta}_0), \boldsymbol{\theta} - \boldsymbol{\theta}_0 \rangle - \langle g(\boldsymbol{x}_t; \boldsymbol{\theta}_0), \boldsymbol{\theta}^* - \boldsymbol{\theta}_0 \rangle \\
&\le \max_{\boldsymbol{\theta} \in \widetilde{\mathcal{C}}_{t-1}} \langle g(\boldsymbol{x}_t; \widetilde{\boldsymbol{\theta}}_0), \boldsymbol{\theta} - \widetilde{\boldsymbol{\theta}}_{t-1} \rangle - \langle g(\boldsymbol{x}_t; \widetilde{\boldsymbol{\theta}}_{t-1}), \boldsymbol{\theta}^* - \widetilde{\boldsymbol{\theta}}_{t-1} \rangle \\
&\quad + \mathcal{O}(Sm^{-1/6}\sqrt{\log(m)}t^{1/6}\lambda^{-1/6}L^{2/7}) \\
&\le \max_{\boldsymbol{\theta} \in \widetilde{\mathcal{C}}_{t-1}} \|\boldsymbol{\theta} - \widetilde{\boldsymbol{\theta}}_{t-1}\|_{\widetilde{\boldsymbol{\Sigma}}_{t-1}} \|g(\boldsymbol{x}_t; \widetilde{\boldsymbol{\theta}}_0)\|_{\widetilde{\boldsymbol{\Sigma}}_{t-1}^{-1}} + \|g(\boldsymbol{x}_t; \widetilde{\boldsymbol{\theta}}_{t-1})\|_{\widetilde{\boldsymbol{\Sigma}}_{t-1}^{-1}} \|\boldsymbol{\theta}^* - \widetilde{\boldsymbol{\theta}}_{t-1}\|_{\widetilde{\boldsymbol{\Sigma}}_{t-1}} \\
&\quad + \mathcal{O}(Sm^{-1/6}\sqrt{\log(m)}t^{1/6}\lambda^{-1/6}L^{2/7}) \\
&\le 2\widetilde{\gamma}_{t-1} \cdot \|g(\boldsymbol{x}_t; \widetilde{\boldsymbol{\theta}}_{t-1})/\sqrt{m}\|_{\widetilde{\boldsymbol{\Sigma}}_{t-1}^{-1}} + \mathcal{O}(Sm^{-1/6}\sqrt{\log(m)}t^{1/6}\lambda^{-1/6}L^{2/7}) \\
&\quad + \mathcal{O}(m^{-1/6}\sqrt{\log(m)}t^{1/6}\lambda^{-7/6}L^{2/7})
\end{aligned}$$

(C.20)

where the first inequality is by Lemma G.4, second inequality is by Holder's inequality, and the last inequality is by applying the definition of confidence ellipsoid $\widetilde{\mathcal{C}}_{t-1}$, Lemma G.4, and Lemma G.6. Next, recall that the above $\widetilde{UCB}_t$ is one term composing the single-round regret $R_t$, and for the cumulative regret, it will be summed up across $T$ rounds. In this case, for the first term on the RHS

above, we have

$$\sum_{t\in[T]} 2\cdot\min\big\{\widetilde{\gamma}_{t-1}\cdot\|g(\boldsymbol{x}_t;\widetilde{\boldsymbol{\theta}}_{t-1})/\sqrt{m}\|_{\widetilde{\boldsymbol{\Sigma}}_{t-1}^{-1}},\ 1\big\} = \sum_{t\in[T]} 2\cdot\min\big\{\frac{\widetilde{\gamma}_{t-1}}{\sqrt{w_t}}\cdot\|\sqrt{w_t}g(\boldsymbol{x}_t;\widetilde{\boldsymbol{\theta}}_{t-1})/\sqrt{m}\|_{\widetilde{\boldsymbol{\Sigma}}_{t-1}^{-1}},\ 1\big\}$$

$$\leq 2(1+\frac{\widetilde{\gamma}_T}{\sqrt{w_t^{\mathsf{min}}}})\cdot\sum_{t\in[T]}\min\big\{\|\sqrt{w_t}g(\boldsymbol{x}_t;\widetilde{\boldsymbol{\theta}}_{t-1})/\sqrt{m}\|_{\widetilde{\boldsymbol{\Sigma}}_{t-1}^{-1}},\ 1\big\}$$

$$\leq 2(1+\frac{\widetilde{\gamma}_T}{\sqrt{w_t^{\mathsf{min}}}})\cdot\sqrt{T\cdot\sum_{t\in[T]}\min\big\{\|\sqrt{w_t}g(\boldsymbol{x}_t;\widetilde{\boldsymbol{\theta}}_{t-1})/\sqrt{m}\|_{\widetilde{\boldsymbol{\Sigma}}_{t-1}^{-1}}^2,\ 1\big\}}$$

where we have $w_t^{\mathsf{min}} < 1$. Then, applying Lemma G.7 and by the definition of $\gamma_T$ as well as supposing that $\frac{\widetilde{\gamma}_T}{\sqrt{w_t^{\mathsf{min}}}} \geq 1$, we will have

$$\sum_{t\in[T]} 2\min\big\{\widetilde{\gamma}_{t-1}\cdot\|g(\boldsymbol{x}_t;\widetilde{\boldsymbol{\theta}}_{t-1})/\sqrt{m}\|_{\widetilde{\boldsymbol{\Sigma}}_{t-1}^{-1}},\ 1\big\}$$

$$\leq \frac{1}{\sqrt{w_t^{\mathsf{min}}}}\mathcal{O}\bigg(\nu\sqrt{\log\frac{\det(\widetilde{\boldsymbol{\Sigma}}_T)}{\det(\lambda\mathbf{I})} - 2\log(\delta)} + \lambda^{1/2}S\bigg)\cdot\sqrt{2T\cdot\log\frac{\det(\widetilde{\boldsymbol{\Sigma}}_T)}{\det(\lambda\mathbf{I})}}.$$

$$\square$$

Next, for the corrupted parameters $\boldsymbol{\theta}_{i,t-1}$, we consider the confidence ellipsoid $\mathcal{C}_{i,t-1} := \{\boldsymbol{\theta} : \|\boldsymbol{\theta} - \boldsymbol{\theta}_{i,t-1}\|_{\boldsymbol{\Sigma}_{i,t-1}} \leq \gamma_{i,t-1}/\sqrt{m}\}$ constructed around the corrupted parameters $\boldsymbol{\theta}_{i,t-1}$, with the coefficient $\gamma_{i,t-1} = \mathcal{O}\big(\nu\cdot\sqrt{\log\frac{\det(\boldsymbol{\Sigma}_{i,t-1})}{\det(\lambda\mathbf{I})} - 2\log(\delta)} + \lambda^{1/2}S\big)$ and the gradient covariance matrix $\boldsymbol{\Sigma}_{i,t-1} = \lambda\mathbf{I} + \sum_{\tau\in[t-1]} w_{i,t}^{(\tau)}\cdot g(\boldsymbol{x}_\tau;\boldsymbol{\theta}_{\tau-1})g(\boldsymbol{x}_\tau;\boldsymbol{\theta}_{\tau-1})^{\mathsf{T}}/m$.

**Lemma C.10.** *In round $t$, with the notation and conditions in Theorem 5.6, suppose the corruption-free parameters $\widetilde{\boldsymbol{\theta}}_{i,t-1}$ associated with a candidate arm $\boldsymbol{x}_{i,t} \in \mathcal{X}_t$ are trained by $\mathcal{L}(\boldsymbol{\theta}) = \frac{1}{2}\sum_{\tau\in[t-1]} w_{i,t}^{(\tau)}\cdot|f(\boldsymbol{x}_\tau;\boldsymbol{\theta}) - \widetilde{r}_\tau|^2 + \frac{m\lambda}{2}\cdot\|\boldsymbol{\theta} - \boldsymbol{\theta}_0\|_2^2$. Then, we have the corresponding confidence ellipsoid*

$$\widetilde{\mathcal{C}}_{i,t-1} := \{\boldsymbol{\theta} : \|\boldsymbol{\theta} - \widetilde{\boldsymbol{\theta}}_{i,t-1}\|_{\widetilde{\boldsymbol{\Sigma}}_{i,t-1}} \leq \widetilde{\gamma}_{i,t-1}/\sqrt{m}\},$$

*such that $\boldsymbol{\theta}^* \in \widetilde{\mathcal{C}}_{i,t-1}$, where $\widetilde{\gamma}_{i,t-1} = \mathcal{O}\big(\nu\cdot\sqrt{\log\frac{\det(\widetilde{\boldsymbol{\Sigma}}_{i,t-1})}{\det(\lambda\mathbf{I})} - 2\log(\delta)} + \lambda^{1/2}S\big)$, and the gradient covariance matrix $\widetilde{\boldsymbol{\Sigma}}_{i,t-1} = \lambda\mathbf{I} + \sum_{\tau\in[t-1]} w_{i,t}^{(\tau)}\cdot g(\boldsymbol{x}_\tau;\widetilde{\boldsymbol{\theta}}_{\tau-1})g(\boldsymbol{x}_\tau;\widetilde{\boldsymbol{\theta}}_{\tau-1})^{\mathsf{T}}/m$.*

**Proof.** The proof of this lemma follows an analogous approach as in Lemma 5.2 in [86]. Recall that based on Lemma C.1, we have the expected reward of an arm $\boldsymbol{x} \in \mathcal{X}_t$ being $\mathbb{E}[r|\boldsymbol{x}] = h(\boldsymbol{x}) = \langle g(\boldsymbol{x};\boldsymbol{\theta}_0),\ \boldsymbol{\theta}^* - \boldsymbol{\theta}_0\rangle$ where there exist parameters $\boldsymbol{\theta}^*$ such that $\|\boldsymbol{\theta}^* - \boldsymbol{\theta}_0\| \leq S/\sqrt{m}, S > 0$. Intuitively, for each previously chosen arm $\boldsymbol{x}_\tau, \tau \in [t-1]$, we can consider an alternative form being

$$\mathbb{E}\left[\sqrt{w_{i,t}^{(\tau)}}\cdot r\ \middle|\ \boldsymbol{x}_\tau\right] = \sqrt{w_{i,t}^{(\tau)}}\cdot h(\boldsymbol{x}_\tau) = \left\langle\sqrt{w_{i,t}^{(\tau)}}\cdot g(\boldsymbol{x}_\tau;\boldsymbol{\theta}_0)/\sqrt{m},\ \sqrt{m}(\boldsymbol{\theta}^* - \boldsymbol{\theta}_0)\right\rangle$$

where $w > 0$ refers to the weight associated with arm $\boldsymbol{x}$ in our settings of R-NeuralUCB. Afterwards, with the weighted sequence of chosen arm gradients as well as their expected corruption-free rewards, we can have

$$\|\sqrt{m}(\boldsymbol{\theta}^* - \boldsymbol{\theta}_0) - (\widetilde{\boldsymbol{\Sigma}}_{i,t-1}^{(0)})^{-1}\widetilde{\boldsymbol{b}}_{i,t-1}^{(0)}\|_{\widetilde{\boldsymbol{\Sigma}}_{i,t-1}^{(0)}} \leq \nu\cdot\sqrt{\log\frac{\det(\widetilde{\boldsymbol{\Sigma}}_{i,t-1}^{(0)})}{\det(\lambda\mathbf{I})} - 2\log(\delta)} + \lambda^{1/2}S,$$

by applying the conclusion of Theorem 2 from [1]. In this case, by triangular inequality, we also have

$$\|\boldsymbol{\theta}^* - \boldsymbol{\theta}_0\|_{\widetilde{\boldsymbol{\Sigma}}_{i,t-1}}$$

$$\leq \|\boldsymbol{\theta}^* - \boldsymbol{\theta}_0 - (\widetilde{\boldsymbol{\Sigma}}_{i,t-1}^{(0)})^{-1}\widetilde{\boldsymbol{b}}_{i,t-1}^{(0)}/\sqrt{m}\|_{\widetilde{\boldsymbol{\Sigma}}_{i,t-1}} + \|\widetilde{\boldsymbol{\theta}}_{i,t-1} - \boldsymbol{\theta}_0 - (\widetilde{\boldsymbol{\Sigma}}_{i,t-1}^{(0)})^{-1}\widetilde{\boldsymbol{b}}_{i,t-1}^{(0)}/\sqrt{m}\|_{\widetilde{\boldsymbol{\Sigma}}_{i,t-1}}.$$

Then, for the first term on the right hand side, we have

$$\|\boldsymbol{\theta}^* - \boldsymbol{\theta}_0 - (\widetilde{\boldsymbol{\Sigma}}_{i,t-1}^{(0)})^{-1}\widetilde{\boldsymbol{b}}_{i,t-1}^{(0)}/\sqrt{m}\|_{\widetilde{\boldsymbol{\Sigma}}_{i,t-1}}^2$$

$$= (\boldsymbol{\theta}^* - \boldsymbol{\theta}_0 - (\widetilde{\boldsymbol{\Sigma}}_{i,t-1}^{(0)})^{-1}\widetilde{\boldsymbol{b}}_{i,t-1}^{(0)}/\sqrt{m})^{\mathsf{T}}\widetilde{\boldsymbol{\Sigma}}_{i,t-1}(\boldsymbol{\theta}^* - \boldsymbol{\theta}_0 - (\widetilde{\boldsymbol{\Sigma}}_{i,t-1}^{(0)})^{-1}\widetilde{\boldsymbol{b}}_{i,t-1}^{(0)}/\sqrt{m})$$

$$= (\boldsymbol{\theta}^* - \boldsymbol{\theta}_0 - (\widetilde{\boldsymbol{\Sigma}}_{i,t-1}^{(0)})^{-1}\widetilde{\boldsymbol{b}}_{i,t-1}^{(0)}/\sqrt{m})^{\mathsf{T}}\widetilde{\boldsymbol{\Sigma}}_{i,t-1}^{(0)}(\boldsymbol{\theta}^* - \boldsymbol{\theta}_0 - (\widetilde{\boldsymbol{\Sigma}}_{i,t-1}^{(0)})^{-1}\widetilde{\boldsymbol{b}}_{i,t-1}^{(0)}/\sqrt{m})$$

$$+ (\boldsymbol{\theta}^* - \boldsymbol{\theta}_0 - (\widetilde{\boldsymbol{\Sigma}}_{i,t-1}^{(0)})^{-1}\widetilde{\boldsymbol{b}}_{i,t-1}^{(0)}/\sqrt{m})^{\mathsf{T}} \cdot (\widetilde{\boldsymbol{\Sigma}}_{i,t-1} - \widetilde{\boldsymbol{\Sigma}}_{i,t-1}^{(0)}) \cdot (\boldsymbol{\theta}^* - \boldsymbol{\theta}_0 - (\widetilde{\boldsymbol{\Sigma}}_{i,t-1}^{(0)})^{-1}\widetilde{\boldsymbol{b}}_{i,t-1}^{(0)}/\sqrt{m})$$

$$\leq (\boldsymbol{\theta}^* - \boldsymbol{\theta}_0 - (\widetilde{\boldsymbol{\Sigma}}_{i,t-1}^{(0)})^{-1}\widetilde{\boldsymbol{b}}_{i,t-1}^{(0)}/\sqrt{m})^{\mathsf{T}}\widetilde{\boldsymbol{\Sigma}}_{i,t-1}^{(0)}(\boldsymbol{\theta}^* - \boldsymbol{\theta}_0 - (\widetilde{\boldsymbol{\Sigma}}_{i,t-1}^{(0)})^{-1}\widetilde{\boldsymbol{b}}_{i,t-1}^{(0)}/\sqrt{m})$$

$$+ \frac{\|\widetilde{\boldsymbol{\Sigma}}_{i,t-1} - \widetilde{\boldsymbol{\Sigma}}_{i,t-1}^{(0)}\|_2}{\lambda} \cdot (\boldsymbol{\theta}^* - \boldsymbol{\theta}_0 - (\widetilde{\boldsymbol{\Sigma}}_{i,t-1}^{(0)})^{-1}\widetilde{\boldsymbol{b}}_{i,t-1}^{(0)}/\sqrt{m})^{\mathsf{T}}\widetilde{\boldsymbol{\Sigma}}_{i,t-1}^{(0)}(\boldsymbol{\theta}^* - \boldsymbol{\theta}_0 - (\widetilde{\boldsymbol{\Sigma}}_{i,t-1}^{(0)})^{-1}\widetilde{\boldsymbol{b}}_{i,t-1}^{(0)}/\sqrt{m})$$

$$\leq (1 + \frac{\|\widetilde{\boldsymbol{\Sigma}}_{i,t-1} - \widetilde{\boldsymbol{\Sigma}}_{i,t-1}^{(0)}\|_2}{\lambda}) \cdot \left(\nu \cdot \sqrt{\log \frac{\det(\widetilde{\boldsymbol{\Sigma}}_{i,t-1}^{(0)})}{\det(\lambda\mathbf{I})} - 2\log(\delta)} + \lambda^{1/2}S\right)/m$$

$$\leq m^{-1/2} \cdot \sqrt{1 + \mathcal{O}(m^{-1/6}\sqrt{\log(m)}L^4 t^{7/6}\lambda^{-7/6})}$$

$$\cdot \left(\nu \cdot \sqrt{\log \frac{\det(\widetilde{\boldsymbol{\Sigma}}_{i,t-1})}{\det(\lambda\mathbf{I})} + \mathcal{O}(m^{-1/6}\sqrt{\log(m)}L^4 t^{5/3}\lambda^{-1/6}) - 2\log(\delta)} + \lambda^{1/2}S\right)$$

where the first inequality is because $\boldsymbol{x}^{\mathsf{T}}\mathbf{A}\boldsymbol{x} \leq \boldsymbol{x}^{\mathsf{T}}\mathbf{B}\boldsymbol{x} \cdot \|\mathbf{A}\|_2/\lambda_{\min}(\mathbf{B})$ for some $0 \prec \mathbf{B}$ and the fact that the minimum eigenvalue $\lambda_{\min}(\widetilde{\boldsymbol{\Sigma}}_{i,t-1}^{(0)}) \geq \lambda$, and the last inequality is by Lemma G.6 as well as due to the fact $w_{i,t}^{(\tau)} \leq 1$. Afterwards, for the second term, we have

$$\|\widetilde{\boldsymbol{\theta}}_{i,t-1} - \boldsymbol{\theta}_0 - (\widetilde{\boldsymbol{\Sigma}}_{i,t-1}^{(0)})^{-1}\widetilde{\boldsymbol{b}}_{i,t-1}^{(0)}/\sqrt{m}\|_{\widetilde{\boldsymbol{\Sigma}}_{i,t-1}} \leq \sqrt{\|\widetilde{\boldsymbol{\Sigma}}_{i,t-1}\|_2} \cdot \|\widetilde{\boldsymbol{\theta}}_{i,t-1} - \boldsymbol{\theta}_0 - (\widetilde{\boldsymbol{\Sigma}}_{i,t-1}^{(0)})^{-1}\widetilde{\boldsymbol{b}}_{i,t-1}^{(0)}/\sqrt{m}\|_2$$

$$\leq \mathcal{O}(\sqrt{\lambda + tL}) \cdot \mathcal{O}\left((1 - \eta m\lambda)^{J/2}\sqrt{t/m\lambda} + m^{-2/3}\sqrt{\log(m)}L^{7/2}t^{5/3}\lambda^{-5/3}(1 + \sqrt{t/\lambda})\right).$$

The first inequality follows from applying Lemma G.6. For the second inequality, we apply a similar approach as in Lemma D.1, with the optimization problem given by (C.12). Since we scale the $\alpha$ parameter to ensure $1 > w_{i,t}^{(\tau)} \geq \kappa^2$, we can follow the proof of Lemma B.2 in [86] to bound the difference between GD-based optimization and gradient-based regression. With the minimum weight value lower bounded by $\kappa^2$ and under the conditions in Theorem 5.6, applying Remark 4.7 in [86] completes the proof.

$$\square$$

**Lemma C.11.** *For candidate arm $\boldsymbol{x}_{i,t} \in \mathcal{X}_t$, suppose its associated neural network $f(\cdot; \boldsymbol{\theta}_{i,t-1})$ has been trained on received records $\{\boldsymbol{x}_\tau, r_\tau\}_{\tau \in [t-1]}$, with $J$ iterations of GD and learning rate $\eta$. Let $f(\cdot; \boldsymbol{\theta}_{i,t-1})$ be an $L$-layer FC network with width $m$. Suppose conditions in Theorem 5.6 are satisfied. Then, given the candidate arm $\boldsymbol{x}_{i,t} \in \mathcal{X}_t$, with a constant $\zeta > 0$ and probability at least $1 - \delta$, we will have*

$$\left|f(\boldsymbol{x}_{i,t}; \boldsymbol{\theta}_{i,t-1}) - h(\boldsymbol{x}_{i,t})\right|$$

$$\leq \|g(\boldsymbol{x}_{i,t}; \boldsymbol{\theta}_{i,t-1})/\sqrt{m}\|_{\boldsymbol{\Sigma}_{i,t-1}^{-1}} \cdot \left[\zeta \cdot \left(\nu\sqrt{\log \frac{\det(\boldsymbol{\Sigma}_{i,t-1})}{\det(\lambda\mathbf{I})} - 2\log(\delta)} + \lambda^{1/2}S\right)\right.$$

$$+ (1 - \eta m\lambda)^{J/2}\sqrt{t/\lambda} + \mathcal{O}(m^{-1/6}\sqrt{\log(m)}L^{7/2}t^{5/3}\lambda^{-5/3}(1 + \sqrt{t/\lambda}))$$

$$+ \mathcal{O}(Cm^{-1/6}\sqrt{\log(m)}t^{1/6}\lambda^{-7/6}L^{7/2}) + \mathcal{O}(Cm^{-1/6}\sqrt{\log(m)}t^{7/6}\lambda^{-13/6}L^{9/2})$$

$$+ \nu \cdot \sqrt{1 + \mathcal{O}(m^{-1/6}\sqrt{\log(m)}L^4 t^{7/6}\lambda^{-7/6})} \cdot \left. \mathcal{O}(m^{-1/12}\log^{1/4}(m)L^2 t^{5/6}\lambda^{-1/12})\right]$$

$$+ \mathcal{O}(\alpha C) \cdot \min_{\boldsymbol{x} \in \mathcal{X}_t}\|g(\boldsymbol{x}; \boldsymbol{\theta}_{i,t-1})/\sqrt{m}\|_{\widetilde{\boldsymbol{\Sigma}}_{t-1}^{-1}}^2$$

$$+ \mathcal{O}(Sm^{-1/6}\sqrt{\log(m)}t^{1/6}\lambda^{-1/6}L^{2/7}) + \mathcal{O}(m^{-1/6}\sqrt{\log(m)}t^{2/3}\lambda^{-2/3}L^3), \tag{C.21}$$

*with notation and definitions in Theorem 5.6. We also have the covariance matrix $\boldsymbol{\Sigma}_{i,t-1} = \lambda\mathbf{I} + \sum_{\tau\in[t]} w_{i,t}^{(\tau)} \cdot g(\boldsymbol{x}_\tau;\boldsymbol{\theta}_{\tau-1})g(\boldsymbol{x}_\tau;\boldsymbol{\theta}_{\tau-1})^\mathsf{T}$, with gradient vector $g(\boldsymbol{x}_{i,t};\boldsymbol{\theta}) = vec(\nabla_{\boldsymbol{\theta}} f(\boldsymbol{x}_{i,t};\boldsymbol{\theta}))$.*

**Proof.** Applying the Lemma C.1, we can transform the objective by substituting the reward mapping function $h(\cdot)$, as

$$
\begin{aligned}
\big|f(\boldsymbol{x}_{i,t};\boldsymbol{\theta}_{i,t-1}) - h(\boldsymbol{x}_{i,t})\big| &= \big|f(\boldsymbol{x}_{i,t};\boldsymbol{\theta}_{i,t-1}) - \langle g(\boldsymbol{x}_{i,t};\boldsymbol{\theta}_0), \boldsymbol{\theta}^* - \boldsymbol{\theta}_0\rangle\big| \\
&\leq \big|f(\boldsymbol{x}_{i,t};\boldsymbol{\theta}_{i,t-1}) - \langle g(\boldsymbol{x}_{i,t};\boldsymbol{\theta}_{i,t-1}), \boldsymbol{\theta}^* - \boldsymbol{\theta}_0\rangle\big| + \mathcal{O}(Sm^{-1/6}\sqrt{\log(m)}t^{1/6}\lambda^{-1/6}L^{2/7}) \\
&\leq \big|f(\boldsymbol{x}_{i,t};\boldsymbol{\theta}_{i,t-1}) - \langle g(\boldsymbol{x}_{i,t};\boldsymbol{\theta}_{i,t-1}), \boldsymbol{\theta}_{i,t-1} - \boldsymbol{\theta}_0\rangle\big| \\
&\quad + \big|\langle g(\boldsymbol{x}_{i,t};\boldsymbol{\theta}_{i,t-1}), \boldsymbol{\theta}_{i,t-1} - \boldsymbol{\theta}_0\rangle - \langle g(\boldsymbol{x}_{i,t};\boldsymbol{\theta}_{i,t-1}), \boldsymbol{\theta}^* - \boldsymbol{\theta}_0\rangle\big| \\
&\quad + \mathcal{O}(Sm^{-1/6}\sqrt{\log(m)}t^{1/6}\lambda^{-1/6}L^{2/7})
\end{aligned}
$$

where the first equality is due to Lemma C.1, while the first inequality is due to Lemma G.4 and Lemma C.1, and the last inequality is because of the triangular inequality. Then, we proceed to separately bound the first and second term on the RHS. For the first term, we will have

$$
\begin{aligned}
\big|f(\boldsymbol{x}_{i,t};&\boldsymbol{\theta}_{i,t-1}) - \langle g(\boldsymbol{x}_{i,t};\boldsymbol{\theta}_{i,t-1}), \boldsymbol{\theta}_{i,t-1} - \boldsymbol{\theta}_0\rangle\big| \\
&= \big|f(\boldsymbol{x}_{i,t};\boldsymbol{\theta}_{i,t-1}) - f(\boldsymbol{x}_{i,t};\boldsymbol{\theta}_0) - \langle g(\boldsymbol{x}_{i,t};\boldsymbol{\theta}_{i,t-1}), \boldsymbol{\theta}_{i,t-1} - \boldsymbol{\theta}_0\rangle\big| \\
&\leq \mathcal{O}(m^{-1/6}\sqrt{\log(m)}t^{2/3}\lambda^{-2/3}L^3)
\end{aligned}
$$

where the first equality is due to the fact that $f(\boldsymbol{x}_{i,t};\boldsymbol{\theta}_0) = 0$ based on our parameter initialization approach, and the inequality is by applying Lemma G.5 and Lemma G.3. Then, for the second term, we will have

$$
\begin{aligned}
\big|\langle g(\boldsymbol{x}_{i,t};&\boldsymbol{\theta}_{i,t-1}), \boldsymbol{\theta}_{i,t-1} - \boldsymbol{\theta}_0\rangle - \langle g(\boldsymbol{x}_{i,t};\boldsymbol{\theta}_{i,t-1}), \boldsymbol{\theta}^* - \boldsymbol{\theta}_0\rangle\big| \\
&= \big|\langle g(\boldsymbol{x}_{i,t};\boldsymbol{\theta}_{i,t-1}), \boldsymbol{\theta}^* - \boldsymbol{\theta}_{i,t-1}\rangle\big| \\
&\leq \|g(\boldsymbol{x}_{i,t};\boldsymbol{\theta}_{i,t-1})/\sqrt{m}\|_{\boldsymbol{\Sigma}_{i,t-1}^{-1}} \cdot \sqrt{m} \cdot \|\boldsymbol{\theta}^* - \boldsymbol{\theta}_{i,t-1}\|_{\boldsymbol{\Sigma}_{i,t-1}} \\
&\leq \gamma_{i,t-1} \cdot \|g(\boldsymbol{x}_{i,t};\boldsymbol{\theta}_{i,t-1})/\sqrt{m}\|_{\boldsymbol{\Sigma}_{i,t-1}^{-1}}
\end{aligned}
$$

where the first inequality is by applying the Holder's inequality, and the last inequality is by applying Lemma 5.2 in [86], Lemma C.1 in terms of the confidence set $\mathcal{C}_{i,t-1}$ and the fact that $\boldsymbol{\theta}^* \in \mathcal{C}_{i,t-1}$. Then, with the confidence ellipsoid introduced and discussed in Lemma C.12, summing up the results above, we will then have

$$
\begin{aligned}
\Big|f(\boldsymbol{x}_{i,t};&\boldsymbol{\theta}_{i,t-1}) - h(\boldsymbol{x}_{i,t})\Big| \\
&\leq \|g(\boldsymbol{x}_{i,t};\boldsymbol{\theta}_{i,t-1})/\sqrt{m}\|_{\boldsymbol{\Sigma}_{i,t-1}^{-1}} \\
&\quad \cdot \Bigg[\mathcal{O}\bigg(\nu\sqrt{\log\frac{\det(\boldsymbol{\Sigma}_{i,t-1})}{\det(\lambda\mathbf{I})} - 2\log(\delta)} + \lambda^{1/2}S\bigg) + \mathcal{O}(\alpha C) \cdot \frac{\min_{\boldsymbol{x}\in\mathcal{X}_t}\|g(\boldsymbol{x};\boldsymbol{\theta}_{i,t-1})/\sqrt{m}\|_{\bar{\boldsymbol{\Sigma}}_{t-1}^{-1}}^2}{\|g(\boldsymbol{x}_{i,t};\boldsymbol{\theta}_{i,t-1})/\sqrt{m}\|_{(\bar{\boldsymbol{\Sigma}}_{t-1}^{(\kappa)})^{-1}}} \\
&\quad + (1 - \eta m\lambda)^{J/2}\sqrt{t/\lambda} + \mathcal{O}(m^{-1/6}\sqrt{\log(m)}L^{7/2}t^{5/3}\lambda^{-5/3}(1 + \sqrt{t/\lambda})) \\
&\quad + \mathcal{O}(Cm^{-2/3}\sqrt{\log(m)}t^{1/6}\lambda^{-7/6}L^{7/2}) + \mathcal{O}(Cm^{-1/6}\sqrt{\log(m)}t^{7/6}\lambda^{-13/6}L^{9/2}) \\
&\quad + \nu \cdot \sqrt{1 + \mathcal{O}(m^{-1/6}\sqrt{\log(m)}L^4t^{7/6}\lambda^{-7/6})} \cdot \mathcal{O}(m^{-1/12}\log^{1/4}(m)L^2t^{5/6}\lambda^{-1/12})\Bigg] \\
&\quad + \mathcal{O}(Sm^{-1/6}\sqrt{\log(m)}t^{1/6}\lambda^{-1/6}L^{2/7}) + \mathcal{O}(m^{-1/6}\sqrt{\log(m)}t^{2/3}\lambda^{-2/3}L^3)
\end{aligned}
$$

$$= \|g(\boldsymbol{x}_{i,t}; \boldsymbol{\theta}_{i,t-1})/\sqrt{m}\|_{\boldsymbol{\Sigma}_{i,t-1}^{-1}}$$

$$\cdot \Bigg[ \mathcal{O}\bigg( \nu\sqrt{\log\frac{\det(\boldsymbol{\Sigma}_{i,t-1})}{\det(\lambda\mathbf{I})} - 2\log(\delta)} + \lambda^{1/2}S \bigg)$$

$$+ (1-\eta m\lambda)^{J/2}\sqrt{t/\lambda} + \mathcal{O}(m^{-1/6}\sqrt{\log(m)}L^{7/2}t^{5/3}\lambda^{-5/3}(1+\sqrt{t/\lambda}))$$

$$+ \mathcal{O}(Cm^{-1/6}\sqrt{\log(m)}t^{1/6}\lambda^{-7/6}L^{7/2}) + \mathcal{O}(Cm^{-1/6}\sqrt{\log(m)}t^{7/6}\lambda^{-13/6}L^{9/2})$$

$$+ \nu \cdot \sqrt{1 + \mathcal{O}(m^{-1/6}\sqrt{\log(m)}L^4 t^{7/6}\lambda^{-7/6})} \cdot \mathcal{O}(m^{-1/12}\log^{1/4}(m)L^2 t^{5/6}\lambda^{-1/12}) \Bigg]$$

$$+ \|g(\boldsymbol{x}_{i,t}; \boldsymbol{\theta}_{i,t-1})/\sqrt{m}\|_{\boldsymbol{\Sigma}_{i,t-1}^{-1}} \cdot \mathcal{O}(\alpha C) \cdot \frac{\min_{\boldsymbol{x}\in\mathcal{X}_t}\|g(\boldsymbol{x}; \boldsymbol{\theta}_{i,t-1})/\sqrt{m}\|_{\bar{\boldsymbol{\Sigma}}_{t-1}^{-1}}^2}{\|g(\boldsymbol{x}_{i,t}; \boldsymbol{\theta}_{i,t-1})/\sqrt{m}\|_{(\bar{\boldsymbol{\Sigma}}_{t-1}^{(\kappa)})^{-1}}}$$

$$+ \mathcal{O}(Sm^{-1/6}\sqrt{\log(m)}t^{1/6}\lambda^{-1/6}L^{2/7}) + \mathcal{O}(m^{-1/6}\sqrt{\log(m)}t^{2/3}\lambda^{-2/3}L^3).$$

where we naturally have Hermitian matrices $\boldsymbol{\Sigma}_{i,t-1} \succeq \bar{\boldsymbol{\Sigma}}_{t-1}^{(\kappa)}$ by definition, which leads to $(\boldsymbol{\Sigma}_{i,t-1})^{-1} \preceq (\bar{\boldsymbol{\Sigma}}_{t-1}^{(\kappa)})^{-1}$ by applying the conclusion from Lemma G.8. Afterwards, due to the fact that the minimum round-wise arm weight $w_t^{\min} = \min\{w_{i,t}^{(\tau)}\}_{i\in[K],\tau\in[t-1]} = \kappa^2 < 1, \forall t \in [T]$ by scaling parameter $\alpha$, we can further have

$$\left| f(\boldsymbol{x}_{i,t}; \boldsymbol{\theta}_{i,t-1}) - h(\boldsymbol{x}_{i,t}) \right|$$

$$\leq \|g(\boldsymbol{x}_{i,t}; \boldsymbol{\theta}_{i,t-1})/\sqrt{m}\|_{\boldsymbol{\Sigma}_{i,t-1}^{-1}}$$

$$\cdot \Bigg[ \mathcal{O}\bigg( \nu\sqrt{\log\frac{\det(\boldsymbol{\Sigma}_{i,t-1})}{\det(\lambda\mathbf{I})} - 2\log(\delta)} + \lambda^{1/2}S \bigg)$$

$$+ (1-\eta m\lambda)^{J/2}\sqrt{t/\lambda} + \mathcal{O}(m^{-1/6}\sqrt{\log(m)}L^{7/2}t^{5/3}\lambda^{-5/3}(1+\sqrt{t/\lambda}))$$

$$+ \mathcal{O}(Cm^{-2/3}\sqrt{\log(m)}t^{1/6}\lambda^{-7/6}L^{7/2}) + \mathcal{O}(Cm^{-1/6}\sqrt{\log(m)}t^{7/6}\lambda^{-13/6}L^{9/2})$$

$$+ \nu \cdot \sqrt{1 + \mathcal{O}(m^{-1/6}\sqrt{\log(m)}L^4 t^{7/6}\lambda^{-7/6})} \cdot \mathcal{O}(m^{-1/12}\log^{1/4}(m)L^2 t^{5/6}\lambda^{-1/12}) \Bigg]$$

$$+ \mathcal{O}(\alpha C) \cdot \min_{\boldsymbol{x}\in\mathcal{X}_t}\|g(\boldsymbol{x}; \boldsymbol{\theta}_{i,t-1})/\sqrt{m}\|_{\bar{\boldsymbol{\Sigma}}_{t-1}^{-1}}^2$$

$$+ \mathcal{O}(Sm^{-1/6}\sqrt{\log(m)}t^{1/6}\lambda^{-1/6}L^{2/7}) + \mathcal{O}(m^{-1/6}\sqrt{\log(m)}t^{2/3}\lambda^{-2/3}L^3),$$

which completes the proof for this lemma.

$$\qquad\qquad\qquad\qquad\qquad\qquad\qquad\qquad\qquad\qquad\qquad\qquad\qquad\qquad\qquad\quad \square$$

**Lemma C.12.** *In round $t \in [T]$, with the notation and conditions from Theorem 5.6, suppose the corrupted parameters $\boldsymbol{\theta}_{i,t-1}$ associated with a candidate arm $\boldsymbol{x}_{i,t} \in \mathcal{X}_t$ are trained by $\mathcal{L}(\boldsymbol{\theta}) = \frac{1}{2}\sum_{\tau\in[t-1]} w_{i,t}^{(\tau)} \cdot |f(\boldsymbol{x}_\tau; \boldsymbol{\theta}) - r_\tau|^2 + \frac{m\lambda}{2} \cdot \|\boldsymbol{\theta} - \boldsymbol{\theta}_0\|_2^2$. Then, we have the confidence ellipsoid*

$$\mathcal{C}_{i,t-1} = \left\{ \boldsymbol{\theta} : \|\boldsymbol{\theta} - \boldsymbol{\theta}_{i,t-1}\|_{\boldsymbol{\Sigma}_{i,t-1}} \leq \gamma_{i,t-1}/\sqrt{m} \right\}$$

*where we have the unknown parameter $\boldsymbol{\theta}^* \in \mathcal{C}_{i,t-1}$, and we denote*

$$\gamma_{i,t-1} = \mathcal{O}\bigg( \nu\sqrt{\log\frac{\det(\boldsymbol{\Sigma}_{i,t-1})}{\det(\lambda\mathbf{I})} - 2\log(\delta)} + \lambda^{1/2}S \bigg) + \mathcal{O}(\alpha C) \cdot \frac{\min_{\boldsymbol{x}\in\mathcal{X}_t}\|g(\boldsymbol{x}; \boldsymbol{\theta}_{i,t-1})/\sqrt{m}\|_{\bar{\boldsymbol{\Sigma}}_{t-1}^{-1}}^2}{\|g(\boldsymbol{x}_{i,t}; \boldsymbol{\theta}_{i,t-1})/\sqrt{m}\|_{(\bar{\boldsymbol{\Sigma}}_{t-1}^{(\kappa)})^{-1}}}$$

$$+ (1-\eta m\lambda)^{J/2}\sqrt{t/\lambda} + \mathcal{O}(m^{-1/6}\sqrt{\log(m)}L^{7/2}t^{5/3}\lambda^{-5/3}(1+\sqrt{t/\lambda}))$$

$$+ \mathcal{O}(Cm^{-1/6}\sqrt{\log(m)}t^{1/6}\lambda^{-7/6}L^{7/2}) + \mathcal{O}(Cm^{-1/6}\sqrt{\log(m)}t^{7/6}\lambda^{-13/6}L^{9/2})$$

$$+ \nu \cdot \sqrt{1 + \mathcal{O}(m^{-1/6}\sqrt{\log(m)}L^4 t^{7/6}\lambda^{-7/6})} \cdot \mathcal{O}(m^{-1/12}\log^{1/4}(m)L^2 t^{5/6}\lambda^{-1/12}).$$

**Proof.** Recall that for the imaginary corruption-free parameters $\widetilde{\boldsymbol{\theta}}_{i,t-1}$, trained on corruption-free records $\widetilde{\mathcal{P}}_{t-1}$, we can construct the confidence interval $\widetilde{\mathcal{C}}_{i,t-1} := \{\boldsymbol{\theta} : \|\boldsymbol{\theta} - \widetilde{\boldsymbol{\theta}}_{i,t-1}\|_{\widetilde{\boldsymbol{\Sigma}}_{i,t-1}} \leq \widetilde{\gamma}_{i,t-1}/\sqrt{m}, \widetilde{\gamma}_{i,t-1} > 0\}$, ensuring that the unknown $\boldsymbol{\theta}^*$ in Lemma C.1 satisfies $\boldsymbol{\theta}^* \in \widetilde{\mathcal{C}}_{i,t-1}$. The corresponding confidence ellipsoid is presented in Lemma C.10.

With the ellipsoid centered at $\boldsymbol{\theta}_{i,t-1}$ and the gradient covariance matrix defined as $\boldsymbol{\Sigma}_{i,t-1} = \lambda \mathbf{I} + \sum_{\tau \in [t-1]} w_{i,t}^{(\tau)} \cdot g(\boldsymbol{x}_\tau; \boldsymbol{\theta}_{\tau-1}) \cdot g(\boldsymbol{x}_\tau; \boldsymbol{\theta}_{\tau-1})^\intercal / m$, we proceed to derive the corresponding radius. Recall that $w_{i,t}^{(\tau)}$ denotes the sample weight associated with the chosen arm $\boldsymbol{x}_\tau$. For reference, we first recall the preliminary bounds: $\|\boldsymbol{\Sigma}_{i,t-1} - \widetilde{\boldsymbol{\Sigma}}_{i,t-1}\|_F \leq \mathcal{O}(m^{-1/6}\sqrt{\log(m)}L^4 t^{7/6}\lambda^{-1/6})$ by Lemma G.6 and $\|\boldsymbol{\theta}_{i,t-1} - \widetilde{\boldsymbol{\theta}}_{i,t-1}\|_2 \leq \mathcal{O}(\sqrt{t/(m\lambda)})$ as shown in Lemma G.3. Next, as we already have $\|\boldsymbol{\theta} - \widetilde{\boldsymbol{\theta}}_{i,t-1}\|_{\widetilde{\boldsymbol{\Sigma}}_{i,t-1}} \leq \widetilde{\gamma}_{i,t-1}/\sqrt{m}$, we then proceed to transform the objective to

$$
\begin{aligned}
\|\boldsymbol{\theta}^* - \boldsymbol{\theta}_{i,t-1}\|_{\boldsymbol{\Sigma}_{i,t-1}} &\leq \|\boldsymbol{\theta}^* - \widetilde{\boldsymbol{\theta}}_{i,t-1}\|_{\boldsymbol{\Sigma}_{i,t-1}} + \|\widetilde{\boldsymbol{\theta}}_{i,t-1} - \boldsymbol{\theta}_{i,t-1}\|_{\boldsymbol{\Sigma}_{i,t-1}} \\
&\leq \|\boldsymbol{\theta}^* - \widetilde{\boldsymbol{\theta}}_{i,t-1}\|_{\boldsymbol{\Sigma}_{i,t-1}} + \|\widetilde{\boldsymbol{\theta}}_{i,t-1} - \boldsymbol{\theta}_{i,t-1}\|_{\boldsymbol{\Sigma}_{i,t-1}} \\
&\leq \|\boldsymbol{\theta}^* - \widetilde{\boldsymbol{\theta}}_{i,t-1}\|_{\boldsymbol{\Sigma}_{i,t-1}-\widetilde{\boldsymbol{\Sigma}}_{i,t-1}+\widetilde{\boldsymbol{\Sigma}}_{i,t-1}} + \|\widetilde{\boldsymbol{\theta}}_{i,t-1} - \boldsymbol{\theta}_{i,t-1}\|_{\boldsymbol{\Sigma}_{i,t-1}} \\
&\leq \|\boldsymbol{\theta}^* - \widetilde{\boldsymbol{\theta}}_{i,t-1}\|_{\widetilde{\boldsymbol{\Sigma}}_{i,t-1}} + \|\boldsymbol{\theta}^* - \widetilde{\boldsymbol{\theta}}_{i,t-1}\|_{\boldsymbol{\Sigma}_{i,t-1}-\widetilde{\boldsymbol{\Sigma}}_{i,t-1}} + \|\widetilde{\boldsymbol{\theta}}_{i,t-1} - \boldsymbol{\theta}_{i,t-1}\|_{\boldsymbol{\Sigma}_{i,t-1}} \\
&\leq \widetilde{\gamma}_{i,t-1}/\sqrt{m} + \|\widetilde{\boldsymbol{\theta}}_{i,t-1} - \boldsymbol{\theta}_{i,t-1}\|_{\boldsymbol{\Sigma}_{i,t-1}} + \mathcal{O}(\sqrt{t/(m\lambda)}) \cdot \mathcal{O}(m^{-1/6}\sqrt{\log(m)}L^4 t^{7/6}\lambda^{-1/6}) \\
&\leq \widetilde{\gamma}_{i,t-1}/\sqrt{m} + \|\widetilde{\boldsymbol{\theta}}_{i,t-1} - \boldsymbol{\theta}_{i,t-1}\|_{\boldsymbol{\Sigma}_{i,t-1}} + \mathcal{O}(m^{-2/3}\sqrt{\log(m)}L^4 t^{13/6}\lambda^{-2/3}).
\end{aligned}
$$

For the first term on the RHS, we can follow the proof flow in Lemma C.6, by applying Lemma G.4 to substitute $\widetilde{\boldsymbol{\Sigma}}_{i,t-1}$ with corresponding $\boldsymbol{\Sigma}_{i,t-1}$, which will consequently lead to $|\gamma_{i,t-1}/\sqrt{m} - \widetilde{\gamma}_{i,t-1}/\sqrt{m}| \leq \nu \cdot \sqrt{1 + \mathcal{O}(m^{-1/6}\sqrt{\log(m)}L^4 t^{7/6}\lambda^{-7/6})} \cdot \mathcal{O}(m^{-7/12}\log^{1/4}(m)L^2 t^{5/6}\lambda^{-1/12})$. Meanwhile, for the second term on the RHS, we first define the gradient-based regression parameters as

$$
\begin{aligned}
\boldsymbol{\Sigma}_{i,t-1}^{(0)} &= \lambda \mathbf{I} + \sum_{\tau \in [t-1]} w_{i,t}^{(\tau)} \cdot g(\boldsymbol{x}_\tau; \boldsymbol{\theta}_0) \cdot g(\boldsymbol{x}_\tau; \boldsymbol{\theta}_0)^\intercal / m, \\
\boldsymbol{b}_{i,t-1}^{(0)} &= \sum_{\tau \in [t-1]} w_{i,t}^{(\tau)} \cdot g(\boldsymbol{x}_\tau; \boldsymbol{\theta}_0) \cdot r_\tau / \sqrt{m}, \\
\widetilde{\boldsymbol{b}}_{i,t-1}^{(0)} &= \sum_{\tau \in [t-1]} w_{i,t}^{(\tau)} \cdot g(\boldsymbol{x}_\tau; \boldsymbol{\theta}_0) \cdot \widetilde{r}_\tau / \sqrt{m},
\end{aligned}
$$

Then, we can proceed to have

$$
\begin{aligned}
&\|\widetilde{\boldsymbol{\theta}}_{i,t-1} - \boldsymbol{\theta}_{i,t-1}\|_{\boldsymbol{\Sigma}_{i,t-1}} \\
&\leq \|\widetilde{\boldsymbol{\theta}}_{i,t-1} - \boldsymbol{\theta}_0 - (\boldsymbol{\Sigma}_{i,t-1}^{(0)})^{-1}\boldsymbol{b}_{i,t-1}^{(0)}/\sqrt{m} + (\boldsymbol{\Sigma}_{i,t-1}^{(0)})^{-1}\boldsymbol{b}_{i,t-1}^{(0)}/\sqrt{m} + \boldsymbol{\theta}_0 - \boldsymbol{\theta}_{i,t-1}\|_{\boldsymbol{\Sigma}_{i,t-1}} \\
&\leq \|\boldsymbol{\theta}_{i,t-1} - \boldsymbol{\theta}_0 - (\boldsymbol{\Sigma}_{i,t-1}^{(0)})^{-1}\boldsymbol{b}_{i,t-1}^{(0)}/\sqrt{m}\|_{\boldsymbol{\Sigma}_{i,t-1}} + \|\widetilde{\boldsymbol{\theta}}_{i,t-1} - \boldsymbol{\theta}_0 - (\boldsymbol{\Sigma}_{i,t-1}^{(0)})^{-1}\boldsymbol{b}_{i,t-1}^{(0)}/\sqrt{m}\|_{\boldsymbol{\Sigma}_{i,t-1}} \\
&\leq \|\boldsymbol{\theta}_{i,t-1} - \boldsymbol{\theta}_0 - (\boldsymbol{\Sigma}_{i,t-1}^{(0)})^{-1}\boldsymbol{b}_{i,t-1}^{(0)}/\sqrt{m}\|_{\boldsymbol{\Sigma}_{i,t-1}} + \|\widetilde{\boldsymbol{\theta}}_{i,t-1} - \boldsymbol{\theta}_0 - (\boldsymbol{\Sigma}_{i,t-1}^{(0)})^{-1}\widetilde{\boldsymbol{b}}_{i,t-1}^{(0)}/\sqrt{m}\|_{\boldsymbol{\Sigma}_{i,t-1}} \\
&\quad + m^{-1}\|(\boldsymbol{\Sigma}_{i,t-1}^{(0)})^{-1} \cdot \big(\sum_{\tau \in [t-1]} w_{i,t}^{(\tau)} \cdot g(\boldsymbol{x}_\tau; \boldsymbol{\theta}_0) \cdot c_\tau\big)\|_{\boldsymbol{\Sigma}_{i,t-1}}.
\end{aligned}
$$

$$\text{(C.22)}$$

**Bounding the first two term in Inequality C.22.** Here, for the first term on the RHS, we can individually apply Lemma D.1, by considering the auxiliary sequence in $j$-th iteration ($j \in [J]$) with $\boldsymbol{\Theta}^{(0)} = \boldsymbol{\theta}_0$, as

$$
\boldsymbol{\Theta}^{(j+1)} = \boldsymbol{\Theta}^{(j)} - \eta \cdot \left[ \mathbf{J}^{(0)} \cdot \mathbf{W} \cdot \big([\mathbf{J}^{(0)}]^\intercal(\boldsymbol{\Theta}^{(j)} - \boldsymbol{\theta}_0) - \boldsymbol{y}\big) + m\lambda(\boldsymbol{\Theta}^{(j)} - \boldsymbol{\theta}_0) \right]
$$

where the Jacobian matrix $\mathbf{J}^{(0)} := \big(g(\boldsymbol{x}_1; \boldsymbol{\theta}_0), g(\boldsymbol{x}_2; \boldsymbol{\theta}_0), \ldots, g(\boldsymbol{x}_{i,t-1}; \boldsymbol{\theta}_0)\big) \in \mathbb{R}^{p \times (t-1)}$, vector $\boldsymbol{y} \in \mathbb{R}^{t-1}$ contains the received arm rewards $r_\tau, \tau \in [t-1]$, and matrix $\mathbf{W} \in \mathbb{R}^{(t-1) \times (t-1)}$ is the

diagonal matrix that contains sample weights $w_{i,t}^{(\tau)}, \tau \in [t-1]$. In particular, we have its norm $\|\mathbf{W}\|_2 \leq 1$ by definition. Here, with $[\mathbf{J}^{(0)}]_\tau$ being the $\tau$-th column of matrix $\mathbf{J}^{(0)}$, the above sequence is expected to solve the following problem

$$\min_{\boldsymbol{\Theta}} \mathcal{L}(\boldsymbol{\Theta}) = \sum_{\tau \in [t-1]} \frac{w_{i,t}^{(\tau)}}{2} \cdot \left\| [\mathbf{J}^{(0)}]_\tau^{\mathsf{T}}(\boldsymbol{\Theta} - \boldsymbol{\theta}_0) - r_\tau \right\|_2^2 + \frac{1}{2} \cdot m\lambda \cdot \left\| \boldsymbol{\Theta} - \boldsymbol{\theta}_0 \right\|_2^2.$$

As a result, following an analogous approach as in Lemma C.4 in [86], we can have $\|\boldsymbol{\Theta}^{(j)} - \boldsymbol{\theta}_0 - (\boldsymbol{\Sigma}_{i,t-1}^{(0)})^{-1}\boldsymbol{b}_{i,t-1}^{(0)}/\sqrt{m}\|_{\boldsymbol{\Sigma}_{i,t-1}} \leq (1 - \eta m\lambda)^{j/2}\sqrt{t/(m\lambda)}$. Furthermore, by applying the conclusion of Lemma D.1, we can have

$$\|\boldsymbol{\theta}_{i,t-1} - \boldsymbol{\theta}_0 - (\boldsymbol{\Sigma}_{i,t-1}^{(0)})^{-1}\boldsymbol{b}_{i,t-1}^{(0)}/\sqrt{m}\|_2$$
$$\leq (1 - \eta m\lambda)^{J/2}\sqrt{t/(m\lambda)} + \mathcal{O}(m^{-2/3}\sqrt{\log(m)}L^{7/2}t^{5/3}\lambda^{-5/3}(1 + \sqrt{t/\lambda})).$$

Similarly, for the second term in Inequality C.22, we also can apply a comparable approach by solving the problem:

$$\min_{\boldsymbol{\Theta}} \mathcal{L}(\boldsymbol{\Theta}) = \sum_{\tau \in [t-1]} \frac{w_{i,t}^{(\tau)}}{2} \left\| [\mathbf{J}^{(0)}]_\tau^{\mathsf{T}}(\boldsymbol{\Theta} - \boldsymbol{\theta}_0) - \widetilde{r}_\tau \right\|_2^2 + \frac{1}{2}m\lambda \cdot \left\| \boldsymbol{\Theta} - \boldsymbol{\theta}_0 \right\|_2^2,$$

and constructing the corresponding auxiliary sequence. This will lead to a similar bound for the second term on the RHS of inequality C.22, such that $\|\widetilde{\boldsymbol{\theta}}_{i,t-1} - \boldsymbol{\theta}_0 - (\boldsymbol{\Sigma}_{i,t-1}^{(0)})^{-1}\widetilde{\boldsymbol{b}}_{i,t-1}^{(0)}/\sqrt{m}\|_2 \leq (1 - \eta m\lambda)^{J/2}\sqrt{t/(m\lambda)} + \mathcal{O}(m^{-2/3}\sqrt{\log(m)}L^{7/2}t^{5/3}\lambda^{-5/3}(1 + \sqrt{t/\lambda}))$.

**Bounding the third term in Inequality C.22.** Then, for the third term on the RHS, we first have

$$m^{-1} \cdot \|(\boldsymbol{\Sigma}_{i,t-1}^{(0)})^{-1} \cdot (\sum_{\tau \in [t-1]} w_{i,t}^{(\tau)} \cdot g(\boldsymbol{x}_\tau; \boldsymbol{\theta}_0) \cdot c_\tau)\|_{\boldsymbol{\Sigma}_{i,t-1}}$$

$$\leq m^{-1}\|(\boldsymbol{\Sigma}_{i,t-1})^{-1}(\sum_{\tau \in [t-1]} w_{i,t}^{(\tau)} \cdot g(\boldsymbol{x}_\tau; \boldsymbol{\theta}_0) \cdot c_\tau)\|_{\boldsymbol{\Sigma}_{i,t-1}}$$
$$+ m^{-1}\|\big((\boldsymbol{\Sigma}_{i,t-1})^{-1} - (\boldsymbol{\Sigma}_{i,t-1}^{(0)})^{-1}\big) \cdot (\sum_{\tau \in [t-1]} w_{i,t}^{(\tau)} \cdot g(\boldsymbol{x}_\tau; \boldsymbol{\theta}_0) \cdot c_\tau)\|_{\boldsymbol{\Sigma}_{i,t-1}}$$

$$\leq m^{-1}\|(\boldsymbol{\Sigma}_{i,t-1})^{-1}(\sum_{\tau \in [t-1]} w_{i,t}^{(\tau)} \cdot g(\boldsymbol{x}_\tau; \boldsymbol{\theta}_0) \cdot c_\tau)\|_{\boldsymbol{\Sigma}_{i,t-1}}$$
$$+ m^{-1}\|(\boldsymbol{\Sigma}_{i,t-1})^{-1}\big(\boldsymbol{\Sigma}_{i,t-1} - \boldsymbol{\Sigma}_{i,t-1}^{(0)}\big)(\boldsymbol{\Sigma}_{i,t-1}^{(0)})^{-1} \cdot (\sum_{\tau \in [t-1]} w_{i,t}^{(\tau)} \cdot g(\boldsymbol{x}_\tau; \boldsymbol{\theta}_0) \cdot c_\tau)\|_{\boldsymbol{\Sigma}_{i,t-1}}$$

$$\leq m^{-1}\|(\boldsymbol{\Sigma}_{i,t-1})^{-1}(\sum_{\tau \in [t-1]} w_{i,t}^{(\tau)} \cdot g(\boldsymbol{x}_\tau; \boldsymbol{\theta}_0) \cdot c_\tau)\|_{\boldsymbol{\Sigma}_{i,t-1}} + \mathcal{O}(Cm^{-2/3}\sqrt{\log(m)}t^{7/6}\lambda^{-13/6}L^{9/2})$$

$$\leq m^{-1}\|(\boldsymbol{\Sigma}_{i,t-1})^{-1}(\sum_{\tau \in [t-1]} w_{i,t}^{(\tau)} \cdot g(\boldsymbol{x}_\tau; \boldsymbol{\theta}_{\tau-1}) \cdot c_\tau)\|_{\boldsymbol{\Sigma}_{i,t-1}} + \mathcal{O}(Cm^{-2/3}\sqrt{\log(m)}t^{7/6}\lambda^{-13/6}L^{9/2})$$
$$+ m^{-1}\|(\boldsymbol{\Sigma}_{i,t-1})^{-1}(\sum_{\tau \in [t-1]} w_{i,t}^{(\tau)} \cdot (g(\boldsymbol{x}_\tau; \boldsymbol{\theta}_{\tau-1}) - g(\boldsymbol{x}_\tau; \boldsymbol{\theta}_0)) \cdot c_\tau)\|_{\boldsymbol{\Sigma}_{i,t-1}}$$

$$\leq m^{-1}\|(\boldsymbol{\Sigma}_{i,t-1})^{-1}(\sum_{\tau \in [t-1]} w_{i,t}^{(\tau)} \cdot g(\boldsymbol{x}_\tau; \boldsymbol{\theta}_{\tau-1}) \cdot c_\tau)\|_{\boldsymbol{\Sigma}_{i,t-1}} + \mathcal{O}(Cm^{-7/6}\sqrt{\log(m)}t^{1/6}\lambda^{-7/6}L^{7/2})$$
$$+ \mathcal{O}(Cm^{-2/3}\sqrt{\log(m)}t^{7/6}\lambda^{-13/6}L^{9/2}),$$

where the third inequality is due to Lemma G.6, and the last inequality is due to Lemma G.4. Then, recall that the weight $w_{i,t}^{(\tau)}$ from (4) for each previously chosen arm $\boldsymbol{x}_\tau$. In this case, we can further

have

$$m^{-1/2}\|(\boldsymbol{\Sigma}_{i,t-1})^{-1}(\sum_{\tau\in[t-1]} w_{i,t}^{(\tau)}\cdot g(\boldsymbol{x}_\tau;\boldsymbol{\theta}_{\tau-1})\cdot c_\tau)/\sqrt{m}\|_{\boldsymbol{\Sigma}_{i,t-1}}$$

$$= m^{-1/2}\|\sum_{\tau\in[t-1]} w_{i,t}^{(\tau)}\cdot g(\boldsymbol{x}_\tau;\boldsymbol{\theta}_{\tau-1})\cdot c_\tau/\sqrt{m}\|_{(\boldsymbol{\Sigma}_{i,t-1})^{-1}}$$

$$\leq m^{-1/2}\sum_{\tau\in[t-1]} w_{i,t}^{(\tau)}c_\tau\cdot\|g(\boldsymbol{x}_\tau;\boldsymbol{\theta}_{\tau-1})/\sqrt{m}\|_{(\boldsymbol{\Sigma}_{i,t-1})^{-1}}$$

where the first inequality is by applying the triangular inequality. By definition, we naturally have $\boldsymbol{\Sigma}_{i,t-1}\succeq\bar{\boldsymbol{\Sigma}}_{\tau-1}^{(\kappa)},\forall\tau\in[t-1]$. Next, we utilize Lemmas G.8 and the fact that $w_t^{\mathsf{min}}=\min\{w_{i,t}^{(\tau)}\}_{i\in[K],\tau\in[t-1]}=\kappa^2<1,\ \forall t\in[T]$ by scaling the parameter $\alpha$, which will lead to

$$m^{-1/2}\|(\boldsymbol{\Sigma}_{i,t-1})^{-1}(\sum_{\tau\in[t-1]} w_{i,t}^{(\tau)}\cdot g(\boldsymbol{x}_\tau;\boldsymbol{\theta}_{\tau-1})\cdot c_\tau)/\sqrt{m}\|_{\boldsymbol{\Sigma}_{i,t-1}}$$

$$\leq\mathcal{O}(m^{-1/2})\sum_{\tau\in[t-1]} w_{i,t}^{(\tau)}c_\tau\cdot\|g(\boldsymbol{x}_\tau;\boldsymbol{\theta}_{\tau-1})/\sqrt{m}\|_{(\boldsymbol{\Sigma}_{i,t-1})^{-1}}$$

$$\leq\mathcal{O}(m^{-1/2})\sum_{\tau\in[t-1]} w_{i,t}^{(\tau)}c_\tau\cdot\|g(\boldsymbol{x}_\tau;\boldsymbol{\theta}_{\tau-1})/\sqrt{m}\|_{(\bar{\boldsymbol{\Sigma}}_{\tau-1}^{(\kappa)})^{-1}}$$

$$\leq\mathcal{O}(m^{-1/2})\sum_{\tau\in[t-1]} c_\tau\cdot\frac{\alpha\cdot\min_{\boldsymbol{x}\in\mathcal{X}_t}\|g(\boldsymbol{x};\boldsymbol{\theta}_{i,t-1})/\sqrt{m}\|_{\bar{\boldsymbol{\Sigma}}_{t-1}^{-1}}^2}{\|g(\boldsymbol{x}_{i,t};\boldsymbol{\theta}_{i,t-1})/\sqrt{m}\|_{(\bar{\boldsymbol{\Sigma}}_{t-1}^{(\kappa)})^{-1}}}$$

$$\leq\mathcal{O}(m^{-1/2})\cdot C\cdot\frac{\alpha\cdot\min_{\boldsymbol{x}\in\mathcal{X}_t}\|g(\boldsymbol{x};\boldsymbol{\theta}_{i,t-1})/\sqrt{m}\|_{\bar{\boldsymbol{\Sigma}}_{t-1}^{-1}}^2}{\|g(\boldsymbol{x}_{i,t};\boldsymbol{\theta}_{i,t-1})/\sqrt{m}\|_{(\bar{\boldsymbol{\Sigma}}_{t-1}^{(\kappa)})^{-1}}},$$

where the third and the last inequalities are due to the definition of $w_{i,t}^{(\tau)}$ and the definition of corruption level $C$. Finally, summing up all the results will give the lemma.

$\square$

## C.9 Discussion on the Minimum Fraction Value $\beta$

In this subsection, we provide an exemplary upper bound of $\mathcal{O}(1/\beta)$. Inspired by [84], the following analysis is based on a scenario where the arm contexts nearly lie within a low-dimensional subspace of the RKHS, induced by the NTK defined in Definition 5.2. To begin, we recall that, as defined in (4), each candidate arm $\boldsymbol{x}_{i,t}\in\mathcal{X}_t$ is associated with the corresponding arm weight:

$$w_{i,t}^{(\tau)}=\min\left\{1,\frac{\alpha\cdot\min_{\boldsymbol{x}\in\mathcal{X}_t}\|g(\boldsymbol{x};\boldsymbol{\theta}_{t-1})/\sqrt{m}\|_{\bar{\boldsymbol{\Sigma}}_{t-1}^{-1}}^2}{g_\tau\cdot\|g(\boldsymbol{x}_{i,t};\boldsymbol{\theta}_{t-1})/\sqrt{m}\|_{(\bar{\boldsymbol{\Sigma}}_{t-1}^{(\kappa)})^{-1}}}\right\}=\min\left\{1,\alpha\cdot\mathsf{frac}_\tau(\boldsymbol{x}_{i,t};\mathcal{X}_t,\bar{\boldsymbol{\Sigma}}_{t-1})\right\},$$

where $\alpha>0$ is a tunable scaling parameter to control the arm weight value range, and $\mathsf{frac}(\cdot)$ is a shorthand for the fraction term. We denote the data-dependent minimum fraction value as $\beta=\min_{t\in[T],\tau\in[t-1]}\left[\min\left\{\mathsf{frac}_\tau(\boldsymbol{x}_t;\mathcal{X}_t,\bar{\boldsymbol{\Sigma}}_{t-1}),\mathsf{frac}_\tau(\widetilde{\boldsymbol{x}}_t;\mathcal{X}_t,\bar{\boldsymbol{\Sigma}}_{t-1})\right\}\right]$. In this case, the lower bound for $\mathsf{frac}_\tau(\boldsymbol{x}_{i,t};\mathcal{X}_t,\bar{\boldsymbol{\Sigma}}_{t-1})$, represented by $\beta$, can be expressed as

$$\frac{\min_{\boldsymbol{x}\in\mathcal{X}_t}\|g(\boldsymbol{x};\boldsymbol{\theta}_{t-1})/\sqrt{m}\|_{\bar{\boldsymbol{\Sigma}}_{t-1}^{-1}}^2}{\|g(\boldsymbol{x}_\tau;\boldsymbol{\theta}_{\tau-1})/\sqrt{m}\|_{(\bar{\boldsymbol{\Sigma}}_{\tau-1}^{(\kappa)})^{-1}}\cdot\|g(\boldsymbol{x}_{i,t};\boldsymbol{\theta}_{t-1})/\sqrt{m}\|_{(\bar{\boldsymbol{\Sigma}}_{t-1}^{(\kappa)})^{-1}}}\geq\frac{\min_{\boldsymbol{x}\in\mathcal{X}_t}\|g(\boldsymbol{x};\boldsymbol{\theta}_{t-1})/\sqrt{m}\|_{\bar{\boldsymbol{\Sigma}}_{t-1}^{-1}}^2}{\mathcal{O}(L\cdot\lambda^{-1})}$$

$$\geq\frac{\min_{\boldsymbol{x}\in\mathcal{X}_t}\|g(\boldsymbol{x};\boldsymbol{\theta}_{t-1})/\sqrt{m}\|_2^2\cdot\lambda_{\mathsf{min}}(\bar{\boldsymbol{\Sigma}}_{t-1}^{-1})}{\mathcal{O}(L\cdot\lambda^{-1})}$$

where the first inequality is because of Lemma G.2 and Lemma G.6. The second inequality is by applying Rayleigh-Ritz theorem.

Then, we let $\breve{\mathcal{P}}_{t-1} \subseteq \mathcal{P}_{t-1}$ being the collection of received records that only consists of unique chosen arms from $\mathcal{P}_{t-1}$. With $\breve{\boldsymbol{\theta}}_{t-1}$ be the parameters trained on $\breve{\mathcal{P}}_{t-1}$, applying Lemma G.4 and Lemma G.3, we have $\|g(\boldsymbol{x}';\boldsymbol{\theta}_{t-1})/\sqrt{m} - g(\boldsymbol{x}';\breve{\boldsymbol{\theta}}_{t-1})/\sqrt{m}\|_2 \leq \mathcal{O}(m^{-1/6}\lambda^{-1/6}t^{1/6}L^{7/2}\log(m))$, where $\boldsymbol{x}' = \arg\min_{\boldsymbol{x}\in\mathcal{X}_t}\|g(\boldsymbol{x};\boldsymbol{\theta}_{t-1})/\sqrt{m}\|_2$. Given the over-parameterization settings in Theorem 5.6 with sufficiently large $m$, we can have $\|g(\boldsymbol{x}';\boldsymbol{\theta}_{t-1})/\sqrt{m} - g(\boldsymbol{x}';\breve{\boldsymbol{\theta}}_{t-1})/\sqrt{m}\|_2 \ll \mathcal{O}(1)$. As a result, it leads to

$$
\frac{\min_{\boldsymbol{x}\in\mathcal{X}_t}\|g(\boldsymbol{x};\boldsymbol{\theta}_{t-1})/\sqrt{m}\|^2_{\bar{\boldsymbol{\Sigma}}_{t-1}^{-1}}}{\|g(\boldsymbol{x}_\tau;\boldsymbol{\theta}_{\tau-1})/\sqrt{m}\|_{(\bar{\boldsymbol{\Sigma}}_{\tau-1}^{(\kappa)})^{-1}} \cdot \|g(\boldsymbol{x}_{i,t};\boldsymbol{\theta}_{t-1})/\sqrt{m}\|_{(\bar{\boldsymbol{\Sigma}}_{t-1}^{(\kappa)})^{-1}}} \geq \frac{\min_{\boldsymbol{x}\in\mathcal{X}_t}\|g(\boldsymbol{x};\boldsymbol{\theta}_{t-1})/\sqrt{m}\|^2_2 \cdot \lambda_{\mathsf{min}}(\bar{\boldsymbol{\Sigma}}_{t-1}^{-1})}{\mathcal{O}(L \cdot \lambda^{-1})}
$$

$$
\geq \frac{\left(\|g(\boldsymbol{x}';\breve{\boldsymbol{\theta}}_{t-1})/\sqrt{m}\|^2_2 + \|g(\boldsymbol{x}';\boldsymbol{\theta}_{t-1})/\sqrt{m}\|^2_2 - \|g(\boldsymbol{x}';\breve{\boldsymbol{\theta}}_{t-1})/\sqrt{m}\|^2_2\right) \cdot \lambda_{\mathsf{min}}(\bar{\boldsymbol{\Sigma}}_{t-1}^{-1})}{\mathcal{O}(L \cdot \lambda^{-1})}
$$

$$
\geq \frac{\mathcal{O}(1) \cdot \lambda_{\mathsf{min}}(\bar{\boldsymbol{\Sigma}}_{t-1}^{-1})}{\mathcal{O}(L \cdot \lambda^{-1})}
$$

where the second inequality is derived based on $\|g(\boldsymbol{x}';\boldsymbol{\theta}_{t-1})/\sqrt{m}\|^2_2 - \|g(\boldsymbol{x}';\breve{\boldsymbol{\theta}}_{t-1})/\sqrt{m}\|^2_2 \geq -1 \cdot (\|g(\boldsymbol{x}';\boldsymbol{\theta}_{t-1})/\sqrt{m} - g(\boldsymbol{x}';\breve{\boldsymbol{\theta}}_{t-1})/\sqrt{m}\|_2 \cdot \|g(\boldsymbol{x}';\boldsymbol{\theta}_{t-1})/\sqrt{m} + g(\boldsymbol{x}';\breve{\boldsymbol{\theta}}_{t-1})/\sqrt{m}\|_2)$, and we bound the two multipliers separately with choice of $m$. The last inequality is by applying Theorem 3 of [3] with the fact that $f(\cdot;\boldsymbol{\theta}_{t-1}) = \mathcal{O}(1)$ (Lemma B.2 in [21]).

Afterwards, regarding the lower bound of $\lambda_{\mathsf{min}}(\bar{\boldsymbol{\Sigma}}_{t-1}^{-1})$, since $\lambda_{\mathsf{min}}(\bar{\boldsymbol{\Sigma}}_{t-1}^{-1}) = \frac{1}{\lambda_{\mathsf{max}}(\bar{\boldsymbol{\Sigma}}_{t-1})}$, we need to find the upper bound for $\|\bar{\boldsymbol{\Sigma}}_{t-1}\|_2$. Denoting $\mathbf{G}_{t-1}^{(0)} = [g(\boldsymbol{x}_1;\boldsymbol{\Theta}_0), g(\boldsymbol{x}_2;\boldsymbol{\Theta}_0), \ldots, g(\boldsymbol{x}_{t-1};\boldsymbol{\Theta}_0)]^\intercal \in \mathbb{R}^{(t-1)\times p}$ as the gradient matrix, as well as $\mathbf{H}_{t-1} \in \mathbb{R}^{(t-1)\times(t-1)}$ as the NTK matrix constructed from the chosen arms $\{\boldsymbol{x}_\tau\}_{\tau\in[t-1]}$, with sufficient network width $m$ (Theorem 5.6), we first have

$$
\|\mathbf{G}_{t-1}^{(0)}(\mathbf{G}_{t-1}^{(0)})^\intercal/m - \mathbf{H}_{t-1}\|_2 \leq \mathcal{O}(1)
$$

and this inequality is because of Lemma B.1 in [86] by setting $\epsilon = t - 1$. In this case, we will have

$$
\|\bar{\boldsymbol{\Sigma}}_{t-1}\|_2 - \lambda_{\mathsf{max}}(\mathbf{H}_{t-1}) \leq \|\bar{\boldsymbol{\Sigma}}_{t-1}\|_2 - \|(\mathbf{G}_{t-1}^{(0)})^\intercal\mathbf{G}_{t-1}^{(0)}/m\|_2 + \|\mathbf{G}_{t-1}^{(0)}(\mathbf{G}_{t-1}^{(0)})^\intercal/m\|_2 - \lambda_{\mathsf{max}}(\mathbf{H}_{t-1})
$$

$$
\leq \|\bar{\boldsymbol{\Sigma}}_{t-1} - (\mathbf{G}_{t-1}^{(0)})^\intercal\mathbf{G}_{t-1}^{(0)}/m\|_2 + \|\mathbf{G}_{t-1}^{(0)}(\mathbf{G}_{t-1}^{(0)})^\intercal/m - \mathbf{H}_{t-1}\|_2
$$

$$
\leq \mathcal{O}(m^{-1/6}\sqrt{\log(m)}L^4 t^{7/6}\lambda^{-1/6}) + \mathcal{O}(\lambda) + \mathcal{O}(1)
$$

where the last inequality is by applying Lemma G.6. This leads to the result that $\|\bar{\boldsymbol{\Sigma}}_{t-1}\|_2 \leq \lambda_{\mathsf{max}}(\mathbf{H}_{t-1}) + \mathcal{O}(m^{-1/6}\sqrt{\log(m)}L^4 t^{7/6}\lambda^{-1/6}) + \mathcal{O}(\lambda) + \mathcal{O}(1)$.

Then, inspired by Section D in [84], based on the formulation from [14] and [20], we can consider that each entry of $\mathbf{H}_{t-1}$ is generated by

$$
[\mathbf{H}_{t-1}]_{i,i'} = \sum_{k=0}^\infty \mu_k \sum_{j=1}^{N(d,k)} Y_{k,j}(\boldsymbol{x}_i)Y_{k,j}(\boldsymbol{x}_{i'}),
$$

where $Y_{k,j}$ are linearly independent spherical harmonics, w.r.t. degree $k$ and $d$ variables. $N(d,k) = \frac{2k+d-2}{k} \cdot C_{k+d-3}^{d-2}, \mu_k = \Theta(\max\{k^{-d}, (d-1)^{1-k}\})$. With the above feature mapping, if we have a subspace of the RKHS, such that the feature mapping in the RKHS is close enough to its projection onto this subspace, we can have $\lambda_{\mathsf{max}}(\mathbf{H}_{t-1}) = \|\mathbf{H}_{t-1}\|_2 \leq \mathcal{O}(1)$. As a result, summing up the results and with sufficiently large $m$, we can have $\frac{1}{\beta} \leq \mathcal{O}(L\lambda^{-1})$.

## D  Bounding the Difference of Trained Parameters and Regression Parameters

In section, with weighted Gradient Descent, we provide the upper bound in terms of the distance between GD trained parameters $\boldsymbol{\theta}_{t-1}$ and the regression parameters $(\boldsymbol{\Sigma}_{t-1}^{(0)})^{-1}\boldsymbol{b}_{t-1}^{(0)}$. The results will be applied to the proof flow of both NeuralUCB-WGD and R-NeuralUCB. First, with the definitions

in Subsec. C.2, we have the gradient-based regression parameters specified to arm $\boldsymbol{x}_t \in \mathcal{X}_t$ as

$$\boldsymbol{\Sigma}_{t-1}^{(0)} = \lambda \mathbf{I} + \sum_{\tau \in [t-1]} w_t^{(\tau)} g(\boldsymbol{x}_\tau; \boldsymbol{\theta}_0) g(\boldsymbol{x}_\tau; \boldsymbol{\theta}_0)^\mathsf{T}/m,$$

$$\boldsymbol{\Sigma}_{t-1} = \lambda \mathbf{I} + \sum_{\tau \in [t-1]} w_t^{(\tau)} g(\boldsymbol{x}_\tau; \boldsymbol{\theta}_{\tau-1}) g(\boldsymbol{x}_\tau; \boldsymbol{\theta}_{\tau-1})^\mathsf{T}/m,$$

$$\boldsymbol{b}_{t-1}^{(0)} = \sum_{\tau \in [t-1]} w_t^{(\tau)} g(\boldsymbol{x}_\tau; \boldsymbol{\theta}_0) \cdot r_\tau/\sqrt{m}, \qquad\qquad \boldsymbol{b}_{t-1} = \sum_{\tau \in [t-1]} w_t^{(\tau)} g(\boldsymbol{x}_\tau; \boldsymbol{\theta}_{\tau-1}) \cdot r_\tau/\sqrt{m}$$

where $\{\boldsymbol{x}_\tau, r_\tau\}, \tau \in [t]$ respectively stands for the chosen arms as well as their rewards. For notation simplicity, we also use $w_t^{(\tau)}, \tau \in [t-1]$ to denote the arm weights for the chosen arm $\boldsymbol{x}_t$ respectively.

Analogously, given an candidate arm $\boldsymbol{x}_{i,t} \in \mathcal{X}_t$ with arm weight $w_{i,t}^{(\tau)}$ defined in (4), we have a series auxiliary gradient sequences $\{\boldsymbol{\Theta}^{(0)}, \boldsymbol{\Theta}^{(1)}, \ldots, \boldsymbol{\Theta}^{(J)}\}$ as in G.3, such that for $j$-th iteration

$$\boldsymbol{\Theta}^{(j+1)} = \boldsymbol{\Theta}^{(j)} - \eta \cdot \left[ \mathbf{J}^{(0)} \cdot \mathbf{W} \cdot \left([\mathbf{J}^{(0)}]^\mathsf{T}(\boldsymbol{\Theta}^{(j)} - \boldsymbol{\theta}_0) - \boldsymbol{y}\right) + m\lambda(\boldsymbol{\Theta}^{(j)} - \boldsymbol{\theta}_0) \right]$$

where $\mathbf{J}^{(0)} := \left(g(\boldsymbol{x}_1; \boldsymbol{\theta}_0), g(\boldsymbol{x}_2; \boldsymbol{\theta}_0), \ldots, g(\boldsymbol{x}_{t-1}; \boldsymbol{\theta}_0)\right) \in \mathbb{R}^{p \times (t-1)}$, and $\mathbf{W}$ refers to the diagonal matrix of arm weights $\{w_{i,t}^{(\tau)}\}_{\tau \in [t-1]}$, along with the reward vectors $\boldsymbol{y} \in \mathbb{R}^{t-1}$ separately being the vector of received rewards and corruption-free rewards. In particular, the auxiliary sequence $\boldsymbol{\Theta}^{(j)}$ can be deemed as applying Gradient Descent to solve the following optimization problem

$$\min_{\boldsymbol{\Theta}} \mathcal{L}(\boldsymbol{\Theta}) = \sum_{\tau \in [t-1]} \frac{1}{2} \cdot w_{i,t}^{(\tau)} \cdot \left\| [\mathbf{J}^{(0)}]_\tau^\mathsf{T}(\boldsymbol{\Theta} - \boldsymbol{\theta}_0) - y_\tau \right\|_2^2 + \frac{1}{2} \cdot m\lambda \cdot \left\| \boldsymbol{\Theta} - \boldsymbol{\theta}_0 \right\|_2^2$$

where $[\mathbf{J}^{(0)}]_\tau$ refers to the $\tau$-th column of the matrix $\mathbf{J}^{(0)}$. Analogously, we can also derive the optimization problem for the sequence of corruption-free auxiliary parameters $\widetilde{\boldsymbol{\Theta}}^{(j)}$, by applying the same definition of weight matrix $\mathbf{W}$. By the definition of arm weights, we will also have $\|\mathbf{W}\|_2 \leq 1$.

**Lemma D.1** (Lemma B.2 of [86]). *With the notation and conditions in Theorem 5.6, consider* $m \geq \Omega(poly(T, L, \check{\lambda}_0^{-1}, \lambda^{-1})$ *and* $\eta \leq \mathcal{O}(\frac{1}{m\lambda + tmL})$. *For round* $t \in [T]$, *we have*

$$\|\boldsymbol{\theta}_{t-1} - \boldsymbol{\theta}_0 - (\boldsymbol{\Sigma}_{t-1}^{(0)})^{-1}\boldsymbol{b}_{t-1}^{(0)}\|_2$$
$$\leq (1 - \eta m\lambda)^{J/2}\sqrt{t/(m\lambda)} + \mathcal{O}(m^{-2/3}t^{5/3}\sqrt{\log(m)}L^{7/2}\lambda^{-5/3}(1 + \sqrt{t/\lambda}))$$

*with the $J$ iterations of Gradient Descent process.*

**Proof.** The proof of this lemma follows an analogous approach as in Lemma B.2 of [86]. Here, similar to the previous sequence $\{\boldsymbol{\Theta}^{(0)}, \boldsymbol{\Theta}^{(1)}, \ldots, \boldsymbol{\Theta}^{(J)}\}$, we denote another set of auxiliary sequences to simulate the $J$ iterations of GD, by

$$\boldsymbol{\theta}^{(j+1)} = \boldsymbol{\theta}^{(j)} - \eta \cdot \left[ \mathbf{J}^{(j)} \cdot \mathbf{W} \cdot \left(\boldsymbol{f}^{(j)} - \boldsymbol{y}\right) + m\lambda(\boldsymbol{\theta}^{(j)} - \boldsymbol{\theta}_0) \right]$$

where we denote the corresponding gradient matrix at the $j$-th iteration as $\mathbf{J}^{(j)} = \left(g(\boldsymbol{x}_1; \boldsymbol{\theta}^{(j)}), g(\boldsymbol{x}_2; \boldsymbol{\theta}^{(j)}), \ldots, g(\boldsymbol{x}_{t-1}; \boldsymbol{\theta}^{(j)})\right) \in \mathbb{R}^{p \times (t-1)}$, as well as the vector of network outputs $\boldsymbol{f}^{(j)} = \left(f(\boldsymbol{x}_1; \boldsymbol{\theta}^{(j)}), f(\boldsymbol{x}_2; \boldsymbol{\theta}^{(j)}), \ldots, f(\boldsymbol{x}_{t-1}; \boldsymbol{\theta}^{(j)})\right) \in \mathbb{R}^{t-1}$. In this case, we can have the difference between parameter sequences as

$$\|\boldsymbol{\theta}^{(j+1)} - \boldsymbol{\Theta}^{(j+1)}\|$$
$$= \|(1 - \eta m\lambda) \cdot (\boldsymbol{\theta}^{(j)} - \boldsymbol{\Theta}^{(j)}) - \eta\mathbf{W}(\mathbf{J}^{(j)} - \mathbf{J}^{(0)})(\boldsymbol{f}^{(j)} - \boldsymbol{y}) - \eta\mathbf{W}(\boldsymbol{f}^{(j)} - [\mathbf{J}^{(0)}]^\mathsf{T}(\boldsymbol{\Theta}^{(j)} - \boldsymbol{\theta}_0))\|$$
$$\leq \|(1 - \eta m\lambda) \cdot (\boldsymbol{\theta}^{(j)} - \boldsymbol{\Theta}^{(j)})\| + \eta\|\mathbf{W}(\mathbf{J}^{(j)} - \mathbf{J}^{(0)})(\boldsymbol{f}^{(j)} - \boldsymbol{y})\| + \eta\|\mathbf{J}^{(0)}\mathbf{W}(\boldsymbol{f}^{(j)} - [\mathbf{J}^{(0)}]^\mathsf{T}(\boldsymbol{\Theta}^{(j)} - \boldsymbol{\theta}_0))\|$$
$$\leq \|(1 - \eta m\lambda)(\boldsymbol{\theta}^{(j)} - \boldsymbol{\Theta}^{(j)})\| + \eta\|\mathbf{W}\|\|(\mathbf{J}^{(j)} - \mathbf{J}^{(0)})(\boldsymbol{f}^{(j)} - \boldsymbol{y})\| + \eta\|\mathbf{J}^{(0)}\mathbf{W}\|\|\boldsymbol{f}^{(j)} - [\mathbf{J}^{(0)}]^\mathsf{T}(\boldsymbol{\Theta}^{(j)} - \boldsymbol{\theta}_0)\|$$
$$\leq \underbrace{\|(1 - \eta m\lambda) \cdot (\boldsymbol{\theta}^{(j)} - \boldsymbol{\Theta}^{(j)})\|}_{I_4} + \underbrace{\eta \cdot \|(\mathbf{J}^{(j)} - \mathbf{J}^{(0)})(\boldsymbol{f}^{(j)} - \boldsymbol{y})\|}_{I_5} + \underbrace{\eta \cdot \|\mathbf{J}^{(0)}\|\|\boldsymbol{f}^{(j)} - [\mathbf{J}^{(0)}]^\mathsf{T}(\boldsymbol{\Theta}^{(j)} - \boldsymbol{\theta}_0)\|}_{I_6}$$

where the first inequality is by applying the triangular inequality. The second inequality is by using Cauchy-Schwartz inequality, and the third inequality is due to the fact that $\|\mathbf{W}\|_2 \leq 1$. Here, the first term $I_4$ can be bounded recursively. For the second term $I_5$ on the RHS, we have

$$I_5 = \eta\|(\mathbf{J}^{(j)} - \mathbf{J}^{(0)})(\boldsymbol{f}^{(j)} - \boldsymbol{y})\| \leq \eta\|(\mathbf{J}^{(j)} - \mathbf{J}^{(0)})\|\|(\boldsymbol{f}^{(j)} - \boldsymbol{y})\| \leq \mathcal{O}(\eta t^{7/6} m^{1/3} \sqrt{\log(m)} L^{7/2} \lambda^{-1/6})$$

by extending Lemma G.4 to the matrix $\mathbf{J}$. Since we have the arm weights $w_{i,t}^{(\tau)} \leq \mathcal{O}(1)$ due to the maximum cap as well as the choice of parameter $\alpha$, we can also directly apply the conclusion of Lemma C.3 in [86] for the second inequality. Meanwhile, for the third term $I_6$, we will have

$$
\begin{aligned}
I_6 &= \eta \cdot \|\mathbf{J}^{(0)}\|\|\boldsymbol{f}^{(j)} - [\mathbf{J}^{(0)}]^\mathsf{T}(\boldsymbol{\Theta}^{(j)} - \boldsymbol{\theta}_0)\| \\
&\leq \eta \cdot \|\mathbf{J}^{(0)}\| \cdot \max_{\tau \in [t-1]} \sqrt{t} \cdot |f(\boldsymbol{x}_\tau; \boldsymbol{\theta}^j) - f(\boldsymbol{x}_\tau; \boldsymbol{\theta}_0) - \langle g(\boldsymbol{x}_\tau; \boldsymbol{\theta}_0), \boldsymbol{\theta}^j - \boldsymbol{\theta}_0 \rangle| \\
&\leq \mathcal{O}(\eta t^{5/3} m^{1/3} \sqrt{\log(m)} L^{7/2} \lambda^{-2/3})
\end{aligned}
$$

by extending Lemma G.2 to the gradient matrix setting, as well as applying the Lemma G.5 on the absolute value. Afterwards, we can integrate these three terms, which will lead to

$$
\begin{aligned}
\|\boldsymbol{\theta}^{(j+1)} - \boldsymbol{\Theta}^{(j+1)}\| &\leq \|(1 - \eta m \lambda) \cdot (\boldsymbol{\theta}^{(j)} - \boldsymbol{\Theta}^{(j)})\| \\
&\quad + \mathcal{O}(\eta t^{7/6} m^{1/3} \sqrt{\log(m)} L^{7/2} \lambda^{-1/6}) + \mathcal{O}(\eta t^{5/3} m^{1/3} \sqrt{\log(m)} L^{7/2} \lambda^{-2/3}) \\
&\leq (1 - \eta m \lambda)\|\boldsymbol{\theta}^{(j)} - \boldsymbol{\Theta}^{(j)}\| + \mathcal{O}(\eta t^{7/6} m^{1/3} \sqrt{\log(m)} L^{7/2} \lambda^{-1/6}) + \mathcal{O}(\eta t^{5/3} m^{1/3} \sqrt{\log(m)} L^{7/2} \lambda^{-2/3}) \\
&\leq \mathcal{O}(m^{-2/3} t^{5/3} \sqrt{\log(m)} L^{7/2} \lambda^{-5/3}(1 + \sqrt{t/\lambda}))
\end{aligned}
$$

where for term $I_4$, the last inequality is obtained by recursively applying the process to $\|\boldsymbol{\theta}^{(0)} - \boldsymbol{\Theta}^{(0)}\| = 0$ and substituting the chosen upper bound for the learning rate $\eta$. Then, with a sufficiently large network width $m$ as indicated in the lemma, we have

$$
\begin{aligned}
\|\boldsymbol{\theta}_{t-1} - \boldsymbol{\theta}_0 - (\boldsymbol{\Sigma}_{t-1}^{(0)})^{-1}\boldsymbol{b}_{t-1}^{(0)}\|_2 &\leq \|\boldsymbol{\theta}^{(j+1)} - \boldsymbol{\Theta}^{(j+1)}\| + \|\boldsymbol{\Theta}^{(j+1)} - \boldsymbol{\theta}_0 - (\boldsymbol{\Sigma}_{t-1}^{(0)})^{-1}\boldsymbol{b}_{t-1}^{(0)}\|_2 \\
&\leq \|\boldsymbol{\Theta}^{(j+1)} - \boldsymbol{\theta}_0 - (\boldsymbol{\Sigma}_{t-1}^{(0)})^{-1}\boldsymbol{b}_{t-1}^{(0)}\|_2 + \mathcal{O}(m^{-2/3} t^{5/3} \sqrt{\log(m)} L^{7/2} \lambda^{-5/3}(1 + \sqrt{t/\lambda})) \\
&\leq (1 - \eta m \lambda)^{j/2} \sqrt{t/(m\lambda)} + \mathcal{O}(m^{-2/3} t^{5/3} \sqrt{\log(m)} L^{7/2} \lambda^{-5/3}(1 + \sqrt{t/\lambda}))
\end{aligned}
$$

where the first inequality is by applying triangular inequality, and the second inequality is by applying the previous conclusion. The last inequality is by applying Lemma C.4 in [86] with the fact that $\|\boldsymbol{\theta}^{(j+1)} - \boldsymbol{\Theta}^{(j+1)}\| \leq \sqrt{t/(m\lambda)}$. Since we have the arm weights $w_{i,t}^{(\tau)} \leq 1$ due to its maximum bound, we apply the conclusion of Lemma C.4 in [86] for the last inequality. $\qquad\square$

# E  A Base Algorithm: NeuralUCB-WGD

Recall that for each candidate arm $\boldsymbol{x}_{i,t} \in \mathcal{X}_t$, its corruption-free expected reward is generated by an unknown reward mapping function $h(\cdot)$. Following existing neural bandit approaches, we use a neural network $f(\cdot)$ to approximate $h(\cdot)$ for estimating arm rewards. Consistent with the main text and R-NeuralUCB, we consider the network $f(\cdot; \boldsymbol{\theta})$ to be a fully connected (FC) network with depth $L \geq 2$ and width $m \in \mathbb{N}^+$:

$$f(\boldsymbol{x}; \boldsymbol{\theta}) := \sqrt{m}\boldsymbol{\theta}_L \sigma(\boldsymbol{\theta}_{L-1}\sigma(\boldsymbol{\theta}_{L-2}\ldots\sigma(\boldsymbol{\theta}_1\boldsymbol{x}))),$$

where $\sigma(\cdot)$ denotes the ReLU activation function, and the trainable weight matrices are $\boldsymbol{\theta}_1 \in \mathbb{R}^{m \times d}$, $\boldsymbol{\theta}_l \in \mathbb{R}^{m \times m}$ for $2 \leq l \leq L - 1$, and $\boldsymbol{\theta}_L \in \mathbb{R}^{1 \times m}$. For simplicity, we also denote the vectorized parameters as

$$\boldsymbol{\theta} := [\text{vec}(\boldsymbol{\theta}_1)^\intercal, \text{vec}(\boldsymbol{\theta}_2)^\intercal, \ldots, \text{vec}(\boldsymbol{\theta}_L)]^\intercal \in \mathbb{R}^p,$$

with dimensionality $p$ and randomly initialized parameters $\boldsymbol{\theta}_0$. Similar to R-NeuralUCB, we also define $g(\boldsymbol{x}; \boldsymbol{\theta}) = \text{vec}(\nabla_{\boldsymbol{\theta}} f(\boldsymbol{x}; \boldsymbol{\theta})) \in \mathbb{R}^p$ as the vectorized network gradients, for input $\boldsymbol{x}$ and parameters $\boldsymbol{\theta}$.

## E.1  NeuralUCB-WGD: Neural-UCB with Weighted GD

Then, we introduce the workflow of our base algorithm NeuralUCB-WGD (Algorithm 3), which stands for Neural-UCB with Weighted GD. Here, NeuralUCB-WGD can be considered as a simplified version of R-NeuralUCB, where all the candidate arms $\mathcal{X}_t$ in round $t$ will share the same neural network for decision making. The idea is that although we do not know which training samples are corrupted, we can reduce the effects caused by the potential corruption instead, by paying relatively more attention on the samples with low uncertainty for a stable training process.

---

**Algorithm 3** Neural-UCB with Weighted GD (NeuralUCB-WGD)

---

1: **Input:** Time horizon $T$. GD steps $J$. Learning rate $\eta$. Exploration coefficient $\nu \geq 0$. Scaling coefficient $\alpha > 0$. Norm parameter $S$, regularization parameter $\lambda$.
2: **Initialization:** Initialized parameters $\boldsymbol{\theta}_0$. Covariance matrix $\boldsymbol{\Gamma}_0 = \lambda\mathbf{I}$. Received records $\mathcal{P}_0 = \emptyset$.
3: **for** each round $t \in [T]$ **do**
4:     Observe candidate arms $\mathcal{X}_t = \{\boldsymbol{x}_{i,t}\}_{i \in [K]}$.
5:     **for** each arm $\boldsymbol{x}_{i,t} \in \mathcal{X}_t$ **do**
6:         Calculate its benefit score $U(\boldsymbol{x}_{i,t})$, based on reward estimation $f(\boldsymbol{x}_{i,t}; \boldsymbol{\theta}_{i,t-1})$ and the UCB-type exploration score for arm $\boldsymbol{x}_{i,t}$, (E.1).
7:     **end for**
8:     Recommend arm based on benefit scores $\boldsymbol{x}_t = \arg\max_{\boldsymbol{x}_{i,t} \in \mathcal{X}_t} \left[U(\boldsymbol{x}_{i,t})\right]$.
9:     Receive arm reward $r_t$, and update the records, such that $\mathcal{P}_t = \mathcal{P}_{t-1} \cup \{(\boldsymbol{x}_t, r_t)\}$. Then, save the corresponding weight for chosen arm $\boldsymbol{x}_t$, as $w_t = \min\{1, \alpha/\|g(\boldsymbol{x}_t; \boldsymbol{\theta}_{t-1})/\sqrt{m}\|_{\boldsymbol{\Gamma}_{t-1}^{-1}}\}$.
10:    Update the gradient covariance matrix $\boldsymbol{\Gamma}_t = \boldsymbol{\Gamma}_{t-1} + w_t \cdot g(\boldsymbol{x}_t; \boldsymbol{\theta}_{t-1}) \cdot g(\boldsymbol{x}_t; \boldsymbol{\theta}_{t-1})^\intercal/m$.
11:    Starting from random initialization $\boldsymbol{\theta}_0$, update the network parameters to $\boldsymbol{\theta}_t$, based on $J$ iterations of Gradient Descent and training data $\mathcal{P}_t$.
12: **end for**

---

**Arm Selection.** In each round $t \in [T]$, after observing the candidate arms $\mathcal{X}_t$ for selection, we calculate the reward estimation and the UCB score for arm selection (lines 5-7, Algorithm 3). Similar to R-NeuralUCB, in terms of the arm selection (line 8, Algorithm 3), we determine the chosen arm $\boldsymbol{x}_t \in \mathcal{X}_t$ with the highest benefit score, by $\boldsymbol{x}_t = \arg\max_{\boldsymbol{x}_{i,t} \in \mathcal{X}_t} \left[U(\boldsymbol{x}_{i,t})\right]$. Here, with the NTK norm parameter $S > 0$ and the probability at least $1 - \delta$ given probability parameter $\delta \in (0, 1)$, we formulate the benefit score $U(\boldsymbol{x}_{i,t})$, along with a UCB-type exploration strategy (motivated by Lemma F.1), as

$$U(\boldsymbol{x}_{i,t}) = f(\boldsymbol{x}_{i,t}; \boldsymbol{\theta}_{t-1}) + \mathcal{O}\Big(\nu\sqrt{\log\frac{\det(\boldsymbol{\Gamma}_{t-1})}{\det(\lambda\mathbf{I})} - 2\log(\delta)} + \lambda^{1/2}S\Big) \cdot \|g(\boldsymbol{x}_{i,t}; \boldsymbol{\theta}_{t-1})/\sqrt{m}\|_{\boldsymbol{\Gamma}_{t-1}^{-1}}$$

$$\text{(E.1)}$$

where $\boldsymbol{\theta}_{t-1}$ refer to network parameters in round $t$ before GD, which have been trained with received records $\mathcal{P}_{t-1}$, and $\lambda > 0$ is regularization parameter. Different from conventional

neural bandit works, our gradient covariance matrix is defined as $\mathbf{\Gamma}_{t-1} = \lambda\mathbf{I} + \sum_{\tau\in[t-1]} w_\tau \cdot g(\boldsymbol{x}_\tau;\boldsymbol{\theta}_{\tau-1})g(\boldsymbol{x}_\tau;\boldsymbol{\theta}_{\tau-1})^{\mathsf{T}}/m$. Here, to quantify the arm uncertainty level, inspired by [42], we define the sample weight as $w_\tau = \min\{1, \alpha/\|g(\boldsymbol{x}_\tau;\boldsymbol{\theta}_{\tau-1})/\sqrt{m}\|_{\mathbf{\Gamma}_{\tau-1}^{-1}}\}$ based on the gradient vector $g(\boldsymbol{x}_\tau;\boldsymbol{\theta}_{\tau-1}) = \text{vec}\big(\nabla_{\boldsymbol{\theta}} f(\boldsymbol{x}_\tau;\boldsymbol{\theta}_{\tau-1})\big) \in \mathbb{R}^p$, and it is scaled by a tunable parameter $\alpha > 0$. Notice that the arm weight $w_\tau$ is inversely proportional to our UCB-type exploration score in (E.1). Since the UCB-based exploration score can be considered to quantify reward estimation uncertainty levels [24, 72, 86], we thus assign small weights to training samples with high uncertainty. Recall that in (1), we apply $\nu$ to characterize the random noise $\epsilon$. Similar to R-NeuralUCB, when this value is unknown, we alternatively deem $\nu \geq 0$ as a tunable exploration parameter to control the exploration intensity analogous to existing works (e.g., [86]).

After receiving the reward $r_t$ for the chosen arm $\boldsymbol{x}_t$, we update the records to $\mathcal{P}_t$. The arm context $\boldsymbol{x}_t$ and its received reward $r_t$ are added to the collection $\mathcal{P}_t$, along with their weight $w_t = \min\{1, \alpha/\|g(\boldsymbol{x}_t;\boldsymbol{\theta}_{t-1})/\sqrt{m}\|_{\mathbf{\Gamma}_{t-1}^{-1}}\}$ (line 9, Algorithm 3).

**Model Training.** Afterwards, we perform $J$ iterations of GD to update the network parameters (line 11, Algorithm 3). With $\mathcal{P}_t = \{\boldsymbol{x}_\tau, r_\tau\}_{\tau\in[t]}$ up to round $t$, we train the model parameters $\boldsymbol{\theta}_t$, through $\boldsymbol{\theta}_t^{(j)} = \boldsymbol{\theta}_t^{(j-1)} - \eta\nabla_{\boldsymbol{\theta}}\mathcal{L}(\mathcal{P}_t;\boldsymbol{\theta}_t^{(j-1)})$. Here, $\boldsymbol{\theta}_t^{(j)}, j \in [J]$ are the parameters after the $j$-th GD iteration, starting from randomly initialized ones $\boldsymbol{\theta}_t^{(0)} = \boldsymbol{\theta}_0$, and $\eta > 0$ refers to the learning rate. Different from existing neural bandit works (e.g., [86, 84]) which consider all received records to be equally important with the ordinary $L_2$ loss function, we alternatively define the weighted loss function as

$$\mathcal{L}(\mathcal{P};\boldsymbol{\theta}) = \sum_{(\boldsymbol{x}_\tau, r_\tau)\in\mathcal{P}} \frac{w_\tau}{2}\big|f(\boldsymbol{x}_\tau;\boldsymbol{\theta}) - r_\tau\big|^2 + \frac{m\lambda}{2}\|\boldsymbol{\theta} - \boldsymbol{\theta}_0\|_2^2$$

where $\lambda > 0$ is the regularization parameter as in (E.1), and we have previously defined sample weights $w_\tau, \tau \in [t]$ associated with each chosen arm-reward pair $(\boldsymbol{x}_\tau, r_\tau) \in \mathcal{P}_t$. In summary, the intuition is that, when defining the loss function for the received records $\mathcal{P}_t = \{\boldsymbol{x}_\tau, r_\tau\}_{\tau\in[t]}$ up to round $t$, we aim to give extra emphasis to samples with low estimation uncertainty. Intuitively, if samples with high uncertainty are indeed corrupted by the adversary, they are more likely to significantly disrupt the internal decision-making process, thereby impacting the stability of reward estimation. To mitigate this risk, even though we cannot identify which training samples are corrupted, we conservatively assign smaller weights to high-uncertainty samples, supporting a stable and robust GD training process.

## E.2 Regret Analysis for NeuralUCB-WGD

For the theoretical analysis, different from that of R-NeuralUCB (Subsection 5.1), we focus on the case where the corruption level $C$ is known, a common setting in existing works [16, 42, 76, 17]. This assumption allows us to appropriately select the parameter $\alpha$ in Algorithm 3 to achieve a tighter regret bound. Additionally, we briefly discuss potential outcomes if $C$ is unknown. Recall that our objective (2) is to minimize the overall pseudo-regret in terms of the corruption-free expected reward over a finite horizon of $T$ rounds: $R(T) = \sum_{t=1}^T \mathbb{E}[\widetilde{r}_t^* - \widetilde{r}_t]$, where $\mathbb{E}[\widetilde{r}_t] = h(\boldsymbol{x}_t)$ denotes the expected corruption-free reward from the chosen arm $\boldsymbol{x}_t$, and $\mathbb{E}[\widetilde{r}_t^*] = \max_{\boldsymbol{x}_{i,t}\in\mathcal{X}_t}[h(\boldsymbol{x}_{i,t})]$ represents the expected reward of the optimal arm.

To address challenges in regret analysis, we define two sets of regression parameters corresponding to the corrupted model and the corruption-free model. Using the corruption-free model $f(\cdot;\widetilde{\boldsymbol{\theta}})$, we derive the confidence ellipsoid around its parameters $\widetilde{\boldsymbol{\theta}}$, which serves as a proxy for updating the confidence ellipsoid around the trained corrupted model parameters $\boldsymbol{\theta}$. With the updated confidence ellipsoid and concentration results, we then finalize the regret upper bound. Here, without carefully designing the arm weights $w_\tau, \tau \in [t]$ (Algorithm 3) and structuring the regret analysis workflow, deriving the regret upper bound under adversarial corruption settings would be impractical. The following Theorem E.1 provides a bound on the cumulative pseudo-regret for NeuralUCB-WGD.

**Theorem E.1.** *Given the finite horizon* $T \in \mathbb{N}^+$, *denote* $S \geq \sqrt{2\breve{\boldsymbol{h}}^{\mathsf{T}}\breve{\mathbf{H}}^{-1}\breve{\boldsymbol{h}}}$. *Suppose probability parameter* $\delta \in (0,1)$, *network width* $m \geq \Omega(poly(T, L, C, \breve{\lambda}_0^{-1}, \lambda^{-1}, S^{-1}) \cdot \log(1/\delta))$, $\eta \leq \mathcal{O}((TmL + m\lambda)^{-1})$, $J \geq \widetilde{\mathcal{O}}(TL/\lambda)$, *and* $\lambda \geq S^{-2}$. *Let* $f(\cdot)$ *be the L-layer FC network with*

*width $m$, and set $\alpha = 1/C$ without the prior knowledge of $\widetilde{d}$. Then, with probability at least $1 - \delta$ over random initialization, NeuralUCB-WGD achieves the regret upper bound:*

$$R(T) \leq \widetilde{\mathcal{O}}\left(\sqrt{\widetilde{d}T}\right) \cdot \widetilde{\mathcal{O}}\left(\nu\sqrt{\widetilde{d} - 2\log(\delta)} + \lambda^{1/2}S\right) + \mathcal{O}\left(C\widetilde{d}\log(1 + TK/\lambda)\right) + \mathcal{O}(C\widetilde{d} \cdot \lambda^{1/2}S)$$

$$+ \mathcal{O}\left(C\widetilde{d} \cdot \left(\nu\sqrt{\widetilde{d}\log(1 + TK/\lambda) - 2\log(\delta)}\right)\right)$$

The proof of Theorem E.1 is provided in Appendix F. The first term on the RHS represents the corruption-independent regret upper bound, which matches the bound $\widetilde{\mathcal{O}}(\widetilde{d}\sqrt{T})$ in corruption-free neural bandit studies [86, 84]. For terms that depend on the corruption level $C$, we obtain $\widetilde{\mathcal{O}}(C\widetilde{d}\lambda^{1/2}S + C\widetilde{d}^{3/2})$ by omitting logarithmic terms. When aligning the definition of information gain [16] with the effective dimension $\widetilde{d}$, our results are consistent with the latest kernelized bandit research in terms of the horizon $T$ and effective dimension $\widetilde{d}$, given the NTK-induced RKHS and an indefinite arm space (Corollary 7 in [16]). Additionally, we can bound the NTK norm term $S$ by a constant if $h(\cdot)$ belongs to the RKHS norm induced by NTK (Subsection B.4). The regularization parameter $\lambda$ can also be tuned to account for the NTK norm $S$. Different from the vanilla Neural-UCB [86], we quantify the impact of corruption by deriving a new confidence ellipsoid around the corrupted parameters $\boldsymbol{\theta}_{t-1}$, ensuring that the corruption-related terms remain independent of the non-logarithmic $T$ term.

Inspired by [42], when $C$ is unknown to the learner, an estimated corruption level $\bar{C} > 0$ can be utilized based on prior knowledge of the adversary. In this case, we can set the scaling parameter $\alpha = 1/\bar{C}$. If the actual $C \leq \bar{C}$, then the corresponding regret upper bound in Theorem 5.6 still holds. Conversely, if $C > \bar{C}$, the regret bound will no longer hold, leading to a trivial upper bound of $R(T) \leq \mathcal{O}(T)$, similar to existing works (e.g., [42]). Here, practically, one approach is to set $\bar{C} = \sqrt{T}$. For $C \leq \sqrt{T}$, the overall regret is then bounded by $\widetilde{\mathcal{O}}(\widetilde{d}^{3/2}\sqrt{T})$, which matches [16] and improves upon $\widetilde{\mathcal{O}}(\widetilde{d}T)$ from [15]. When $C > \sqrt{T}$, our trivial regret bound of $\mathcal{O}(T)$ also aligns with the state-of-the-art kernelized method [15] with unknown $C$, which has a bound of $\widetilde{\mathcal{O}}(\widetilde{d}\sqrt{T} + C\widetilde{d}\sqrt{T}) \implies \widetilde{\mathcal{O}}(\widetilde{d}\sqrt{T} + \widetilde{d}T)$. Thus, the regret bound for NeuralUCB-WGD comparably matches the latest theoretical results from the kernelized bandit research [16, 15] under the indefinite arm space setting. While the problem definition of neural contextual bandits (1) is more general and reduce restrictions from the reward mapping function aspect, it is also significantly distinct from the problem definitions of linear or kernelized bandits, which makes lots of techniques from existing works on tackling adversarial corruptions (e.g., [42, 16]) infeasible.

## F   Proof of Regret Bound for NeuralUCB-WGD (Proof of Theorem E.1)

By definition in (1), recall that we aim to minimize the pseudo-regret for $T$ rounds, denoted by

$$\begin{aligned}
R(T) &= \sum_{t=1}^{T}\left[h(\boldsymbol{x}_t^*) - h(\boldsymbol{x}_t)\right] = \sum_{t=1}^{T} R_t \\
&= \sum_{t=1}^{T}\left[\langle g(\boldsymbol{x}_t^*; \boldsymbol{\theta}_0),\ \boldsymbol{\theta}^* - \boldsymbol{\theta}_0\rangle - \langle g(\boldsymbol{x}_t; \boldsymbol{\theta}_0),\ \boldsymbol{\theta}^* - \boldsymbol{\theta}_0\rangle\right]
\end{aligned}$$

where $\boldsymbol{x}_t$ is the chosen arm and $\boldsymbol{x}_t^* = \arg\max_{\boldsymbol{x}_{i,t} \in \mathcal{X}_t}[h(\boldsymbol{x}_{i,t})]$ being the optimal arm in round $t$. The third equality is due to Lemma C.1. Then, we denote $f(\cdot)$ as the bandit model we currently possess, which is trained with corrupted records $\mathcal{P}_{t-1}$ up to round $t$, and also suppose an imaginary corruption-free bandit model accordingly, which is trained with corruption-free records $\widehat{\mathcal{P}}_{t-1}$. The corresponding model parameters of $f(\cdot)$ will be denoted as $\boldsymbol{\theta}$, while the parameters of the imaginary corruption-free model will be denoted as $\widetilde{\boldsymbol{\theta}}$.

**Proof sketch.** To begin with, we first analyze the single-round pseudo-regret $R_t$ for $t \in [T]$, where the cumulative regret is given by $R(T) = \sum_{t \in [T]} R_t$. Here, we demonstrate that the single-round regret with corruption can be upper bounded by using the updated confidence ellipsoid (Lemma F.1) around

the corrupted network parameters $\theta_t$. Since we cannot directly apply the self-regularized martingale concentration results from existing studies [1, 86], we instead derive updated concentration results that account for adversarial corruptions, as shown in Lemma F.2. These results are then combined to establish the cumulative regret over $T$ rounds. Additionally, we provide the optimal value for the scaling parameter $\alpha$ based on the known corruption level $C$, demonstrating why setting $\alpha = 1/C$ yields the desired regret bound.

### F.1 Bounding the Single-round Regret

Following analogous approach as the proof of Lemma 5.3 in [86], we can transform the regret for a single round $t \in [T]$ to

$$
\begin{aligned}
R_t &= \langle g(\boldsymbol{x}_t^*; \boldsymbol{\theta}_0), \, \boldsymbol{\theta}^* - \boldsymbol{\theta}_0 \rangle - \langle g(\boldsymbol{x}_t; \boldsymbol{\theta}_0), \, \boldsymbol{\theta}^* - \boldsymbol{\theta}_0 \rangle \\
&\leq \langle g(\boldsymbol{x}_t^*; \boldsymbol{\theta}_{t-1}), \, \boldsymbol{\theta}^* - \boldsymbol{\theta}_0 \rangle - \langle g(\boldsymbol{x}_t; \boldsymbol{\theta}_{t-1}), \, \boldsymbol{\theta}^* - \boldsymbol{\theta}_0 \rangle + \mathcal{O}(Sm^{-1/6}\sqrt{\log(m)}t^{1/6}\lambda^{-1/6}L^{2/7}) \\
&\leq \max_{\boldsymbol{\theta} \in \mathcal{C}_{t-1}} \langle g(\boldsymbol{x}_t^*; \boldsymbol{\theta}_{t-1}), \, \boldsymbol{\theta} - \boldsymbol{\theta}_0 \rangle - \langle g(\boldsymbol{x}_t; \boldsymbol{\theta}_{t-1}), \, \boldsymbol{\theta}^* - \boldsymbol{\theta}_0 \rangle + \mathcal{O}(Sm^{-1/6}\sqrt{\log(m)}t^{1/6}\lambda^{-1/6}L^{2/7}).
\end{aligned}
$$

where the first inequality is by the gradient difference in Lemma G.4, and the bound of $\|\boldsymbol{\theta}^* - \boldsymbol{\theta}_0\|$ in Lemma C.1. Here, based on Lemma F.1, we have the unknown parameter $\boldsymbol{\theta}^* \in \mathcal{C}_{t-1}$ with the confidence ellipsoid $\mathcal{C}_{t-1} = \{\boldsymbol{\theta} : \|\boldsymbol{\theta} - \boldsymbol{\theta}_{t-1}\|_{\boldsymbol{\Gamma}_{t-1}^{-1}} \leq \gamma_{t-1}/\sqrt{m}\}$ induced by our currently possessed parameters $\boldsymbol{\theta}_{t-1}$ as well as the chosen arms $\{\boldsymbol{x}_\tau\}_{\tau \in [t-1]}$. Here, with $\widetilde{\gamma}_{t-1}$ being the corresponding corruption-free radius term, we have

$$
\begin{aligned}
\gamma_{t-1} &= \widetilde{\gamma}_{t-1} + \alpha \cdot C + (1 - \eta m \lambda)^{J/2}\sqrt{t/\lambda} + \mathcal{O}(m^{-1/6}\sqrt{\log(m)}L^{7/2}t^{5/3}\lambda^{-5/3}(1 + \sqrt{t/\lambda})) \\
&\quad + \mathcal{O}(Cm^{-2/3}\sqrt{\log(m)}t^{1/6}\lambda^{-7/6}L^{7/2}) + \mathcal{O}(Cm^{-1/6}\sqrt{\log(m)}t^{7/6}\lambda^{-13/6}L^{9/2}).
\end{aligned}
$$

Then, based on the arm pulling mechanism, we denote the estimated arm benefit score as $U(\boldsymbol{x}_{i,t}) = f(\boldsymbol{x}_{i,t}; \boldsymbol{\theta}_{t-1}) + \gamma_{t-1} \cdot \sqrt{g(\boldsymbol{x}_{i,t}; \boldsymbol{\theta}_{t-1})^\mathsf{T} \boldsymbol{\Gamma}_{t-1}^{-1} g(\boldsymbol{x}_{i,t}; \boldsymbol{\theta}_{t-1})}$, and we also define its alternative based on the confidence ellipsoid as

$$
\begin{aligned}
V(\boldsymbol{x}_{i,t}) &= \langle g(\boldsymbol{x}_{i,t}; \boldsymbol{\theta}_{t-1}), \, \boldsymbol{\theta}_{t-1} - \boldsymbol{\theta}_0 \rangle + \gamma_{t-1} \cdot \sqrt{g(\boldsymbol{x}_{i,t}; \boldsymbol{\theta}_{t-1})^\mathsf{T} \boldsymbol{\Gamma}_{t-1}^{-1} g(\boldsymbol{x}_{i,t}; \boldsymbol{\theta}_{t-1})/m} \\
&= \max_{\boldsymbol{\theta} \in \mathcal{C}_{t-1}} \langle g(\boldsymbol{x}_{i,t}; \boldsymbol{\theta}_{t-1}), \, \boldsymbol{\theta} - \boldsymbol{\theta}_0 \rangle
\end{aligned}
$$

based on the confidence interval $\mathcal{C}_{t-1}$ induced by the corrupted parameters $\boldsymbol{\theta}_{t-1}$. Regarding their distance, we can further derive $|U(\boldsymbol{x}_{i,t}) - V(\boldsymbol{x}_{i,t})| \leq \mathcal{O}(m^{-1/6}\sqrt{\log(m)}t^{2/3}\lambda^{-2/3}L^3)$ by applying Lemma G.5, as well as the fact that $f(\boldsymbol{x}; \boldsymbol{\theta}_0) = 0$ based on random initialization. It then leads to

$$
\begin{aligned}
R_t &\leq \max_{\boldsymbol{\theta} \in \mathcal{C}_{t-1}} \langle g(\boldsymbol{x}_t^*; \boldsymbol{\theta}_{t-1}), \, \boldsymbol{\theta} - \boldsymbol{\theta}_0 \rangle - \langle g(\boldsymbol{x}_t; \boldsymbol{\theta}_{t-1}), \, \boldsymbol{\theta}^* - \boldsymbol{\theta}_0 \rangle + \mathcal{O}(Sm^{-1/6}\sqrt{\log(m)}t^{1/6}\lambda^{-1/6}L^{2/7}) \\
&= V(\boldsymbol{x}_t^*) - \langle g(\boldsymbol{x}_t; \boldsymbol{\theta}_{t-1}), \, \boldsymbol{\theta}^* - \boldsymbol{\theta}_0 \rangle + \mathcal{O}(Sm^{-1/6}\sqrt{\log(m)}t^{1/6}\lambda^{-1/6}L^{2/7}) \\
&\leq U(\boldsymbol{x}_t^*) - \langle g(\boldsymbol{x}_t; \boldsymbol{\theta}_{t-1}), \, \boldsymbol{\theta}^* - \boldsymbol{\theta}_0 \rangle \\
&\quad + \mathcal{O}(Sm^{-1/6}\sqrt{\log(m)}t^{1/6}\lambda^{-1/6}L^{2/7}) + \mathcal{O}(m^{-1/6}\sqrt{\log(m)}t^{2/3}\lambda^{-2/3}L^3) \\
&\leq U(\boldsymbol{x}_t) - \langle g(\boldsymbol{x}_t; \boldsymbol{\theta}_{t-1}), \, \boldsymbol{\theta}^* - \boldsymbol{\theta}_0 \rangle \\
&\quad + \mathcal{O}(Sm^{-1/6}\sqrt{\log(m)}t^{1/6}\lambda^{-1/6}L^{2/7}) + \mathcal{O}(m^{-1/6}\sqrt{\log(m)}t^{2/3}\lambda^{-2/3}L^3) \\
&\leq V(\boldsymbol{x}_t) - \langle g(\boldsymbol{x}_t; \boldsymbol{\theta}_{t-1}), \, \boldsymbol{\theta}^* - \boldsymbol{\theta}_0 \rangle \\
&\quad + \mathcal{O}(Sm^{-1/6}\sqrt{\log(m)}t^{1/6}\lambda^{-1/6}L^{2/7}) + \mathcal{O}(m^{-1/6}\sqrt{\log(m)}t^{2/3}\lambda^{-2/3}L^3) \\
&\leq \max_{\boldsymbol{\theta} \in \mathcal{C}_{t-1}} \langle g(\boldsymbol{x}_t; \boldsymbol{\theta}_{t-1}), \, \boldsymbol{\theta} - \boldsymbol{\theta}_0 \rangle - \langle g(\boldsymbol{x}_t; \boldsymbol{\theta}_{t-1}), \, \boldsymbol{\theta}^* - \boldsymbol{\theta}_0 \rangle \\
&\quad + \mathcal{O}(Sm^{-1/6}\sqrt{\log(m)}t^{1/6}\lambda^{-1/6}L^{2/7}) + \mathcal{O}(m^{-1/6}\sqrt{\log(m)}t^{2/3}\lambda^{-2/3}L^3)
\end{aligned}
$$

where the third inequality is due to the arm pulling mechanism. The second and the fourth inequality is due to the distance between $V(\cdot)$ and $U(\cdot)$. Since we have $\boldsymbol{\theta}^* \in \mathcal{C}_{t-1}$, by Holder's inequality, it

will further lead to

$$R_t \leq \max_{\boldsymbol{\theta} \in \mathcal{C}_{t-1}} \langle g(\boldsymbol{x}_t; \boldsymbol{\theta}_{t-1}), \boldsymbol{\theta} - \boldsymbol{\theta}_0 \rangle - \langle g(\boldsymbol{x}_t; \boldsymbol{\theta}_{t-1}), \boldsymbol{\theta}^* - \boldsymbol{\theta}_0 \rangle + \mathcal{O}(Sm^{-1/6}\sqrt{\log(m)}t^{1/6}\lambda^{-1/6}L^{2/7})$$

$$+ \mathcal{O}(m^{-1/6}\sqrt{\log(m)}t^{2/3}\lambda^{-2/3}L^3)$$

$$\leq \max_{\boldsymbol{\theta} \in \mathcal{C}_{t-1}} \|g(\boldsymbol{x}_t; \boldsymbol{\theta}_{t-1})\|_{\boldsymbol{\Gamma}_{t-1}^{-1}} \cdot \|\boldsymbol{\theta} - \boldsymbol{\theta}_0\|_{\boldsymbol{\Gamma}_{t-1}} + \|g(\boldsymbol{x}_t; \boldsymbol{\theta}_{t-1})\|_{\boldsymbol{\Gamma}_{t-1}^{-1}} \cdot \|\boldsymbol{\theta}^* - \boldsymbol{\theta}_0\|_{\boldsymbol{\Gamma}_{t-1}}$$

$$+ \mathcal{O}(Sm^{-1/6}\sqrt{\log(m)}t^{1/6}\lambda^{-1/6}L^{2/7}) + \mathcal{O}(m^{-1/6}\sqrt{\log(m)}t^{2/3}\lambda^{-2/3}L^3)$$

$$\leq 2\gamma_{t-1} \cdot \|g(\boldsymbol{x}_t; \boldsymbol{\theta}_{t-1})/\sqrt{m}\|_{\boldsymbol{\Gamma}_{t-1}^{-1}} + \mathcal{O}(Sm^{-1/6}\sqrt{\log(m)}t^{1/6}\lambda^{-1/6}L^{2/7}) + \mathcal{O}(m^{-1/6}\sqrt{\log(m)}t^{2/3}\lambda^{-2/3}L^3)$$

where the last inequality is due to the definition of confidence ellipsoid $\mathcal{C}_{t-1}$. This gives the upper bound for our single-round regret.

## F.2 Bounding the cumulative regret

On the other hand, based on the conclusion from Subsec. F.1, the cumulative regret upper bound can be transformed to

$$R(T) = \sum_{t=1}^{T} \left[ h(\boldsymbol{x}_t^*) - h(\boldsymbol{x}_t) \right] = \sum_{t=1}^{T} R_t$$

$$\leq \sum_{t=1}^{T} \left[ 2 \cdot \min \left\{ \gamma_{t-1} \cdot \|g(\boldsymbol{x}_t; \boldsymbol{\theta}_{t-1})/\sqrt{m}\|_{\boldsymbol{\Gamma}_{t-1}^{-1}}, \; 1 \right\} + \mathcal{O}(Sm^{-1/6}\sqrt{\log(m)}t^{1/6}\lambda^{-1/6}L^{2/7}) \right.$$

$$\left. + \mathcal{O}(m^{-1/6}\sqrt{\log(m)}t^{2/3}\lambda^{-2/3}L^3) \right]$$

$$\leq \sum_{t=1}^{T} \left[ 2 \cdot \min \left\{ \gamma_{t-1} \cdot \|g(\boldsymbol{x}_t; \boldsymbol{\theta}_{t-1})/\sqrt{m}\|_{\boldsymbol{\Gamma}_{t-1}^{-1}}, \; 1 \right\} \right] + \mathcal{O}(1)$$

where the last inequality is because of sufficiently large network width $m$ that satisfies conditions in Theorem E.1. Then, for the first term on the RHS, we can further have

$$\sum_{t=1}^{T} 2 \cdot \min \left\{ \gamma_{t-1} \cdot \|g(\boldsymbol{x}_t; \boldsymbol{\theta}_{t-1})/\sqrt{m}\|_{\boldsymbol{\Gamma}_{t-1}^{-1}}, \; 1 \right\}$$

$$\leq 2\gamma_T \cdot \sqrt{\widetilde{d}T \log(1 + TK/\lambda)} + (1 + \frac{\gamma_T}{\alpha}) \cdot \widetilde{d}\log(1 + TK/\lambda)$$

$$= \widetilde{d}\log(1 + TK/\lambda) + 2\gamma_T \cdot \left( \sqrt{\widetilde{d}T \log(1 + TK/\lambda)} + \frac{1}{\alpha} \cdot \widetilde{d}\log(1 + TK/\lambda) \right)$$

$$\leq \widetilde{d}\log(1 + TK/\lambda) + \left( \sqrt{\widetilde{d}T \log(1 + TK/\lambda)} + \frac{1}{\alpha} \cdot \widetilde{d}\log(1 + TK/\lambda) \right)$$

$$\cdot \left( 2\widetilde{\gamma}_{t-1} + 2\alpha \cdot C + 2(1 - \eta m\lambda)^{J/2}\sqrt{t/\lambda} + \mathcal{O}(m^{-1/6}\sqrt{\log(m)}L^{7/2}t^{5/3}\lambda^{-5/3}(1 + \sqrt{t/\lambda})) \right.$$

$$\left. + \mathcal{O}(Cm^{-2/3}\sqrt{\log(m)}t^{1/6}\lambda^{-7/6}L^{7/2}) + \mathcal{O}(Cm^{-1/6}\sqrt{\log(m)}t^{7/6}\lambda^{-13/6}L^{9/2}) \right)$$

$$\leq \widetilde{d}\log(1 + TK/\lambda) + \left( \sqrt{\widetilde{d}T \log(1 + TK/\lambda)} + \frac{1}{\alpha} \cdot \widetilde{d}\log(1 + TK/\lambda) \right)$$

$$\cdot \left( \left( \nu\sqrt{\widetilde{d}\log(1 + TK/\lambda) - 2\log(\delta)} + \lambda^{1/2}S \right) + 2\alpha \cdot C \right.$$

$$+ 2(1 - \eta m\lambda)^{J/2}\sqrt{t/\lambda} + \mathcal{O}(m^{-1/6}\sqrt{\log(m)}L^{7/2}t^{5/3}\lambda^{-5/3}(1 + \sqrt{t/\lambda}))$$

$$\left. + \mathcal{O}(Cm^{-2/3}\sqrt{\log(m)}t^{1/6}\lambda^{-7/6}L^{7/2}) + \mathcal{O}(Cm^{-1/6}\sqrt{\log(m)}t^{7/6}\lambda^{-13/6}L^{9/2}) \right)$$

$$\leq \left( \sqrt{\widetilde{d}T \log(1 + TK/\lambda)} + \frac{\widetilde{d}}{\alpha}\log(1 + TK/\lambda) \right) \cdot \mathcal{O}\left( \nu\sqrt{\widetilde{d}\log(1 + TK/\lambda) - 2\log(\delta)} + \lambda^{1/2}S + 2\alpha C \right) + \mathcal{O}(1)$$

where the first inequality is due to Lemma F.2, and the second inequality is due to Lemma F.1. The third inequality can be derived following an analogous approach as Lemma 5.2 in [86], and the last inequality is due to the sufficiently large network width $m$ as mentioned in Theorem E.1, as well as the sufficient number of GD iterations $J = \widetilde{\mathcal{O}}(TL/\lambda)$.

**Discussion on the value of $\alpha$.** Here, notice that we have the tunable parameter $\alpha > 0$ nested in the regret bound. Taking some more steps, we can have

$$\sum_{t=1}^{T} 2 \min \left\{ \gamma_{t-1} \cdot \|g(\boldsymbol{x}_t; \boldsymbol{\theta}_{t-1})/\sqrt{m}\|_{\boldsymbol{\Gamma}_{t-1}^{-1}}, \ 1 \right\}$$

$$\leq \left( \sqrt{\widetilde{d}T \log(1 + TK/\lambda)} + \frac{1}{\alpha} \cdot \widetilde{d} \log(1 + TK/\lambda) \right)$$

$$\cdot \mathcal{O}\left( \nu \sqrt{\widetilde{d} \log(1 + TK/\lambda) - 2\log(\delta)} + \lambda^{1/2} S + 2\alpha \cdot C \right)$$

$$\leq \widetilde{O}\left( \sqrt{\widetilde{d}T} + \frac{1}{\alpha} \cdot \widetilde{d} \right) \cdot \left( \nu \sqrt{\widetilde{d} - 2\log(\delta)} + \lambda^{1/2} S + 2\alpha \cdot C \right)$$

$$\leq \widetilde{\mathcal{O}}\left( \sqrt{\widetilde{d}T} + \frac{1}{\alpha} \cdot \widetilde{d} \right) \cdot \left( \nu \sqrt{\widetilde{d} - 2\log(\delta)} + \lambda^{1/2} S \right) + \widetilde{\mathcal{O}}\left( \alpha C \sqrt{\widetilde{d}T} + C\widetilde{d} \right)$$

Since we have no prior knowledge of effective dimension $\widetilde{d}$, we can set $\alpha = \frac{1}{C}$. It will then lead to

$$\sum_{t=1}^{T} 2 \min \left\{ \gamma_{t-1} \cdot \|g(\boldsymbol{x}_t; \boldsymbol{\theta}_{t-1})/\sqrt{m}\|_{\boldsymbol{\Gamma}_{t-1}^{-1}}, \ 1 \right\}$$

$$\leq \widetilde{\mathcal{O}}\left( \sqrt{\widetilde{d}T} + \frac{1}{\alpha} \cdot \widetilde{d} \right) \cdot \left( \nu \sqrt{\widetilde{d} - 2\log(\delta)} + \lambda^{1/2} S \right) + \widetilde{\mathcal{O}}\left( \alpha C \sqrt{\widetilde{d}T} + C\widetilde{d} \right)$$

$$\leq \widetilde{\mathcal{O}}\left( \sqrt{\widetilde{d}T} \right) \cdot \left( \nu \sqrt{\widetilde{d} - 2\log(\delta)} + \lambda^{1/2} S \right) + \widetilde{\mathcal{O}}\left( C\widetilde{d} + C\widetilde{d} \cdot \left( \nu \sqrt{\widetilde{d} - 2\log(\delta)} + \lambda^{1/2} S \right) \right).$$

Finally, summing up all the results above will give the conclusion.

### F.3 Confidence ellipsoid for corrupted parameters

**Lemma F.1.** *With the notation and conditions in Theorem E.1, train the network parameters $\boldsymbol{\theta}_{t-1}$ based on received records $\mathcal{P}_{t-1}$. The confidence ellipsoid around the corrupted network parameters $\boldsymbol{\theta}_{t-1}$ can be defined as*

$$\mathcal{C}_{t-1} = \left\{ \boldsymbol{\theta} : \|\boldsymbol{\theta} - \boldsymbol{\theta}_{t-1}\|_{\boldsymbol{\Gamma}_{t-1}} \leq \gamma_{t-1}/\sqrt{m} \right\}$$

*where we have the unknown parameter $\boldsymbol{\theta}^* \in \mathcal{C}_{t-1}$, and we denote*

$$\gamma_{t-1} = \widetilde{\gamma}_{t-1} + \alpha \cdot C + (1 - \eta m\lambda)^{J/2} \sqrt{t/\lambda} + \mathcal{O}(m^{-1/6} \sqrt{\log(m)} L^{7/2} t^{5/3} \lambda^{-5/3} (1 + \sqrt{t/\lambda}))$$
$$+ \mathcal{O}(Cm^{-2/3} \sqrt{\log(m)} t^{1/6} \lambda^{-7/6} L^{7/2}) + \mathcal{O}(Cm^{-1/6} \sqrt{\log(m)} t^{7/6} \lambda^{-13/6} L^{9/2})$$

**Proof.** The proof follows an analogous approach as in Lemma C.12. For the imaginary corruption-free parameters $\widetilde{\boldsymbol{\theta}}_{t-1}$, which are trained on corruption-free records $\widetilde{\mathcal{P}}_{t-1}$, we can construct the confidence interval $\widetilde{\mathcal{C}}_{t-1} := \{\boldsymbol{\theta} : \|\boldsymbol{\theta} - \widetilde{\boldsymbol{\theta}}_{t-1}\|_{\widetilde{\boldsymbol{\Gamma}}_{t-1}} \leq \widetilde{\gamma}_{t-1}/\sqrt{m}, \widetilde{\gamma}_{t-1} > 0\}$, such that the unknown $\boldsymbol{\theta}^*$ in Lemma C.1 satisfies $\boldsymbol{\theta}^* \in \widetilde{\mathcal{C}}_{t-1}$. However, since we do not possess $\widetilde{\boldsymbol{\theta}}_{t-1}$, we need to alternatively derive the confidence ellipsoid $\mathcal{C}_{t-1}$ around the possessed corrupted parameters $\boldsymbol{\theta}_{t-1}$, such that $\boldsymbol{\theta}^* \in \mathcal{C}_{t-1}$.

With the ellipsoid center $\boldsymbol{\theta}_{t-1}$ as well as the weighted gradient covariance matrix $\boldsymbol{\Gamma}_{t-1} = \lambda \mathbf{I} + \sum_{\tau \in [t-1]} w_\tau \cdot g(\boldsymbol{x}_\tau; \boldsymbol{\theta}_{\tau-1}) \cdot g(\boldsymbol{x}_\tau; \boldsymbol{\theta}_{\tau-1})^\mathsf{T}/m$, we need to derive the corresponding radius. Recall that $w_\tau$ is the sample weight associated with chosen arm $\boldsymbol{x}_\tau$. Here, we first recall some preliminaries that $\|\boldsymbol{\Gamma}_{t-1} - \widetilde{\boldsymbol{\Gamma}}_{t-1}\|_F \leq \mathcal{O}(m^{-1/6} \sqrt{\log(m)} L^4 t^{7/6} \lambda^{-1/6})$ due to Lemma G.6, as well as $\|\boldsymbol{\theta}_{t-1} - \widetilde{\boldsymbol{\theta}}_{t-1}\|_2 \leq \mathcal{O}(\sqrt{t/(m\lambda)})$ due to Lemma G.3.

Next, as we already have $\|\boldsymbol{\theta} - \widetilde{\boldsymbol{\theta}}_{t-1}\|_{\widetilde{\boldsymbol{\Gamma}}_{t-1}} \le \widetilde{\gamma}_{t-1}/\sqrt{m}$, we proceed to transform the objective to

$$
\begin{aligned}
\|\boldsymbol{\theta}^* - \boldsymbol{\theta}_{t-1}\|_{\boldsymbol{\Gamma}_{t-1}} &\le \|\boldsymbol{\theta}^* - \widetilde{\boldsymbol{\theta}}_{t-1}\|_{\boldsymbol{\Gamma}_{t-1}} + \|\widetilde{\boldsymbol{\theta}}_{t-1} - \boldsymbol{\theta}_{t-1}\|_{\boldsymbol{\Gamma}_{t-1}} \\
&\le \|\boldsymbol{\theta}^* - \widetilde{\boldsymbol{\theta}}_{t-1}\|_{\boldsymbol{\Gamma}_{t-1}} + \|\widetilde{\boldsymbol{\theta}}_{t-1} - \boldsymbol{\theta}_{t-1}\|_{\boldsymbol{\Gamma}_{t-1}} \\
&\le \|\boldsymbol{\theta}^* - \widetilde{\boldsymbol{\theta}}_{t-1}\|_{\boldsymbol{\Gamma}_{t-1} - \widetilde{\boldsymbol{\Gamma}}_{t-1} + \widetilde{\boldsymbol{\Gamma}}_{t-1}} + \|\widetilde{\boldsymbol{\theta}}_{t-1} - \boldsymbol{\theta}_{t-1}\|_{\boldsymbol{\Gamma}_{t-1}} \\
&\le \|\boldsymbol{\theta}^* - \widetilde{\boldsymbol{\theta}}_{t-1}\|_{\widetilde{\boldsymbol{\Gamma}}_{t-1}} + \|\boldsymbol{\theta}^* - \widetilde{\boldsymbol{\theta}}_{t-1}\|_{\boldsymbol{\Gamma}_{t-1} - \widetilde{\boldsymbol{\Gamma}}_{t-1}} + \|\widetilde{\boldsymbol{\theta}}_{t-1} - \boldsymbol{\theta}_{t-1}\|_{\boldsymbol{\Gamma}_{t-1}} \\
&\le \widetilde{\gamma}_{t-1}/\sqrt{m} + \|\widetilde{\boldsymbol{\theta}}_{t-1} - \boldsymbol{\theta}_{t-1}\|_{\boldsymbol{\Gamma}_{t-1}} + \mathcal{O}(\sqrt{t/(m\lambda)}) \cdot \mathcal{O}(m^{-1/6}\sqrt{\log(m)}L^4 t^{7/6}\lambda^{-1/6}) \\
&\le \widetilde{\gamma}_{t-1}/\sqrt{m} + \|\widetilde{\boldsymbol{\theta}}_{t-1} - \boldsymbol{\theta}_{t-1}\|_{\boldsymbol{\Gamma}_{t-1}} + \mathcal{O}(m^{-2/3}\sqrt{\log(m)}L^4 t^{13/6}\lambda^{-2/3}).
\end{aligned}
$$

For the second term on the RHS, we first define the gradient-based regression parameters as

$$
\begin{aligned}
\boldsymbol{\Gamma}_{t-1}^{(0)} &= \lambda \mathbf{I} + \sum_{\tau \in [t-1]} w_\tau \cdot g(\boldsymbol{x}_\tau; \boldsymbol{\theta}_0) \cdot g(\boldsymbol{x}_\tau; \boldsymbol{\theta}_0)^\mathsf{T}/m, \\
\boldsymbol{b}_{t-1}^{(0)} &= \sum_{\tau \in [t-1]} w_\tau \cdot g(\boldsymbol{x}_\tau; \boldsymbol{\theta}_0) \cdot r_\tau/\sqrt{m}, \\
\widetilde{\boldsymbol{b}}_{t-1}^{(0)} &= \sum_{\tau \in [t-1]} w_\tau \cdot g(\boldsymbol{x}_\tau; \boldsymbol{\theta}_0) \cdot \widetilde{r}_\tau/\sqrt{m},
\end{aligned}
$$

Then, with the triangular inequality, we can proceed to have

$$
\begin{aligned}
\|\widetilde{\boldsymbol{\theta}}_{t-1} - \boldsymbol{\theta}_{t-1}\|_{\boldsymbol{\Gamma}_{t-1}} &\le \|\widetilde{\boldsymbol{\theta}}_{t-1} - \boldsymbol{\theta}_0 - (\boldsymbol{\Gamma}_{t-1}^{(0)})^{-1}\boldsymbol{b}_{t-1}^{(0)}/\sqrt{m} + (\boldsymbol{\Gamma}_{t-1}^{(0)})^{-1}\boldsymbol{b}_{t-1}^{(0)}/\sqrt{m} + \boldsymbol{\theta}_0 - \boldsymbol{\theta}_{t-1}\|_{\boldsymbol{\Gamma}_{t-1}} \\
&\le \|\boldsymbol{\theta}_{t-1} - \boldsymbol{\theta}_0 - (\boldsymbol{\Gamma}_{t-1}^{(0)})^{-1}\boldsymbol{b}_{t-1}^{(0)}/\sqrt{m}\|_{\boldsymbol{\Gamma}_{t-1}} + \|\widetilde{\boldsymbol{\theta}}_{t-1} - \boldsymbol{\theta}_0 - (\boldsymbol{\Gamma}_{t-1}^{(0)})^{-1}\boldsymbol{b}_{t-1}^{(0)}/\sqrt{m}\|_{\boldsymbol{\Gamma}_{t-1}} \\
&\le \|\boldsymbol{\theta}_{t-1} - \boldsymbol{\theta}_0 - (\boldsymbol{\Gamma}_{t-1}^{(0)})^{-1}\boldsymbol{b}_{t-1}^{(0)}/\sqrt{m}\|_{\boldsymbol{\Gamma}_{t-1}} + \|\widetilde{\boldsymbol{\theta}}_{t-1} - \boldsymbol{\theta}_0 - (\boldsymbol{\Gamma}_{t-1}^{(0)})^{-1}\widetilde{\boldsymbol{b}}_{t-1}^{(0)}/\sqrt{m}\|_{\boldsymbol{\Gamma}_{t-1}} \\
&\quad + m^{-1}\|(\boldsymbol{\Gamma}_{t-1}^{(0)})^{-1} \cdot \big(\sum_{\tau \in [t-1]} w_\tau \cdot g(\boldsymbol{x}_\tau; \boldsymbol{\theta}_0) \cdot c_\tau\big)\|_{\boldsymbol{\Gamma}_{t-1}}.
\end{aligned}
$$

(F.1)

**Bounding the first two term on the RHS of Inequality F.1.** Here, for the first term on the RHS, we can individually apply Lemma D.1, by considering the auxiliary sequence in $j$-th iteration ($j \in [J]$) with $\boldsymbol{\Theta}^{(0)} = \boldsymbol{\theta}_0$, as

$$
\boldsymbol{\Theta}^{(j+1)} = \boldsymbol{\Theta}^{(j)} - \eta \cdot \left[\mathbf{J}^{(0)} \cdot \mathbf{W} \cdot \big([\mathbf{J}^{(0)}]^\mathsf{T}(\boldsymbol{\Theta}^{(j)} - \boldsymbol{\theta}_0) - \boldsymbol{y}\big) + m\lambda(\boldsymbol{\Theta}^{(j)} - \boldsymbol{\theta}_0)\right]
$$

where the Jacobian matrix $\mathbf{J}^{(0)} := \big(g(\boldsymbol{x}_1; \boldsymbol{\theta}_0), g(\boldsymbol{x}_2; \boldsymbol{\theta}_0), \ldots, g(\boldsymbol{x}_{t-1}; \boldsymbol{\theta}_0)\big) \in \mathbb{R}^{p \times (t-1)}$, and vector $\boldsymbol{y} \in \mathbb{R}^{t-1}$ contains the received arm rewards $r_\tau, \tau \in [t-1]$, while matrix $\mathbf{W} \in \mathbb{R}^{(t-1) \times (t-1)}$ is the diagonal matrix that contains sample weights $w_\tau, \tau \in [t-1]$. We have its norm $\|\mathbf{W}\|_2 \le 1$ by definition. Here, the above sequence is expected to solve the following problem with gradient descent

$$
\min_{\boldsymbol{\Theta}} \mathcal{L}(\boldsymbol{\Theta}) = \sum_{\tau \in [t-1]} \frac{w_\tau}{2} \cdot \left\|[\mathbf{J}^{(0)}]^\mathsf{T}_\tau(\boldsymbol{\Theta} - \boldsymbol{\theta}_0) - r_\tau\right\|_2^2 + \frac{1}{2} \cdot m\lambda \cdot \left\|\boldsymbol{\Theta} - \boldsymbol{\theta}_0\right\|_2^2.
$$

As a result, following an analogous approach as in Lemma C.4 in [86], we can have $\|\boldsymbol{\Theta}^{(j)} - \boldsymbol{\theta}_0 - (\boldsymbol{\Gamma}_{t-1}^{(0)})^{-1}\boldsymbol{b}_{t-1}^{(0)}/\sqrt{m}\|_{\boldsymbol{\Gamma}_{t-1}} \le (1 - \eta m\lambda)^{j/2}\sqrt{t/(m\lambda)}$. Furthermore, by applying the conclusion of Lemma D.1, we can have

$$
\|\boldsymbol{\theta}_{t-1} - \boldsymbol{\theta}_0 - (\boldsymbol{\Gamma}_{t-1}^{(0)})^{-1}\boldsymbol{b}_{t-1}^{(0)}/\sqrt{m}\|_2 \le (1 - \eta m\lambda)^{J/2}\sqrt{t/(m\lambda)} + \mathcal{O}(m^{-2/3}\sqrt{\log(m)}L^{7/2}t^{5/3}\lambda^{-5/3}(1 + \sqrt{t/\lambda})).
$$

Similarly, for the second term in Inequality F.1, we also can apply a comparable approach by solving the problem:

$$
\min_{\boldsymbol{\Theta}} \mathcal{L}(\boldsymbol{\Theta}) = \sum_{\tau \in [t-1]} \frac{w_\tau}{2}\left\|[\mathbf{J}^{(0)}]^\mathsf{T}_\tau(\boldsymbol{\Theta} - \boldsymbol{\theta}_0) - \widetilde{r}_\tau\right\|_2^2 + \frac{1}{2}m\lambda \cdot \left\|\boldsymbol{\Theta} - \boldsymbol{\theta}_0\right\|_2^2,
$$

and constructing the corresponding auxiliary sequence. Following an analogous approach, it will lead to a similar bound for the second term, such that $\|\widetilde{\boldsymbol{\theta}}_{t-1} - \boldsymbol{\theta}_0 - (\boldsymbol{\Gamma}_{t-1}^{(0)})^{-1}\widetilde{\boldsymbol{b}}_{t-1}^{(0)}/\sqrt{m}\|_2 \leq (1 - \eta m\lambda)^{J/2}\sqrt{t/(m\lambda)} + \mathcal{O}(m^{-2/3}\sqrt{\log(m)}L^{7/2}t^{5/3}\lambda^{-5/3}(1 + \sqrt{t/\lambda}))$.

**Bounding the third term on the RHS of Inequality F.1.** Afterwards, for the third term on the RHS, we first have

$$m^{-1} \cdot \|(\boldsymbol{\Gamma}_{t-1}^{(0)})^{-1} \cdot (\sum_{\tau \in [t-1]} w_\tau \cdot g(\boldsymbol{x}_\tau; \boldsymbol{\theta}_0) \cdot c_\tau)\|_{\boldsymbol{\Gamma}_{t-1}}$$

$$\leq m^{-1}\|(\boldsymbol{\Gamma}_{t-1})^{-1}(\sum_{\tau \in [t-1]} w_\tau \cdot g(\boldsymbol{x}_\tau; \boldsymbol{\theta}_0) \cdot c_\tau)\|_{\boldsymbol{\Gamma}_{t-1}}$$

$$+ m^{-1}\|((\boldsymbol{\Gamma}_{t-1})^{-1} - (\boldsymbol{\Gamma}_{t-1}^{(0)})^{-1}) \cdot (\sum_{\tau \in [t-1]} w_\tau \cdot g(\boldsymbol{x}_\tau; \boldsymbol{\theta}_0) \cdot c_\tau)\|_{\boldsymbol{\Gamma}_{t-1}}$$

$$\leq m^{-1}\|(\boldsymbol{\Gamma}_{t-1})^{-1}(\sum_{\tau \in [t-1]} w_\tau \cdot g(\boldsymbol{x}_\tau; \boldsymbol{\theta}_0) \cdot c_\tau)\|_{\boldsymbol{\Gamma}_{t-1}}$$

$$+ m^{-1}\|(\boldsymbol{\Gamma}_{t-1})^{-1}(\boldsymbol{\Gamma}_{t-1} - \boldsymbol{\Gamma}_{t-1}^{(0)})(\boldsymbol{\Gamma}_{t-1}^{(0)})^{-1} \cdot (\sum_{\tau \in [t-1]} w_\tau \cdot g(\boldsymbol{x}_\tau; \boldsymbol{\theta}_0) \cdot c_\tau)\|_{\boldsymbol{\Gamma}_{t-1}}$$

$$\leq m^{-1}\|(\boldsymbol{\Gamma}_{t-1})^{-1}(\sum_{\tau \in [t-1]} w_\tau \cdot g(\boldsymbol{x}_\tau; \boldsymbol{\theta}_0) \cdot c_\tau)\|_{\boldsymbol{\Gamma}_{t-1}} + \mathcal{O}(Cm^{-2/3}\sqrt{\log(m)}t^{7/6}\lambda^{-13/6}L^{9/2})$$

$$\leq m^{-1}\|(\boldsymbol{\Gamma}_{t-1})^{-1}(\sum_{\tau \in [t-1]} w_\tau \cdot g(\boldsymbol{x}_\tau; \boldsymbol{\theta}_{\tau-1}) \cdot c_\tau)\|_{\boldsymbol{\Gamma}_{t-1}} + \mathcal{O}(Cm^{-2/3}\sqrt{\log(m)}t^{7/6}\lambda^{-13/6}L^{9/2})$$

$$+ m^{-1}\|(\boldsymbol{\Gamma}_{t-1})^{-1}(\sum_{\tau \in [t-1]} w_\tau \cdot (g(\boldsymbol{x}_\tau; \boldsymbol{\theta}_{\tau-1}) - g(\boldsymbol{x}_\tau; \boldsymbol{\theta}_0)) \cdot c_\tau)\|_{\boldsymbol{\Gamma}_{t-1}}$$

$$\leq m^{-1}\|(\boldsymbol{\Gamma}_{t-1})^{-1}(\sum_{\tau \in [t-1]} w_\tau \cdot g(\boldsymbol{x}_\tau; \boldsymbol{\theta}_{\tau-1}) \cdot c_\tau)\|_{\boldsymbol{\Gamma}_{t-1}}$$

$$+ \mathcal{O}(Cm^{-7/6}\sqrt{\log(m)}t^{1/6}\lambda^{-7/6}L^{7/2}) + \mathcal{O}(Cm^{-2/3}\sqrt{\log(m)}t^{7/6}\lambda^{-13/6}L^{9/2})$$

where the third inequality is due to Lemma G.6, and the last inequality is due to Lemma G.4. Then, recall that the weight $w_\tau = \{1, \alpha/\|g(\boldsymbol{x}_\tau; \boldsymbol{\theta}_{\tau-1})/\sqrt{m}\|_{\boldsymbol{\Gamma}_{\tau-1}^{-1}}\}$ for each chosen arm $\boldsymbol{x}_\tau$. In this case, we can further have

$$m^{-1/2}\|(\boldsymbol{\Gamma}_{t-1})^{-1}(\sum_{\tau \in [t-1]} w_\tau \cdot g(\boldsymbol{x}_\tau; \boldsymbol{\theta}_{\tau-1}) \cdot c_\tau)/\sqrt{m}\|_{\boldsymbol{\Gamma}_{t-1}}$$

$$= m^{-1/2}\|\sum_{\tau \in [t-1]} w_\tau \cdot g(\boldsymbol{x}_\tau; \boldsymbol{\theta}_{\tau-1}) \cdot c_\tau/\sqrt{m}\|_{(\boldsymbol{\Gamma}_{t-1})^{-1}}$$

$$\leq m^{-1/2}\sum_{\tau \in [t-1]} w_\tau |c_\tau| \cdot \|g(\boldsymbol{x}_\tau; \boldsymbol{\theta}_{\tau-1})/\sqrt{m}\|_{(\boldsymbol{\Gamma}_{t-1})^{-1}}$$

$$\leq m^{-1/2}\sum_{\tau \in [t-1]} w_\tau |c_\tau| \cdot \|g(\boldsymbol{x}_\tau; \boldsymbol{\theta}_{\tau-1})/\sqrt{m}\|_{(\boldsymbol{\Gamma}_{\tau-1})^{-1}}$$

$$\leq \alpha \cdot C/\sqrt{m}.$$

where the first inequality is by applying the Cauchy-Schwartz inequality, while the second inequality is due to $\boldsymbol{\Gamma}_{\tau-1} \preceq \boldsymbol{\Gamma}_{t-1}$ and Lemma G.8. The last inequality is by the definition of corruption level $C$. Finally, summing up all the results will give the desired lemma.

$\square$

### F.4 Self-regularized Martingale Sequence with Weighted Matrix

Recall that we have the weighted gradient covariance matrix $\boldsymbol{\Gamma}_{t-1} = \lambda\mathbf{I} + \sum_{\tau \in [t-1]} w_\tau \cdot g(\boldsymbol{x}_\tau; \boldsymbol{\theta}_{\tau-1}) \cdot g(\boldsymbol{x}_\tau; \boldsymbol{\theta}_{\tau-1})^\intercal/m$, where $g(\boldsymbol{x}_\tau; \boldsymbol{\theta}_{\tau-1})$ is the vectorized gradient vector, and $w_\tau \leq 1$ refers to the sampled associated with chosen arm $\boldsymbol{x}_\tau$.

By existing works [1, 86], we can have the self-normalized martingale such that

$$\sum_{\tau \in [t]} \min\left\{ \left\| g(\boldsymbol{x}_\tau; \boldsymbol{\theta}_{\tau-1})/\sqrt{m} \right\|^2_{\boldsymbol{\Gamma}^{-1}_{\tau-1}}, 1 \right\} \le 2 \log \frac{\det(\boldsymbol{\Gamma}_t)}{\det(\lambda \mathbf{I})}.$$

if weights in $\boldsymbol{\Gamma}_t$ are all set to 1. However, since in our settings the gradient covariance matrix $\boldsymbol{\Gamma}_t$ involves sample weights, we will need to further discuss the upper bound for this sequence summation.

**Lemma F.2.** *With the definition of $\gamma_{t-1}$ from Lemma F.1 as well as the notation and conditions in Theorem E.1, we have the following inequality*

$$\sum_{t \in [T]} \min\left\{ \gamma_{t-1} \cdot \left\| g(\boldsymbol{x}_t; \boldsymbol{\theta}_{t-1})/\sqrt{m} \right\|_{\boldsymbol{\Gamma}^{-1}_{t-1}}, 1 \right\} \le \gamma_T \sqrt{\widetilde{d} T \log(1 + TK/\lambda)} + (1 + \frac{\gamma_T}{\alpha}) \cdot \widetilde{d} \log(1 + TK/\lambda),$$

*where $\boldsymbol{\Gamma}_{t-1} = \lambda \mathbf{I} + \sum_{\tau \in [t-1]} w_\tau \cdot g(\boldsymbol{x}_\tau; \boldsymbol{\theta}_{\tau-1}) \cdot g(\boldsymbol{x}_\tau; \boldsymbol{\theta}_{\tau-1})^\mathsf{T}/m.$*

**Proof.** The proof of this lemma follows an analogous approach as Theorem 4.2 in [42]. Recall that sample weights $w_\tau \le 1, \tau \in [T]$. In this case, we separately consider two scenarios when (i) $w_\tau = 1$, then $\tau \in \mathcal{T}^{(T)}_{w=1}$; and (ii) the scenario when $w_\tau < 1$, then $\tau \in \mathcal{T}^{(T)}_{w<1}$. In this case, the original objective will become the following inequality

$$\sum_{t \in [T]} \min\left\{ \left\| g(\boldsymbol{x}_t; \boldsymbol{\theta}_{t-1})/\sqrt{m} \right\|_{\boldsymbol{\Gamma}^{-1}_{t-1}}, 1 \right\}$$

$$= \sum_{t \in \mathcal{T}^{(T)}_{w=1}} \min\left\{ \left\| g(\boldsymbol{x}_t; \boldsymbol{\theta}_{t-1})/\sqrt{m} \right\|_{\boldsymbol{\Gamma}^{-1}_{t-1}}, 1 \right\} + \sum_{t \in \mathcal{T}^{(T)}_{w<1}} \min\left\{ \left\| g(\boldsymbol{x}_t; \boldsymbol{\theta}_{t-1})/\sqrt{m} \right\|_{\boldsymbol{\Gamma}^{-1}_{t-1}}, 1 \right\}$$

First, for the scenario where $w_\tau = 1, \tau \in \mathcal{T}^{(T)}_{w=1}$, we have

$$\sum_{t \in \mathcal{T}^{(T)}_{w=1}} \min\left\{ \left\| g(\boldsymbol{x}_t; \boldsymbol{\theta}_{t-1})/\sqrt{m} \right\|_{\boldsymbol{\Gamma}^{-1}_{t-1}}, 1 \right\} \le \sqrt{\sum_{t \in \mathcal{T}^{(T)}_{w=1}} \min\left\{ \left\| g(\boldsymbol{x}_t; \boldsymbol{\theta}_{t-1})/\sqrt{m} \right\|^2_{\boldsymbol{\Gamma}^{-1}_{t-1}}, 1 \right\}}$$

$$\le \sqrt{\sum_{t \in \mathcal{T}^{(T)}_{w=1}} \min\left\{ \left\| g(\boldsymbol{x}_t; \boldsymbol{\theta}_{t-1})/\sqrt{m} \right\|^2_{(\boldsymbol{\Gamma}^{w=1}_{t-1})^{-1}}, 1 \right\}}.$$

Here, we define an extra auxiliary matrix $\boldsymbol{\Gamma}^{w=1}_{t-1} = \lambda \mathbf{I} + \sum_{\tau \in \mathcal{T}^{(t-1)}_{w=1}} w_\tau \cdot g(\boldsymbol{x}_\tau; \boldsymbol{\theta}_{\tau-1}) \cdot g(\boldsymbol{x}_\tau; \boldsymbol{\theta}_{\tau-1})^\mathsf{T}/m$. Compared with the original gradient covariance matrix $\boldsymbol{\Gamma}_{t-1}$, since we have $\boldsymbol{\Gamma}^{w=1}_{t-1} \preceq \boldsymbol{\Gamma}_{t-1}$ and they are both Hermitian matrices, we can derive the last inequality based on Lemma G.8. Then, by applying Lemma G.7 and Lemma C.2, it will lead to

$$\sum_{t \in \mathcal{T}^{(T)}_{w=1}} \min\left\{ \left\| g(\boldsymbol{x}_t; \boldsymbol{\theta}_{t-1})/\sqrt{m} \right\|_{\boldsymbol{\Gamma}^{-1}_{t-1}}, 1 \right\} \le \sqrt{|\mathcal{T}^{(T)}_{w=1}| \cdot \sum_{t \in \mathcal{T}^{(T)}_{w=1}} \min\left\{ \left\| g(\boldsymbol{x}_t; \boldsymbol{\theta}_{t-1})/\sqrt{m} \right\|^2_{(\boldsymbol{\Gamma}^{w=1}_{t-1})^{-1}}, 1 \right\}}$$

$$\le \sqrt{|\mathcal{T}^{(T)}_{w=1}| \cdot \widetilde{d} \log(1 + TK/\lambda)}$$

$$\le \sqrt{\widetilde{d} T \log(1 + TK/\lambda)}.$$

Then, since we have $\gamma_{t-1} \le \gamma_T$, plugging in the $\gamma_T$ will complete the proof.

Afterwards, for the second scenario when $w_\tau < 1, \tau \in \mathcal{T}^{(T)}_{w<1}$, we will have

$$\sum_{t \in \mathcal{T}^{(T)}_{w<1}} \min\left\{ \gamma_{t-1} \cdot \left\| g(\boldsymbol{x}_t; \boldsymbol{\theta}_{t-1})/\sqrt{m} \right\|_{\boldsymbol{\Gamma}^{-1}_{t-1}}, 1 \right\} = \sum_{t \in \mathcal{T}^{(T)}_{w<1}} \left\{ \gamma_{t-1} \cdot \frac{w_t}{\alpha} \cdot \left\| g(\boldsymbol{x}_t; \boldsymbol{\theta}_{t-1})/\sqrt{m} \right\|^2_{\boldsymbol{\Gamma}^{-1}_{t-1}}, 1 \right\}$$

$$\le (1 + \frac{\gamma_T}{\alpha}) \cdot \sum_{t \in \mathcal{T}^{(T)}_{w<1}} \left\{ w_t \cdot \left\| g(\boldsymbol{x}_t; \boldsymbol{\theta}_{t-1})/\sqrt{m} \right\|^2_{\boldsymbol{\Gamma}^{-1}_{t-1}}, 1 \right\}$$

$$\le (1 + \frac{\gamma_T}{\alpha}) \cdot \sum_{t \in \mathcal{T}^{(T)}_{w<1}} \left\{ w_t \cdot \left\| g(\boldsymbol{x}_t; \boldsymbol{\theta}_{t-1})/\sqrt{m} \right\|^2_{(\boldsymbol{\Gamma}^{w<1}_{t-1})^{-1}}, 1 \right\}$$

where the inequality is because we have $\gamma_{t-1} \leq \gamma_T, \forall t \in [T]$.

Following the previous approach, we define the auxiliary matrix $\Gamma_{t-1}^{w<1} = \lambda \mathbf{I} + \sum_{\tau \in \mathcal{T}_{w=1}^{(t-1)}} w_\tau \cdot g(\boldsymbol{x}_\tau; \boldsymbol{\theta}_{\tau-1}) \cdot g(\boldsymbol{x}_\tau; \boldsymbol{\theta}_{\tau-1})^\intercal / m$. Here, since we also have $\Gamma_{t-1}^{w<1} \preceq \Gamma_{t-1}$ and they are both Hermitian matrices, we can derive the last inequality based on Lemma G.8. In addition, we consider an alternative form of the original gradient vector as $g'(\boldsymbol{x}_\tau; \boldsymbol{\theta}_{\tau-1}) = \sqrt{w_\tau} \cdot g(\boldsymbol{x}_\tau; \boldsymbol{\theta}_{\tau-1}), \tau \in \mathcal{T}_{w=1}^{(t-1)}$. In this case, the auxiliary gradient covariance matrix can be alternatively represented as $\Gamma_{t-1}^{w<1} = \lambda \mathbf{I} + \sum_{\tau \in \mathcal{T}_{w=1}^{(t-1)}} g'(\boldsymbol{x}_\tau; \boldsymbol{\theta}_{\tau-1}) \cdot g'(\boldsymbol{x}_\tau; \boldsymbol{\theta}_{\tau-1})^\intercal / m$, and the RHS will become

$$\sum_{t \in \mathcal{T}_{w<1}^{(T)}} \min \left\{ \gamma_{t-1} \cdot \left\| g(\boldsymbol{x}_t; \boldsymbol{\theta}_{t-1})/\sqrt{m} \right\|_{\Gamma_{t-1}^{-1}}, 1 \right\} \leq (1 + \frac{\gamma_T}{\alpha}) \cdot \sum_{t \in \mathcal{T}_{w<1}^{(T)}} \left\{ \left\| g'(\boldsymbol{x}_t; \boldsymbol{\theta}_{t-1})/\sqrt{m} \right\|_{(\Gamma_{t-1}^{w<1})^{-1}}^2, 1 \right\}$$

$$\leq (1 + \frac{\gamma_T}{\alpha}) \cdot \widetilde{d} \log(1 + TK/\lambda).$$

where the last inequality is by applying Lemma G.7 and Lemma C.2. Summing up the results will finish the proof. $\qquad\square$

# G  Lemmas for Over-parameterized FC Neural Networks

Given the input arm context vector $\boldsymbol{x} \in \mathbb{R}^d$, we denote the $L$-layer FC neural network with width $m$ as

$$f(\boldsymbol{x}; \boldsymbol{\theta}) = \boldsymbol{\theta}_L (\prod_{l=1}^{L-1} \mathbf{D}_l \boldsymbol{\theta}_l) \cdot \boldsymbol{x}, \tag{G.1}$$

where with $\sigma$ being the ReLU activation, we define the intermediate hidden representations $\boldsymbol{h}_l, l \in \{0, \ldots, L-1\}$ as

$$\boldsymbol{h}_0 = \boldsymbol{x}, \quad \boldsymbol{h}_l = \sigma(\boldsymbol{\theta}_l \boldsymbol{h}_{l-1}), l \in [L-1].$$

and we also have the binary diagonal matrix functioning as the ReLU activation being

$$\mathbf{D}_l = \mathrm{diag}(\mathbb{I}\{(\boldsymbol{\theta}_l \boldsymbol{h}_{l-1})_1\}, \ldots, \mathbb{I}\{(\boldsymbol{\theta}_l \boldsymbol{h}_{l-1})_m\}), l \in [L-1].$$

where $\mathbb{I}(\cdot)$ is the indicator function. Afterwards, the corresponding gradients will become

$$\nabla_{\boldsymbol{\theta}_l} f(\boldsymbol{x}; \boldsymbol{\theta}) = \begin{cases} [\boldsymbol{h}_{l-1} \boldsymbol{\theta}_L (\prod_{\tau=l+1}^{L-1} \mathbf{D}_\tau \boldsymbol{\theta}_\tau)]^\intercal, l \in [L-1] \\ \boldsymbol{h}_{L-1}^\intercal, l = L. \end{cases} \tag{G.2}$$

**Lemma G.1.** *There exists a positive constant $C > 0$ such that with probability at least $1 - \delta$, if $m \geq CT^4 L^6 \log(T^2 L/\delta)/\lambda^4$ for each arbitrary $\boldsymbol{x}_\tau \in \bigcup_{\tau \in [t]} \mathcal{X}_\tau$, there exists a set of parameters $\boldsymbol{\theta}^*$ such that with the neural network parameters $\boldsymbol{\theta}_{t-1}$ trained on $\{\boldsymbol{x}_\tau\}_{\tau=1}^{t-1}$, we have*

$$|\langle g(\boldsymbol{x}; \boldsymbol{\theta}_0), \boldsymbol{\theta}^* - \boldsymbol{\theta}_0 \rangle - \langle g(\boldsymbol{x}; \boldsymbol{\theta}_{t-1}), \boldsymbol{\theta}^* - \boldsymbol{\theta}_0 \rangle| \leq \mathcal{O}(Sm^{-1/6}\sqrt{\log(m)}t^{1/6}\lambda^{-1/6}L^{2/7})$$

*where parameters $\boldsymbol{\theta}^*$ satisfy $\|\boldsymbol{\theta}^* - \boldsymbol{\theta}_0\| \leq S/\sqrt{m}, S > 0$ as shown in Lemma C.1.*

**Proof.** This lemma is based on Lemma C.1. Here, our objective can be reformed into

$$|\langle g(\boldsymbol{x}; \boldsymbol{\theta}_0), \boldsymbol{\theta}^* - \boldsymbol{\theta}_0 \rangle - \langle g(\boldsymbol{x}; \boldsymbol{\theta}_{t-1}), \boldsymbol{\theta}^* - \boldsymbol{\theta}_0 \rangle| = \|\boldsymbol{\theta}^* - \boldsymbol{\theta}_0\|_2 \cdot \left( g(\boldsymbol{x}; \boldsymbol{\theta}_0) - g(\boldsymbol{x}; \boldsymbol{\theta}_{t-1}) \right)$$

$$\leq S/\sqrt{m} \cdot \left( g(\boldsymbol{x}; \boldsymbol{\theta}_0) - g(\boldsymbol{x}; \boldsymbol{\theta}_{t-1}) \right)$$

$$\leq \mathcal{O}(Sm^{-1/6}\sqrt{\log(m)}t^{1/6}\lambda^{-1/6}L^{2/7}),$$

where the first inequality is due to Lemma C.1, and the second inequality is due to Lemmas G.2, G.3, and Lemma G.4. $\qquad\square$

**Lemma G.2** (Lemma B.3 in [21] ). *There exist constants $\{C_1, C_2\}$ such that for any $\delta > 0$, if we have*

$$\omega \leq C_1 L^{-6} (\log m)^{-3/2},$$

*then with probability at least $1 - \delta$, for any $\|\boldsymbol{\theta} - \boldsymbol{\theta}_0\| \leq \omega$ and for $\boldsymbol{x} \in \{\mathcal{X}_t\}_{t=1}^T$ we have $\|g(\boldsymbol{x}; \boldsymbol{\theta})\|_2 \leq C_2\sqrt{mL}$.*

**Proof.** In terms of the gradient upper bound, directly applying Lemma B.3 in [21] will give the desired result that $\|g(\boldsymbol{x}; \boldsymbol{\theta})\|_2 \leq \mathcal{O}(\sqrt{mL})$. $\qquad\square$

**Lemma G.3** (Lemma B.2 in [86] )**.** *For the L-layer full-connected network f trained with J iterations of GD, there exist constants $\{C_i\}_{i=1}^5 \geq 0$ such that for $\delta > 0$, if for all $t \in [T]$, $\eta, m$ satisfy*

$$2\sqrt{t/(m\lambda)} \geq C_1 m^{-3/2} L^{-3/2} [\log(TL^2/\delta)]^{3/2},$$
$$2\sqrt{t/(m\lambda)} \leq C_2 \min\{L^{-6}[\log m]^{-3/2}, (m(\lambda\eta)^2 L^{-6} t^{-1}(\log m)^{-1})^{3/8}\},$$
$$\eta \leq C_3 (m\lambda + tmL)^{-1},$$
$$m^{1/6} \geq C_4 \sqrt{\log m} L^{7/2} t^{7/6} \lambda^{-7/6}(1 + \sqrt{t/\lambda}),$$

*then, with probability at least $1 - \delta$, we have*

$$\|\boldsymbol{\theta}_t - \boldsymbol{\theta}_0\| \leq 2\sqrt{t/(m\lambda)}$$
$$\|\boldsymbol{\theta}_t - \boldsymbol{\theta}_0 - \bar{\boldsymbol{\Sigma}}_t^{-1} \bar{\boldsymbol{b}}_t / \sqrt{m}\| \leq (1 - \eta m\lambda)^{J/2} \sqrt{t/(m\lambda)} + C_5 m^{-2/3} \sqrt{\log m} L^{7/2} t^{5/3} \lambda^{-5/3}(1 + \sqrt{t/\lambda}).$$

*where the unweighted regression parameters are $\bar{\boldsymbol{\Sigma}}_t = \lambda\mathbf{I} + \sum_{\tau \in [t]} g(\boldsymbol{x}_\tau; \boldsymbol{\theta}_{\tau-1}) g(\boldsymbol{x}_\tau; \boldsymbol{\theta}_{\tau-1})^\intercal / m$ and $\bar{\boldsymbol{b}}_t = \sum_{\tau \in [t]} g(\boldsymbol{x}_\tau; \boldsymbol{\theta}_{\tau-1}) r_\tau / \sqrt{m}$.*

**Lemma G.4** (Theorem 5 in [3])**.** *With probability at least $1 - \delta$, there exist constants $C_1, C_2$ such that if $\omega \leq C_1 L^{-9/2} \log^{-3} m$, for $\|\boldsymbol{\theta}_t - \boldsymbol{\theta}_0\|_2 \leq \omega$, we have*

$$\|g(\boldsymbol{x}; \boldsymbol{\theta}_t) - g(\boldsymbol{x}; \boldsymbol{\theta}_0)\|_2 \leq C_2 \sqrt{\log m} \omega^{1/3} L^3 \|g(\boldsymbol{x}; \boldsymbol{\theta}_0)\|_2.$$

**Lemma G.5** (Lemma 4.1 in [21])**.** *There exist constants $\{\bar{C}_{i=1}^3\} \geq 0$ such that for any $\delta \geq 0$, if $\tau$ satisfies that*
$$\tau \leq \bar{C}_2 L^{-6} [\log m]^{-3/2},$$
*then with probability at least $1 - \delta$, for all $\boldsymbol{\theta}^1, \boldsymbol{\theta}^2$ satisfying $\|\boldsymbol{\theta}^1 - \boldsymbol{\theta}_0\| \leq \tau$, $\|\boldsymbol{\theta}^2 - \boldsymbol{\theta}_0\| \leq \tau$ and for any $\boldsymbol{x} \in \{\boldsymbol{x}_t\}_{t=1}^T$, we have*

$$|f(\boldsymbol{x}; \boldsymbol{\theta}^1) - f(\boldsymbol{x}; \boldsymbol{\theta}^2) - \langle(g(\boldsymbol{x}; \boldsymbol{\theta}^2), \boldsymbol{\theta}^1 - \boldsymbol{\theta}^2)\rangle| \leq \bar{C}_3 \tau^{4/3} L^3 \sqrt{m \log m}.$$

**Lemma G.6.** *Suppose $m$ satisfies the conditions in Theorem 5.6. Suppose the gradient matrix can be represented by $\boldsymbol{\Sigma} = \lambda\mathbf{I} + \sum_{t \in [T]} g(\boldsymbol{x}_{i,t}; \boldsymbol{\theta}_{t-1}) \cdot g(\boldsymbol{x}_{i,t}; \boldsymbol{\theta}_{t-1})^\intercal / m$, with an arbitrary arm $\boldsymbol{x}_{i,t} \in \mathcal{X}_t$ from each time step $t$. With probability at least $1 - \delta$ over the initialization, the following results hold:*
$$\|\boldsymbol{\Sigma}\|_2 \leq \lambda + \mathcal{O}(TL),$$
$$\|\boldsymbol{\Sigma} - \boldsymbol{\Sigma}^{(0)}\|_F \leq \mathcal{O}(m^{-1/6} \sqrt{\log(m)} L^4 t^{7/6} \lambda^{-1/6})$$
$$\|\log \frac{\det(\boldsymbol{\Sigma})}{\det(\lambda\mathbf{I})} - \log \frac{\det(\boldsymbol{\Sigma}^{(0)})}{\det(\lambda\mathbf{I})}\|_F \leq \mathcal{O}(m^{-1/6} \sqrt{\log(m)} L^4 t^{5/6} \lambda^{-1/6}),$$

*where the gradient matrix defined based on randomly initialized parameters $\boldsymbol{\theta}_0$ is $\boldsymbol{\Sigma}^{(0)} = \lambda\mathbf{I} + \sum_{t \in [T]} g(\boldsymbol{x}_{i,t}; \boldsymbol{\theta}_0) \cdot g(\boldsymbol{x}_{i,t}; \boldsymbol{\theta}_0)^\intercal / m$.*

**Proof.** Based on the Lemma G.2, for any $t \in [T]$, $\|g(\boldsymbol{x}_{i,t}; \boldsymbol{\Theta}_0)\|_2 \leq \mathcal{O}(\sqrt{mL})$. Then, for the first inequality:

$$\|\boldsymbol{\Sigma}^{(0)}\|_2 = \|\lambda\mathbf{I} + \sum_{t=1}^T g(\boldsymbol{x}_{i,t}; \boldsymbol{\Theta}_0) g(\boldsymbol{x}_{i,t}; \boldsymbol{\Theta}_0)^\intercal / m\|_2$$

$$\leq \|\lambda\mathbf{I}\|_2 + \|\sum_{t=1}^T g(\boldsymbol{x}_{i,t}; \boldsymbol{\Theta}_0) g(\boldsymbol{x}_{i,t}; \boldsymbol{\Theta}_0)^\intercal / m\|_2$$

$$\leq \lambda + \sum_{t=1}^T \|g(\boldsymbol{x}_{i,t}; \boldsymbol{\Theta}_0)/\sqrt{m}\|_2^2 \leq \lambda + \mathcal{O}(TL).$$

Then, the second and third inequalities in this lemma are the direct application of Lemma B.3 of [86].

$\qquad\square$

**Lemma G.7** (Lemma 11 in [1], Lemma B.7 in [86])**.** *Suppose a sequence of arms* $\{\boldsymbol{x}'_\tau\}_{\tau \in [t]}$, *with an arbitrary arm* $\boldsymbol{x}'_\tau \in \mathcal{X}_\tau$ *from each time step* $\tau \in [t]$. *The gradient matrix is denoted by* $\boldsymbol{\Sigma}_t = \lambda \mathbf{I} + \sum_{\tau \in [t]} g(\boldsymbol{x}'_\tau; \boldsymbol{\theta}_{\tau-1}) \cdot g(\boldsymbol{x}'_\tau; \boldsymbol{\theta}_{\tau-1})^{\mathsf{T}}/m$, *where* $\boldsymbol{\theta}_{\tau-1}$ *refer to network parameters in round* $\tau \in [t]$. *We can have*

$$\sum_{\tau \in [t]} \min \left\{ \| g(\boldsymbol{x}_\tau; \boldsymbol{\theta}_{\tau-1})/\sqrt{m} \|^2_{\boldsymbol{\Sigma}^{-1}_{\tau-1}}, \ 1 \right\} \leq 2 \log \frac{\det(\boldsymbol{\Sigma}_t)}{\det(\lambda \mathbf{I})}.$$

## G.1 Auxiliary Lemma

**Lemma G.8** ((Corollary 7.7.4. (a) from [45])**.** *Let* $\mathbf{A}, \mathbf{B}$ *be Hermitian matrices of the same shape, and suppose they are positive semi-definite. Then, we have* $\mathbf{A} \succeq \mathbf{B}$ *iff* $\mathbf{A}^{-1} \preceq \mathbf{B}^{-1}$.

