# OpenReview forum: "Robust Neural Contextual Bandit against Adversarial Corruptions"
_NeurIPS.cc/2024/Conference — NeurIPS 2024 poster_

### Official Review · Reviewer_1FL3 · 2024-07-09

**Soundness:** 2
**Presentation:** 2
**Contribution:** 3
**Rating:** 5
**Confidence:** 3

**Summary:**

This paper studies the problem of contextual bandits with neural function approximation faced with adversarial corruptions. It proposes an algorithm named R-NeuralUCB, which can improve the robustness of neural contextual bandit training. It provides regret analysis and  conducts experiments to show the advantage of the new algorithm.

**Strengths:**

1. This paper proposes a new algorithm for neural contextual bandits, which is based on a new technique that customizes individual sets of network parameters for each candidate arm. It can improve the robustness under adversarial corruption.
2. Theoretical analysis has been provided, with a robust regret bound dependent on the effective dimensions of the neural network.
3. Experiments are conducted on publicly available real-world data sets to show the better performance of the proposed algorithm.

**Weaknesses:**

1. The computational cost is huge. Compared with NeuralUCB-WGD, the main difference lies in the separate neural networks for each candidate arm, which greatly increases the computation cost, as it needs to compute the gradient descent for every arm in each round separately.

2. In [1], Theorem 4.12 shows that if when no corruption, $Regret(T) \le R_T$, then when $C > \Omega(R_T/d)$, the algorithm will suffer $\Omega(T)$ regret. It seems contradictory with Theorem 5.6 for unknown $C$.

3. Theorem 5.6 relies on the tuning of parameter $\alpha$, to achieve that $\min w_{i,t}^\tau = \kappa^2, \forall t$, which is unclear how to achieve.


[1]  He et al. Nearly optimal algorithms for linear contextual bandits with adversarial corruptions Neurips 2022

**Questions:**

See Weaknesses

**Limitations:**

The authors have addressed the limitations and potential societal impact of their work.

---

> ### Author Rebuttal · Authors · 2024-08-07
>
> We would like to sincerely thank the reviewer for your valuable questions and comments.
> Given the page limit of 6000 characters, we will try our best to provide our detailed response in the form of Q\&A. Please also see our manuscript for cited papers. *Thank you!*
>
>
>
> **Q1: Overall discussion on computational efficiency?**
>
>
>
>
>
> Please kindly refer to our "Global Response" for complementary discussions.
>
> To reduce the computational cost for obtaining arm-specific network parameters (line 9, Algorithm 1) in practice, for our experiments, we apply the warm-start GD strategy to strike a good balance between performances and computational costs. Pseudo-code and detailed descriptions are in Appendix Subsection B.6 due to page limit.
>
> In round $t\in \\{3, \dots, T\\}$, for each candidate arm $x_{i, t}, i\in [K]$, we can acquire its arm-specific parameters $\theta_{i, t-1}$ by fine-tuning existing trained parameters $\theta_{t-2}$ with a small number of training samples (received arm-reward pairs), instead of obtaining $\theta_{i, t-1}$ by training from scratch, i.e., starting from randomly initialized $\theta_{0}$ and training with a large number of samples.
> The intuition is inspired by meta-learning works [Finn et al.], where we consider each candidate arm as a "task". In this case, for round $t$, we can start from previously trained model parameters $\theta_{t-2}$, and adapt $\theta_{t-2}$ to each candidate arm (task), with a small number of training samples to obtain arm(task)-specific parameters.
>
> This helps R-NeuralUCB (1) reduce computational costs of calculating arm weights for GD, and keep inference time relatively stable across horizon $T$, since we apply a fixed number of samples for adaptation; (2) reduce GD iterations needed, as starting from $\theta_{0}$ will require more GD iterations for model convergence.
>
>
> Meanwhile, it is also a common practice for neural bandit works to adopt warm-start training in practice.
> Here, on one hand, neural bandit works generally formulate their GD training process by starting from randomly initialized $\theta_{0}$ (e.g, [76,74,7,8,60]), in compliance with theoretical analysis requirements. On the other hand, however, for their experiments and source code, it is common to train model parameters incrementally, by updating trained parameters from previous rounds, since it is more computationally efficient for real applications.
>
>
>
> Please see our "Global Response" PDF file for added experiments: (1) With experiments on warm-start vs. restarting from $\theta_{0}$ (Figure 1) given the same number of training (adaptation) samples, we see the warm-start process can improve performances. (2) With experiments based on increased number of arms $K$ ($K=50, 100$, Figure 2), we see that the warm-start strategy can lead to good performances while helping the inference time stay relatively stable.
>
>
>
> *[Finn et al.] Model-agnostic meta-learning for fast adaptation
> of deep networks.*
>
>
>
>
>
>
>
>
>
> **Q2: Relationship with the lower bound in (He et al., 2022)?**
>
>
> First, we would like to first recall that Theorem 4.12 of [35] relies on Assumption 2.1 of [35], which formulates a learning problem under linear contextual bandit settings.
> Then, in our Theorem 5.6, when $C=0$, we have the corruption-free regret of $\tilde{\mathcal{O}}(\tilde{d}\sqrt{T} + S\sqrt{\tilde{d}T})$. In this case, after setting $C = \Omega(R_{T}/d)$, we will end up with a regret bound of $\tilde{\mathcal{O}}\left( (\tilde{d}^{2}\sqrt{T} + S\tilde{d}^{3/2}\sqrt{T}) \cdot C \beta^{-1}d^{-1} \right)$.
> Here, we note that with the proof flow of Theorem 4.12 in [35] and learning problem specified by Assumption 2.1 of [35], our effective dimension term $\tilde{d}^{2}$ can possibly depend on horizon $T$, and grow along with $T$ [22].
> As a result, although the regret bound only contains $\sqrt{T}$ terms, the overall order of regret bound can be equal or greater than $\mathcal{O}(T)$ due to effective dimension $\tilde{d}^{2}$, as discussed in [22].
> This is distinct from linear bandits works, where regret bounds generally only consist of horizon $T$ and other $T$-independent terms, such as context dimension $d$ and fixed $T$-independent linear parameter norm $\|\theta^{*}\|_{2}$.
> Therefore, our Theorem 5.6 will not contradict with Theorem 4.12 of [35].
>
>
> Meanwhile, we also note that for stochastic neural bandit works, the effective dimension term $\tilde{d}$ (or information gain for kernelized bandits [65,12]) is generally inevitable [76,74,7,43], since we need to bridge over-parameterized networks with the NTK-based regression model. Comparably, kernelized bandits [12,65,24] will also generally involve similar terms as costs of modeling the unknown non-linear reward mapping function.
>
> We appreciate the reviewer for your question. For readers' reference, we have included above discussions to Appendix Subsection B.7, as the supplement to our current discussion of the lower bound.
>
>
>
>
>
>
> **Q3: How to tune parameter $\alpha$ to satisfy the requirements of $\kappa$?**
>
> We first recall that in round $t\in [T]$, we have arm weights $w\_{i, t}^{(\tau)}, \tau\in [t-1], i\in [K]$. Based on line 184, we can also denote $w\_{i, t}^{(\tau)} = \min\\{ 1,\alpha\cdot \textsf{frac}\_{\tau}(x\_{i, t}; \mathcal{X}\_{t}, \bar{\Sigma}\_{t-1})  \\}$.
> Here, instead of deeming $\alpha$ as a fixed value across horizon $T$, we can consider $\alpha$ to be varying across different rounds, denoted by $\alpha\_{t}, t\in [T]$.
> With $\textsf{frac}\_{t}^{\textsf{min}} = \min\_{i\in [K], \tau\in [t-1]}[
> \textsf{frac}\_{\tau}(x\_{i, t}; \mathcal{X}\_{t}, \bar{\Sigma}\_{t-1}) ]$, we can set each $\alpha\_{t} = \kappa^{2} / \textsf{frac}\_{t}^{\textsf{min}}, \kappa\in (0, 1)$.
> In this way, we can consequently have $\min\\{w\_{i, t}^{(\tau)}\\}\_{i\in [K], \tau\in [t-1]} = \kappa^{2}, \forall t\in [T]$.
>
> To improve the paper presentation, we have also added above explanations to the manuscript for readers' reference.

---

> > ### Comment · Reviewer_1FL3 · 2024-08-12
> >
> > Thanks for your detailed responses, which are really helpful. I will keep my scores.

---

> > > ### Author Response · Authors · 2024-08-12
> > > **Thank you for the review and feedback!**
> > >
> > > Dear Reviewer 1FL3,
> > >
> > > Thank you again for your insightful comments and questions, and we will definitely integrate our discussions into the manuscript as readers' reference. Meanwhile, please also kindly let us know if you have any further concerns or questions. We will be more than glad to provide additional explanations in a timely manner. Thank you again!
> > >
> > > Best regards,
> > >
> > > Authors

---

### Official Review · Reviewer_Fhvw · 2024-07-12

**Soundness:** 3
**Presentation:** 3
**Contribution:** 3
**Rating:** 6
**Confidence:** 3

**Summary:**

This paper proposes a novel neural contextual bandit algorithm, called R-NeuralUCB, to improve robustness against adversarial reward corruptions. The authors provide regret analysis for R-NeuralUCB under over-parameterized neural network settings, without the commonly adopted arm separateness assumption. The authors also empirically compare R-NeuralUCB with baseline algorithms on three real data sets, under different adversarial corruption scenarios.

**Strengths:**

The paper is well-written. To the best of my knowledge, this work proposes the first theoretical result for neural bandits with adversarial corruptions. The theoretical analysis seems solid and the experiments show that R-NeuralUCB outperforms baseline algorithms in the corruption setting.

**Weaknesses:**

My main concern is about the computational costs. Notice that R-NeuralUCB need to maintain a neural network for each arm. Thus I think R-NeuralUCB could be very expensive when $K$ is very large. I will be happy if the authors could provide some discussions on the computational costs and show some empirical results for larger $K$.

**Questions:**

In the experiments, NeuralUCB and NeuralTS have similar performance as R-NeuralUCB. Is it possible that these two methods also have sublinear regret under adversarial attacks?

---

> ### Author Rebuttal · Authors · 2024-08-07
>
> We would like to sincerely thank the reviewer for your valuable questions and comments.
> Given the page limit of 6000 characters, we will try our best to provide our detailed response in the form of Q\&A. Please also see our manuscript for cited papers. *Thank you!*
>
>
>
>
>
> **Q1: Overall discussion on computational efficiency? Performance with large $K$?**
>
>
>
>
> Please see our "Global Response" PDF file for added experiments. (1) With experiments on warm-start vs. restarting from $\theta_{0}$ given the same number of training (adaptation) samples (Figure 1), we see the warm-start process can improve performances. (2) With experiments based on increased number of arms $K$ ($K=50, 100$, Figure 2), we see that the warm-start strategy can lead to good performances while helping the inference time stay relatively stable.
>
> In practice, for our experiments, we can adopt the warm-start GD process (details and pseudo-code are in Appendix Subsection B.6) to reduce the computational cost, in terms of deriving the arm-specific network parameters (line 9, Algorithm 1).
> In round $t\in \\{3, \dots, T\\}$, for each candidate arm $x_{i, t}, i\in [K]$, we can acquire its arm-specific parameters $\theta_{i, t-1}$ by fine-tuning existing trained parameters $\theta_{t-2}$ with a small number of training samples (received arm-reward pairs), instead of obtaining $\theta_{i, t-1}$ by training from scratch, i.e., starting from randomly initialized $\theta_{0}$ with a large number of samples.
> Similar ideas are also applied by other bandit-related works (e.g., [9]).
> The intuition is inspired by meta-learning [Finn et al.] where we consider each candidate arm as a "task". In this case, for round $t$, we can start from previously trained model parameters $\theta_{t-2}$, and adapt $\theta_{t-2}$ to each candidate arm (task) with a small number of training samples to obtain arm(task)-specific parameters. This also reduces our computational costs of calculating arm weights for GD, and keep the inference time relatively stable across rounds.
>
>
>
> Meanwhile, it is also a common practice for neural bandit works to adopt warm-start training in practice. On one hand, neural bandit works generally formulate their GD training process by starting from randomly initialized $\theta_{0}$, in compliance with theoretical analysis requirements. On the other hand, however, for their experiments and source code (e.g, [76,74,7,8,60]), it is a common technique to train model parameters incrementally, by updating trained parameters from previous rounds, since it is more practical and computationally efficient for real applications.
>
>
>
>
> Therefore, we adopt the following approaches to strike a good balance between model performance and computational costs:
>
> - (1) Inspired by meta-learning ideas [Finn et al.], we utilize warm-start GD by adapting previously trained network parameters $\theta_{t-2}$ for each candidate arm with a small number of training samples, instead of training from $\theta_{0}$ with a large number of samples. Details are in Appendix Subsection B.6.
>
> - (2) Based on our formulation of the warm-start GD process (Algorithm 2, Subsection B.6), we will sample a fixed number of mini-batch training samples (i.e., received arm-reward pairs) for each candidate arm to calculate arm weights and perform GD. This is inspired by the idea of mini-batch warm-start GD training [9]. In this case, a fixed number of training samples can help the round-wise inference time stay relatively stable, without growing drastically along with $T$.
>
>
>
> *[Finn et al.] Model-agnostic meta-learning for fast adaptation
> of deep networks.*
>
>
>
>
>
>
>
>
> **Q2: Is it possible that Neural-UCB and Neural-TS also have sub-linear regret under adversarial attacks?**
>
>
> We agree with the reviewer that it is interesting and important to investigate the regret bound of vanilla neural bandit algorithms, in order to better understand the benefit of proposed methods. In particular, to offer some insights on regret bound of Neural-UCB, one possible route is to follow an analogous approach as the regret analysis of NeuralUCB-WGD.
> On the other hand, regarding Thompson Sampling based approaches, their analysis can be significantly different from our UCB-based methods, where we consider UCB-based analysis in this paper. Therefore, we consider the regret analysis of Neural-TS under corruptions as a part of future works.
>
> Here, the key idea is to quantify the impact of adversarial corruptions upon the confidence ellipsoid around trained parameters.
> By following the derivations in Lemma F.1, denoting the corruption-free confidence radius as $\tilde{\gamma}\_{t-1}$ in round $t$, we can have the corrupted confidence ellipsoid for Neural-UCB as
> $\mathcal{C}\_{t-1} = \\{ \theta: \\|\theta - \theta\_{t-1}\\|\_{\Gamma\_{t-1}} \leq \gamma\_{t-1} / \sqrt{m} \\}$, where $\gamma\_{t-1} = \tilde{\gamma}\_{t-1} + \mathcal{O}(CL\lambda^{-1/2})$.
> This result is derived by applying $w\_{\tau}=1, \tau\in [t-1]$ and the fact $\sum\_{\tau\in [t]}|c\_{\tau}| \leq C$, as well as Lemma G.2 and the initialization of the gradient covariance matrix $\Gamma$.
> As a result, following the proof flow of Lemma 5.3 in [76], we can have regret upper bound $\tilde{\mathcal{O}}(\tilde{d}\sqrt{T} + \sqrt{S\tilde{d}T} + CL\sqrt{\tilde{d}T/\lambda})$, which will introduce an additional $\tilde{\mathcal{O}}(\sqrt{T})$ to the corruption-dependent term.
> Note that although it is possible for Neural-UCB to have a sub-linear cumulative regret when $C$ is a small constant, our R-NeuralUCB and NeuralUCB-WGD can remove the direct dependency of horizon term $T$, and manage to achieve non-trivial $\tilde{\mathcal{O}}(C\tilde{d}\beta^{-1})$ and $\tilde{\mathcal{O}}(C\tilde{d}^{3/2} + C\tilde{d}\sqrt{\lambda}S)$ respectively for corruption-dependent terms.
> For readers' reference, we have also added above discussions to the manuscript Appendix.

---

> > ### Comment · Reviewer_Fhvw · 2024-08-11
> >
> > Thank you for the detailed response. I will keep my score. I hope the authors will include a discussion of computational efficiency in the final version.

---

> ### Author Response · Authors · 2024-08-11
> **Thank you for the review!**
>
> Dear Reviewer Fhvw,
>
> Thank you again for your insightful review and comments, and we will definitely include detailed discussions related to computational efficiency to our final manuscript. Thanks again.
>
> Best regards,
>
> Authors

---

### Official Review · Reviewer_MEPN · 2024-07-13

**Soundness:** 3
**Presentation:** 4
**Contribution:** 3
**Rating:** 7
**Confidence:** 4

**Summary:**

This paper presents R-NeuralUCB, a neural-based network UCB algorithm for robustness under adversarial rewards corruptions in stochastic $K$ multi-armed contextual bandits. Based on NeuralUCB [1], before the arm pulling, R-NeuralUCB additionally optimizes a context-aware gradient descent for each arm by using an objective function. The first term of this function is the weighted-arm uncertainty information from the cumulative observed training samples. The second term is inspired by existing works on enhancing model robustness with the regularization technique between current and initial weights. Contribution includes:
- A novel R-NeuralUCB algorithm, where each arm weight is formulated by the weight between gradient norm UCB w.r.t. context $x$ and weights $\theta$ of training and candidate arm. This intuitively means when the uncertainty w.r.t. candidate's arm is high, this weight will be low, encouraging the model to focus on the regularizer to avoid over-fitting. Conversely, if the uncertainty w.r.t. candidate's arm is low, yielding the weight will be high, pushing the model focus on the observed training samples
- A theoretical upper bound for the proposed algorithm, where the bound for R-NeralUCB is $\tilde{\mathcal{O}}(\tilde{d}\sqrt{T}+C\beta^{-1}\tilde{d}\)$, where $\tilde{d}\$ is the dimensions of NTK matrix, $T$ is the time-horizon, C is the adversarial corruption level C, and $\beta$ is the data-dependent gradient deviation term. Additionally, the proof of this bound does not require the NTK matrix to be positive-definite.
- Experiments show the proposed algorithm has a lower cumulative regret than Neural-UCB and other baselines across different datasets and reward corruption types.

**Strengths:**

- This paper has clear mathematical notations, is very well-written, and is clear to understand the important aspects of the algorithm.
- I like the motivation to improve the robustness of neural-UCB-based methods under reward corruption because of their real-world applications.
- The proposed algorithm also makes sense to me, the weighing arm is based on the uncertainty of the upper confidence bounds, reflecting the reward estimation uncertainty of the model. Therefore, we can use these weights to control the fitting of the cumulative observed training data and prevent impacts caused by adversarial corruption.
- The theoretical contribution of this paper is good, yielding a solid R-NeralUCB algorithm. I appreciate the theory of this paper for two main reasons. Firstly, in the scope of the neural-based contextual bandits, I think this is the first work to provide a regret upper bounds with the reward corruption value. Secondly, while neural-based UCB often assumes the NTK matrix in Def. 5.1 is positive-definite, the theoretical analysis of this paper does not require this assumption.
- The authors additionally provide a base algorithm NeuralUCB-WGD, which uses the weighted GD with Neural-UCB. Theoretical analysis and experimental results of NeuralUCB-WGD are also provided sufficiently.
- Experiments are also elevated across different settings, including datasets, reward corruption types, and running seeds.

**Weaknesses:**

- My biggest concern about R-NeuralUCB is its computational inefficiency, causing a big limitation in the real world with neural-based contextual bandits. Specifically, while NeuralUCB has been criticized be inefficient by the exploration with UCB performed over the entire DNN parameter space and its gradient [2], R-NeuralUCB is even less inefficient than NeuralUCB for the following reasons:
  - Firstly, R-NeuralUCB requires different neural network model weights for different arms, so R-NeuralUCB is very expensive when $K$ is high.
  - Secondly, for each time $t \in [T]$, R-NeuralUCB requires additionally optimizing the Context-aware GD objective function in Eq. 5 $K$ times.
  - Thirdly, this Context-aware GD objective function requires computing each weight for each cumulative data, and the numerator of this weight requires finding the min across $\mathcal{X}_t$, meaning that when $t$ and $K$ increase, the computational cost also increases.
- The proposed algorithm requires to define some non-trivial hyper-parameters such as $\alpha$ in Eq. 4 and $\lambda$ in Eq. 5.
- Similar to other neural-networking-based contextual UCB papers, the proofs must assume the activation of the network structures is ReLU, the neural network model is over-parameterized, and the dimension of the weight matrix at middle layers $l$-th has the same size $m \times m$. As also mentioned by the authors, the theory lacks the lower bound for the regret of R-NeuralUCB.
- Lack of experiment settings, e.g., comparison with other baselines [2, 3, 4], synthetic (i.e., the true reward function is known) and other real-world datasets [1], and a longer time horizon $T$. In some experiments (e.g., Amazon dataset), R-NeuralUCB does not significantly outperform other baselines.
- Miscellaneous: In Figure 1 on MovieLes Dataset right, the cumulative looks are not significantly lower than others. I would recommend the authors zoom in on it by limiting the y-axis value to 400, so the readers can see your better regret more easily.

**Questions:**

1. Can you draw two figures, including the first bar chart for the model size (i.e., number of neural network parameters) comparison, and the second line graph for the inference speed (i.e., latency) comparison at each time $t$?
2. While NeuralUCB [1] set $T=15,000$ in MNIST, why do the authors only limit $T=10,000$? Is this because of the computational reason? Also, do you have any comments to improve the computational efficiency of your algorithms?
3. The paragraph from L-207 to L-212 does not make sense to me. Specifically, why replacing the initial weight $\theta_0$ by the warm-start GD can reduce computational cost? Do you mean the computational cost regarding the update step in L-15 in Algorithm 1? Does this replacement affect the loss function in Eq. 5? Do you have any empirical results in this regard (i.e., cumulative regret results between with and without the replacement)?
4. Can you make a comparison in corruption-free experiments, i.e., set $C=0$? I think this will test whether your Context-aware GD in Eq. (5) hurt NeuralUCB or not. Also in your experiments, at each time $t$, how do you set whether an arm is attacked or not? How do you think about the context corruption like using MNIST-C?
5. Have you considered the stochasticity of $C$ in your regret upper bound? Empirically, instead of fixing $C$ across time horizon $T$, $C$ can be stochastic, we also can empirically test them by (e.g., set $C$ to be monotonic increase, decrease, or randomly distributed by some mean and variance).
6. How do you make the neural network model in the experiment to be over-parameterized? Since the true reward function is unknown, how do you select the sub-gaussian value $\nu$ in your algorithm? Since $\beta$ in Theorem 5.6 depends on the selection of $\alpha$ in Eq. 4, does this mean that we can fine-tune the regret in the experiment by searching $\alpha$?

References:

[1] Zhou et al., Neural Contextual Bandits with UCB-based Exploration, ICML, 2020.

[2] Xu et al., Neural Contextual Bandits with Deep Representation and Shallow Exploration, ICLR, 2022.

[3] Bogunovi et al., Corruption-Tolerant Gaussian Process Bandit Optimization, AISTATS, 2020.

[4] Zhang et al., Contextual Gaussian Process Bandits with Neural Networks, NeurIPS, 2023.

**Limitations:**

Please see my comments on the Weaknesses.

---

> ### Author Rebuttal · Authors · 2024-08-07
>
> We would like to sincerely thank the reviewer for your valuable questions and comments.
> Given the page limit of 6000 characters, we will try our best to provide our detailed response in the form of Q\&A. Please also see our manuscript for cited papers. *Thank you!*
>
>
> **Q1: Discussion on computational efficiency? Why use warm-start GD?**
>
> *Due to page limit, please kindly refer to our "Global Response" for complementary discussions.*
>
> First, we would like to mention that the warm-start GD process (detailed descriptions and pseudo-code are in Appendix Subsection B.6) here is to reduce the computational cost for calculating arm-specific network parameters (line 9, Algorithm 1) in practice.
>
> In round $t$, for each candidate arm $x_{i, t}, i\in [K]$, we can acquire its arm-specific parameters $\theta_{i, t-1}$ by fine-tuning existing trained parameters $\theta_{t-2}$ with a small number of training samples (received arm-reward pairs), instead of obtaining $\theta_{i, t-1}$ by training from scratch, i.e., starting from randomly initialized $\theta_{0}$ with a large number of samples.
> The intuition is analogous to meta-learning works [Finn et al.], where we can consider each candidate arm as a "task". In this case, for round $t$, we can start from previously trained model parameters $\theta_{t-2}$, and adapt $\theta_{t-2}$ to each candidate arm (task) with a small number of training samples to obtain arm(task)-specific parameters.
> This helps R-NeuralUCB (1) reduce computational costs of calculating arm weights for GD, and keep inference time relatively stable across horizon $T$, since we are applying a fixed number of samples for adaptation; (2) reduce GD iterations needed, as starting from $\theta_{0}$ requires more GD iterations for model convergence.
>
>
> In the "Global Response" PDF file: (1) Experiments on warm-start vs. restarting from $\theta_{0}$ (Figure 1) *given the same number of training (adaptation) samples*. We see the warm-start process can improve performances. (2) Experiments based on increased number of arms $K$ ($K=50, 100$, Figure 2). We see that the warm-start strategy can lead to good performances while helping the inference time stay relatively stable.
>
>
> *[Finn et al.] Model-agnostic meta-learning for fast adaptation
> of deep networks.*
>
>
> **Q2: Additional experiments: inference speed with different model sizes? $T=15000$? Corruption-free experiments? Stochastic magnitude of corruption $C$?**
>
>
>
> Due to page limit, please kindly see our "Global Response" and attached PDF file for these experiments.
>
>
> - *Experiment 2* (Figure 1): Experiments with $T=15000$.
>
> - *Experiment 4* (Figure 3): Model size comparison, and line graph for inference speed.
>
> - *Experiment 5* (Figure 4): Corruption-free experiments.
>
> - *Experiment 6* (Figure 4): Stochastic magnitude of corruption experiment (increasing corruption probability).
>
>
>
>
> **Q3: Neural network over-parameterization?**
>
> Analogous to existing works (e.g. [76,74,7,8]), we utilize a two-layer fully-connected (FC) neural network ($L=2$) with hidden dimension $m=200$ for experiments, as in Appendix Subsection A.1.
>
>
> In terms of over-parameterization, we admit that for basically all the neural bandit works with experiments (e.g. [74,43,21,6,7,8]), there exists a gap between experiments and theoretical analysis.
> On one hand, along with more levels $L$ and larger hidden dimension $m$, neural networks will become increasingly difficult to train, more time consuming for inference, and more demanding for computational resources.
> In this case, to make neural bandits practical for real applications,
> neural bandit works generally use an ordinary-size neural network for experiments. It is also shown that with ordinary-size neural networks, neural bandit algorithms can already achieve significant performance gains over linear and kernel methods [76,74,7,8,60].
> On the other hand, from the theoretical perspective, neural networks need to be over-parameterized with $m\geq \mathcal{O}(Poly(T))$, so that they can approximate an arbitrary  reward mapping function $h(\cdot)$. Meanwhile, with over-parameterization, the difference between NTK-based regression models and neural networks will be sufficiently small for regret analysis, which makes it necessary for neural bandit papers.
> Therefore, we adopt a two-layer FC network for experiments, while performing analysis under over-parameterization settings, as in most existing neural bandit works (e.g., [76,74,7,8,60]).
>
>
> We have also include above discussions in the manuscript as reference to readers.
>
>
>
>
>
> **Q4:  Can be fine-tune $\alpha$ to control $\beta$ value in the regret bound?**
>
>
>
>
> Based on line 184 in the manuscript, for each arm $x\_{i, t}$, we can represent its arm weight as $w\_{i, t}^{(\tau)} = \min\\{ 1,\alpha\cdot \textsf{frac}\_{\tau}(x\_{i, t}; \mathcal{X}\_{t}, \bar{\Sigma}\_{t-1})  \\}, \tau\in [t-1]$, while the minimum fraction value $\beta$ is defined as $\beta = \min\_{ t\in [T], \tau\in [t-1]} [ \min\\{ \textsf{frac}\_{\tau}(x\_{t}; \mathcal{X}\_{t}, \bar{\Sigma}\_{t-1}), \textsf{frac}\_{\tau}(\tilde{x}\_{t}; \mathcal{X}\_{t}, \bar{\Sigma}\_{t-1})  \\} ]$ (line 301).
> In this case, tuning parameter $\alpha$ will not directly alter the $\beta$ value, since $\beta$ only explicitly depends on the fraction term $\textsf{frac}(\cdot)$ which is determined by candidate arms.
> On the other hand, different $\alpha$ can correspond to distinct $\kappa^{2}$ values in Theorem 5.6, which is the round-wise minimum arm weight. This will consequently affect the regret bound, as in line 307 of Theorem 5.6.
>
>
>
>
>
>
>
> **Q5:  Comments for Figure 1?**
>
> We sincerely appreciate your comments on improving the paper presentation, and we have integrated these comments into the manuscript.
>
>
>
>
>
> **Q6: Questions about implementation?**
>
> - "How arms are corrupted?": Please see "Global Response" Q3.
>
>
> - "Context corruption?": Please see "Global Response" Q4.
>
>
> - "How to choose $\nu$?": Please see "Global Response" Q5.

---

> > ### Comment · Reviewer_MEPN · 2024-08-11
> >
> > I thank the authors for the detailed rebuttal. I appreciate your explanation and additional results. I keep my original rating for this paper. I hope the authors can add a detailed discussion regarding computational efficiency in the final version. Additionally, I hope the authors can add more detail to the gap between choosing the exploration rate in the theorem and the practical algorithm in Q5. I understand this is a gap in the literature on neural bandits in general, but I believe it is worth discussing to help the community understand the gap between theory and experiments in this direction. Good luck!

---

> ### Author Response · Authors · 2024-08-11
> **Thank you for the feedback!**
>
> Dear Reviewer MEPN,
>
> We would like to sincerely thank you again for your valuable and insightful comments on improving our paper. We will definitely integrate these detailed discussions regarding computational efficiency, parameter selection, and theoretical analysis gap to our final manuscript. Thank you again for your invaluable review and feedback.
>
> Best regards,
>
> Authors

---

### Official Review · Reviewer_xUbn · 2024-07-31

**Soundness:** 4
**Presentation:** 4
**Contribution:** 4
**Rating:** 6
**Confidence:** 5

**Summary:**

The paper studied neural contextual bandits under adversarial reward corruptions. The adversary is allowed to perturb the reward after observing the action selected by the bandit player, but is subject to the constraint that the total reward corruption must be bounded by some budget C. Within this attack framework, the paper proposed a robust neural contextual bandit algorithm called R-NeuralUCB that formulates a novel context-aware gradient descent process, by taking the uncertainty information of both candidate arms and training samples into
consideration. When the neural network is over-parametrized, the paper provided theoretical study on the regret using the NTK technique. The authors proved sublinear regret that scales linearly as the total corruption level C grows. Empirical experiments demonstrated the effectiveness of the proposed robust bandit algorithm.

**Strengths:**

The paper studied a popular topic of robust bandit under corruption. Furthermore, different from existing works, this is the first paper that investigates the robustness of "neural" contextual bandit as far as I know, which significantly pushes the frontier of this area.

The authors provided solid theoretical analysis for the proposed R-NeuralUCB algorithm using the NTK technique, which is a great use case of NTK in the bandit domain. The results are interesting and appealing.

The authors further demonstrated the effectiveness and efficiency of the R-NeuralUCB algorithm on real-world dataset. The results are convincing and validated the theoretical findings discovered in the paper.

**Weaknesses:**

The idea to make the UCB algorithm robust is not novel. Prior works have applied similar idea that performs more exploration on arms with lower confidence. This work is mostly relying on the same idea to achieve robustness. Also, the final regret bound is not surprising to me. It is common that the regret scales linearly as the total corruption level C grows. That said, I was wondering if the authors have thought about providing a lower bound on how the regret scales as C grows? Is the upper bound tight?

**Questions:**

I was wondering if the authors have thought about providing a lower bound on how the regret scales as C grows. Is the upper bound tight?

**Limitations:**

Yes

---

> ### Author Rebuttal · Authors · 2024-08-07
>
> We would like to sincerely thank the reviewer for your valuable questions and comments. We will try our best to provide our detailed response in the form of Q\&A. Please also see our manuscript for cited papers. *Thank you!*
>
>
>
> **Q1: The regret lower bound in terms of corruption level $C$ under neural bandit settings?**
>
>
> We appreciate the reviewer for your question on the regret lower bound.
> Recall that in our Conclusion section as well as Appendix Subsection B.7, we mention that one limitation of this paper is lacking a theoretical regret lower bound, and we consider this task as a promising future direction of our work.
> This is because deriving the lower bound under neural bandits with adversarial corruption settings is significantly non-trivial, and it can lead to a different line of research work by proving these results themselves.
>
> Here, the non-linear bandit settings tend to pose extra challenges compared with linear bandit settings.
> For instance, for kernelized bandit works, their lower bounds will depend on kernel characteristics, such as $\Omega(C(\log(T))^{d/2})$ for the SE kernel and $\Omega(C^{\frac{v}{d+v}}T^{\frac{v}{d+v}})$ for the $v$-Mat\'ern kernel [63,12]. We see that the order of corruption level $C$ and whether the lower bound is related to non-logarithmic $T$, will both depend on kernel properties.
>
> As few restrictions are imposed for reward mapping $h(\cdot)$ for neural bandits, we will need to use regression models based on Neural Tangent Kernel (NTK) to perform theoretical analysis.
> In this case, without a well-established existing knowledge base for NTK (e.g., number of functions $M$ needed for the functional separateness condition [63,15] for NTK), it will require significant efforts to obtain such a lower bound for neural bandit cases, and can lead to a different line of research work by proving these results.
> In this case, we consider deriving such a theoretic regret lower bound as one promising and challenging future direction of this paper.

---

### Author Rebuttal · Authors · 2024-08-07

We would like to sincerely thank reviewers for your valuable questions and comments. *Please refer to attached PDF for added experiments.*
Given the page limit of 6000 characters, we will try our best to provide our detailed response.  Please also see our manuscript for cited papers. *Thank you!*





**Q1: Overall discussion on computational efficiency?**


In practice, for our experiments, we can adopt the warm-start GD process (details and pseudo-code are in Appendix Subsection B.6) to reduce the computational cost, in terms of deriving the arm-specific network parameters (line 9, Algorithm 1).
In round $t\in \\{3, \dots, T\\}$, for each candidate arm $x_{i, t}, i\in [K]$, we can acquire its arm-specific parameters $\theta_{i, t-1}$ by fine-tuning existing trained parameters $\theta_{t-2}$ with a small number of training samples (received arm-reward pairs), instead of obtaining $\theta_{i, t-1}$ by training from scratch, i.e., starting from randomly initialized $\theta_{0}$ with a large number of samples. Similar ideas on mini-batch warm-start training are also applied by other bandit-related works (e.g., [9]).



Meanwhile, it is also a common practice for neural bandit works to adopt warm-start training. On one hand, neural bandit works generally formulate their GD training process by starting from randomly initialized $\theta_{0}$ (e.g, [76,74,7,8,60]), in compliance with theoretical analysis requirements. On the other hand, however, for their experiments and source code, it is common to train model parameters incrementally, by updating trained parameters from previous rounds, since it is more practical and computationally efficient for real applications.





In this case, for our experiments, in order to strike a balance between computational costs and model performances:



- (1) Inspired by meta-learning ideas [Finn et al.], we utilize warm-start GD by adapting previously trained network parameters $\theta_{t-2}$ for each candidate arm with a small number of training samples, instead of training from $\theta_{0}$ with a large number of samples. Details are in Appendix Subsection B.6.

- (2) Based on our formulation of the warm-start GD process (Algorithm 2, Subsection B.6), we will sample a fixed number of mini-batch training samples (i.e., received arm-reward pairs) for each candidate arm to calculate arm weights and perform GD. This is inspired by the idea of mini-batch warm-start GD training [9]. In this case, a fixed number of training samples can help the round-wise inference time stay relatively stable, without growing drastically along with $T$.




*[Finn et al.] Model-agnostic meta-learning for fast adaptation
of deep networks.*



**Q2: Complementary experiment results?**




- **Experiment 1** (Figure 1): Warm-start vs. restarting from $\theta_{0}$ for each round, *given the same number of training (adaptation) samples*. We see the warm-start process can improve performances, as it can leverage previous trained parameters while restarting from $\theta_{0}$ can be hard to converge.

- **Experiment 2** (Figure 1): Here, we use $T=10000$ for both recommendation data sets (MovieLens, Amazon) and MNIST to maintain consistency, similar to existing neural bandits for recommendation works, which apply $T=10000$ for their recommendation data sets (e.g., [60,7]). Here, we include experiments with $T=15000$, where our proposed methods can maintain good performances.

- **Experiment 3** (Figure 2): Experimental results with increased number of arms $K$. We see that the warm-start strategy can lead to good performances while helping the inference time stay relatively stable.


- **Experiment 4** (Figure 3): Number of parameters and inference time, given different hidden dimensions. We can see that the inference time stays relatively stable, due to a fixed number of adaptation samples for warm-start GD.

- **Experiment 5** (Figure 4): MNIST data set under corruption-free setting, where our proposed methods can maintain competitive performances compared with baselines.


- **Experiment 6** (Figure 4): MovieLens data set with increasing corruption probability, and the probability of corrupting a chosen arm is $p_{t}$, where $p_{t}$ starts from $p_{0} = 20\%$ and increases by $\Delta p = 0.2\%$ for every 100 rounds. NeuralUCB-WGD and R-NeuralUCB can still enjoy an advantage over baselines.




**Q3: How arms are corrupted for our experiments?**

As in the caption of Figure 1: (1) For MovieLens and Amazon, the adversary will randomly decide whether to corrupt the reward of each chosen arm with a probability of 20\% or 50\%; (2) For MNIST data set, we consider a fixed budget $C=2000$ or $C=4000$, and randomly sample 2000 or 4000 rounds across $T=10000$ as corrupted.




**Q4: Context corruption?**

Recall that we use NTK-based regression to bridge reward mapping function $h(\cdot)$ and over-parameterized neural network $f(\cdot)$, where we quantify impacts of reward corruptions by decomposing NTK-based regression parameters as in Appendix Subsection C.4.
However, this approach will fail under arm context corruption settings. With gradient-based mapping being $g(\cdot)$, it can be difficult to measure the difference between projected context with corruption $g(x+\Delta x)$ and the original projection $g(x)$.
In this case, we may need additional assumptions, such as Lipschitz-smoothness assumption for the neural network [Du et al.] or Lipschitz reward mapping function [42]. We propose to explore this direction in our future works.


*[Du et al.] Gradient descent finds global minima of deep neural networks.*



**Q5: How to choose $\nu$?**

Similar to existing stochastic neural bandit works (e.g. [76,74,7]), when the random noise variance proxy $\nu$ is unknown to the learner, we consider it as a tunable parameter controlling exploration intensity, and use grid search to find a good $\nu$. Please also find related parameter study in Appendix Subsection A.2.

---

### Decision · Program_Chairs · 2024-09-25

**Decision:**

Accept (poster)

**Comment:**

The paper examined neural contextual bandits in the presence of adversarial reward corruptions. In this setting, the adversary can alter the reward after the bandit player has chosen an action but is constrained by a total corruption budget of size C.
The paper introduced a robust neural contextual bandit algorithm named R-NeuralUCB, which incorporates a novel context-aware gradient descent process. This process accounts for the uncertainty associated with both candidate arms and training samples. Additionally, when the neural network is over-parameterized, the paper offers a theoretical analysis of regret using the Neural Tangent Kernel (NTK) technique.
The authors provide regret analysis for R-NeuralUCB under over-parameterized neural network settings, without the commonly adopted arm separateness assumption. The authors also empirically compare R-NeuralUCB with baseline algorithms on three real data sets, under different adversarial corruption scenarios.

All reviewers agreed that the paper features a solid theoretical and empirical study for the study of robust stochastic bandits under adversarial corruptions. On the other hand, reviewers raised a common concern about computational cost. The authors presented an approach that deviated from the proposed algorithm, which is a balance between cost and performance guarantees, which I believe is reasonable.